

# Complexity and entanglement for thermofield double states

**Shira Chapman**[1,2⋆], **Jens Eisert**[3,4†], **Lucas Hackl**[2,5,6‡], **Michal P. Heller**[7,9∘],
**Ro Jefferson**[7§], **Hugo Marrochio**[2,8¶] **and Robert C. Myers**[2∥]

**1** Institute for Theoretical Physics, University of Amsterdam,
Science Park 904, Postbus 94485, 1090 GL Amsterdam, The Netherlands
**2** Perimeter Institute for Theoretical Physics, Waterloo, ON N2L 2Y5, Canada
**3** Dahlem Center for Complex Quantum Systems,
Freie Universität Berlin, D-14195 Berlin, Germany
**4** Department of Mathematics and Computer Science,
Freie Universität Berlin, 14195 Berlin, Germany
**5** Institute for Gravitation and the Cosmos and Department of Physics,
The Pennsylvania State University, University Park, PA 16802, USA
**6** Max Planck Institute of Quantum Optics,
Hans-Kopfermann-Straße 1, D-85748 Garching bei München, Germany
**7** Max Planck Institute for Gravitational Physics (Albert Einstein Institute),
Am Mühlenberg 1, D-14476 Potsdam-Golm, Germany
**8** Department of Physics and Astronomy, University of Waterloo,
Waterloo, ON N2L 3G1, Canada
**9** On leave from: National Centre for Nuclear Research, Hoża 69, 00-681 Warsaw, Poland

⋆ s.chapman@uva.nl, † jense@zedat.fu-berlin.de, ‡ lucas.hackl@lfhs.eu,
∘ michal.p.heller@aei.mpg.de, § rjefferson@aei.mpg.de,
¶ hmarrochio@perimeterinstitute.ca, ∥ rmyers@perimeterinstitute.ca

## Abstract

**Motivated by holographic complexity proposals as novel probes of black hole space-times, we explore circuit complexity for thermofield double (TFD) states in free scalar quantum field theories using the Nielsen approach. For TFD states at $t = 0$, we show that the complexity of formation is proportional to the thermodynamic entropy, in qualitative agreement with holographic complexity proposals. For TFD states at $t > 0$, we demonstrate that the complexity evolves in time and saturates after a time of the order of the inverse temperature. The latter feature, which is in contrast with the results of holographic proposals, is due to the Gaussian nature of the TFD state of the free bosonic QFT. A novel technical aspect of our work is framing complexity calculations in the language of covariance matrices and the associated symplectic transformations, which provide a natural language for dealing with Gaussian states. Furthermore, for free QFTs in 1+1 dimension, we compare the dynamics of circuit complexity with the time dependence of the entanglement entropy for simple bipartitions of TFDs. We relate our results for the entanglement entropy to previous studies on non-equilibrium entanglement evolution following quenches. We also present a new analytic derivation of a logarithmic contribution due to the zero momentum mode in the limit of vanishing mass for a subsystem containing a single degree of freedom on each side of the TFD and argue why a similar logarithmic growth should be present for larger subsystems.**



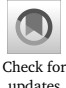

doi:10.21468/SciPostPhys.6.3.034

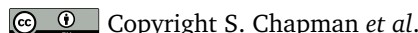

# Contents

## 1  Introduction

In the context of holography [1], *thermofield double (TFD)* states play an especially important role. From the perspective of the boundary theory, such states are formed by entangling two copies of a *conformal field theory (CFT)* in such a manner that tracing out either copy produces the thermal density matrix at inverse temperature $\beta > 0$ for the other, *i.e.,*

$$|\mathrm{TFD}(t_L, t_R)\rangle = \frac{1}{\sqrt{Z_\beta}} \sum_n e^{-\beta E_n/2} e^{-iE_n(t_L + t_R)} |E_n\rangle_L |E_n\rangle_R \, , \tag{1}$$

where $|E_n\rangle_{L,R}$ and $t_{L,R}$ are the energy eigenstate and times of the left/right CFTs respectively, and $Z_\beta$ is the canonical partition function at inverse temperature $\beta$. The TFD state (1) plays a special role in holography due to the fact that it is dual to an eternal black hole in AdS [2]. Hence, it provides a particularly well-controlled setup for studying various aspects of entanglement, black holes, and quantum information, *e.g.,* the time-evolution of entanglement entropy [3], scrambling and quantum chaos [4–6], firewalls [7–9], ER=EPR [8,10], and emergent spacetime [11].

The left and right sides of the geometry associated with the black hole dual to the TFD state are connected by a wormhole, or Einstein-Rosen bridge (ERB), whose volume increases for a time which is exponential in the number of degrees of freedom of the boundary theory [12,13]. This time is much larger than other characteristic times in holography, *e.g.,* the time for the mutual information to saturate or the scrambling time $t_* \sim \frac{\beta}{2\pi} \log S$. This implies that "entanglement (entropy) is not enough" [14] to capture the dynamics behind the horizon, and in particular that there must be some other quantity in the dual field theory which encodes this late-time evolution of the wormhole interior. It has been suggested that this quantity is the *quantum computational complexity* of the boundary state [10].

The concept of circuit complexity is rooted in theoretical computer science [15,16]. In classical computing, one is interested in implementing a given algorithm, *i.e.,* a function that maps an arbitrary set of input bits to specific output bits, using the minimum number of elementary operations or gates. In quantum computing, this function becomes a unitary operation $\widehat{U}$ which maps an input quantum state for some number of qubits to an output quantum state on an equal number of qubits [17–19]. Here, one may adopt a circuit model in which $\widehat{U}$ is

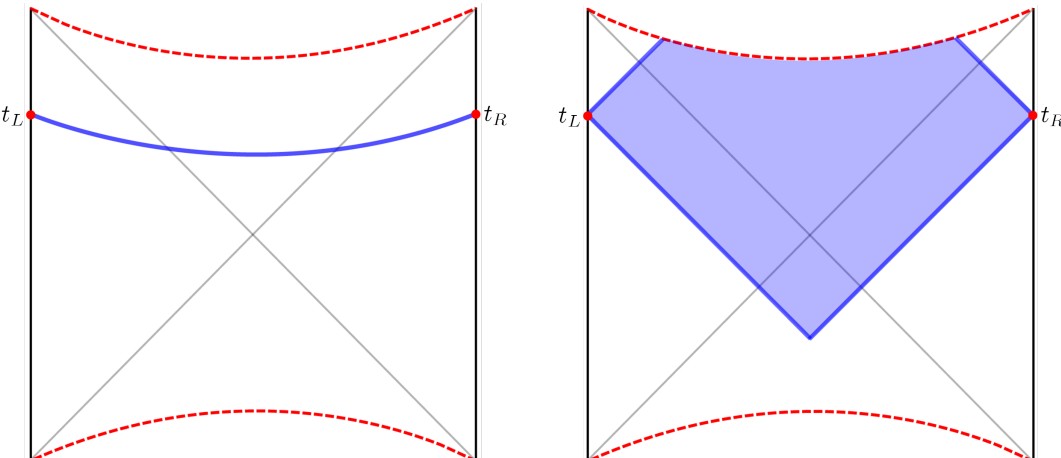

Figure 1: Complexity=volume (CV, left) and complexity=action (CA, right) for the eternal AdS black hole dual to the TFD state (1). Left panel: the blue curve represents the maximal spacelike surface that connects the specified time slices on the left and right boundaries. Right panel: the shaded region is the corresponding WdW patch. Figure and caption taken from ref. [23].

constructed from elementary gates – which typically act on some limited number of qubits – chosen from some fixed set of universal gates $\{\widehat{V}_I\}$. The circuit complexity of the unitary $\widehat{U}$ is then given by the minimal number $D$ of elementary gates $\widehat{V}_{I_k}$ required to construct the desired unitary, i.e., $\widehat{U} = \prod_{k=1}^{D} \widehat{V}_{I_k}$. We note that with discrete elementary gates $\{\widehat{V}_I\}$, most unitaries $\widehat{U}$ can only be obtained up to some accuracy $\varepsilon > 0$ (in operator norm), and therefore the circuit complexity is defined up to some tolerance $\varepsilon$. In the context of the preceding holographic conjectures, these notions should be adapted to define the circuit complexity for the states in the boundary theory [10, 12–14]. That is, we seek the minimum number of elementary gates required to prepare a desired target state $|\psi_{\text{T}}\rangle$ from a simple (unentangled) reference state $|\psi_{\text{R}}\rangle$, i.e.,

$$|\psi_{\text{T}}\rangle = \prod_{k=1}^{D} \widehat{V}_{I_k} |\psi_{\text{R}}\rangle. \tag{2}$$

Hence the complexity of a family of target states will be defined with respect to a particular reference state $|\psi_{\text{R}}\rangle$ and a choice of the gate set $\{\widehat{V}_I\}$, as well as a tolerance $\varepsilon$.

In terms of the gravitational description of the holographic theory, Susskind and collaborators have put forward two proposals for quantifying the size of the ERB dual to holographic complexity. The "complexity=volume" (CV) [20] proposal states that the complexity is given by the codimension-one volume of the maximal spacelike slice that connects the left and right CFTs through the bulk. The "complexity=action" (CA) [21,22] conjecture instead suggests that a more appropriate bulk dual is given by the gravitational action of the bulk region known as the Wheeler-deWitt (WdW) patch bounded by light sheets. These two proposals are illustrated in fig. 1.

Over the past few years, a wide variety of aspects of these proposals have been investigated on the gravitational side of the duality, e.g., see refs. [24–50], but this research program is still in its nascent stages. A particular obstacle to further progress is that a precise definition of complexity is still lacking for the boundary CFT. That is, it remains to construct a concrete formulation of eq. (2) for strongly coupled CFTs, or more broadly for general quantum field theories. Clearly, such a definition will be necessary in order to firmly establish any holographic complexity proposal as a new entry in the holographic dictionary. To that end, some initial

steps towards precisely defining complexity in quantum field theory have recently been made in refs. [23,51–69]. In particular, ref. [23] has considered the complexity of the ground state of a free scalar field theory. As alluded above, one must first identify both an appropriate reference state, and a suitable set of gates with which to construct the target state (*i.e.,* the ground state). However, evaluating the complexity still requires identifying the optimal circuit out of the infinite number of possible circuits which prepare the ground state of the theory. To overcome this challenge, the authors of ref. [23] has adapted a geometric approach developed by Nielsen and collaborators [70–72]. The essential idea in the latter is that the elementary gates acting on a chain of $n$ qubits form a representation of the Lie algebra $\mathfrak{su}(2^n)$, and hence one can define a natural geometry on the associated Lie manifold. This allows one to translate the question of finding the optimal circuit to that of finding the minimal geodesic in the space of unitaries $\widehat{U}$ equipped with a suitable metric. This idea has been applied to free scalar field theories in ref. [23], and subsequently extended to free fermionic theories in refs. [55,56] (see also ref. [57]). Another geometric definition of complexity based on the Fubini-Study metric has been explored for the case of a free scalar theory in ref. [51]. Despite the restriction to free scalar field theories, the results of refs. [23,51] exhibit some surprising similarities with holography in the structure of the UV divergences. While this is not sufficient to either confirm or rule out either the CV or CA proposals, it may suggest some degree of universality.

The purpose of the present work is to extend the construction of ref. [23] to evaluate the complexity of the TFD state in a free scalar field theory. That is, the state (1) will serve as our target state $|\psi_{\text{T}}\rangle$, while the reference state $|\psi_{\text{R}}\rangle$ will consist of two copies of the reference state used in refs. [23,51], namely the product state in which all of the positions are disentangled from each other. It will be sufficient to consider circuits $\widehat{U}$ generated by quadratic operators of the canonical variables to effect the necessary changes in the entanglement structure to produce the desired state (1) via eq. (2). The key feature which permits this construction is that both the reference and target states are Gaussian; furthermore, since all of the gates are generated by quadratic operators, all of the intermediate states will be Gaussian as well. We will use an approach based on the covariance matrix to manipulate these Gaussian states. This will allow us to study the time-dependence of complexity, as well as the complexity of formation of the thermal state, and to compare our results with those of the holographic complexity conjectures. We will find that the complexity of formation of the TFD state is proportional to the thermodynamic entropy, in qualitative agreement with holographic complexity proposals, but that it does not exhibit the late-time growth characteristic of holographic theories/fast scramblers [73] due to the Gaussian nature of the TFD state for free scalar theories. We note that this problem has been studied previously in refs. [52,53]. Our methods differ however, in particular in the choice of the cost function, and hence certain key features of our respective results differ as well, *e.g.,* the time-dependence of the complexity. We will compare the different approaches and results in detail in section 7.

In addition, for QFTs in 1+1 dimensions, we study the time evolution of entanglement entropy for subregions containing an equal number of sites on each side of the TFD state. We adapt the analytic formula of the time evolution of entanglement entropy following a global quench which was constructed based on a quasi-particle picture in refs. [74,75] to the case of the TFD state, and find a good agreement up to the effect of a zero mode which leads to a logarithmic growth for the case of a scalar with vanishing mass. We present an analytic derivation of the effect of this zero mode for a subsystem containing one degree of freedom on each side of the TFD and argue why a similar logarithmic growth is obtained for larger subsystems. We also comment on the comparison between the time evolution of complexity and that of entanglement entropy.

The remainder of this work is organized as follows: In section 2, we introduce the required preliminaries. We briefly review the approach of ref. [23] to circuit complexity, and then

demonstrate how the TFD state of two harmonic oscillators can be generated by quadratic operators. In section 3, we introduce a formalism for manipulating Gaussian states based on their covariance matrices, which serves as an alternate – and indeed, more natural – characterization of Gaussian states. We then proceed to evaluate the complexity for TFD states comprised of a pair of harmonic oscillators in section 4, by considering circuits that form a representation of $Sp(4, \mathbb{R})$. We then explain how to extend this construction to evaluate the complexity of TFD states of a free scalar field theory in section 5. In section 6, we study the dynamics of entanglement entropy of simple subsystems involving an equal number of sites on each side of the TFD in 1+1-dimensional QFTs and compare it to the dynamics of complexity. We close in section 7 with a brief discussion and outlook. Various technical details have been relegated to a series of appendices. In appendix A, we collect information about our conventions and notation. In appendix B we review how to construct the TFD state for a single harmonic oscillator in a unitary way. In appendix C, we collect the matrix generators of $Sp(4, \mathbb{R})$ which are relevant for the discussion in section 4. Appendix D comments on a few details with respect to the basis of operators use to describe the circuits and the complexity in the scalar field theory. In appendix E, we provide further details on the complexity of formation of the TFD state in the diagonal basis. In appendix F, we present the proof that the shortest unpenalized circuit in the full $Sp(2N, \mathbb{R})$ geometry (for the case $\lambda_{\text{R}} = 1$ using the $L^2$ norm) amounts to an independent squeezing of the normal modes. We conclude with appendix G where we derive compact matrix functions for the time-dependent covariance matrix in position space, which we use for the efficient numerical evaluation of the entanglement entropy.

## 2 Preliminaries: circuit complexity and thermofield double states

In this section, we provide the relevant preliminaries for the construction of both the TFD state and the relevant quantum circuits. We start with a brief overview of Nielsen's geometric approach to evaluating quantum circuit complexity. In the second part of this section, we will present the thermofield double state for a simple harmonic oscillator in preparation for our studies of its complexity in the rest of the paper.

### 2.1 Circuit complexity from Nielsen geometry

As mentioned in the introduction, in applying circuit complexity to quantum field theory, we are essentially quantifying the effort needed to prepare a target state $|\psi_{\text{T}}\rangle$ beginning from a certain reference state $|\psi_{\text{R}}\rangle$. That is, we wish to construct the shortest circuit which performs the transformation

$$|\psi_{\text{T}}\rangle = \widehat{U}_{\text{T}} |\psi_{\text{R}}\rangle \,, \tag{3}$$

where the unitary is constructed as a sequence of elementary unitary gates, $\widehat{U}_{\text{T}} = \prod_{k=1}^{D} \widehat{V}_{I_k}$, and the complexity corresponds to the number of gates $D$ in the optimal construction. A generic gate $\widehat{V}_I$ can be written as an exponential $\widehat{V}_I = e^{-i\varepsilon \widehat{K}_I}$, where $\widehat{K}_I$ is some Hermitian operator which only acts on a few degrees of freedom, and $\varepsilon$ is a small parameter which ensures that each gate only produces a small change on the state. In general however, there may be arbitrarily many different circuits, *i.e.,* different sequences of elementary gates, which yield the same target state, and so the primary challenge in defining complexity for the latter is identifying the optimal circuit from amongst the infinite family of possible constructions of $\widehat{U}_{\text{T}}$.

To overcome this challenge, Nielsen and collaborators [70–72] introduced a geometric approach to identify the optimal circuit, which ref. [23] adapted to define the complexity of

Gaussian states in free scalar field theory. While the above discussion phrased the construction of $\widehat{U}_\text{T}$ as a string of discrete gates, Nielsen's approach begins by introducing a continuous parametrization of the unitary operators, namely

$$\widehat{U} = \overleftarrow{\mathcal{P}} \exp\left[-i \int_0^1 \mathrm{d}s \, H(s)\right] \qquad \text{with} \qquad H(s) = \sum_I Y^I(s) \widehat{K}_I, \tag{4}$$

where the Hermitian operators $\widehat{K}_I$, which appeared in the unitary gates above, now form a basis for the interactions appearing in the "time"-dependant Hamiltonian $H(s)$. The path-ordering symbol $\overleftarrow{\mathcal{P}}$ indicates that the circuit is constructed from right to left as $s$ increases. Hence we can think of the control functions $Y^I(s)$ as indicating which gates are inserted at a given point in the circuit, *i.e.,* at a given "time" $s$. One can then extend this construction such that the circuit (4) represents a trajectory through the space of unitaries,

$$\widehat{U}(\sigma) = \overleftarrow{\mathcal{P}} \exp\left[-i \int_0^\sigma \mathrm{d}s \, H(s)\right]. \tag{5}$$

From this perspective, $Y^I(\sigma)$ becomes the tangent vector to the trajectory $U(\sigma)$, with

$$Y^I(\sigma) \widehat{K}_I = \frac{\mathrm{d}U(\sigma)}{\mathrm{d}\sigma} U^{-1}(\sigma). \tag{6}$$

Our problem is then to find the "shortest" path which starts at $\widehat{U}(\sigma=0) = \mathbb{1}$, and ends at $\widehat{U}(\sigma=1) = \widehat{U}_\text{T}$, where the latter effects the desired transformation in eq. (3).[1] Of course, there are still (infinitely) many paths which will produce the desired unitary $\widehat{U}_\text{T}$, and so the question remains how to identify the optimal trajectory.

Nielsen's approach [70–72] is to define the circuit depth

$$D(U) = \int_0^1 \mathrm{d}s \, F(U(s), Y^I(s)), \tag{7}$$

where the *cost function* $F(U(s), Y^I(s))$ is a local functional of the position $U(s)$ and the tangent vector $Y^I(s)$. This functional must satisfy a number of conditions:

1. *Smoothness*: $F \in C^\infty$.

2. *Positive definiteness*: $F \geq 0 \;\; \forall \, U, Y^I$, with equality if and only if $Y^I = 0$.

3. *Triangle inequality*: $F(U, Y^I + \tilde{Y}^I) \leq F(U, Y^I) + F(U, \tilde{Y}^I) \;\; \forall \, U, Y^I$, and $\tilde{Y}^I$.

4. *Homogeneity*: $F(U, \lambda Y^I) = \lambda F(U, Y^I) \;\; \forall \, U, Y^I$ and $0 \leq \lambda \in \mathbb{R}$.

Note that this last condition is the requirement for the cost function to be an asymmetric norm.[2] These conditions still leave us with an enormous freedom in choosing the cost function. In the following, we will focus on some simple choices, *e.g.,*

$$F_1 = \sum_I |Y^I|, \qquad F_2 = \left(\sum_I (Y^I)^2\right)^{1/2}, \tag{8}$$

---

[1] We might add that there is no need to consider a tolerance $\varepsilon$ here, since in this framework of continuous circuits (4) we can always adjust $Y^I(s)$ to produce exactly the desired transformation.

[2] It is called asymmetric since the requirement only applies for $\lambda \geq 0$, and so there is no particular relation between $F(U, Y^I)$ and $F(U, -Y^I)$.

which represent the $L^1$- and $L^2$-norms with respect to the basis $\widehat{K}_I$, and satisfy the conditions (1)–(4) above. Another simple set of cost functions introduced in ref. [23] are

$$D_\kappa = \sum_I |Y^I|^\kappa \,, \tag{9}$$

where $\mathbb{R} \ni \kappa > 1$, which however do not satisfy the condition (4). In any event, Nielsen's approach reduces the technical problem of identifying the optimal trajectory to the familiar physics problem of studying the motion of a particle along geodesics, albeit perhaps governed by an unusual Lagrangian.

To make this problem tractable, one typically focuses on a limited basis of operators $\widehat{K}_I$ to construct the unitary circuit (4). Of course, this limits the family of target states that can be constructed from a given reference state $|\psi_{\text{R}}\rangle$ in eq. (3), but it admits a powerful group-theoretic structure if these operators form a closed algebra, *i.e.,* a Lie algebra $\mathfrak{g}$ with $[\widehat{K}_I, \widehat{K}_J] = i f_{I,J}{}^K \widehat{K}_K$. Recent studies of the complexity in free quantum field theories (see, *e.g.,* refs. [23,55,56,58,59]) focused on circuits which remain within the space of Gaussian states, and hence the group of Bogoliubov transformations played an important role. For example, a $GL(N, \mathbb{R})$ algebra appeared in the construction of the free scalar ground state using a lattice of $N$ bosonic degrees of freedom in [23], while the analogous group structure for free (non-interacting) fermions was $O(2N)$ in refs. [55,56]. The group-theoretic perspective proves to be quite powerful in evaluating the circuit complexity as seen in these examples. One key benefit of this perspective is that the specific physical details of the operators $\widehat{K}_I$ become unimportant once the underlying group structure is identified. Rather, we can think of these generators as the elements of the corresponding Lie algebra $\mathfrak{g}$, and of the circuits (5) as trajectories on the corresponding group manifold $\mathcal{G}$. In particular, we are free to choose whichever representation is most convenient for our calculations. In the present paper, we again consider bosonic Gaussian states, but it will be necessary to expand the approach of [23] to the full group of Bogoliubov transformations, namely the symplectic group $Sp(2N, \mathbb{R})$. Additionally, it will turn out to be very useful to characterize the Gaussian states by their covariance matrices $G^{a,b}$, as explained in section 3, whereupon the unitary operators $\widehat{U}$ are represented as symplectic matrices $U^a{}_b$ acting on these covariance matrices. Similarly, in this representation, we can construct matrix generators $K^a{}_b$ associated to the operators $\widehat{K}$.[3]

## 2.2 TFD state for the simple harmonic oscillator

As motivated in the introduction, we wish to study the complexity of the TFD state (1) by applying the notion of circuit complexity for Gaussian states introduced in ref. [23]. The complexity of states in QFTs is in general divergent, due to the need to introduce correlations up to arbitrarily short length scales when building the states. In order to study the complexity of the ground state of a free bosonic QFT, ref. [23] regulated the theory by discretizing it on a spatial lattice. The theory then takes the form of a set of coupled simple harmonic oscillators. We will follow the same approach here, and begin by considering the TFD state (1) constructed from two simple harmonic oscillators. We will later explain how this allows us to evaluate the complexity of the TFD formed from two copies of a free bosonic field theory in section 5.

Let us introduce some basic notation before we proceed. The Hamiltonian for a single oscillator is given by[4]

$$H = \frac{1}{2M} P^2 + \frac{1}{2} M \omega^2 Q^2 \,, \tag{10}$$

---

[3]We will denote the matrix representation of the group elements $U \in Sp(2N, \mathbb{R})$ and the corresponding generators $K \in \mathfrak{sp}(2N, \mathbb{R})$. When we want to refer to them explicitly as quantum operators acting on the Hilbert space $\mathcal{H}$, we will use hats, *e.g.,* $\widehat{U}$ and $\widehat{K}$.

[4]Note that in ref. [23], the mass of the oscillators was set to unity. We avoid doing so here, for reasons that will become clear below.

where $M$ is the mass of the oscillator, $\omega$ is its frequency, and $Q$ and $P$ are the position and momentum operators satisfying the standard commutation relations. In terms of the canonical annihilation and creation operators,

$$a = \sqrt{\frac{M\omega}{2}}\left(Q + i\,\frac{P}{M\omega}\right), \qquad a^\dagger = \sqrt{\frac{M\omega}{2}}\left(Q - i\,\frac{P}{M\omega}\right), \tag{11}$$

which satisfy $[a, a^\dagger] = 1$, the Hamiltonian (10) becomes

$$H = \omega\left(N + \frac{1}{2}\right) = \omega\left(a^\dagger a + \frac{1}{2}\right), \tag{12}$$

where $N \equiv a^\dagger a$ is the number operator. In the following, it will be useful to invert (11) via

$$Q = \frac{1}{\sqrt{2M\omega}}\left(a^\dagger + a\right), \qquad P = i\sqrt{\frac{M\omega}{2}}\left(a^\dagger - a\right). \tag{13}$$

The (normalized) energy eigenstates are given by acting on the vacuum $|0\rangle$ with creation operators,

$$|n\rangle = \frac{(a^\dagger)^n}{\sqrt{n!}}|0\rangle, \tag{14}$$

where

$$a^\dagger|n\rangle = \sqrt{n+1}|n+1\rangle, \qquad a|n\rangle = \sqrt{n}|n-1\rangle, \qquad N|n\rangle = n|n\rangle. \tag{15}$$

Of course, one then has

$$H|n\rangle = \omega\left(n + \frac{1}{2}\right)|n\rangle, \tag{16}$$

as usual.

In the following, we consider two identical copies of a simple harmonic oscillator, which we denote by subscripts $L$ and $R$ in analogy to the left- and right- copies of the CFT in the Penrose diagram in fig. 1. We start by considering the case of the TFD state (1) at $t_L + t_R = 0$ and explain how to express it as a quadratic operator acting on the tensor product of the vacua of two harmonic oscillators. We then consider the generalization to the time-dependent TFD.

### 2.2.1  TFD state at $t = 0$

The TFD state (1) at $t_L = 0 = t_R$ can be constructed from two copies of the vacuum state by acting with creation operators in the following manner, *e.g.*, see ref. [76],

$$
\begin{aligned}
|\text{TFD}\rangle &= \left(1 - e^{-\beta\omega}\right)^{1/2}\sum_{n=0}^{\infty}e^{-n\beta\omega/2}|n\rangle_L|n\rangle_R \\
&= \left(1 - e^{-\beta\omega}\right)^{1/2}\sum_{n=0}^{\infty}\frac{e^{-n\beta\omega/2}}{n!}\left(a_L^\dagger a_R^\dagger\right)^n|0\rangle_L|0\rangle_R \\
&= \left(1 - e^{-\beta\omega}\right)^{1/2}\exp\left[e^{-\beta\omega/2}a_L^\dagger a_R^\dagger\right]|0\rangle_L|0\rangle_R,
\end{aligned}
\tag{17}
$$

where we have taken $E_n = \omega(n + \frac{1}{2})$. However, since we wish to construct our circuit – and hence the state – from unitary operators, the form (17) is not quite the desired result. Rather, we wish to express the TFD state by acting with a unitary operator on the vacuum $|0\rangle_L|0\rangle_R$. Such a rearrangement of eq. (17) can be achieved, as explained in appendix B, using the decomposition formula (252), which yields

$$|\text{TFD}\rangle = \exp\left[\alpha\left(a_L^\dagger a_R^\dagger - a_L a_R\right)\right]|0\rangle_L|0\rangle_R, \tag{18}$$

where we have defined

$$\tanh \alpha := \exp(-\beta\omega/2). \tag{19}$$

For later purposes, it is convenient to express $\alpha$ in the form

$$\alpha = \frac{1}{2}\log\left(\frac{1+e^{-\beta\omega/2}}{1-e^{-\beta\omega/2}}\right), \tag{20}$$

where of course $\alpha > 0$. Note that the normalization $Z_\beta^{-1/2} = \left(1-e^{-\beta\omega}\right)^{1/2}$ in eq. (18) has been absorbed into the exponential. Now, observe that by using eq. (11), we can re-express the generator in eq. (18) as

$$a_L^\dagger a_R^\dagger - a_L a_R = -i\left(Q_R P_L + Q_L P_R\right). \tag{21}$$

Thus we see that, in the language of ref. [23], the TFD state (18) can be interpreted as the result of acting with an entangling operator on the product state of the vacua of the two oscillators.

As we will explain below, a key feature of our approach is that the TFD factorizes in a particular basis, which we refer to as the *diagonal basis*.[5] One can see this by expressing the generator (21) in terms of the diagonal coordinates for the combined system

$$Q_\pm = \frac{1}{\sqrt{2}}\left(Q_L \pm Q_R\right), \qquad P_\pm = \frac{1}{\sqrt{2}}\left(P_L \pm P_R\right), \tag{22}$$

which yields

$$a_L^\dagger a_R^\dagger - a_L a_R = -i\left(Q_+ P_+ - Q_- P_-\right). \tag{23}$$

In other words, the generator can be re-expressed in terms of the scaling operators of the individual diagonal modes. The advantage now is that these two components commute, and hence eq. (18) factorizes as

$$|\text{TFD}\rangle = \exp\left[-\frac{i\alpha}{2}\left(Q_+ P_+ + P_+ Q_+\right)\right]|0\rangle_+ \ \otimes\ \exp\left[\frac{i\alpha}{2}\left(Q_- P_- + P_- Q_-\right)\right]|0\rangle_-, \tag{24}$$

where $|0\rangle_\pm$ denotes the vacuum of the Hamiltonian of each diagonal mode. Of course, these Hamiltonians look the same as in eq. (10), cf. (25) below. Note that we have rearranged the expression in eq. (24) in such a way that the operator in each factor is unitary.

At this point, the set-up for the time-independent ($t = 0$) TFD is essentially complete. To facilitate our circuit computations below however, let us examine the corresponding wavefunctions that serve as our reference and target states. We began with two independent harmonic oscillators, and hence the total Hamiltonian is given by (cf. (10))

$$H_{\text{total}} = \frac{1}{2M}\left[P_L^2 + P_R^2 + M^2\omega^2\left(Q_L^2 + Q_R^2\right)\right] = \frac{1}{2M}\left[P_+^2 + P_-^2 + M^2\omega^2\left(Q_+^2 + Q_-^2\right)\right], \tag{25}$$

where in the second equality, we have used the diagonal basis (22). The ground-state wavefunction for this Hamiltonian takes the simple form

$$\psi_0(Q_+, Q_-) = \psi_0(Q_+)\psi_0(Q_-) \simeq \exp\left[-\frac{M\omega}{2}\left(Q_+^2 + Q_-^2\right)\right]. \tag{26}$$

---

[5]As we will see momentarily, the distinguishing feature of the diagonal basis is that the modes are decoupled in the generator (21) producing the TFD. To avoid confusion, we reserve the term "normal mode" to denote the modes which diagonalize the Hamiltonian for a single copy of the QFT in section 5.

We might characterize the analysis in refs. [23, 51] as defining the complexity of this ground state given the reference state[6]

$$\psi_{\text{R}}(Q_+, Q_-) \simeq \exp\left[-\frac{M\mu}{2}\left(Q_+^2 + Q_-^2\right)\right].$$

(27)

We can think of this reference state as being the ground state of a Hamiltonian as eq. (25), but where the frequency is fixed to be $\mu$, the reference frequency of our complexity model – compare with eq. (182) in our analysis of the scalar field theory. In the present work, we extend these results to define the complexity of the TFD state relative to the same reference state $|\psi_{\text{R}}\rangle$. Examination of eq. (24) indicates that the wavefunction for the TFD is obtained by acting on the vacuum wavefunction (26) with the appropriate scaling operators in the diagonal basis. For example, using the results of ref. [23], we have

$$\exp\left[-\frac{i\alpha}{2}(Q_+P_+ + P_+Q_+)\right]\psi_0(Q_+) \simeq \psi_0(e^{-\alpha}Q_+) \simeq \exp\left[-\frac{1}{2}e^{-2\alpha}M\omega Q_+^2\right].$$

(28)

Therefore the desired wavefunction is simply given by

$$\begin{aligned}
\psi_{\text{TFD}} &\simeq \exp\left[-\frac{M\omega}{2}\left(e^{-2\alpha}Q_+^2 + e^{2\alpha}Q_-^2\right)\right] \\
&= \exp\left[-\frac{M\omega}{2}\left(\cosh(2\alpha)(Q_L^2 + Q_R^2) - 2\sinh(2\alpha)Q_LQ_R\right)\right].
\end{aligned}$$

(29)

Thus, the target state is indeed a simple Gaussian wavefunction of the same form studied in refs. [23, 51].

### 2.2.2 Time-dependent TFD state

It is relatively straightforward to extend the manipulations of the previous subsection to the full time-dependent case. For simplicity, we shall follow the common convention in holography (see, *e.g.,* refs. [20, 28]) and set $t_L = t_R = t/2$ in eq. (1). Then the generalization of eq. (17) becomes

$$\begin{aligned}
|\text{TFD}(t)\rangle &= \left(1 - e^{-\beta\omega}\right)^{1/2}\sum_{n=0}^{\infty}e^{-n\beta\omega/2}e^{-i\left(n+\frac{1}{2}\right)\omega t}|n\rangle_L|n\rangle_R \\
&= e^{-\frac{i}{2}\omega t}\left(1 - e^{-\beta\omega}\right)^{1/2}\sum_{n=0}^{\infty}\frac{e^{-n\beta\omega/2}e^{-in\omega t}}{n!}\left(a_L^{\dagger}a_R^{\dagger}\right)^n|0\rangle_L|0\rangle_R \\
&= e^{-\frac{i}{2}\omega t}\left(1 - e^{-\beta\omega}\right)^{1/2}\exp\left[e^{-\beta\omega/2}e^{-i\omega t}a_L^{\dagger}a_R^{\dagger}\right]|0\rangle_L|0\rangle_R.
\end{aligned}$$

(30)

Note that in what follows we will simply drop the global time-dependent phase given by the $e^{-\frac{i}{2}\omega t}$ factor, since this does not change the physical state.[7] As above, we wish to express this

---

[6]Compare with eq. (2.10) of ref. [23]. Note that in order to avoid potential confusion with the ground state $\psi_0$, we shall here denote the reference scale set by the variance of the canonical variables by $M\mu$, rather than $M\omega_0$ (with $M = 1$) as in *ibid*.

[7]One could consider adding a contribution to the complexity to account for this phase using a phase gate as in [23]. This contribution to the complexity would just be some weight or penalty factor times a simple function of the phase, *i.e.,* one would take the absolute value of the portion of the phase between $-\pi$ and $\pi$. However, when considering the vacuum state, one would acquire an identical phase contribution to the complexity as it evolves in time (recall we are evolving with $t_L = t_R = t/2$. Note that no such contribution arises in holographic complexity. Furthermore, if we compare the complexity of the TFD to that of the vacuum at general times, this contribution would cancel out.

state in terms of a unitary operator acting on the vacuum state. As explained in appendix B, using eq. (257), one finds

$$|\text{TFD}(t)\rangle = \exp\!\big(z\,a_L^\dagger a_R^\dagger - z^*\,a_L a_R\big)|0\rangle_L|0\rangle_R,\tag{31}$$

where we have defined

$$z = \alpha\,e^{-i\omega t}\tag{32}$$

and $\alpha$ is given by eq. (19). Note that this reduces to eq. (18) upon setting $t = 0$. As before, we use eq. (13) to re-express the generator in eq. (31) as

$$
\begin{aligned}
&e^{-i\omega t}\,a_L^\dagger a_R^\dagger - e^{i\omega t}\,a_L a_R\\
&= -i\cos(\omega t)(Q_R P_L + Q_L P_R) - i\sin(\omega t)\left(M\omega Q_L Q_R - \frac{1}{M\omega}P_L P_R\right).
\end{aligned}\tag{33}
$$

At this point, we make the important observation that producing the time-dependent TFD requires moving beyond the scaling and entangling operators of the $\mathfrak{gl}(2,\mathbb{R})$ algebra considered in ref. [23]. Rather, we must build our unitaries using generators from the full algebra formed by all possible bilinears of $Q_L, Q_R, P_L, P_R$, which form the algebra $\mathfrak{sp}(4,\mathbb{R})$ [77]. As we demonstrate in the following section, such unitaries describe the set of transformations of two-mode Gaussian states with vanishing one-point functions among themselves. In fact, we will be mostly using the diagonal basis and the corresponding $\mathfrak{sp}(2,\mathbb{R})$ subalgebras for each diagonal mode formed by bilinears of $P_+, Q_+$ or $P_-, Q_-$ respectively.

Rotating to the diagonal basis (22) as above, the generator (33) becomes

$$
\begin{aligned}
e^{-i\omega t}\,a_L^\dagger a_R^\dagger - e^{i\omega t}\,a_L a_R ={}& -i\cos(\omega t)Q_+ P_+ - \frac{i}{2}\sin(\omega t)\left(M\omega Q_+^2 - \frac{1}{M\omega}P_+^2\right)\\
&+ i\cos(\omega t)Q_- P_- + \frac{i}{2}\sin(\omega t)\left(M\omega Q_-^2 - \frac{1}{M\omega}P_-^2\right),
\end{aligned}\tag{34}
$$

where we see that as in the time-independent case, the generator can be decomposed into two (anti-Hermitian) operators,

$$\widehat{\mathcal{O}}_\pm(t) = \frac{1}{2}\cos(\omega t)(Q_\pm P_\pm + P_\pm Q_\pm) + \frac{1}{2}\sin(\omega t)\left(M\omega Q_\pm^2 - \frac{1}{M\omega}P_\pm^2\right),\tag{35}$$

which act separately in the '+' or '−' Hilbert spaces. Therefore the time-dependent TFD continues to factorizes as

$$|\text{TFD}(t)\rangle = \exp\!\big[-i\alpha\,\widehat{\mathcal{O}}_+(t)\big]|0\rangle_+ \otimes \exp\!\big[i\alpha\,\widehat{\mathcal{O}}_-(t)\big]|0\rangle_-.\tag{36}$$

This state can also be expressed as a Gaussian wavefunction of the general form (29), and serves as our target state for the time-dependent case. In this case the explicit expression for the wavefunction is somewhat complicated, so we will not write it out here. Rather, below we will introduce a more elegant formalism for characterizing these Gaussian states by their covariance matrices, which then allows us to represent the symplectic transformations of $\mathfrak{sp}(4,\mathbb{R})$ (or of the $\mathfrak{sp}(2,\mathbb{R})$ subalgebras) acting on them as matrix operations.

### 2.2.3 Gate scale

Looking ahead, we note that in constructing the quantum circuits (4), our generators, *i.e.,* the $\widehat{K}_I$, will consist of all quadratic combinations of the $Q$'s and $P$'s – see section 3.2. However, since as usual these operators are dimensionful, we will need to include an additional scale

in our complexity model. The simplest approach, which we follow here, is to introduce the dimensionless position and momenta

$$q_a := \omega_g Q_a, \qquad p_a := \frac{P_a}{\omega_g}, \tag{37}$$

where we refer to $\omega_g$ as the *gate scale*. In particular, we can then construct dimensionless generators $\widehat{K}_I$ as all quadratic combinations of these dimensionless $q$'s and $p$'s. For example, we might think of some typical elementary gates represented as

$$U(\varepsilon) = e^{i\varepsilon\, q_a q_b} = e^{i\varepsilon\, \omega_g^2 Q_a Q_b}, \qquad \text{or} \qquad \widetilde{U}(\varepsilon) = e^{i\varepsilon\, p_a p_b} = e^{i\varepsilon\, \frac{P_a P_b}{\omega_g^2}}, \tag{38}$$

where $\varepsilon$ is a dimensionless infinitesimal parameter.[8] Hence the gate scale implicitly ensures that all of the components of the control functions $Y^I(s)$ are dimensionless and are readily combined in cost functions, such as those given in eqs. (8) and (9).

In terms of these dimensionless variables, the reference (27), vacuum (26), and time-independent TFD (29) states become, respectively,

$$\psi_{\text{R}}(q_+, q_-) = \sqrt{\frac{\lambda_{\text{R}}}{\pi}} \exp\left[-\frac{\lambda_{\text{R}}}{2}\left(q_+^2 + q_-^2\right)\right],$$

$$\psi_0(q_+, q_-) = \sqrt{\frac{\lambda}{\pi}} \exp\left[-\frac{\lambda}{2}\left(q_+^2 + q_-^2\right)\right], \tag{39}$$

$$\psi_{\text{TFD}}(q_+, q_-) = \sqrt{\frac{\lambda}{\pi}} \exp\left[-\frac{\lambda}{2}\left(e^{-2\alpha}q_+^2 + e^{2\alpha}q_-^2\right)\right],$$

where we have defined the dimensionless ratios

$$\lambda_{\text{R}} := M\mu/\omega_g^2 \quad \text{and} \quad \lambda := M\omega/\omega_g^2. \tag{40}$$

## 3 Covariance matrix approach

So far we have described Gaussian states of bosonic systems in terms of their associated wavefunctions. It turns out, however, that this representation is more complicated than necessary; in particular, as alluded above, the wavefunction for the time-dependent TFD is rather unwieldy. Fortunately, an equivalent and simpler representation is available at the level of covariance matrices, which greatly facilitates our analysis. In this section, we review the relevant aspects of this approach.

### 3.1 From quantum states to covariance matrices

A bosonic system with $N$ degrees of freedom can be described by $2N$ linear observables $\xi = (q_1, \cdots, q_N, p_1, \cdots, p_N)$ consisting of canonical coordinates. Given an arbitrary quantum state $|\psi\rangle$, its two-point function may be expressed as

$$\langle\psi|\xi^a \xi^b|\psi\rangle = \frac{1}{2}(G^{a,b} + i\Omega^{a,b}), \tag{41}$$

where $G^{a,b} = G^{(a,b)}$ is symmetric and $\Omega^{a,b} = \Omega^{[a,b]}$ is antisymmetric. Such a decomposition can always be performed for an arbitrary matrix, but we included an $i$ in front of $\Omega^{a,b}$ to ensure that

---

[8]The previous discussion in [23] did not require a gate scale because it only considered gates constructed with the generators $P_a Q_b$, which are dimensionless in natural units.

both $G^{a,b}$ and $\Omega^{a,b}$ are real linear forms. This follows directly from the fact that $\xi^a \xi^b + \xi^b \xi^a$ is a Hermitian operator with real eigenvalues, while $\xi^a \xi^b - \xi^b \xi^a$ is anti-Hermitian with purely imaginary eigenvalues. In fact, the latter is completely fixed by the canonical commutation relations to

$$\Omega^{a,b} = \begin{pmatrix} 0 & \mathbb{1} \\ -\mathbb{1} & 0 \end{pmatrix}, \tag{42}$$

with respect to the coordinates $\xi$ introduced above.

If $|\psi\rangle$ is a pure Gaussian state with $\langle\psi|\xi^a|\psi\rangle = 0$, then it is completely characterized by its symmetric two-point function, often referred to as its (symmetric) *covariance matrix G* with entries

$$G^{a,b} = \langle\psi|(\xi^a \xi^b + \xi^b \xi^a)|\psi\rangle. \tag{43}$$

For a mixed state $\rho$ with vanishing first moments, one can in the same way define [78, 79]

$$G^{a,b} = \text{tr}(\rho(\xi^a \xi^b + \xi^b \xi^a)). \tag{44}$$

In particular, we can use Wick's theorem to compute any higher order $n$-point functions from $G^{a,b}$. Hence we can without ambiguity use $G$ as label for these Gaussian states. For example, let us consider a general pure Gaussian state in the Hilbert space of a single degree of freedom. Such a state can be parameterized by its wavefunction as

$$\psi(q) = \langle q|\psi\rangle = \left(\frac{a}{\pi}\right)^{1/4} \exp\left[-\frac{1}{2}(a+ib)q^2\right], \tag{45}$$

where $a, b \in \mathbb{R}$, and $a > 0$ for normalizability. The two-point function (43) may be explicitly evaluated, and the results encapsulated in the covariance matrix

$$G = \begin{pmatrix} \frac{1}{a} & -\frac{b}{a} \\ -\frac{b}{a} & \frac{a^2+b^2}{a} \end{pmatrix}. \tag{46}$$

One sees that it is straightforward to extract the parameters $a$ and $b$ of the wavefunction from the covariance matrix $G$: $a = 1/G^{1,1}$ and $b = -G^{2,1}/G^{1,1}$.[9] This illustrates our claim above that the covariance matrix provides a complete characterization of Gaussian states, and can therefore be used as an alternative description thereof. In particular, the wavefunctions (39) each decouple into a product of wavefunctions for the $\pm$ modes, and hence the associated covariance matrices for the total states are block-diagonal. The covariance matrices for the $+$ mode read

$$G_{\text{R}} = \begin{pmatrix} \frac{1}{\lambda_{\text{R}}} & 0 \\ 0 & \lambda_{\text{R}} \end{pmatrix}, \qquad G_0 = \begin{pmatrix} \frac{1}{\lambda} & 0 \\ 0 & \lambda \end{pmatrix}, \qquad G_{\text{TFD}}^{+} = \begin{pmatrix} \frac{e^{2\alpha}}{\lambda} & 0 \\ 0 & e^{-2\alpha}\lambda \end{pmatrix}, \tag{47}$$

and similarly for the $-$ mode, with $\alpha \to -\alpha$.

## 3.2 Trajectories between states and their generators

The power of the covariance matrix formalism lies in the fact that we can study trajectories in state space purely in terms of $G^{a,b}$, provided that we do not leave the sector of Gaussian states.

---

[9]For pure states, $\det(G) = 1$, so the remaining component $G^{2,2}$ does not contain any new information. In fact, for pure states, the eigenvalues of $G \cdot \Omega$ must appear in pairs $\pm i$, as the symplectic spectrum of covariance matrices of pure quantum states consists of unit elements only [78, 79]. More generally, however, the covariance matrix for a mixed state has $\det(G) > 1$, and hence all three components of $G^{a,b}$ are independent.

Restricting to this class of states can be easily achieved by focusing on a natural subgroup of unitary transformations which evolves Gaussian states into Gaussian states, *i.e.,* the Bogoliubov transformations. This subgroup is generated by Hermitian operators that are quadratic in the canonical coordinates $\xi$ introduced above. The most general quadratic operator can be written as

$$\widehat{K} = \frac{1}{2}\xi^a k_{a,b}\xi^b = \frac{1}{2}\xi k \xi^{\mathsf{T}}, \tag{48}$$

where $k = k_{(a,b)}$ is chosen to be a real symmetric form. This is because any antisymmetric part in $k$ does not affect $\widehat{K}$ due to $\xi^a\xi^b$ being symmetric.[10] Such a general operator $\widehat{K}$ generates unitaries

$$\widehat{U}(\sigma) = e^{-i\sigma\widehat{K}}, \qquad |G_\sigma\rangle = \widehat{U}(\sigma)|G_0\rangle, \tag{49}$$

that map Gaussian states into Gaussian states [80,81]. To find the covariance matrix associated with the new state $|G_\sigma\rangle$, we start by computing the operation of $\widehat{U}(\sigma)$ on $\xi^a$ given by

$$\widehat{U}^\dagger(\sigma)\xi^a\widehat{U}(\sigma) = \sum_{n=0}^{\infty}\frac{\sigma^n}{n!}[i\widehat{K}, \xi^a]_{(n)}, \tag{50}$$

where $[i\widehat{K}, \xi^a]_{(n)}$ is defined recursively via $[i\widehat{K}, \xi^a]_{(n)} = [i\widehat{K}, [i\widehat{K}, \xi^a]_{(n-1)}]$ and $[i\widehat{K}, \xi^a]_{(0)} = \xi^a$, where we have used the well-known Baker-Campbell-Hausdorff formula. Using the commutation relation $[\xi^a, \xi^b] = i\Omega^{a,b}$, one finds

$$[i\widehat{K}, \xi^a] = \Omega^{a,b}k_{b,c}\xi^c = K^a{}_b\xi^b, \tag{51}$$

where we have defined the matrix generator associated with $\widehat{K}$,

$$K^a{}_b = \Omega^{a,c}k_{c,b}. \tag{52}$$

One can check that $K \in \mathfrak{sp}(2N,\mathbb{R})$, and that $U(\sigma) = e^{\sigma K}$ belongs to the *symplectic group* $\mathrm{Sp}(2N,\mathbb{R})$, namely[11]

$$U(\sigma) = e^{\sigma K}, \qquad U(\sigma)\Omega U^{\mathsf{T}}(\sigma) = \Omega. \tag{53}$$

Satisfaction of the last condition is most transparent when expressed in terms of the algebra, *i.e.,*

$$K\Omega + \Omega K^T = 0, \tag{54}$$

which can be verified by the use of eq. (52) and the fact that $\Omega^T = -\Omega$.

The symplectic group $\mathrm{Sp}(2N,\mathbb{R})$ is the group of elements satisfying the relations (53), which amounts to linear transformations on the canonical variables $\xi^a$ that preserve the canonical commutation relations; $\mathfrak{sp}(2N,\mathbb{R})$ is the associated algebra of generators. We can express the operation of $\widehat{U}(\sigma)$ on $\xi^a$ as

$$\widehat{U}^\dagger(\sigma)\xi^a\widehat{U}(\sigma) = U(\sigma)^a{}_b\xi^b. \tag{55}$$

---

[10]Note that $\widehat{K}$ is a positive operator if and only if $k$ is a positive definite bilinear form. In this case it can be viewed as a physical Hamiltonian which is bounded from below.

[11]Note that we have dropped the hats in moving from the operators to the matrix representation, and that despite the suggestive notation, $U(\sigma)$ is *not* a unitary matrix.

$\widehat{U}$ reflects the so-called metaplectic representation of $U$ [77]. With the above in hand, the covariance matrix of $|G_\sigma\rangle$ may be computed as

$$
\begin{aligned}
G_\sigma^{a,b} &= \langle G_\sigma|(\xi^a\xi^b + \xi^b\xi^a)|G_\sigma\rangle \\
&= \langle G_0|e^{i\sigma\widehat{K}}(\xi^a\xi^b + \xi^b\xi^a)e^{-i\sigma\widehat{K}}|G_0\rangle \\
&= U(\sigma)^a{}_c\, U(\sigma)^b{}_d\, \langle G_0|(\xi^c\xi^d + \xi^d\xi^c)|G_0\rangle \\
&= U(\sigma)^a{}_c\, G_0^{c,d}\, U^\mathsf{T}(\sigma)_d{}^b\,.
\end{aligned}
\tag{56}
$$

Note that the transformation of the covariance matrix is linear, *i.e.,* each of the components of $G_\sigma$ is a linear combination of the entries in $G_0$. This allows for the very compact notation

$$
G_\sigma = U(\sigma)\,G_0\,U^\mathsf{T}(\sigma) \qquad \text{and} \qquad |G_\sigma\rangle = \widehat{U}(\sigma)|G_0\rangle = |U(\sigma)G_0 U^\mathsf{T}(\sigma)\rangle\,.
\tag{57}
$$

Of course, we can express any Gaussian state $|G\rangle$ also as a Gaussian wavefunction of the form

$$
\psi_G(q_1,\cdots,q_N) = \sqrt[4]{\det(A/\pi)}\exp\left(-\frac{1}{2}q^\alpha(A_{\alpha,\beta} + iB_{\alpha,\beta})q^\beta\right),
\tag{58}
$$

with $q := (q_1,\cdots,q_N)$, where $A$ and $B$ are real bilinear forms that can be computed from $G$ as explained in ref. [81]. However, the action of $\widehat{U}(\sigma)$ on these bilinear forms – namely $A(\sigma)$ and $B(\sigma)$ for the sequence of states $|G_\sigma\rangle$ – is much more cumbersome than the simple expression (57) above. Only if one enforces $B = 0$ with respect to a specific splitting of the classical phase space into positions $q_i$ and their conjugate momenta $p_i$ does one find that $A(\sigma)$ transforms in a simple way under the subgroup $\mathrm{GL}(2N,\mathbb{R})$. This was the case studied in ref. [23], but if we want to extend our analysis to the full symplectic group (as required for the TFD state), then it is much more convenient to label Gaussian states by their covariance matrices rather than by the parameters $A$ and $B$ in the wavefunction.

Let us consider the special case $N = 1$ in order to gain some intuition for the generators of the corresponding group $\mathrm{Sp}(2,\mathbb{R})$. In this case, eq. (48) yields only three independent generators, which we denote

$$
\widehat{W} = \frac{1}{2}(qp + pq), \qquad \widehat{V} = \frac{q^2}{\sqrt{2}}, \qquad \widehat{Z} = \frac{p^2}{\sqrt{2}}\,.
\tag{59}
$$

These close to form the algebra $\mathfrak{sp}(2,\mathbb{R})$, with commutation relations

$$
[\widehat{V},\widehat{W}] = -2i\,\widehat{V}\,, \qquad [\widehat{W},\widehat{Z}] = 2i\,\widehat{Z}\,, \qquad [\widehat{V},\widehat{Z}] = 2i\widehat{W}\,.
\tag{60}
$$

Note that these are also the generators that serve as building blocks for the operators in eq. (35). The associated matrices $k_{(a,b)}$ in eq. (48) are given by

$$
k_{(a,b)}(\widehat{W}) = \begin{pmatrix} 0 & 1 \\ 1 & 0 \end{pmatrix}, \qquad k_{(a,b)}(\widehat{V}) = \begin{pmatrix} \sqrt{2} & 0 \\ 0 & 0 \end{pmatrix}, \qquad k_{(a,b)}(\widehat{Z}) = \begin{pmatrix} 0 & 0 \\ 0 & \sqrt{2} \end{pmatrix}.
\tag{61}
$$

We can then use eq. (52) to obtain the relevant matrix generators

$$
W = \begin{pmatrix} 1 & 0 \\ 0 & -1 \end{pmatrix}, \qquad V = \begin{pmatrix} 0 & 0 \\ -\sqrt{2} & 0 \end{pmatrix}, \qquad Z = \begin{pmatrix} 0 & \sqrt{2} \\ 0 & 0 \end{pmatrix},
\tag{62}
$$

which satisfy

$$
[V,W] = 2V\,, \qquad [W,Z] = 2Z\,, \qquad [V,Z] = 2W\,.
\tag{63}
$$

Finally, exponentiating these generators yields the group elements that will serve as the elementary gates used in the construction of our quantum circuits below:

$$U_W = e^{\epsilon W} = \begin{pmatrix} e^\epsilon & 0 \\ 0 & e^{-\epsilon} \end{pmatrix}, \quad U_V = e^{\epsilon V} = \begin{pmatrix} 1 & 0 \\ -\sqrt{2}\epsilon & 1 \end{pmatrix}, \quad U_Z = e^{\epsilon Z} = \begin{pmatrix} 1 & \sqrt{2}\epsilon \\ 0 & 1 \end{pmatrix}, \quad (64)$$

where $\epsilon$ is a real parameter with $|\epsilon| \ll 1$. To see how these $Sp(2, \mathbb{R})$ gates modify the state, we evaluate the change in the covariance matrix effected by these gates via eq. (57). Suppose we start with the generic pure Gaussian state $|\psi\rangle$ given in eq. (45). Then, denoting the state after acting with the $U_Z$ gate by $\tilde{\psi}_Z$, i.e., $|\tilde{\psi}_Z\rangle = e^{-i\epsilon\widehat{Z}}|\psi\rangle$, one finds[12]

$$G^{1,1} = \langle\tilde{\psi}_Z|2q^2|\tilde{\psi}_Z\rangle = \frac{1 - 2\sqrt{2}\epsilon\, b + 2\epsilon^2(a^2 + b^2)}{a},$$

$$G^{1,2} = G^{2,1} = \langle\tilde{\psi}_Z|(qp + pq)|\tilde{\psi}_Z\rangle = \frac{-b + \sqrt{2}\epsilon(a^2 + b^2)}{a}, \quad (65)$$

$$G^{2,2} = \langle\tilde{\psi}_Z|2p^2|\tilde{\psi}_Z\rangle = \frac{a^2 + b^2}{a}.$$

The parameters $\tilde{a}, \tilde{b}$ of the transformed wavefunction $\tilde{\psi}_Z$ are therefore

$$U_Z : \qquad \tilde{a} = \frac{a}{1 - 2\sqrt{2}\epsilon\, b + 2\epsilon^2(a^2 + b^2)}, \qquad \tilde{b} = b\,\frac{1 - \sqrt{2}\epsilon(a^2 + b^2)/b}{1 - 2\sqrt{2}\epsilon\, b + 2\epsilon^2(a^2 + b^2)}. \quad (66)$$

Similarly, the changes effected by the other two gates are

$$
\begin{aligned}
U_W : & \qquad \tilde{a} = e^{-2\epsilon}\, a, & \tilde{b} = e^{-2\epsilon}\, b, \\
U_V : & \qquad \tilde{a} = a, & \tilde{b} = b + \sqrt{2}\epsilon.
\end{aligned}
\qquad (67)
$$

These last two expressions are relatively simple; and indeed, in $U_W$ we recognize the action of a scaling gate as in ref. [23].[13] In contrast, one sees from (66) that, while the state remains Gaussian under the action of $U_Z$, this gate produces a nonlinear change in the parameters of the wavefunction,[14] in contrast to the simple transformation of the covariance matrix in eq. (56). At a practical level, the fact that this action can be deduced straightforwardly from the change in the two-point function is the main advantage of working with the covariance matrix.

Now, returning to the quantum circuits introduced in eq. (5), we can use the covariance matrix language to replace the circuits $\widehat{U}(\sigma)$ by their matrix representation $U(\sigma)$. In particular, we can ask how a state changes under the evolution $\widehat{U}(\sigma) = \bar{\mathcal{P}}e^{-i\int_0^\sigma \widehat{K}(s)\mathrm{d}s}$ of a varying quadratic operator

$$\widehat{K}(\sigma) = \frac{1}{2}\,\xi\, k(\sigma)\,\xi^{\mathsf{T}}. \quad (68)$$

We can use the same arguments as before to find

$$\widehat{U}^\dagger(\sigma)\,\xi^a\,\widehat{U}(\sigma) = U(\sigma)^a{}_b\,\xi^b, \quad (69)$$

---

[12] Alternatively, one may use the Baker-Campbell-Hausdorff formula directly to obtain the same relations.

[13] This was also referred to as a squeezing gate in ref. [51], since it shrinks the variance of the position operator $q$ at the expense of increasing the variance of the momentum operator $p$, while keeping the expectation value of the cross-product fixed.

[14] In fact, the action in eq. (66) is somewhat akin to a special conformal transformation in conformally-invariant systems, insofar as it can be obtained from an inversion $a' = a/(a^2 + b^2)$ and $b' = b/(a^2 + b^2)$, followed by a translation in the $b$-direction (i.e., applying $U_V$), $a'' = a''$ and $b'' = b' - 2\epsilon$, and finally by another inversion $a''' = a''/(a''^2 + b''^2)$ and $b''' = b''/(a''^2 + b''^2)$.

where the matrix representation of the path $U(\sigma)$ satisfies the equation

$$U'(\sigma) = [K(\sigma), U(\sigma)] \quad \text{with} \quad K^a{}_b(\sigma) = \Omega^{a,c} k_{c,b}(\sigma), \tag{70}$$

whose solution is given by the path-ordered exponential

$$U(\sigma) = \tilde{\mathcal{P}} e^{\int_0^\sigma K(s)ds} \tag{71}$$

that acts on the covariance matrix (or the state) as in eq. (57).

### 3.3 Covariance matrix for the time-dependent TFD state

In this section, we demonstrate how the above machinery may be employed to evaluate the covariance matrix for the time-dependent thermofield double state. We saw in eqs. (35)-(36) that this state can be written as a tensor product decomposition in the diagonal basis. Hence we focus our attention on only one of these Gaussian factors, *e.g.,* the state formed by acting with $\widehat{\mathcal{O}}_+(t)$ in eq. (36). The Gaussian state formed by acting with $\widehat{\mathcal{O}}_-(t)$ is obtained simply by replacing $\alpha \mapsto -\alpha$.

The most straightforward way to obtain the covariance matrix of the time-dependent TFD state is by evolving the covariance matrix of the TFD state at the $t = 0$ in eq. (47) forward in time. More concretely, we would like to apply to the latter state the unitary

$$\widehat{U}_+(t) = e^{-i\frac{t}{2}H_+}, \qquad H_+ = \frac{1}{2M} P_+^2 + \frac{1}{2} M \omega^2 Q_+^2 = \frac{p_+^2}{2} \frac{\omega}{\lambda} + \lambda \omega \frac{q_+^2}{2}, \tag{72}$$

where we have used eqs. (1), (25), (37), and (40), such that the state whose covariance matrix we wish to obtain is given by

$$|\text{TFD}(t)\rangle_+ = \widehat{U}_+(t)|\text{TFD}(0)\rangle_+. \tag{73}$$

This problem falls precisely within the formalism of the last subsection, where the evolution is given by the operator

$$\widehat{K} = \frac{1}{2} H_+, \qquad k_{(a,b)} = \frac{\omega}{2} \begin{pmatrix} \lambda & 0 \\ 0 & \frac{1}{\lambda} \end{pmatrix}. \tag{74}$$

Translating this to the level of matrix operators, (52) and (53) become

$$K = \frac{\omega}{2} \begin{pmatrix} 0 & \frac{1}{\lambda} \\ -\lambda & 0 \end{pmatrix}, \qquad U(t) = e^{tK} = \begin{pmatrix} \cos\left(\frac{t\omega}{2}\right) & \sin\left(\frac{t\omega}{2}\right)/\lambda \\ -\lambda \sin\left(\frac{t\omega}{2}\right) & \cos\left(\frac{t\omega}{2}\right) \end{pmatrix}, \tag{75}$$

whereupon the action of $U(t)$ on the covariance matrix (57) is found to be

$$\begin{aligned}
G^+_{\text{TFD}}(t) = U(t) G^+_{\text{TFD}} U^{\mathsf{T}}(t) &= \begin{pmatrix} \frac{1}{\lambda}(\cosh 2\alpha + \sinh 2\alpha \cos \omega t) & -\sinh 2\alpha \sin \omega t \\ -\sinh 2\alpha \sin \omega t & \lambda(\cosh 2\alpha - \sinh 2\alpha \cos \omega t) \end{pmatrix} \\
&= \cosh^2 \alpha \begin{pmatrix} \frac{1}{\lambda}\left(1 + 2\tanh\alpha \cos\omega t + \tanh^2\alpha\right) & -2\tanh\alpha \sin\omega t \\ -2\tanh\alpha \sin\omega t & \lambda\left(1 - 2\tanh\alpha \cos\omega t + \tanh^2\alpha\right) \end{pmatrix},
\end{aligned} \tag{76}$$

where $G^+_{\text{TFD}}$ was given in (47). In the second line, we have presented the result in a way that is easily related to the physical variables via (19), *i.e.,* $\tanh\alpha = \exp(-\beta\omega/2)$ and $\cosh\alpha = (1 - e^{-\beta\omega})^{-1/2}$. The time-dependence of this expression is of course periodic. Note also that we recover $G_{\text{TFD}}$ from eq. (47) upon setting $t = 0$, as expected.

Alternatively, one may obtain this covariance matrix by using the operation of $\widehat{\mathcal{O}}_+(t)$ on the vacuum state according to eqs. (35)-(36), *i.e.,* $|\psi(t)\rangle := \exp\left[-i\alpha\widehat{\mathcal{O}}_+(t)\right]|0\rangle$, where

$$\widehat{\mathcal{O}}_+(t) = \frac{1}{2} \cos\omega t\, (q_+ p_+ + p_+ q_+) + \frac{1}{2} \sin\omega t \left(\lambda q_+^2 - \frac{1}{\lambda} p_+^2\right), \tag{77}$$

and where we have rewritten the generator $\widehat{O}_+(t)$ in eq. (35) in terms of the rescaled variables $q, p$ defined in eq. (37) and the parameter $\lambda$ which was defined in eq. (40). To simplify the notation, we drop the $+$ subscript in the following. The relevant matrix operator that obtains the TFD at time $t$ from the vacuum state, which we denote $U_{\text{TFD}}(t)$, is again obtained according to eqs. (52), (53), and (57):

$$
\begin{aligned}
U_{\text{TFD}}(t) &= \exp\begin{pmatrix} \alpha \cos(t\omega) & -\frac{\alpha}{\lambda}\sin(t\omega) \\ -\alpha\lambda\sin(t\omega) & -\alpha\cos(t\omega) \end{pmatrix} \\
&= \begin{pmatrix} \cosh\alpha + \sinh\alpha\,\cos\omega t & -\frac{1}{\lambda}\sinh\alpha\,\sin\omega t \\ -\lambda\sinh\alpha\,\sin\omega t & \cosh\alpha - \sinh\alpha\,\cos\omega t \end{pmatrix}.
\end{aligned}
\tag{78}
$$

We can then obtain the covariance matrix of the time-dependent TFD state by acting on the vacuum covariance matrix $G_0$ in eq. (47) with $U_{\text{TFD}}(t)$ according to eq. (57), *i.e.,*

$$
G_{\text{T}}(t) = U_{\text{TFD}}(t)\,G_0\,U_{\text{TFD}}(t)^{\intercal},
\tag{79}
$$

which of course reproduces eq. (76) above.

To close this section, let us also give the transformation $U_{\text{vac}}$ that brings the reference state $G_{\text{R}}$ to the vacuum state $G_0$ (also given in eq. (47)), since we will need this in the following sections. This transformation was studied in ref. [23], and is related to the following quantum operator (again focusing on the $+$ mode and dropping the subscripts):

$$
|\psi_0\rangle = e^{-\frac{i}{2}\alpha_{\text{R}}(qp+pq)}|\psi_{\text{R}}\rangle, \qquad \text{with} \qquad \alpha_{\text{R}} = -\frac{1}{2}\log(\lambda/\lambda_{\text{R}}).
\tag{80}
$$

Following the same steps as above, one finds

$$
U_{\text{vac}} = \exp[\alpha_{\text{R}}W] = \begin{pmatrix} \sqrt{\frac{\lambda_{\text{R}}}{\lambda}} & 0 \\ 0 & \sqrt{\frac{\lambda}{\lambda_{\text{R}}}} \end{pmatrix},
\tag{81}
$$

and one can readily verify that

$$
G_0 = U_{\text{vac}}\,G_{\text{R}}\,U_{\text{vac}}^{\intercal}.
\tag{82}
$$

### 3.4 Relative covariance matrices and stabilizer group

The stabilizer subgroup $\text{Sta}_G \subset \text{Sp}(2N, \mathbb{R})$ associated to a Gaussian state $G$ is defined as

$$
\text{Sta}_G = \{U \in \text{Sp}(2N, \mathbb{R}) \,|\, UGU^{\intercal} = G\}.
\tag{83}
$$

The importance of the stabilizer group lies in the fact that it allows one to relate different unitaries that map a given reference state to the same target state. Explicitly, if we have a matrix $U$ satisfying

$$
G_{\text{T}} = UG_{\text{R}}U^{\intercal},
\tag{84}
$$

then for any $U_{\text{R}} \in \text{Sta}_{G_{\text{R}}}$ the operator $UU_{\text{R}}$ will also obtain the same state $G_{\text{T}}$. Thus when minimizing over circuits to compute the complexity, we must also minimize over this family of transformations. As a group, we have $\text{Sta}_G \simeq U(N)$, but different choices of $G$ will lead to different embeddings of this subgroup within $\text{Sp}(2N, \mathbb{R})$.[15]

---

[15]In fact, up to a deformation, the elements of the group are nothing but the elements of the passive subgroup $\text{Sp}(2N, \mathbb{R}) \cap O(2N)$. This, in turn, is isomorphic to $U(N)$, as this subgroup reflects unitary transformations of vectors of bosonic operators.

For the case $N = 1$, we consider the stabilizer group which leaves invariant the reference state $G_R$ in eq. (47). This is an $SO(2) \simeq U(1)$ subgroup of $Sp(2, \mathbb{R})$ of the form

$$U_\phi = e^{\phi H_R} = \begin{pmatrix} \cos(\phi) & -\frac{\sin(\phi)}{\lambda_R} \\ \lambda_R \sin(\phi) & \cos(\phi) \end{pmatrix} \quad \text{with} \quad H_R = -\frac{1}{2}\left(\lambda_R V + \frac{Z}{\lambda_R}\right) = \phi \begin{pmatrix} 0 & -\frac{1}{\lambda_R} \\ \lambda_R & 0 \end{pmatrix}. \tag{85}$$

In this case, if $U_T$ achieves the desired transformation between the reference and target states, then we must minimize over the family of circuits given by the transformation $U_T U_\phi$ (subject to the same boundary conditions) for all values of the rotation angle $\phi$.

If we are interested in the relation of two Gaussian states $G$ and $\widetilde{G}$, then we can express the relation between them in a basis independent way in terms of the *relative covariance matrix*

$$\Delta^a{}_b = \widetilde{G}^{a,c} g_{c,b}, \tag{86}$$

where $g$ is the inverse of $G$, such that $G^{a,c} g_{c,b} = \delta^a{}_b$. Any quantity which is invariant under the $Sp(2N, \mathbb{R})$ group is necessarily a pure function of the spectrum of $\Delta$.[16] To show this, we may first use the full group $Sp(2N, \mathbb{R})$ to choose a basis, such that the matrix representation of $G$ becomes the identity, *i.e.*, $G = \mathbb{1}$. We can then diagonalize the covariance matrix $\widetilde{G}$ using only transformations within the stabilizer subgroup $Sta_G$, which provides the freedom to change basis without affecting $G$. This is equivalent to finding the spectrum of the relative covariance matrix. An example of such an invariant function is the inner product $|\langle G|\widetilde{G}\rangle|$, which can be computed as [81]

$$|\langle G|\widetilde{G}\rangle|^2 = \det \frac{\sqrt{2}\Delta^{1/4}}{\sqrt{\mathbb{1} + \Delta}}. \tag{87}$$

In the case of complexity, we can make a choice for the cost function that is defined in terms of the reference state $G_R$.[17] As this implies that the complexity only depends on $G_R$ and $G_T$, we will find a simple formula for the $F_2$ complexity in terms of $\Delta$, namely

$$\mathcal{C}_2(G_R, G_T) = \frac{1}{2\sqrt{2}} \sqrt{\text{Tr}[(\log \Delta)^2]}, \tag{88}$$

or for the $\kappa = 2$ complexity, we have $\mathcal{C}_{\kappa=2}(G_R, G_T) = [\mathcal{C}_2(G_R, G_T)]^2 = \frac{1}{8} \text{Tr}[(\log \Delta)^2]$. Both of these expressions are derived in appendix F – see eqs. (321) and (322). However, we will also consider other cost functions that explicitly depend on a choice of basis, in which case knowledge of $\Delta$ does not suffice to compute the complexity.

To make the above more concrete, let us write the relative covariance matrix between the reference state $G_R$ and the time-independent TFD state $G_{TFD}^+$ given in eq. (47):

$$\Delta = G_{TFD}^+ G_R^{-1} = \begin{pmatrix} \frac{\lambda_R}{\lambda} e^{2\alpha} & 0 \\ 0 & \frac{\lambda}{\lambda_R} e^{-2\alpha} \end{pmatrix}. \tag{89}$$

Hence in this case, any cost function (*i.e.*, definition of complexity) which is invariant under the $Sp(2N, \mathbb{R})$ group must only depend on the combination $e^{2(\alpha_R + \alpha)} = \frac{\lambda_R}{\lambda} e^{2\alpha}$. For example, the inner product is

$$|\langle G_R|G_{TFD}^+\rangle| = \sqrt[4]{\frac{\lambda_R \lambda e^{-2\alpha}}{\pi^2}} \int dq_+ \exp\left(-\frac{\lambda_R + \lambda e^{-2\alpha}}{2} q_+^2\right) = \frac{\sqrt{2}\left(\frac{\lambda}{\lambda_R} e^{-2\alpha}\right)^{1/4}}{\sqrt{1 + \frac{\lambda}{\lambda_R} e^{-2\alpha}}}, \tag{90}$$

---

[16]The eigenvalues of $\Delta$ come in pairs where each eigenvalue is accompanied by its inverse. We refer to the set containing the first of each pair as the spectrum.

[17]This is not actually the choice that we make in our complexity calculations, see discussion in section 4.1. Generally, we distinguish the reference state (47) from the state $G = \mathbb{1}$ appearing in the definition of the cost functions, *e.g.*, see eq. (108). Therefore, the complexity expression in eq. (88) only applies for the choice $\lambda_R = 1$, as we make clear in appendix F.

which indeed depends only on the spectrum of $\Delta$.

## 3.5 Generators of $\mathrm{Sp}(2N, \mathbb{R})$

In examining the complexity, we replace the circuits (5) with their matrix-valued counterparts (71), but we will still need to decompose the exponential in terms of some basis in order to evaluate the appropriate cost function, *e.g.,* in eq. (8) or (9). Hence we will find it useful to have explicit expressions for the generators of the group $\mathrm{Sp}(2N, \mathbb{R})$, in particular for $N = 1$ and $N = 2$. Accordingly, here we give a list of the generators of $\mathrm{Sp}(2N, \mathbb{R})$ for general values of $N$. We may start with the quantum generators, $i, j, k \in (1, \dots, N)$

$$
\begin{aligned}
\widehat{W}_{i,j} &= \frac{1}{2}\left(Q_i P_j + P_j Q_i\right) = \frac{1}{2}\left(q_i p_j + p_j q_i\right), \\
\widehat{V}_{i,j} &= \begin{cases} \frac{\omega_g^2}{\sqrt{2}} Q_i^2 = \frac{1}{\sqrt{2}} q_i^2 & i = j \\ \omega_g^2 Q_i Q_j = q_i q_j & i \neq j \end{cases}, \\
\widehat{Z}_{i,j} &= \begin{cases} \frac{1}{\sqrt{2}\omega_g^2} P_i^2 = \frac{1}{\sqrt{2}} p_i^2 & i = j \\ \frac{1}{\omega_g^2} P_i P_j = p_i p_j & i \neq j \end{cases}.
\end{aligned}
\tag{91}
$$

We note that the number of $\widehat{W}_{i,j}$, $\widehat{V}_{i,j}$, and $\widehat{Z}_{i,j}$ operators is $N^2$, $\frac{1}{2}(N^2 + N)$, and $\frac{1}{2}(N^2 + N)$, respectively, for a total of $N(2N + 1)$ generators. In these expressions, $\omega_g$ is the gate scale introduced in eq. (37) in order to render the generators $\widehat{V}_{i,j}$ and $\widehat{Z}_{i,j}$ dimensionless. This scale does not enter $\widehat{W}_{i,j}$, since it is invariant under a rescaling of $Q_i$ and $P_j$. The generators $\widehat{W}_{i,j}$ span the subalgebra $\mathfrak{gl}(N, \mathbb{R})$, which was analyzed in ref. [23], and hence the gate scale did not enter into the complexity calculations considered therein. However, as shown above, the preparation of the time-evolve TFD states will also involve the $\widehat{V}_{i,j}$ and $\widehat{Z}_{i,j}$ generators, and so *a priori* the complexity of these states will depend on the choice of the gate scale.

The above generators have been chosen to be orthonormal according to the Frobenius inner product

$$
\langle K, \widetilde{K} \rangle = \frac{1}{2}\mathrm{tr}\left(KG\widetilde{K}^{\mathsf{T}}g\right),
\tag{92}
$$

where $g_{a,b}$ is the inverse of $G^{a,b}$ as above. We will choose $G = \mathbb{1}$ in the basis spanned by our dimensionless variables $(q_i, p_j)$. In particular, this normalization is responsible for the extra factors of $1/\sqrt{2}$ appearing in eq. (91) for the diagonal generators. This inner product will play an important role in what follows.

We can translate these generators into the relevant matrix representation by using eq. (48) and eq. (52), which leads to

$$
\begin{aligned}
k_{(a,b)}\left(\widehat{W}_{i,j}\right) &= \left(\delta_{a,i}\delta_{b,j+N} + \delta_{a,j+N}\delta_{b,i}\right), \\
k_{(a,b)}\left(\widehat{V}_{i,j}\right) &= \begin{cases} \sqrt{2}\,\delta_{a,i}\delta_{b,i} & i = j \\ \delta_{a,i}\delta_{b,j} + \delta_{a,j}\delta_{b,i} & i \neq j \end{cases}, \\
k_{(a,b)}\left(\widehat{Z}_{i,j}\right) &= \begin{cases} \sqrt{2}\,\delta_{a,i+N}\delta_{b,i+N} & i = j \\ \delta_{a,i+N}\delta_{b,j+N} + \delta_{a,j+N}\delta_{b,i+N} & i \neq j \end{cases}.
\end{aligned}
\tag{93}
$$

Following eq. (52), the associated matrix generators are obtained by multiplying with the symplectic form

$$
\Omega^{a,b} = \sum_{k=1}^{N}\left(\delta^{a,k}\delta^{b,k+N} - \delta^{a,k+N}\delta^{b,k}\right),
\tag{94}
$$

which yields

$$
\begin{aligned}
(W_{i,j})^a{}_b &= \delta^{a,j}\delta_{b,i} - \delta^{a,i+N}\delta_{b,j+N}\,, \\
(V_{i,j})^a{}_b &= \begin{cases} -\sqrt{2}\delta^{a,i+N}\delta_{b,i} & i = j \\ -\delta^{a,i+N}\delta_{b,j} - \delta^{a,j+N}\delta_{b,i} & i \neq j \end{cases}, \\
(Z_{i,j})^a{}_b &= \begin{cases} \sqrt{2}\delta^{a,i}\delta_{b,i+N} & i = j \\ \delta^{a,i}\delta_{b,j+N} + \delta^{a,j}\delta_{b,i+N} & i \neq j \end{cases}.
\end{aligned}
\tag{95}
$$

For $N = 1$, these expressions reproduce the generators $W$, $V$, $Z$ in eq. (62). For the purposes of this paper, we will mainly use the generators of $\mathrm{Sp}(4,\mathbb{R})$ which we list explicitly in appendix C.

When computing the complexity of a particular circuit described by eq. (71), we may need to expand a given generator $K(s)$ with respect to different bases of generators, say $K_I$ and $\widetilde{K}_I$. Provided that these bases are both orthonormal with respect to the Frobenius inner product (92), *i.e.*,

$$
\langle K_I, K_J \rangle = \langle \tilde{K}_I, \tilde{K}_J \rangle = \delta_{I,J}\,,
\tag{96}
$$

we can accomplish this by computing

$$
Y^I(s) = \langle K_I | K(s) \rangle \quad \text{and} \quad \widetilde{Y}^I(s) = \langle \widetilde{K}_I | K(s) \rangle\,.
\tag{97}
$$

In the following, we will work with two bases that have already appeared in section 2.2. In particular, we have the $L, R$ basis referring to the two copies of the physical degrees of freedom entangled in the TFD state, and the diagonal or $\pm$ basis in which the TFD state factorizes. As indicated in eq. (22), these two bases are related by a simple rotation[18]

$$
R_2 = \frac{1}{\sqrt{2}} \begin{pmatrix} 1 & 1 \\ 1 & -1 \end{pmatrix} \implies \begin{pmatrix} q_+ \\ q_- \end{pmatrix} = R \begin{pmatrix} q_L \\ q_R \end{pmatrix}\,,
\tag{98}
$$

but it will be useful to systematize the transformation for general expressions. For example, eq. (98) extends to an analogous equation for the momenta, and so the transformation on the full phase space reads

$$
\xi^a = [R_4]^a{}_b\,\xi^b, \quad \text{where} \quad R_4 = \mathbb{1}_2 \otimes R_2 = \frac{1}{\sqrt{2}} \begin{pmatrix} 1 & 1 & 0 & 0 \\ 1 & -1 & 0 & 0 \\ 0 & 0 & 1 & 1 \\ 0 & 0 & 1 & -1 \end{pmatrix}.
\tag{99}
$$

The two-point function (43) then transforms as

$$
\widetilde{G} = R_4\, G\, R_4^{\mathsf{T}}.
\tag{100}
$$

It is now straightforward to see that if we have a circuit acting in the diagonal basis as $\widetilde{G}_{\mathrm{T}} = \widetilde{U}\,\widetilde{G}_{\mathrm{R}}\,\widetilde{U}^{\mathsf{T}}$, then the same circuit in the physical basis is

$$
G_{\mathrm{T}} = U\, G_{\mathrm{R}}\, U^{\mathsf{T}}, \qquad \text{with} \quad U = R_4^{\mathsf{T}}\,\widetilde{U}\, R_4\,.
\tag{101}
$$

Furthermore, the matrix generators $K_I$ in eq. (95) are also simply transformed to[19]

$$
K_I = R_4^{\mathsf{T}}\,\widetilde{K}_I\, R_4\,.
\tag{102}
$$

---

[18]We use the subscript 2 here to indicate that this is a 2×2 matrix and distinguish this rotation matrix from $R_4$ below. Furthermore, while as a numerical matrix $R_2$ is symmetric, we nonetheless distinguish $R_2$ and $R_2^{\mathsf{T}}$ in the following to emphasize the fact that $R_2$ provides a mapping from the physical to the diagonal coordinates, while $R_2^{-1} = R_2^{\mathsf{T}}$ provides the inverse mapping. In other words, the columns of $R_2$ are labeled $L, R$ while the rows are labeled $+, -$, and vice versa for $R^{\mathsf{T}}$.

[19]The transformation (99) is special since it is orthogonal. Generally, such coordinate transformations on the phase space have a similar effect, except that eqs. (101) and (102) are replaced with $U = R^{-1}\widetilde{U}R$ and $K = R^{-1}\widetilde{K}R$, respectively.

For example we may use these transformation rules to transform the covariance matrix from the $(q_+, q_-, p_+, p_-)$ basis to the $(q_L, q_R, p_L, p_R)$ basis. We start from the covariance matrix in the $(q_+, q_-, p_+, p_-)$ basis

$$G(t) = G_{\text{TFD}}^+(t) \oplus G_{\text{TFD}}^-(t), \tag{103}$$

where the direct sum inputs the + and minus components in a 4 by 4 combined matrix and where the $G_{\text{TFD}}^+(t)$ was defined in eq. (76) and $G_{\text{TFD}}^-(t)$ is a similar matrix with $\alpha \to -\alpha$. The time dependent covariance matrix with respect to the $(q_L, q_R, p_L, p_R)$ basis is given according to the rotation (100) or explicitly

$$G(t) = \begin{pmatrix} \frac{\cosh(2\alpha)}{\lambda} & \frac{\cos(t\omega)\sinh(2\alpha)}{\lambda} & 0 & -\sin(t\omega)\sinh(2\alpha) \\ \frac{\cos(t\omega)\sinh(2\alpha)}{\lambda} & \frac{\cosh(2\alpha)}{\lambda} & -\sin(t\omega)\sinh(2\alpha) & 0 \\ 0 & -\sin(t\omega)\sinh(2\alpha) & \lambda\cosh(2\alpha) & -\lambda\cos(t\omega)\sinh(2\alpha) \\ -\sin(t\omega)\sinh(2\alpha) & 0 & -\lambda\cos(t\omega)\sinh(2\alpha) & \lambda\cosh(2\alpha) \end{pmatrix}. \tag{104}$$

This expression will come handy later on in section 6.

## 4 Complexity of TFD states

In this section, we apply the tools developed above to study the complexity of the TFD state comprised of two harmonic oscillators. We will later use the results of this section to evaluate the complexity of the TFD state of a free scalar field theory in section 5.

### 4.1 Circuit geometry for the TFD state

In subsection 2.1, we introduced the definition of complexity for general groups. In the present subsection, we specialize to the group $\mathcal{G} = \text{Sp}(2N, \mathbb{R})$, and the associated algebra $\mathfrak{g} = \mathfrak{sp}(2N, \mathbb{R})$ of matrices acting on the covariance matrix. We will explain how the general definitions presented in section 2.1 can be applied in this specific case. In general, our matrix-valued circuits will take the form given in eq. (71), *i.e.*,

$$U(\sigma) = \bar{\mathcal{P}} \exp \int_0^\sigma \mathrm{d}s\, K(s), \qquad K(\sigma) = \partial_\sigma U\, U^{-1} = Y^I(\sigma) K_I, \tag{105}$$

where $K_I$ are the generators of the algebra $\mathfrak{sp}(2N, \mathbb{R})$ in a given basis. We will start with the covariance matrix associated with our reference state $G_{\text{R}}$, and follow a path in the space of Gaussian states represented by the covariance matrices

$$G(\sigma) = U(\sigma)\, G_{\text{R}}\, U(\sigma)^T, \qquad \sigma \in [0, 1]. \tag{106}$$

We will be interested in trajectories that end on a given Gaussian target state $G_{\text{T}}$, *i.e.*, our circuits satisfy the boundary conditions

$$U(0) = \mathbb{1}, \qquad G_{\text{T}} = U(1)\, G_{\text{R}}\, U(1)^T. \tag{107}$$

The length of a given circuit will be given by integrating certain cost functions along the path, as we have discussed in subsection 2.1. We introduced some examples with the $F_1$, $F_2$, and $D_\kappa$ cost functions in eqs. (8) and (9) above. Note however that we still have an enormous amount of freedom, since different choices of basis vectors $K_I$ will in general lead to different results for the total cost [23]. The $F_2$ cost function, as well as $D_{\kappa=2}$, is invariant when the two

bases are related by an orthogonal transformation.[20] However, even these cost functions are implicitly defined in terms of a particular reference state [82].[21]

We can also view the $F_2$ cost function as arising from a natural construction using a positive-definite matrix $G^{a,b}$ (for instance, by taking it from the covariance matrix $G$ of a Gaussian state), namely

$$F_2(K) = \sqrt{\frac{\mathrm{Tr}(KGK^\intercal g)}{2}} = \sqrt{\frac{K^a{}_b\, G^{b,c}\, (K^\intercal)_c{}^d\, g_{d,a}}{2}}\,, \qquad (108)$$

where again $g_{a,b}$ denotes the inverse of $G^{a,b}$, i.e., $G^{a,c} g_{c,b} = \delta^a{}_b$. This cost function coincides with the norm induced by the inner product (92). Extending this inner product to the full group turns $\mathrm{Sp}(2N,\mathbb{R})$ into a Riemannian manifold, whose metric at the point $U \in \mathrm{Sp}(2N,\mathbb{R})$ can be computed as

$$\mathrm{d}s^2 = \frac{1}{2}\mathrm{Tr}\big(\mathrm{d}U\,U^{-1}G(\mathrm{d}U\,U^{-1})^\intercal g\big) = \frac{1}{2}\mathrm{Tr}(K(\sigma)\,G\,K(\sigma)^\intercal g)\,\mathrm{d}\sigma^2\,. \qquad (109)$$

If we choose $G = \mathbb{1}$, this metric reproduces that given in eq. (8). This is the choice that we will make in the following. Recall from eq. (97) that we can use the inner product to extract the coefficients $Y_I$ in a given generator $K = Y^I K_I$ with respect to an orthonormal basis $K_I$ by simply evaluating

$$Y^I = \langle K, K_I\rangle = \frac{1}{2}\mathrm{Tr}\big(KGK_I^\intercal g\big)\,. \qquad (110)$$

If we are interested in the circuit complexity defined with respect to a given reference state $G_{\mathrm{R}}$, then a great simplification occurs when the matrix $G$ used to define the geometry above and the covariance matrix $G_{\mathrm{R}}$ of the reference state coincide. This is the case when we choose $G = \mathbb{1}$, and the gate scale $\omega_g$ is chosen to be equal to the characteristic scale of the reference state $\sqrt{M\mu}$, equivalently when setting $\lambda_{\mathrm{R}} = 1$, cf. eq. (40). In this case the covariance matrix of the reference state is simply the identity, and is hence equal to the matrix $G$ used in defining the geometry above. For the cost function $F_1$, we also have considerable freedom in choosing the basis of generators $K_I$. We will impose that our generators be orthonormal under the inner product inducing the $F_2$ cost function. Even then we retain quite a bit of freedom, e.g., the rotation between the $LR$ basis and the $\pm$ basis in the group $\mathrm{Sp}(4,\mathbb{R})$. This change of basis does not affect the $F_2$ cost function, but it does affect $F_1$.

Let us now consider the case of a single degree of freedom. In this case, we have the group $\mathrm{Sp}(2,\mathbb{R}) = \mathrm{SL}(2,\mathbb{R})$, whose algebra is given by the traceless matrices $K_I \in \{W, V, Z\}$ given in eq. (62), where we have chosen the coordinates $\xi = (q, p)$ (which will later represent the conjugate pairs $(q_\pm, p_\pm)$ that mix the left and right sides of the TFD). With respect to these coordinates, our spatially unentangled reference state $G_{\mathrm{R}}$ is given in eq. (47). To obtain the complexity of a given target state $G_{\mathrm{T}}$, we will then study geodesics in the geometry (109) which satisfy the boundary conditions (106). More precisely, we will have to minimize over the family of geodesics that end at $U(\sigma = 1) = U_{\mathrm{T}}U_\phi$, cf. eq. (85), all of which transform the reference state to the same target state.

## 4.2 Minimal geodesics in $\mathrm{Sp}(2,\mathbb{R})$ with Riemannian metric

In this section, we focus again on the case of a single degree of freedom and explain how the metric distance of the last subsection can be applied to this specific case. We consider the

---

[20]For instance, the transformation between the $q_{L,R}$ basis and the diagonal basis $q_\pm$ in eq. (22) is such an orthogonal transformation, and hence the $F_2$ and $D_{\kappa=2}$ cost functions will be invariant under this change.

[21]This state-dependence does not occur for fermions [56].

symplectic group $\mathrm{Sp}(2, \mathbb{R})$ which is isomorphic to $\mathbb{R}^2 \times S^1$. A general element $U \in \mathrm{Sp}(2, \mathbb{R})$ can be parameterized by three coordinates $(\rho, \theta, \tau)$ as

$$U(\rho, \theta, \tau) = \begin{pmatrix} \cos\tau\cosh\rho - \sin\theta\sinh\rho & -\sin\tau\cosh\rho + \cos\theta\sinh\rho \\ \sin\tau\cosh\rho + \cos\theta\sinh\rho & \cos\tau\cosh\rho + \sin\theta\sinh\rho \end{pmatrix}, \tag{111}$$

cf. ref. [23], in which these coordinates parameterized the isomorphic group $\mathrm{SL}(2, \mathbb{R})$. In particular, $(\rho, \theta)$ serve as polar coordinates in the plane $\mathbb{R}^2$, while $\tau \in [-\pi, \pi)$ is periodic and parameterizes the $S^1$. As this parametrization appeared in ref. [23], we can import much of the technology developed therein to our current problem.

For concreteness, we will consider the cost function to be either $F_2$ from eq. (8) or $D_{\kappa=2}$ from eq. (9). In either case, the cost function is associated to the Riemannian metric from eq. (109) which, translated to our coordinates $(\rho, \theta, \tau)$, becomes

$$\mathrm{d}s^2 = \mathrm{d}\rho^2 + \cosh(2\rho)\cosh^2\rho\,\mathrm{d}\tau^2 + \cosh(2\rho)\sinh^2\rho\,\mathrm{d}\theta^2 - \sinh^2(2\rho)\,\mathrm{d}\tau\,\mathrm{d}\theta. \tag{112}$$

The relevant geodesics for this metric were already worked out in ref. [23]. In particular, if we denote the geometric coordinates $\mathbf{x}(\sigma) = (\rho(\sigma), \theta(\sigma), \tau(\sigma))$, then the initial condition $U(\sigma=0) = \mathbb{1}$ in eq. (107) fixes

$$\mathbf{x}(\sigma=0) = (0, \theta_0, 0), \tag{113}$$

cf. (111), where $\theta_0 := \theta(\sigma=0)$. The freedom in specifying the initial angle $\theta_0$ is simply the freedom to specify the initial direction in which the geodesic moves away from the origin. Now, denote the coordinates at the endpoint of the geodesic by

$$\mathbf{x}(\sigma=1) = (\rho_1, \theta_1, \tau_1). \tag{114}$$

These will be fixed in terms of the physical quantities using eq. (107) when we consider specific cases below. Of course, there will still be some residual freedom from the stabilizer group of the reference state as discussed in subsection 3.4, in particular around eq. (85). The geodesics with these general boundary conditions are

$$\begin{aligned}
\rho(\sigma) &= \sinh^{-1}\left( \frac{c}{\sqrt{c^2 - \Delta\theta^2}} \sinh\left( \frac{\sigma}{2}\sqrt{c^2 - \Delta\theta^2} \right) \right), \\
\theta(\sigma) &= \Delta\theta\,\sigma + \theta_0, \\
\tau(\sigma) &= \Delta\theta\,\sigma - \tan^{-1}\left( \frac{\Delta\theta}{\sqrt{c^2 - \Delta\theta^2}} \tanh\left( \frac{\sigma}{2}\sqrt{c^2 - \Delta\theta^2} \right) \right).
\end{aligned} \tag{115}$$

Eq. (114) then allows us to extract the values of $(\theta_0, \Delta\theta, c)$ in terms of the boundary condition $(\rho_1, \theta_1, \tau_1)$; in general, this inversion has to be done numerically. The geodesics in eq. (115) are affinely parameterized such that the line element is constant and equal to

$$\mathrm{d}s^2 = \frac{1}{4}\left( c^2 + \Delta\theta^2 \right)\mathrm{d}\sigma^2. \tag{116}$$

Given a reference state $G_{\mathrm{R}}$ and a target state $G_{\mathrm{T}}$, we wish to find the minimal geodesic in the group that takes us from $\mathbb{1}$ to an element of the family

$$\mathcal{F}_{\mathrm{R}\to\mathrm{T}} = \left\{ U \in \mathrm{Sp}(2, \mathbb{R}) \,\middle|\, U G_{\mathrm{R}} U^{\mathsf{T}} = G_{\mathrm{T}} \right\}. \tag{117}$$

In order to perform this minimization, we need to understand this family $\mathcal{F}_{\mathrm{R}\to\mathrm{T}}$ for different target states, so that we can use them as boundary condition for our solutions of the geodesic equation. In particular, we will be interested in the family of minimal geodesics starting from

the reference state $G_R$ in eq. (47). Recall from subsection 3.4 that the stabilizer subgroup $\mathrm{Sta}_{G_R}$ associated with the reference state $G_R$ is given in terms of $U_\phi$ in eq. (85). Given a group element $U_T$ that prepares the target state $G_T = U_T G_R U_T^\mathsf{T}$, we find

$$\mathcal{F}_{R\to T} = \{U_T U_\phi \mid -\pi \le \phi \le \pi\}. \tag{118}$$

In order to better understand the geometry of $\mathcal{F}_{R\to T}$ for different $U_T$, we can plot a selection of them in fig. 3 for various values of $\lambda_R$. Minimizing the geodesic distance over all end points $U_T(\phi) := U_T U_\phi \in \mathcal{F}_{R\to T}$ is a difficult task for general $\lambda_R$, and we will discuss it further in subsection 4.5 below.

As we have already mentioned, a significant simplification occurs for the special case of $\lambda_R = 1$, in which the reference and gate scales are equivalent, $\sqrt{M\mu} = \omega_g$. The reference-state covariance matrix becomes $G_R = \mathbb{1}$, and the transformation $U(\rho, \theta, \tau)$ takes us to the target state with covariance matrix

$$\begin{aligned}
G_T(\rho, \theta+\tau) &= U(\rho, \theta, \tau)\, G_R\, U^\mathsf{T}(\rho, \theta, \tau) \\
&= \begin{pmatrix} \cosh(2\rho) - \sin(\theta+\tau)\sinh(2\rho) & \cos(\theta+\tau)\sinh(2\rho) \\ \cos(\theta+\tau)\sinh(2\rho) & \cosh(2\rho) + \sin(\theta+\tau)\sinh(2\rho) \end{pmatrix}.
\end{aligned} \tag{119}$$

We immediately observe that this expression only depends on the two coordinates $\theta$ and $\tau$ through the combination $\theta+\tau$, leading to a one-parameter family of solutions. This is precisely the U(1) invariance of the stabilizer group of the reference state (note that in this case the matrix $U_\phi$ in eq. (85) is a simple rotation matrix). Let us define $\chi := \theta + \tau$, so that we can label the covariance matrix $G_T(\rho, \chi)$. Then the equivalence class of group elements that prepare the same state $G_T(\rho, \chi)$ is a spiral, which can be parameterized by $\tau$ with

$$U(\rho, \chi - \tau, \tau). \tag{120}$$

We illustrate this equivalence class in fig. 3.

As alluded above, obtaining the explicit parameters $(c, \Delta\theta, \theta_0)$ for a general boundary condition $(\rho_1, \theta_1, \tau_1)$ at $\sigma = 1$ is difficult, but for $\Delta\theta = 0$ the expressions simplify to

$$\rho(\sigma) = \rho_1\sigma, \quad \theta(\sigma) = \theta_0 = \theta_1, \quad \tau(\sigma) = 0. \tag{121}$$

This means that the geodesics in the plane with $\tau = 0$ are just given by straight lines. The line element associated with the trajectory is given by eq. (116), which for this specific case reads

$$ds = \rho_1 d\sigma, \qquad c = 2\rho_1. \tag{122}$$

Ref. [23] showed that in certain cases the optimal trajectory is indeed the one associated with $\Delta\theta = 0$. The analysis is based on a series of inequalities which appear in eqs. (3.39)-(3.44) therein. For completeness, we give a sketch of the derivation here: first, recall that evaluating the norm of the velocity along the geodesics using the metric (112) yields a constant, i.e., $|\partial_\sigma \mathbf{x}|^2 = k^2$. Using the geodesic solution (115) to evaluate $\partial_\sigma\theta$ and $\partial_\sigma\tau$, and averaging the resulting expression for $k^2$ over the geodesic, we can then show that

$$k^2 = \int_0^1 d\sigma\, \dot\rho^2 + \frac{\Delta\theta^2}{2}\int_0^1 d\sigma\left(1 - \frac{1}{2\cosh^2\rho}\right). \tag{123}$$

We may now use the two inequalities

$$\int_0^1 d\sigma\,(\dot\rho - \rho_1)^2 \ge 0 \implies \int_0^1 d\sigma\,\dot\rho^2 \ge \rho_1^2, \quad \text{and} \quad 1 \ge \int_0^1 d\sigma\left(1 - \frac{1}{2\cosh^2\rho}\right) \ge \frac{1}{2} \tag{124}$$

to prove the geodesic inequality

$$k^2 \geq \rho_1^2 + \frac{\Delta\theta^2}{4} \, . \tag{125}$$

It is then obvious that in cases where $\rho_1$ is a constant independent of $\Delta\theta$ (as in ref. [23]), the geodesic with the minimal value of $k^2$ indeed has $\Delta\theta = 0$. In the present case, with the $F_2$ or $D_{\kappa=2}$ cost functions, we have simply $F_2 = k$ or $D_{\kappa=2} = k^2$, and so in both cases minimizing $k^2$ corresponds to minimizing the corresponding circuit depth. Therefore we conclude that the optimal geodesic also corresponds to that with $\Delta\theta = 0$.

It is straightforward to extract the value of $\rho_1(\phi)$ associated to the end point of the trajectory $UU_\phi$—see eq. (168) below. In the particular case when $\lambda_{\rm R} = 1$, the matrix $U_\phi$ in eq. (85) is simply a rotation matrix, and we will find that the value of $\rho_1(\phi)$ does not depend on $\phi$, i.e., $\rho_1$ becomes a constant with $\lambda_{\rm R} = 1$. We are therefore able to apply the above result and conclude that for $\lambda_{\rm R} = 1$, the straight-line geodesics with $\Delta\theta = 0$ describe the optimal circuits (where again we have chosen either the $F_2$ or $D_{\kappa=2}$ cost functions).

In appendix F, we prove a similar result for $\mathrm{Sp}(2N, \mathbb{R})$ for general $N$, again for the special case that $\lambda_{\rm R} = 1$. In particular, we demonstrate that the optimal circuit that prepares the target state $G_{\rm T}$ from the reference state $G_{\rm R}$ whose covariance matrix is the identity is given by the straight-line geodesic,

$$\gamma(\sigma) := U(\sigma) = e^{\sigma K} \, , \tag{126}$$

generated by a single generator determined by the relative covariance matrix introduced in eq. (86), i.e., $K = \frac{1}{2} \log \Delta$ with $\Delta^a{}_b = (G_{\rm T})^{a,c}(g_{\rm R})_{c,b}$. The proof of this general result requires some Lie group techniques together with a well-known decomposition of group elements of $\mathrm{Sp}(2N, \mathbb{R})$. In particular, this Lie group can be represented as a $\mathrm{U}(N)$ fiber bundle over the symmetric space $\mathrm{Sp}(2N, \mathbb{R})/\mathrm{U}(N)$. Here the fiber is nothing but the stabilizer group (83), and the base manifold can be interpreted as the space of Gaussian quantum states. We build on this decomposition to produce a generalized cylindrical foliation of the group manifold of the form $e^A u$ with $u \in \mathrm{U}(N)$, where $\|A\|$ plays the role of the radius. One can show that all geodesics that prepare the desired state will end on the cylinder of radius $\|K\|$. The final step is then to show that the minimal geodesic connecting $\mathbb{1}$ with this cylinder is the one which moves in a purely radial direction, i.e., it is the geodesic given in eq. (126), as we saw in the previous discussion of the special case $N = 1$.

It now remains to find the geodesics that produce the particular target states of interest, namely the TFD states introduced above. We shall first consider the special case of the time-independent TFD with $t = 0$ given by eq. (47). This case is relatively straightforward, and also enables us to make contact with the holographic results on the complexity of formation [27]. We shall then move on to the full time-dependent TFD (76) in the subsequent subsection.

## 4.3 Complexity of the TFD at $t = 0$ with $\lambda_{\rm R} = 1$

In this subsection, we focus on the complexity of one of the diagonal modes comprising the TFD at $t = 0$. Specifically, the covariance matrix $G_{\rm TFD}^+$ in eq. (47) for the mode associated with the $x^+$ coordinate will serve as our target state. Our reference state is given by $G_{\rm R}$ in eq. (47). As mentioned above, setting $\lambda_{\rm R} = 1$ provides a significant simplification, so we will focus on this case first. For convenience, we restate the relevant covariance matrices here:

$$G_{\rm R} = \begin{pmatrix} 1 & 0 \\ 0 & 1 \end{pmatrix}, \qquad G_{\rm T} = \begin{pmatrix} \frac{e^{2\alpha}}{\lambda} & 0 \\ 0 & e^{-2\alpha}\lambda \end{pmatrix} . \tag{127}$$

Using eq. (107) together with eq. (119), we obtain the boundary conditions for our circuit

$$
\begin{aligned}
\frac{1}{\lambda} e^{2\alpha} &= \cosh 2\rho_1 - \sinh 2\rho_1 \sin(\theta_1 + \tau_1)\,, \\
\lambda e^{-2\alpha} &= \cosh 2\rho_1 + \sinh 2\rho_1 \sin(\theta_1 + \tau_1)\,, \\
0 &= \sinh 2\rho_1 \cos(\theta_1 + \tau_1)\,.
\end{aligned}
\tag{128}
$$

The determinant of $G_{\mathrm{T}}$ in eq. (119) is one, and so the above equations only represent two independent relations for $\theta_1 + \tau_1$ and $\rho_1$ in terms of the physical parameters of the problem (*i.e.*, $e^{-2\alpha}\lambda$). Of course, this leaves the linear combination $\theta_1 - \tau_1$ unfixed. This remaining freedom is due to the equivalence class of circuits which produce the same target state, as discussed in the previous subsection. Additionally, as explained above, when $\rho_1$ is a fixed constant, the optimal geodesic is the straight line moving at a fixed angle in the $\tau = 0$ plane, as given in eq. (121). Hence we have $\tau_1 = 0$, whereupon solving eq. (165) yields

$$
\rho_1 = \left| \frac{1}{2} \log \lambda - \alpha \right|\,, \qquad\qquad \theta_0 = \theta_1 = \frac{\pi}{2} \, \mathrm{sgn}\!\left( \frac{1}{2} \log \lambda - \alpha \right)\,.
\tag{129}
$$

Substituting into eq. (111) and using eq. (121), we thus obtain the optimal circuit

$$
U_+(\sigma) = \begin{pmatrix} \lambda^{-\sigma/2} e^{\sigma\alpha} & 0 \\ 0 & \lambda^{\sigma/2} e^{-\sigma\alpha} \end{pmatrix} = \exp\!\left[ -\left( \frac{1}{2} \log \lambda - \alpha \right) \sigma W_+ \right]\,,
\tag{130}
$$

where $W_+$ is the generator for the scaling gate acting on $(x_+, p_+)$ in eq. (62). Repeating the analysis for the $x_-$ mode,[22] we simply get $U_-(\sigma) = \exp\!\left[ -\left( \frac{1}{2} \log \lambda + \alpha \right) \sigma W_- \right]$, where $W_-$ is the scaling generator of the $-$ modes. The simple form of these circuits, each containing only a single generator, allows us to easily compute the complexity $\mathcal{C}$ according to the cost functions in eq. (8). Since the two circuits commute, we can think of $\left( \frac{1}{2} \log \lambda \pm \alpha \right)$ as the number of times each scaling gate was applied, and simply combine these numbers using the chosen norm. The $F_2$ cost function yields the complexity[23]

$$
\mathcal{C}_2 = \sqrt{ \left( \frac{1}{2} \log \lambda + \alpha \right)^2 + \left( \frac{1}{2} \log \lambda - \alpha \right)^2 } = \sqrt{ \frac{1}{2} (\log \lambda)^2 + 2\alpha^2 }\,,
\tag{131}
$$

or, in terms of the physical variables, using eqs. (20) and (40),

$$
\mathcal{C}_2 = \sqrt{ \frac{1}{2} \log^2 \frac{\omega}{\mu} + \frac{1}{2} \log^2 \!\left( \frac{1 + e^{-\beta\omega/2}}{1 - e^{-\beta\omega/2}} \right) }\,.
\tag{132}
$$

Of course, the complexity for the $\kappa = 2$ cost function is simply related to the above result, *i.e.*,

$$
\mathcal{C}_{\kappa=2} = (\mathcal{C}_2)^2\,.
\tag{133}
$$

We can also evaluate the length of this circuit with the $F_1$ cost function, which yields

$$
\begin{aligned}
\mathcal{C}_1^{(\pm)} &= \left| \frac{1}{2} \log \lambda + \alpha \right| + \left| \frac{1}{2} \log \lambda - \alpha \right| \\
&= \frac{1}{2} \left| \log\!\left( \frac{\omega}{\mu} \frac{1 + e^{-\beta\omega/2}}{1 - e^{-\beta\omega/2}} \right) \right| + \frac{1}{2} \left| \log\!\left( \frac{\omega}{\mu} \frac{1 - e^{-\beta\omega/2}}{1 + e^{-\beta\omega/2}} \right) \right|\,,
\end{aligned}
\tag{134}
$$

---

[22] Recall that the state associated with the $x_-$ coordinate is obtained by replacing $\alpha \mapsto -\alpha$ in the $+$ state.

[23] Alternatively, we can use eq. (109) to read the line element directly, which for a circuit generated by a constant generator reduces to $\mathrm{d}s^2 = \frac{1}{2} \mathrm{tr}(K \cdot K^{\mathsf{T}}) d\sigma^2$.

where we have included the superscript ± to indicate that this complexity was evaluated with respect to the diagonal basis of generators, see eq. (22). Note that we did not prove that this trajectory was optimal for the $F_1$ cost function. It may be that a proof can be formulated, but it seems likely that with the $F_1$ cost function, many different circuits will be assigned the same, minimal circuit depth—see discussion in ref. [23]. However, we have not pursued this possibility in any detail at this point, and hence we may only state that our results for the $F_1$ cost function are an upper bound on the complexity of the target state.

Additionally, we noted above that the $F_1$ cost function depends on the basis chosen for the generators—again, see discussion in ref. [23]. It is therefore interesting to explore how the result (134) changes when we consider the basis of generators which naturally act on the physical (left (L) and right (R)) modes, rather than the diagonal (±) modes—see eq. (22). For this purpose, we must first combine the two circuits $U_\pm(\sigma)$ into a single $4 \times 4$ matrix (rather than two independent $2 \times 2$ matrices). The relevant transformation is then given by the combined $U(\sigma)$ acting on the covariance matrix describing the two ± oscillators. Organizing the diagonal modes as $\tilde{\xi} = (q_+, q_-, p_+, p_-)$, the transformation is block-diagonal, and takes the form

$$U(\sigma) = \exp\left[\sigma \widetilde{K}\right] , \qquad \widetilde{K} = \frac{1}{2}\log\lambda \begin{pmatrix} -1 & 0 & 0 & 0 \\ 0 & -1 & 0 & 0 \\ 0 & 0 & 1 & 0 \\ 0 & 0 & 0 & 1 \end{pmatrix} - \alpha \begin{pmatrix} -1 & 0 & 0 & 0 \\ 0 & 1 & 0 & 0 \\ 0 & 0 & 1 & 0 \\ 0 & 0 & 0 & -1 \end{pmatrix}. \tag{135}$$

Organizing the physical modes as $\xi = (q_L, q_R, p_L, p_R)$, the transformation of the circuit to the left-right basis via eq. (102) yields

$$U(\sigma) = \exp\left[\sigma K\right] , \qquad K = \frac{1}{2}\log\lambda \begin{pmatrix} -1 & 0 & 0 & 0 \\ 0 & -1 & 0 & 0 \\ 0 & 0 & 1 & 0 \\ 0 & 0 & 0 & 1 \end{pmatrix} + \alpha \begin{pmatrix} 0 & 1 & 0 & 0 \\ 1 & 0 & 0 & 0 \\ 0 & 0 & 0 & -1 \\ 0 & 0 & -1 & 0 \end{pmatrix}. \tag{136}$$

To compute the complexity in this basis, we need to decompose this generator $K$ in terms of the generators of $Sp(4,\mathbb{R})$ defined in eq. (95), see also appendix C. It is straightforward to extract the coefficients of these basis generators by taking the inner product (110) of eq. (136) with each of the generators. In this way, we find that

$$K = -\frac{1}{2}\log\lambda \left(W_{L,L} + W_{R,R}\right) + \alpha \left(W_{L,R} + W_{R,L}\right) , \tag{137}$$

and so only four components of the tangent vector $Y^I$ are nonvanishing. Evaluating the complexity with the $F_1$ cost function (8) then yields

$$\mathcal{C}_1^{(\text{LR})} = |Y_{L,L}| + |Y_{L,R}| + |Y_{R,L}| + |Y_{R,R}| = |\log\lambda| + 2|\alpha|, \tag{138}$$

which clearly differs from the result in eq. (134) in the diagonal basis.

It is also interesting to compare the complexity of the entangled TFD state of the two oscillators with that of the unentangled vacuum state, *i.e.,* the $\beta \to \infty$ or $\alpha \to 0$ limit of the TFD. The difference between these complexities for the present case of two oscillators serves as a precursor to the *complexity of formation* for the free scalar field in the next section. This quantity was originally defined in the context of holographic complexity, see, *e.g.,* ref. [27]. For the $F_2$ and $\kappa = 2$ complexities above, one finds

$$\Delta\mathcal{C}_2 = \sqrt{\frac{1}{2}(\log\lambda)^2 + 2\alpha^2} - \frac{1}{\sqrt{2}}|\log\lambda| , \qquad \Delta\mathcal{C}_{\kappa=2} = 2\alpha^2 . \tag{139}$$

Hence we see that there is a more complete cancellation in the complexity with the $\kappa = 2$ cost function. In particular, the difference only depends on the combination $\beta\omega$ through $\alpha$ from eq. (20), and is independent of the reference state scale $\mu$ or the mass $M$. For the $F_1$ complexity, we can compare the difference for the diagonal and physical bases, respectively:

$$\Delta C_1^{(\pm)} = \left| \frac{1}{2}\log\lambda + \alpha \right| + \left| \frac{1}{2}\log\lambda - \alpha \right| - |\log\lambda| \,, \qquad \Delta C_1^{(\mathrm{LR})} = 2|\alpha| \,. \tag{140}$$

We note that a similar cancellation arises for $\Delta C_1^{(\mathrm{LR})}$ as in $\Delta C_{\kappa=2}$, where the result only depends on $\beta\omega$ (but not $\mu$ or $M$).

In the following section, we will apply the above results to evaluate the complexity of formation for a free quantum field theory. These calculations will be based on the fact that the field theory can be represented as a collection of momentum modes, where each momentum mode is essentially entangled with its counterpart in the TFD to form a product of two-oscillator TFD states of the form studied in this section. We will find that the there are some interesting similarities between the $F_1$ result in the physical basis and previous results obtained for holographic complexity [27].

## 4.4 Complexity of the TFD at general $t$ with $\lambda_{\mathrm{R}} = 1$

Here we explore the complexity of the target state $G_{\mathrm{TFD}}^+(t)$ given in eq. (76), starting from the reference state $G_{\mathrm{R}}$ in (47) with $\lambda_{\mathrm{R}} = 1$. We focus again on the $+$ mode; the $-$ mode is then easily obtained under the replacement $\alpha \mapsto -\alpha$. The boundary conditions are determined by comparing eq. (119) with $(\rho, \tau, \theta) = (\rho_1, \tau_1, \theta_1)$ at $\sigma = 1$ to the covariance matrix of the target state (76). This comparison yields

$$
\begin{aligned}
\frac{1}{\lambda}\left(\cosh 2\alpha + \sinh 2\alpha \cos\omega t\right) &= \cosh 2\rho_1 - \sinh 2\rho_1 \sin(\theta_1 + \tau_1), \\
\lambda\left(\cosh 2\alpha - \sinh 2\alpha \cos\omega t\right) &= \cosh 2\rho_1 + \sinh 2\rho_1 \sin(\theta_1 + \tau_1), \\
-\sinh 2\alpha \sin\omega t &= \sinh 2\rho_1 \cos(\theta_1 + \tau_1) \,.
\end{aligned}
\tag{141}
$$

Of course, as in eq. (165), the right-hand side only depends on $\rho_1$ and $\theta_1 + \tau_1$, while the combination $\theta_1 - \tau_1$ is left undetermined. We can solve eq. (141) explicitly to obtain

$$
\begin{aligned}
\cosh 2\rho_1 &= \frac{1+\lambda^2}{2\lambda}\cosh 2\alpha + \frac{1-\lambda^2}{2\lambda}\sinh 2\alpha \cos\omega t \,, \\
\tan(\theta_1 + \tau_1) &= \frac{1+\lambda^2}{2\lambda}\cot\omega t + \frac{1-\lambda^2}{2\lambda}\frac{1}{\tanh 2\alpha \sin\omega t} \,.
\end{aligned}
\tag{142}
$$

Now, for the minimal straight-line geodesic (121) in the $\tau = 0$ plane, the circuit (111) is given by

$$
\begin{aligned}
U(\sigma) &= \begin{pmatrix} \cosh(\rho_1\sigma) - \sin\theta_1 \sinh(\rho_1\sigma) & \cos\theta_1 \sinh(\rho_1\sigma) \\ \cos\theta_1 \sinh(\rho_1\sigma) & \cosh(\rho_1\sigma) + \sin\theta_1 \sinh(\rho_1\sigma) \end{pmatrix} \\
&= \exp\left[ \begin{pmatrix} -\sin\theta_1 & \cos\theta_1 \\ \cos\theta_1 & \sin\theta_1 \end{pmatrix} \rho_1\sigma \right] = \exp\left[ -\rho_1 \sin\theta_1 \,\sigma\, W + \frac{\rho_1}{\sqrt{2}}\cos\theta_1 \,\sigma\,(Z - V) \right],
\end{aligned}
\tag{143}
$$

where in the last expression we have used the matrix generators in eq. (62).

Now, for the $F_2$ cost function (8), we can use eq. (122) to evaluate the complexity, *i.e.,* the length of this optimal circuit, which yields

$$C_2^{(+)} = \int_0^1 \rho_1 = \rho_1 \,, \tag{144}$$

which is independent of $\theta_1$, as expected from our analysis above. Of course, for the $\kappa = 2$ cost function (9), we find

$$\mathcal{C}_{\kappa=2}^{(+)} = \mathcal{C}_2^2 = \rho_1^2, \tag{145}$$

where $\rho_1$ was given in eq. (142). Alternatively, the $F_1$ cost function yields[24]

$$\mathcal{C}_1^{(+)} = \left|Y^W\right| + \left|Y^V\right| + \left|Y^Z\right| = \rho_1\left(\sqrt{2}|\cos\theta_1| + |\sin\theta_1|\right). \tag{146}$$

As we commented above, we are not assured that the straight-line trajectory (121) is the shortest geodesic for this cost function, but it does at least provide an upper bound on the complexity. Substituting in the boundary conditions $\rho_1$ and $\theta_1$ given in eq. (142) (with $\tau_1 = 0$) into the above expressions then yields the complexity of $|\text{TFD}(t)\rangle$ in terms of the physical parameters of the state, *i.e.,* $\omega$ and $\beta$. The contribution to complexity from the $-$ mode are obtained from the above by simply replacing $\alpha \mapsto -\alpha$ in eqs. (141) and (142).

Next, we would like to look at the $F_1$ complexity in the physical $LR$ basis. By construction, in the diagonal basis $\tilde{\xi} = (q_+, q_-, p_+, p_-)$ the full optimal circuit $\widetilde{U}(\sigma) = \exp[\widetilde{M}\,\sigma]$ is

$$\widetilde{M} = \begin{pmatrix} -\rho_{1+}\sin\theta_{1+} & 0 & \rho_{1+}\cos\theta_{1+} & 0 \\ 0 & -\rho_{1-}\sin\theta_{1-} & 0 & \rho_{1-}\cos\theta_{1-} \\ \rho_{1+}\cos\theta_{1+} & 0 & \rho_{1+}\sin\theta_{1+} & 0 \\ 0 & \rho_{1-}\cos\theta_{1-} & 0 & \rho_{1-}\sin\theta_{1-} \end{pmatrix}. \tag{147}$$

Now we apply the transformation (102) to obtain the relevant generator $M = R_4^{\mathsf{T}}\widetilde{M}R_4$ in the $LR$ basis. We can then use eq. (97) to extract the coefficients of the $SP(4, \mathbb{R})$ generators in this basis (see appendix C for the full list of generators). We finally obtain the following decomposition:

$$\begin{aligned} M = & -\frac{\rho_{1+}\cos\theta_{1+} + \rho_{1-}\cos\theta_{1-}}{2\sqrt{2}}\left(V_{L,L} + V_{R,R} - Z_{L,L} - Z_{R,R}\right) \\ & -\frac{\rho_{1+}\cos\theta_{1+} - \rho_{1-}\cos\theta_{1-}}{2}\left(V_{L,R} - Z_{L,R}\right) \\ & -\frac{\rho_{1+}\sin\theta_{1+} + \rho_{1-}\sin\theta_{1-}}{2}\left(W_{L,L} + W_{R,R}\right) \\ & -\frac{\rho_{1+}\sin\theta_{1+} - \rho_{1-}\sin\theta_{1-}}{2}\left(W_{L,R} + W_{R,L}\right). \end{aligned} \tag{148}$$

If we measure the complexity with the $F_2$ cost function (8), combining the two modes, we find

$$\mathcal{C}_2 = \sqrt{\rho_{1+}^2 + \rho_{1-}^2}, \tag{149}$$

while the $\kappa = 2$ cost function (9) instead yields

$$\mathcal{C}_{\kappa=2} = \rho_{1+}^2 + \rho_{1-}^2. \tag{150}$$

Both these results are invariant under the orthogonal basis transformation (102), *i.e.,* these complexities are the same in the diagonal and the physical bases, as well as being independent of $\theta_{1\pm}$. In contrast, using the $F_1$ cost function in the physical basis, we arrive at a different result

$$\begin{aligned} \mathcal{C}_1^{(\text{LR})} = & \sqrt{2}\,|\rho_{1+}\cos\theta_{1+} + \rho_{1-}\cos\theta_{1-}| + |\rho_{1+}\sin\theta_{1+} + \rho_{1-}\sin\theta_{1-}| \\ & + |\rho_{1+}\cos\theta_{1+} - \rho_{1-}\cos\theta_{1-}| + |\rho_{1+}\sin\theta_{1+} - \rho_{1-}\sin\theta_{1-}|. \end{aligned} \tag{151}$$

---

[24]We might note that the results here and in the following are simplified somewhat if we replace $\{W, V, Z\} \mapsto \{W, \frac{1}{\sqrt{2}}(V \pm Z)\}$. With this new basis, eq. (146) becomes $\mathcal{C}_1 = \rho_1(|\cos\theta_1| + |\sin\theta_1|)$, but the results are qualitatively unchanged.

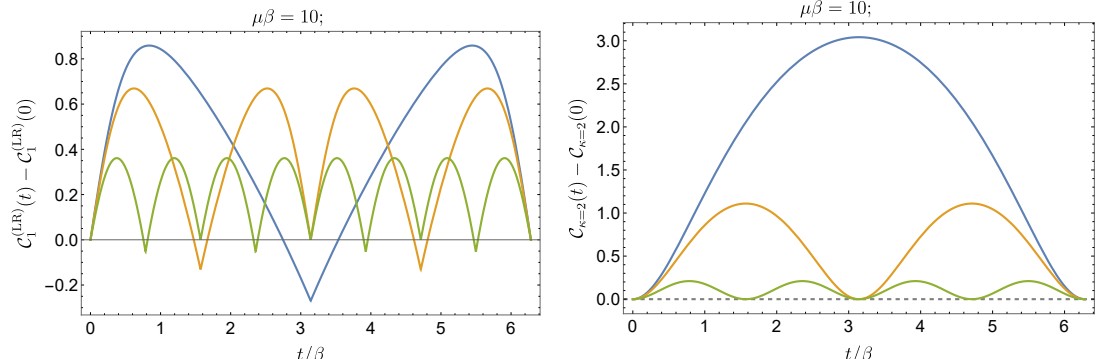

Figure 2: Single mode complexity as a function of time for $\beta\omega = 0.5$ (blue), 1 (yellow), and 2 (green). We note that the complexity oscillates in time with periodicity $\delta t = \pi/\omega$ as expected from the explicit expressions and with an amplitude that decreases (approximately exponentially for large $\beta\omega$) for increasing $\beta\omega$.

Translating eq. (142) for the $\pm$ boundary conditions then allows us to express the coefficients above in terms of the physical parameters, namely

$$
\begin{aligned}
\cosh 2\rho_{1+} &= \cosh 2\hat{\alpha}\cosh 2\alpha - \sinh 2\hat{\alpha}\sinh 2\alpha\cos\omega t\,, \\
\tan\theta_{1+} &= \cosh 2\hat{\alpha}\cot\omega t - \frac{\sinh 2\hat{\alpha}}{\tanh 2\alpha\sin\omega t}\,, \\
\cosh 2\rho_{1-} &= \cosh 2\hat{\alpha}\cosh 2\alpha + \sinh 2\hat{\alpha}\sinh 2\alpha\cos\omega t\,, \\
\tan\theta_{1-} &= \cosh 2\hat{\alpha}\cot\omega t + \frac{\sinh 2\hat{\alpha}}{\tanh 2\alpha\sin\omega t}\,,
\end{aligned}
\tag{152}
$$

where to simplify these expressions, we have introduced

$$
\lambda := \exp(2\hat{\alpha})\,.
\tag{153}
$$

In fig. 2, we plot the results for $\mathcal{C}_{\kappa=2}$ and $\mathcal{C}_1^{(\mathrm{LR})}$ in eqs. (150) and (151), respectively, for various values of the parameters. If one expands the above expressions for small $t$, one finds that the growth of $\mathcal{C}_1^{(\mathrm{LR})}$ is initially linear, while that for $\mathcal{C}_2$ and $\mathcal{C}_{\kappa=2}$ is quadratic. The examples depicted in fig. 2 exhibit this behaviour. We also see that the complexity oscillates in time, but that the amplitude of the oscillations decreases as $\beta\omega$ increases. From eq. (20), we see that for large $\beta\omega$, $\alpha \simeq \exp[-\beta\omega/2]$, and expanding the expressions in eq. (152) shows that the oscillations are indeed exponentially suppressed in this regime. This fact will allow our results for the field theory to be integrated with respect to the frequency in the next section.

**A simple limit**

Recall that above we have set $\lambda_{\mathrm{R}} = 1$, which amounts to setting the gate scale and reference-state scales equal, *i.e.,* $\omega_g^2 = M\mu$. Hence from eq. (40) we have

$$
\exp(2\hat{\alpha}) = \lambda = \omega/\mu\,.
\tag{154}
$$

Now let us consider the limit in which $\mu$ is much bigger than any other scale, *i.e.,* $\lambda \to 0$ or $\hat{\alpha} \to -\infty$. In this limit, we have

$$
\cosh 2\hat{\alpha} \simeq \frac{1}{2}\frac{\mu}{\omega} \simeq -\sinh 2\hat{\alpha}\,,
\tag{155}
$$

in which case eq. (152) simplifies to

$$\cosh 2\rho_{1\pm} \simeq \frac{1}{2}\frac{\mu}{\omega}\left(\cosh 2\alpha \pm \sinh 2\alpha \cos \omega t\right),$$

$$\tan \theta_{1\pm} \simeq \frac{1}{2}\frac{\mu}{\omega}\left(\cot \omega t \pm \frac{1}{\tanh 2\alpha \sin \omega t}\right). \tag{156}$$

These expressions yield the simple solution

$$\rho_{1\pm} \simeq \frac{1}{2}\log\frac{\mu}{\omega} + \frac{1}{2}\log\left(\cosh 2\alpha \pm \sinh 2\alpha \cos \omega t\right), \qquad \theta_{1\pm} \simeq \pm\mathrm{sgn}(\sin(\omega t))\frac{\pi}{2}. \tag{157}$$

Then substituting into the $\kappa = 2$ cost in eq. (150) yields

$$\Delta C_{\kappa=2} = \frac{1}{2}\log\frac{\mu}{\omega}\log\left(\cosh^2 2\alpha - \sinh^2 2\alpha \cos^2 \omega t\right)$$

$$+ \frac{1}{4}\log^2\left(\cosh 2\alpha + \sinh 2\alpha \cos \omega t\right) + \frac{1}{4}\log^2\left(\cosh 2\alpha - \sinh 2\alpha \cos \omega t\right),$$

$$= \frac{1}{2}\log\frac{\mu}{\omega}\log\left(1 + \sinh^2 2\alpha \sin^2 \omega t\right)$$

$$+ \frac{1}{4}\log^2\left(e^{2\alpha} - 2\sinh 2\alpha \sin^2\left(\frac{\omega t}{2}\right)\right) + \frac{1}{4}\log^2\left(e^{-2\alpha} + 2\sinh 2\alpha \sin^2\left(\frac{\omega t}{2}\right)\right), \tag{158}$$

where we have subtracted the zero-temperature complexity, *i.e.,* $C_{\kappa=2}(\alpha \to 0) = \frac{1}{2}\log^2\frac{\mu}{\omega}$. Note that only the first term depends on $\mu$, and that this contribution vanishes for $t = 0$. Of course, this is in agreement with our expression (139) for the "complexity of formation", which is independent of this reference frequency. However, $\mu$ appears as a new scale in our result for $\Delta C_{\kappa=2}$ as soon as the time is nonvanishing.

We can also substitute the simplified expressions from eq. (157) into the $F_2$ complexity in eq. (149) to find

$$\Delta C_2 = \frac{1}{2\sqrt{2}}\log\left(\cosh^2 2\alpha - \sinh^2 2\alpha \cos^2 \omega t\right) = \frac{1}{2\sqrt{2}}\log\left(1 + \sinh^2 2\alpha \sin^2 \omega t\right), \tag{159}$$

where we have again subtracted the zero-temperature contribution, which in this case is $C_2(\alpha \to 0) = \frac{1}{\sqrt{2}}\log\frac{\mu}{\omega}$. We have also dropped terms which are suppressed by inverse powers of $\log\frac{\mu}{\omega}$. Finally, substituting the simplified result (157) into the $F_1$ complexity (151), we find

$$\Delta C_1^{(\mathrm{LR})} = \log\left(\cosh 2\alpha + \sinh 2\alpha \,|\cos \omega t|\right), \tag{160}$$

where the subtracted zero-temperature contribution is $C_1^{(\mathrm{LR})}(\alpha \to 0) = \log\frac{\mu}{\omega}$. One sees that $\Delta C_1^{(\mathrm{LR})}$ is maximal when $|\cos \omega t| = 1$, *i.e.,* $t = n\pi/\omega$. In particular, we have a maximum at $t = 0$. Note that in this limit, the initial growth of $\Delta C_1^{(\mathrm{LR})}$ in eq. (160) is actually quadratic rather than linear, as found with the full expression (151) – see comment below eq. (153).

We can also consider the opposite limit where $\mu$ is much smaller than any other scale, *i.e.,* $\lambda \to \infty$ or $\hat{\alpha} \to \infty$ from eq. (154). Again, we have set $\omega_g^2 = M\mu$. Then in this limit, we have

$$\cosh 2\hat{\alpha} \simeq \frac{1}{2}\frac{\omega}{\mu} \simeq \sinh 2\hat{\alpha}, \tag{161}$$

in which case eq. (152) simplifies to

$$\cosh 2\rho_{1\pm} \simeq \frac{1}{2}\frac{\omega}{\mu}\left(\cosh 2\alpha \mp \sinh 2\alpha \cos \omega t\right),$$

$$\tan \theta_{1\pm} \simeq \frac{1}{2}\frac{\omega}{\mu}\left(\cot \omega t \mp \frac{1}{\tanh 2\alpha \sin \omega t}\right). \tag{162}$$

These expressions then produce the simple solution

$$\rho_{1\pm} \simeq \frac{1}{2}\log\frac{\omega}{\mu} + \frac{1}{2}\log\left(\cosh 2\alpha \mp \sinh 2\alpha \cos\omega t\right), \qquad \theta_{1\pm} \simeq \mp\mathrm{sgn}(\sin(\omega t))\frac{\pi}{2}, \qquad (163)$$

which is quite similar to that found above in eq. (157). We can substitute these simplified expressions into all of the various expressions for the complexity from the different cost functions above, but let us focus on the $F_1$ complexity in eq. (151). In this case, we find an identical result to eq. (160), namely

$$\Delta\mathcal{C}_1^{(\mathrm{LR})} = \log\left(\cosh 2\alpha + \sinh 2\alpha \,|\cos\omega t|\right). \qquad (164)$$

Thus, even though we are considering the opposite limit here (*i.e.*, $\lambda \to \infty$ rather than $\lambda \to 0$), $\Delta\mathcal{C}_1^{(\mathrm{LR})}$ is unchanged. In particular, we still find that the complexity has a maximum at $t = 0$. The results for $\Delta\mathcal{C}_2$ and $\Delta\mathcal{C}_{\kappa=2}$ are very similar to those obtained using the previous limit.

## 4.5 Complexity of the TFD at general $t$ with $\lambda_{\mathrm{R}} \neq 1$

For the general case $\lambda_{\mathrm{R}} \neq 1$, the boundary conditions (107) for the circuits leading to the time-dependent TFD state (76) read

$$\frac{1}{\lambda}\left(\cosh 2\alpha + \sinh 2\alpha \cos\omega t\right)$$
$$= \lambda_{\mathrm{R}}\left(\cosh 2\rho_1 - \sinh 2\rho_1 \sin(\theta_1 + \tau_1)\right) + \frac{1 - \lambda_{\mathrm{R}}^2}{\lambda_{\mathrm{R}}}\left(\cosh\rho_1 \cos\tau_1 - \sinh\rho_1 \sin\theta_1\right)^2,$$

$$\lambda\left(\cosh 2\alpha - \sinh 2\alpha \cos\omega t\right)$$
$$= \lambda_{\mathrm{R}}\left(\cosh 2\rho_1 + \sinh 2\rho_1 \sin(\theta_1 + \tau_1)\right) + \frac{1 - \lambda_{\mathrm{R}}^2}{\lambda_{\mathrm{R}}}\left(\cosh\rho_1 \sin\tau_1 + \sinh\rho_1 \cos\theta_1\right)^2, \qquad (165)$$

$$-\sinh 2\alpha \sin\omega t = \lambda_{\mathrm{R}} \sinh 2\rho_1 \cos(\theta_1 + \tau_1)$$
$$+ \frac{1 - \lambda_{\mathrm{R}}^2}{2\lambda_{\mathrm{R}}}\left(\sinh 2\rho_1 \cos(\theta_1 + \tau_1) - \sinh^2\rho_1 \sin 2\theta_1 + \cosh^2\rho_1 \sin 2\tau_1\right).$$

Here we have used the parametrization in eq. (111) in terms of the values $(\rho_1, \theta_1, \tau_1)$ at the end point of the trajectory, and as usual $G_{\mathrm{R}}$ is given in eq. (47). We have also focused on the $+$ mode in the TFD. The TFD state at $t = 0$ with general $\lambda_{\mathrm{R}}$ is obtained by simply setting $t = 0$ in the results of this section. Of course, since the determinant of the matrices on either side of eq. (107) is one, there are again only two independent relations in eq. (165), and as a result there will be a one-parameter family geodesics (*i.e.*, circuits) that produce the desired target state. As before, this family of end points maps out a spiral in the space of unitaries spanned by $(\rho, \theta, \tau)$. However, an important feature is that for $\lambda_{\mathrm{R}} \neq 1$, $\rho_1$ is no longer a fixed constant, but rather varies as we move along the spiral. Hence our previous arguments that the optimal circuit corresponds to $\Delta\theta = 0 = \tau$ in section 4.1 no longer apply, and we must undertake a more extensive analysis of all possible geodesics ending on the spiral. Given an end point, we may use the geodesic solution (115) evaluated at $\sigma = 1$ to solve for $c$, $\Delta\theta$, and $\theta_1$ for a given boundary condition (in general, this solution is found numerically below). The optimal circuit will be given by eq. (115) for general values of $\sigma$, and its length can be evaluated according to eq. (116). We must then minimize this length over all possible solutions of eq. (165) in order to find the optimal circuit.

One particular solution to eq. (165) arises naturally from our construction of the time-dependent TFD state, namely, that which corresponds to the unitary

$$U_{\mathrm{R}\to\mathrm{TFD}(t)} = U_{\mathrm{vac}\to\mathrm{TFD}(t)}\, U_{\mathrm{R}\to\mathrm{vac}}, \qquad (166)$$

where $U_{\text{vac}\to\text{TFD(t)}}$ is given by eq. (78), and $U_{\text{R}\to\text{vac}}$ by eq. (81). Other end points can then be obtained using the stabilizer group of the reference state by multiplying this transformation on the right by $U_\phi$ in eq. (85) (see also eq. (118)),

$$U_{\text{T}}(\phi) = U_{\text{vac}\to\text{TFD(t)}} U_{\text{R}\to\text{vac}} U_\phi \, . \tag{167}$$

More explicitly, one extracts from the unitary $U_{\text{T}}(\phi)$ the corresponding end point $(\rho_1(\phi), \theta_1(\phi), \tau_1(\phi))$ using eq. (111) via the relations

$$4\cosh^2(\rho_1(\phi)) = \left(U_{\text{T}}(\phi)_{2,1} - U_{\text{T}}(\phi)_{1,2}\right)^2 + \left(U_{\text{T}}(\phi)_{1,1} + U_{\text{T}}(\phi)_{2,2}\right)^2 \, ,$$

$$2\sin(\theta_1(\phi)) = \frac{U_{\text{T}}(\phi)_{2,2} - U_{\text{T}}(\phi)_{1,1}}{\sinh(\rho_1(\phi))}, \qquad 2\cos(\theta_1(\phi)) = \frac{U_{\text{T}}(\phi)_{1,2} + U_{\text{T}}(\phi)_{2,1}}{\sinh(\rho_1(\phi))}, \tag{168}$$

$$2\sin(\tau_1(\phi)) = \frac{U_{\text{T}}(\phi)_{2,1} - U_{\text{T}}(\phi)_{1,2}}{\cosh(\rho_1(\phi))}, \qquad 2\cos(\tau_1(\phi)) = \frac{U_{\text{T}}(\phi)_{1,1} + U_{\text{T}}(\phi)_{2,2}}{\cosh(\rho_1(\phi))} \, .$$

Alternatively, we may derive differential equations for the end points by varying the two sides of eq. (165) with respect to $\phi$, whereupon we find

$$\tau_1'(\phi) = \rho_1'(\phi)\sec(\theta_1(\phi) - \tau_1(\phi))\left[\tanh(\rho_1(\phi))\sin(\theta_1(\phi) - \tau_1(\phi)) + \frac{\lambda_{\text{R}}^2 + 1}{\lambda_{\text{R}}^2 - 1}\right],$$

$$\theta_1'(\phi) = -\rho_1'(\phi)\sec(\theta_1(\phi) - \tau_1(\phi))\left[\coth(\rho_1(\phi))\sin(\theta_1(\phi) - \tau_1(\phi)) + \frac{\lambda_{\text{R}}^2 + 1}{\lambda_{\text{R}}^2 - 1}\right]. \tag{169}$$

These differential equations provide an efficient way of generating end points for geodesics in our numerical solutions below. Once we have the family of end points $(\rho_1(\phi), \theta_1(\phi), \tau_1(\phi))$, we can compute the parameters $(\theta_0(\phi), c(\phi), \Delta\theta(\phi))$ of the corresponding geodesic by inverting the boundary condition numerically and then evaluate the corresponding length (*i.e.,* complexity) of the geodesic ending at $U_{\text{T}}U_\phi$ as

$$\ell(\phi) = \frac{\sqrt{c(\phi)^2 + \Delta\theta(\phi)}}{2} \, . \tag{170}$$

We can gain some intuition for this set-up by plotting the curves of equivalent end points in the three-dimensional space spanned by $(\rho\cos(\theta), \rho\sin(\theta), \tau)$. These curves have a spiral like shape, and their projection on the $\tau = 0$ plane is a closed curve—see the plots in fig. 3. We may use the point where the curves cross the $\tau = 0$ plane as a representative point for each spiral. In the figure, we have chosen two such representative end points, one on the $\rho\sin(\theta)$ axis in the upper half-plane, and a generic point in the upper-right quadrant.

Let us now focus on the example of the TFD state at $t = 0$. Using the explicit expression for the symplectic transformations which bring us to this state, *i.e.,* eq. (166) together with eqs. (78), (81), and (85) evaluated at $t = 0$, as well as the inversions (168), we find a family of end points $(\rho_1(\phi), \theta_1(\phi), \tau_1(\phi))$ associated with this state. We do not write them explicitly here since the expressions are rather lengthy and uninformative. One of these end points, corresponding to $\tau_1 = 0$, turns out to be associated to the value $\phi = 0$ in the unitary $U_\phi$. This means that the natural symplectic transformation coming from our construction of the TFD state was associated with $\tau_1 = 0$. Now, the corresponding value of $\theta_1$ can be read from the inversion formula (168) for the case of $\phi = 0$ in eq. (166), to get

$$\sin(\theta_1) = \text{sgn}(\lambda - \exp(2\alpha)\lambda_{\text{R}}), \qquad \cos(\theta_1) = 0 \, . \tag{171}$$

This means that $\theta_1 = \pi/2$ for $e^{-2\alpha}\lambda/\lambda_{\text{R}} > 1$, while for $e^{-2\alpha}\lambda/\lambda_{\text{R}} < 1$ we have $\theta_1 = -\pi/2$. Now let us consider the expression for $\rho_1(\phi)$. This determines the projection of the spiral of equivalent end points on the $\tau = 0$ plane. We obtain the following expression:

$$\cosh^2(\rho_1(\phi)) = \cosh^2(x + y) - \sinh(2x)\sinh(2y)\cos^2\phi \, , \tag{172}$$

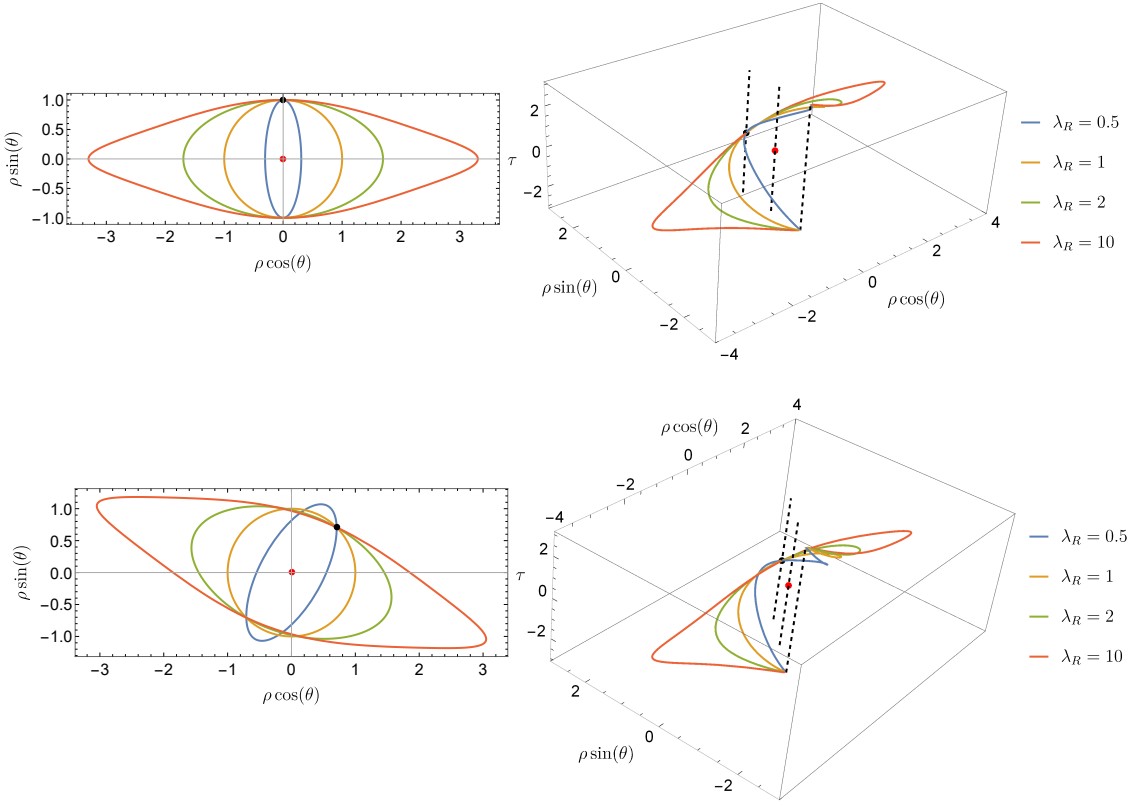

Figure 3: Illustration of the geometry of $\mathcal{F}_{r\to\text{T}}$ given in eq. (118) for different target states $|G_\text{T}\rangle$ and different values of $\lambda_\text{R}$. The black dot indicates the intersection with the $\tau = 0$ plane, and all the curves originate from this same point.

where we have defined $e^{-2y} := \lambda_\text{R}$ and $e^{-2x} := \lambda e^{-2\alpha}$. The above curve is a closed shape with a long and a short axis where and the short axis is either aligned with $\phi = 0$ or $\pi/2$ depending on the signs of $x$ and $y$. In particular, when they both have the same sign (*i.e.*, $xy > 0$), the short axis is at $\phi = 0$, and hence given the previous observations, it is aligned with the point at which the spiral crosses the $\tau = 0$ plane. Recall that the geodesic ending at $\tau_1 = 0$ is still the straight-line geodesic with $\Delta\theta = 0$, which remains in the $\tau = 0$ plane for the entire trajectory.[25] Since for other values of $\phi$ we will have $\rho_1(\phi) \geq \rho_1(0)$ and $\Delta\theta^2 \geq 0$, we can use the inequality (125) to argue that the shortest geodesic is in fact still the simple straight-line geodesic (121), where the previous analysis indicates that $\theta_1 = \pm\pi/2$.

Alternatively, if $x$ and $y$ have opposite signs, then the short axis is aligned with $\phi = \pi/2$, and we should expect that the optimal geodesic will no longer be the simple one which remains in the $\tau = 0$ plane. Numerical testing reveals that this is indeed the case. This means that even for the TFD at $t = 0$, there exist values of the parameters for which the straight line geodesic does not represent the optimal trajectory. This is illustrated in fig. 4. This conclusion continues to hold even in the limit of low temperatures where the two sides of the thermofield double decouple from one another and our target state becomes two copies of the vacuum state. This is simply achieved by setting $\alpha = 0$ in the definition of $x$. Exploring the relevant ranges of $x$ and $y$ reveals that we deviate from the straight line trajectories when $\lambda > 1$ and $\lambda_\text{R} < 1$ or when $\lambda < 1$ and $\lambda_\text{R} > 1$. This is different from the conclusion of [23] where only $\widehat{W}_{i,j}$ gates were used, see eq. (91), and indeed we see that when exploring the full symplectic

---

[25]The analysis of the geometry and the geodesics is not changed by choosing $\lambda_\text{R} \neq 1$, only the positions of the end points.

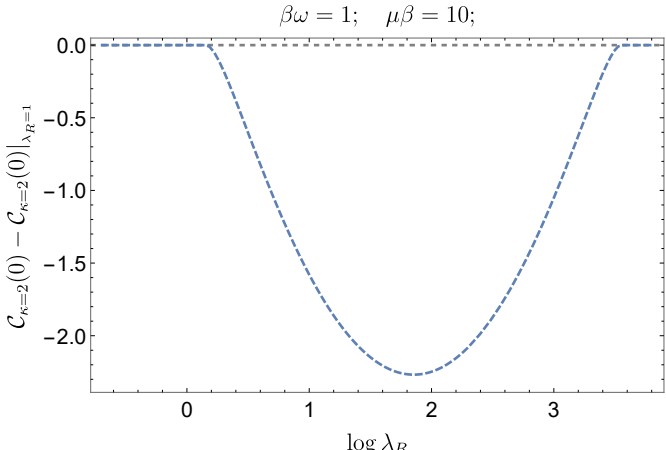

Figure 4: Complexity of a single mode TFD at $t = 0$ with various values of $\lambda_{\text{R}}$ minus that with $\lambda_{\text{R}} = 1$. In this plot we have fixed $\omega_{\text{R}}^2 \beta / M = 10$ and $\beta\omega = 1$ and this means that $\lambda_{\text{R}}/\lambda = 10$ and focused on the $+$ mode. In this case we expect that for $1 < \lambda_{\text{R}} < 40.84$ we will have deviations from the straight line trajectories. Outside this range, we recover straight line trajectories and since the ratio of $\lambda/\lambda_{\text{R}}$ is fixed $((x-y)$ is fixed), eq. (172) predicts no dependence on $\lambda_{\text{R}}$ outside this range and this is indeed what we see in the figure when comparing to the $\lambda_{\text{R}} = 1$ value. We remark that this plot does not have the resolution to show exponentially small deviations around $\Delta\mathcal{C}_{\kappa=2} = 0$.

group, shorter trajectories exist using the full set of gates in eq. (91).

Numerical testing reveals that the optimal geodesics deviate from the $\tau = 0$ plane in all cases where the spirals do not intersect $\tau = 0$ at $\theta = \pm\frac{\pi}{2}$. Of course, the alignment with $\theta = \pm\frac{\pi}{2}$ happens not only at $t = 0$ but more generally for $\omega_k t = \pi n$ where $n \in N$ (since our solutions are periodic in time). When $n$ is even, the conditions are identical to the previous case. When $n$ is odd, one has to substitute $\alpha \mapsto -\alpha$ to obtain the relevant conditions for the trajectories to move in the $\tau = 0$ plane. Of course, if we consider the TFD for oscillators with different values of $\omega$,[26] the time for this alignment will differ for the different oscillators and therefore, in the QFT calculations, the time in which all trajectories move in the $\tau = 0$ plane for all values of $\omega_k$ can only be achieved at $t = 0$.

We have plotted the time dependence of complexity (using the $\kappa = 2$ cost function, *i.e.,* eq. (150)) for the TFD state (76) in fig. 5. We have chosen to plot it as a function of $\omega t$ to account for the periodicity of the result. One striking feature of all of these plots is that the complexity (or rather the difference $\mathcal{C}_{\kappa=2}(t) - \mathcal{C}_{\kappa=2}(0)$) decreases as $\lambda_{\text{R}}$ is varied away from one, *i.e.,* , the complexity decreases as both $\lambda_{\text{R}}$ is increased or decreased. We see that as before the complexity decreases exponentially as we increase $\omega\beta$. We observe that as we increase $\mu/\omega$, the $\lambda_{\text{R}} < 1$ result becomes very close to the one for $\lambda_{\text{R}} = 1$. We do not understand what is the reason for this behaviour at present and leave this issue for future study. The low temperature limit is obtained by taking $\beta\omega \to 0$ while keeping $\mu/\omega$ fixed. Even in this limit, we see that significant deviations are obtained from the previous $\lambda_{\text{R}} = 1$ results. Further, we observe that for certain values of $\lambda_{\text{R}}$, the complexity is decreased compared to that at $t = 0$ for all other times.

In the next section we explain how to use the results of this section for the complexity of the TFD state of two simple harmonic oscillators for the purpose of computing the complexity

---

[26]For example, the regulated scalar field theory in the next section.

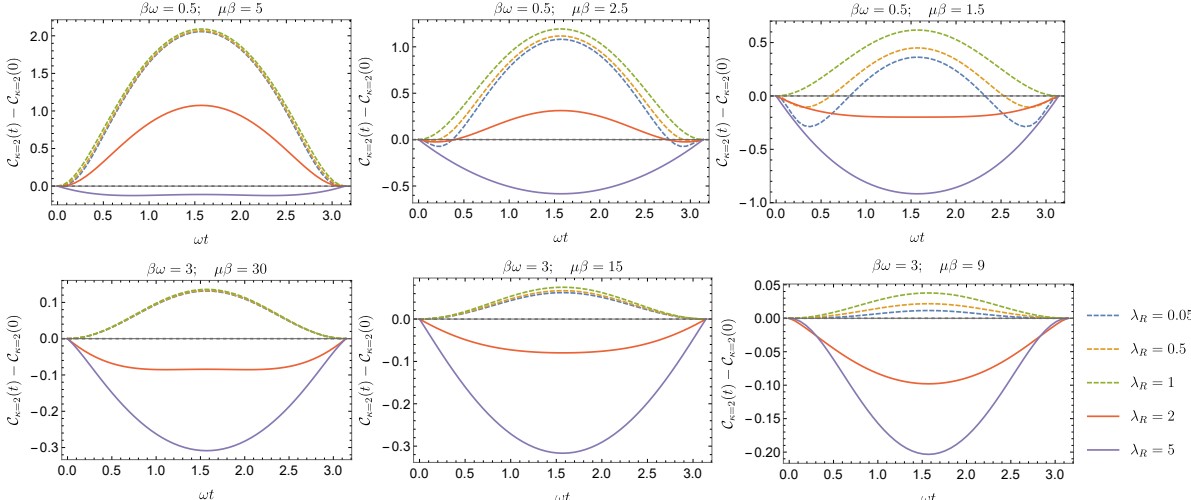

Figure 5: Complexity of a single mode TFD with various values of $\lambda_R$ and of the parameters $\omega\beta$ and $\mu\beta$ as indicated on the different panels. The plot includes the sum of the $+$ and $-$ modes. The plot is always periodic as a function of $\omega t$ with periodicity $\pi$ as previously, and so we have only plotted one period. The result for the complexity is suppressed as we increase $\beta\omega$. When increasing the reference state scale encoded in the parameter $\mu\beta$, the curves for $\lambda_R < 1$ approach the one of $\lambda_R = 1$. We do not fully understand what is the reason for this behavior and leave it for future study. In some instances the (regularized) complexity is negative for all times.

of the TFD state of two copies of a free scalar field theory. We will discretize our field theory on the spatial lattice in such a way that the TFD state becomes a product of these harmonic oscillator TFD states for the normal modes on the lattice.

# 5 Complexity of TFD states in quantum field theory

In this section, we combine the results of section 4 with the lattice discretization and normal mode decomposition of free scalar QFTs explained below in subsection 5.1 in order to obtain the complexity of TFD states of free scalar QFTs. The results of subsection 5.1 also serve as background for the entanglement calculations presented in section 6. In subsection 5.2, we present results for complexity obtained using lattice regularization, which will turn out to be particularly relevant for the comparison with the physics of entanglement entropy. In subsection 5.3, we focus on the complexity of formation of TFD states at $t = 0$ with $\lambda_R = 1$. In subsection 5.4, we use a similar approach to study the complexity of the TFD of a free scalar QFT at $t \neq 0$ with $\lambda_R = 1$. Finally, in subsection 5.5 we comment on the case $\lambda_R \neq 1$.

## 5.1 Normal mode decomposition for a free QFT

Let us start by focusing on a 1+1-dimensional free QFT living on a (Lorentzian) cylinder of circumference $\mathcal{L}$, defined by the Hamiltonian

$$H = \int_{-\mathcal{L}/2}^{\mathcal{L}/2} dx \left( \frac{1}{2}\pi(x)^2 + \frac{1}{2}m^2\phi(x)^2 + \frac{1}{2}(\partial_x\phi(x))^2 \right). \qquad (173)$$

We regulate this theory in the UV by putting it on a lattice with lattice spacing $\delta > 0$. If we assume that the lattice has $N$ sites arranged on the spatial circle, we then have

$$\delta = \mathcal{L}/N. \tag{174}$$

The Hamiltonian then takes the form of $N$ coupled harmonic oscillators,

$$H = \sum_{a=1}^{N} \left( \frac{\delta}{2} P_a^2 + \frac{m^2}{2\delta} Q_a^2 + \frac{1}{2\delta^3} (Q_a - Q_{a+1})^2 \right), \tag{175}$$

with the redefined canonical variables[27]

$$Q_a = \phi(x_a)\,\delta \quad \text{and} \quad P_a = \pi(x_a), \tag{176}$$

where $Q_{N+1} := Q_1$ and $P_{N+1} := P_1$. Passing to Fourier space gives

$$H = \sum_{k=0}^{N-1} \left( \frac{\delta}{2} \tilde{P}_k \tilde{P}_{N-k} + \frac{1}{2\delta} \left( m^2 + \frac{4}{\delta^2} \sin^2 \frac{\pi k}{N} \right) \tilde{Q}_k \tilde{Q}_{N-k} \right), \tag{177}$$

where we have the Fourier transformed variables

$$\tilde{Q}_k = \frac{1}{\sqrt{N}} \sum_{a=1}^{N} e^{\frac{2\pi i k a}{N}} Q_a \quad \text{and} \quad \tilde{P}_k = \frac{1}{\sqrt{N}} \sum_{a=1}^{N} e^{-\frac{2\pi i k a}{N}} P_a. \tag{178}$$

Note that $\tilde{P}_k^\dagger = \tilde{P}_{N-k}$ and $\tilde{Q}_k^\dagger = \tilde{Q}_{N-k}$. Furthermore, we have $\tilde{Q}_N = \tilde{Q}_0$ and $\tilde{P}_N = \tilde{P}_0$, which implies that $\tilde{Q}_0$ and $\tilde{P}_0$ are always real, and $\tilde{Q}_{N-k} = \tilde{Q}_{-k}$ and $\tilde{P}_{N-k} = \tilde{P}_{-k}$. If $N$ is even, we also have $\tilde{Q}_{N/2}^\dagger = \tilde{Q}_{N/2}$ and $\tilde{P}_{N/2}^\dagger = \tilde{P}_{N/2}$, which are therefore also real. The canonical commutation relations are given by $[\tilde{Q}_k, \tilde{P}_\ell^\dagger] = i\,\delta_{k\ell}$. If we define the frequency

$$\omega_k = \left( m^2 + \frac{4}{\delta^2} \sin^2 \frac{\pi k}{N} \right)^{1/2}, \tag{179}$$

our Hamiltonian (177) takes the form

$$H = \sum_{k=0}^{N-1} \left( \frac{\delta}{2} |\tilde{P}_k|^2 + \frac{\omega_k^2}{2\delta} |\tilde{Q}_k|^2 \right), \tag{180}$$

which is a sum of $N$ independent harmonic oscillators[28] of equal mass $M = \delta^{-1}$ (not to be confused with the physical mass $m$), but $k$-dependent frequencies $\omega_k$, cf. eq. (10). It is important to point out that when $m = 0$, the frequency $\omega_0$ vanishes and, as a result, the zero mode ($k = 0$) Hamiltonian does not have a normalizable ground state. Of course, the case of $m = 0$ represents the conformal limit, and as such is the case of primary interest in our paper, since we want to compare with holographic CFTs. The most straightforward means of dealing with the potential problems stemming from the presence of this zero mode is to regulate its behaviour by introducing a small but non-vanishing mass (*e.g.*, $m \ll 1/\mathcal{L}$), which we shall do below. We can then obtain results with decreasing values of this IR regulator.

---

[27]The scaling here ensures that $Q_a$ and $P_a$ have the usual dimensions of positions and momenta, respectively.

[28]Note that our variables $\tilde{Q}_k$ and $\tilde{P}_k$ are complex, except for $k = 0$ and $k = N/2$ (for even $N$). The two complex degrees of freedom labeled by $(\tilde{Q}_k, \tilde{P}_k)$ and $(\tilde{Q}_{N-k}, \tilde{P}_{N-k})$ only contain two real degrees of freedom, because they are subject to the constraints $\tilde{Q}_k^\dagger = \tilde{Q}_{N-k}$ and $\tilde{P}_k^\dagger = \tilde{P}_{N-k}$. When expressing the Hamiltonian in terms of the real and imaginary parts of these variables, we find two independent real harmonic oscillators with common frequency $\omega_k$. This is discussed in more detail in appendix D.

We take the reference state to be the ground state of the ultralocal Hamiltonian

$$H_{\text{R}} = \int_{-\mathcal{L}/2}^{\mathcal{L}/2} dx \left( \frac{1}{2} \pi(x)^2 + \frac{1}{2} \mu^2 \phi(x)^2 \right), \tag{181}$$

where the spatial derivative term is absent, cf. eq. (173) and where $\mu > 0$ takes the role of the mass of this fictitious Hamiltonian. After performing the above discretization and Fourier transform, we arrive at

$$H_{\text{R}} = \sum_{k=0}^{N} \left( \frac{\delta}{2} |\tilde{P}_k|^2 + \frac{\mu^2}{2\delta} |\tilde{Q}_k|^2 \right). \tag{182}$$

Hence the momentum modes remain decoupled and the reference state is a simple product over the momentum modes of Gaussian wavefunctions, all with a fixed width set by $\mu$. Of course, the ground state of the physical Hamiltonian (180) has the same form where the variance of each mode is set by $\omega_k$. Since each mode $k$ is decoupled from the other modes, the respective TFD state will be the product of TFD states for each of the oscillators. This brings us to the setup of subsection 2.2.2, after we make the following identifications for each mode:

$$M = \delta^{-1}, \quad \omega = \omega_k, \quad \alpha_k = \frac{1}{2} \log \left( \frac{1 + e^{-\beta \omega_k/2}}{1 - e^{-\beta \omega_k/2}} \right). \tag{183}$$

The dimensionless ratios $\lambda$ and $\lambda_{\text{R}}$ in eq. (40) for each mode now take the form

$$\lambda = \frac{\omega_k}{\delta \omega_g^2} \quad \text{and} \quad \lambda_{\text{R}} = \frac{\mu}{\delta \omega_g^2}. \tag{184}$$

It may seem curious that these coefficients in eq. (184) seem to depend on the lattice spacing $\delta$. However, it turns out that the natural gate scale must be modified when working with the field theory as follows: for a general spacetime dimension, the relation (176) becomes $Q_a = \delta^{d/2} \phi(x_a)$ and $P_a = \delta^{d/2-1} \pi(x_a)$. Following the structure in eq. (38), we express the quantum circuits of interest as

$$\begin{aligned}
U(\sigma) &= \exp \left[ i \int_0^{\sigma} ds \sum \left\{ Y^{ab}(s) \omega_g^2 Q_a Q_b + \frac{1}{2} \widehat{Y}^{ab}(s) (Q_a P_b + P_b Q_a) + \widetilde{Y}^{ab}(s) \frac{P_a P_b}{\omega_g^2} \right\} \right] \\
&= \exp \left[ i \int_0^{\sigma} ds \int d^{d-1}x_1 \int d^{d-1}x_2 \left\{ y(x_1, x_2, s) \mu_g \phi(x_1) \phi(x_2) \right. \right. \tag{185} \\
&\quad \left. \left. + \frac{1}{2} \hat{y}(x_1, x_2, s) (\phi(x_1) \pi(x_2) + \pi(x_2) \phi(x_1)) + \tilde{y}(x_1, x_2, s) \frac{\pi(x_1) \pi(x_2)}{\mu_g} \right\} \right].
\end{aligned}$$

Note that in going from the lattice to the continuum expressions, we have absorbed a factor of $1/\delta^{d-1}$ into the control functions, and with this choice, the three functions $y$, $\hat{y}$, and $\tilde{y}$ all have the same dimensions, i.e., length$^{-(d-1)}$ for the field theory in $(d-1)$ spatial dimensions.[29] In the continuum, we have also defined the gate scale as

$$\mu_g \equiv \delta \omega_g^2, \tag{186}$$

which naturally appears in the $\phi^2$ and $\pi^2$ gates. This also absorbs the lattice spacing $\delta$ in eq. (184). That is, the dimensionless ratios in eq. (184) reduce to

$$\lambda = \frac{\omega_k}{\mu_g} \quad \text{and} \quad \lambda_{\text{R}} = \frac{\mu}{\mu_g}. \tag{187}$$

---

[29]Hence in the continuum circuit, the control functions have the same dimension as a $\delta$-function.

Additionally, note that our most heavily analyzed case above, namely $\lambda_R = 1$, now equates the new gate scale with the reference scale, *i.e.,* $\mu_g = \mu$, and gives $\lambda = \omega_k/\mu$.

We can also take the limit of a large chain, $\mathcal{L} \gg \delta$ (*i.e.,* $N \gg 1$ while keeping $\delta$ fixed), in which case the system becomes infinite and we obtain a lattice-regularized quantum field theory living on an infinite line,

$$H = \int_{-\frac{\pi}{\delta}}^{\frac{\pi}{\delta}} dp \left( \frac{\delta}{2} |\tilde{P}_p|^2 + \frac{\omega_p^2}{2\delta} |\tilde{Q}_p|^2 \right), \tag{188}$$

where

$$\omega_p^2 = m^2 + \frac{4}{\delta^2} \sin^2 \left( \frac{p\,\delta}{2} \right). \tag{189}$$

In these expressions, we have introduced the continuous label $p \in [-\frac{\pi}{\delta}, \frac{\pi}{\delta}]$, defined as

$$p := \frac{2\pi k}{N\,\delta} \tag{190}$$

for $\tilde{Q}_p$, $\tilde{P}_p$. The range of $p$ corresponds to the Brillouin zone familiar from the physics of crystals, see *e.g.,* ref. [83]. What we have done is introduce a UV-regularization and mode decomposition in which the Hamiltonian of a continuous quantum-many body system becomes a sum over independent harmonic oscillators (bosonic modes). In subsequent parts of this section, we will calculate the complexity for a quantum field theory by simply adding up (or integrating over) contributions for each bosonic mode.

Before we conclude this subsection, it is worth commenting on a different regularization scheme introduced for the purposes of continuous multiscale entanglement renormalization ansatz (cMERA) in ref. [84] (see also refs. [85, 86]), which was used in the context of QFT complexity in ref. [51]. In this approach, the UV regulator is introduced by modifying the Hamiltonian of the free quantum field theory in momentum space, such that the ground state behaves at large momenta $|p| > \Lambda$ as a product state, *i.e.,* the ground state of the ultralocal Hamiltonian (181). In particular, taking the reference state to be the ground state of an ultralocal Hamiltonian (182) and working in momentum space, one can truncate the momentum integrals at $|p| = \Lambda$ when computing the complexity, since for higher momenta the state already has the form of the target state.

We extend this approach to the complexity of the thermofield double by requiring that we only reproduce the TFD state up to a cutoff scale $\Lambda$ in momentum. This means that we can again cut our momentum integrals at $p = |\Lambda|$. The result with this regulator can be easily obtained by starting with the previous lattice regularization and placing the new regulator $\Lambda$ far below the scale of the lattice spacing $\Lambda \ll \pi/\delta$, such that upon sending $\delta \to 0$, the result remains finite and regulated by the new scale $\Lambda$. In light of eq. (188), and using the fact that $\Lambda \ll \frac{\pi}{\delta}$, we may linearize the sine function in eq. (189) for $\omega_p$ as

$$\omega_p = \sqrt{m^2 + p^2} \quad \text{for} \quad |p| \le \Lambda. \tag{191}$$

We may also demand that the frequency of the oscillator is continuous at the transition point, *i.e.,* $\omega_{p=\Lambda} = \mu$ where $\omega_p$ was defined in eq. (189).

Of course, as alluded in the discussion around eq. (185), the analysis discussed here generalizes in a straightforward manner to higher dimensions. In the Hamiltonian (188), one then replaces the integral over the range $[-\frac{\pi}{\delta}, \frac{\pi}{\delta}]$ by an integral over the corresponding $(d-1)$-dimensional hypercube. When calculating complexity in the cMERA-inspired approach, one replaces the one-dimensional integral over momenta by an integral in momentum space confined to the ball of radius $\Lambda$ centered at the origin.

Lastly, we would like to comment that most of the observables in which we will be interested are regularized quantities, and will exhibit exponential suppression for momenta larger than the temperature scale. Hence in this case the details of the regularization scheme will not matter as long as $\beta \gg \delta$ (or $\beta \gg 1/\Lambda$).

## 5.2 Warm-up: complexity as a function of time with $\lambda_{\text{R}} = 1$ on the lattice

As a warm-up for both the complexity calculations in field theory and the analysis of entanglement in section 6, we present here representative lattice results in $(1 + 1)$-dimensions. For concreteness, we again choose the gate scale such that $\lambda_{\text{R}} = 1$ and focus on the $\kappa = 2$ cost function, see eqs. (9) and (150). We will consider three cases: keeping the total size $\mathcal{L}$ of the system fixed, *i.e.,* working on a circle, see figs. 6 and 7; keeping the lattice spacing fixed and increasing the total number of lattice sites; and working with infinite system at fixed lattice spacing—see fig. 8 for these last two cases. The motivation to study these three cases is, first, to isolate finite size effects and, second, to see how well the cMERA-type regularization in eq. (191) agrees with the answer on an infinite lattice.

As mentioned above, when $\lambda_{\text{R}} = 1$ the parameter $\lambda$ for each mode becomes $\omega_k/\mu$, see eq. (187). We now have to add contributions from all modes with each mode contributing as in eq. (150), see also eq. (142). For a finite lattice this gives

$$\mathcal{C}_{\kappa=2} = \frac{1}{4} \sum_{k=0}^{N-1} \sum_{\pm} \log^2 \left\{ f_k^{(\pm)} + \sqrt{\left( f_k^{(\pm)} \right)^2 - 1} \right\}, \tag{192}$$

where for compactness we have used the identity $(\cosh^{-1} x = \log[x + \sqrt{x^2 - 1}])$ and defined

$$f_k^{(\pm)} := \cosh 2\rho_{1\pm} = \frac{1}{2} \left( \frac{\mu}{\omega_k} + \frac{\omega_k}{\mu} \right) \cosh 2\alpha_k \pm \frac{1}{2} \left( \frac{\mu}{\omega_k} - \frac{\omega_k}{\mu} \right) \sinh 2\alpha_k \cos \omega_k t. \tag{193}$$

Here, $\alpha_k$ is given by eq. (183), $\omega_k$ by eq. (179), and $\mu$ is the QFT gate scale defined in terms of eqs. (181) and (183), see also the discussion below eq. (187).

In order to make contact with holography, we are primarily interested in CFTs (*i.e.,* massless models). However, the massless limit of eq. (192) is ill-defined, since the zero mode gives a divergent contribution. As explained above, we regulate this divergence by instead working with a small but non-zero mass. In most of the analyzed cases, we chose it to be $m = 10^{-6}/\mathcal{L}$, where $\mathcal{L}$ is the circumference of the circle. Finally, the continuum limit on a circle can be approached by keeping the size of the circle and other parameters fixed while increasing the number of points. In order to UV regulate our result for the complexity, we subtract its value at the initial time $t = 0$.

In fig. 6, we look at complexity as a function of time for theories close to the conformal limit for different temperatures. We measure time in units of the inverse temperature $\beta$, since ultimately we want this to be the dominant scale. We observe that at low temperatures the intermediate late time behaviour is dominated by a logarithmic growth, *i.e.,* by a term proportional to $\log^2 t/\beta$ (which we associate with the zero mode below), whereas for higher temperatures we see initially propagating wave packets on a circle superimposed with the logarithmic growth and ultimately, at large temperatures, saturation. Subsequent results in subsection 5.4 applicable to a line are consistent with the finding that the zero mode becomes subdominant in the large temperature limit.

In fig. 7, we investigate the effect of the IR regulator mass on the time-dependence at low and intermediate temperatures. What we see is that decreasing the mass allows us to recover more and more of the logarithmic growth, and the transition to the oscillatory regime occurs at times inversely proportional to the mass. Therefore, we interpret the logarithmic growth as a feature associated with the zero mode. Let us analyze how the logarithmic growth arises

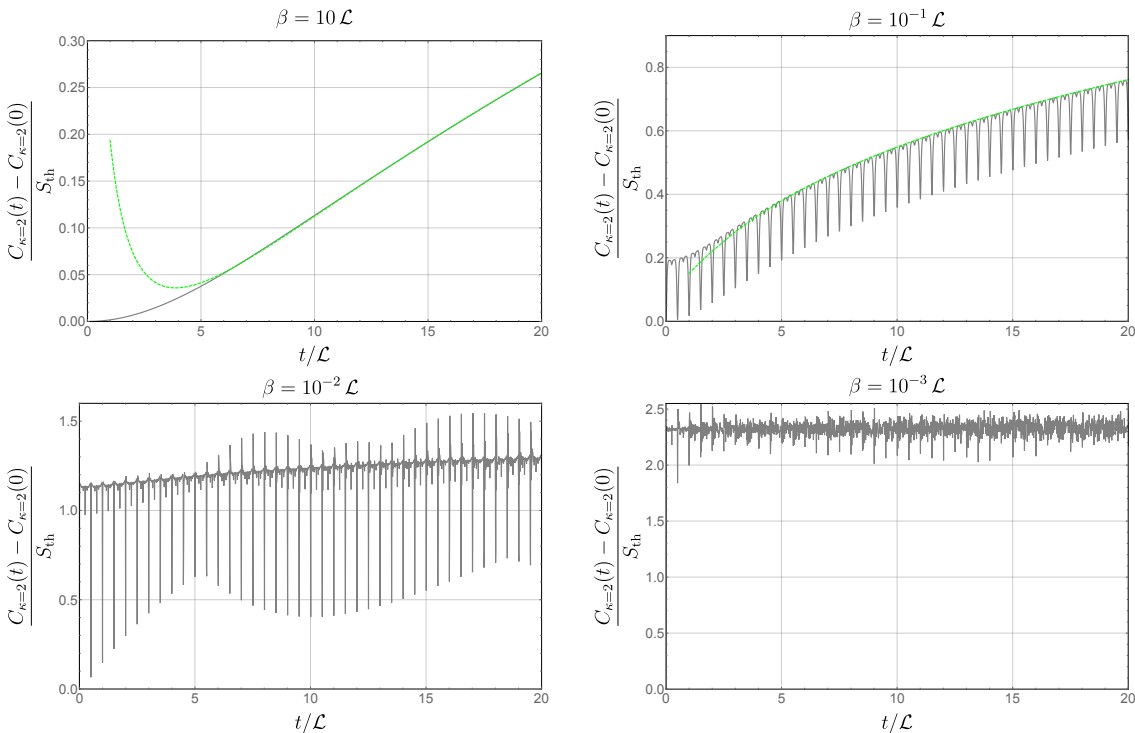

Figure 6: Grey curves represent time dependence of $\kappa = 2$ complexity with the initial value subtracted for the TFD on a circle with circumference $\mathcal{L}$ with $\mu = 1/\mathcal{L}$, $m = 10^{-6}/\mathcal{L}$ and increasing temperatures $T = 1/\beta$. We use 1601 lattice sites on each side. For smaller temperatures (top two plots), we observe a late-time behaviour proportional to $\log^2(t/\mathcal{L})$. The green dotted curve demonstrates an excellent fit of $a_2 \log^2(t/\mathcal{L}) + a_1 \log(t/\mathcal{L}) + a_0$ to the full function (upper left plot) and its maxima (upper right plot) for later values of the time. For higher temperatures (bottom two plots), we observe saturation. Transitioning between these two regimes are oscillations, which occur with a period of half of the circle's circumference, as if two wave packets were propagating on a circle with the speed of light in opposite directions. One should think of the saturation as resulting from the presence of many modes non-trivially contributing to the sum (192) at high temperatures, where the zero mode contribution becomes subdominant when dividing by the thermal entropy which scales with the temperature, see the discussion around eq. (197).

primarily from the zero mode contribution in eq. (192) (*i.e.*, only counting the contribution from $k = 0$). In the limit where both $m/\mu$ and $\beta m$ are small parameters, the coefficients $f_0^{\pm}$ simplify to

$$f_0^{\pm} \approx \frac{\mu}{\beta m^2}\left(1 \pm \cos mt\right). \tag{194}$$

Now, if we expand for large times (for instance, compared to the temperature or the ultralocal mass $\mu$), but still such that $mt$ is small, we have

$$f_0^+ \approx \frac{2\mu}{\beta m^2} - \frac{\mu t^2}{2\beta} \qquad \text{and} \qquad f_0^- \approx \frac{\mu t^2}{2\beta}. \tag{195}$$

Notice that as long as the condition $mt \ll 1$ is satisfied, the effective dimensionless quantity associated with the time dependence becomes to leading order $\frac{\mu t^2}{2\beta}$. Therefore, in this particular

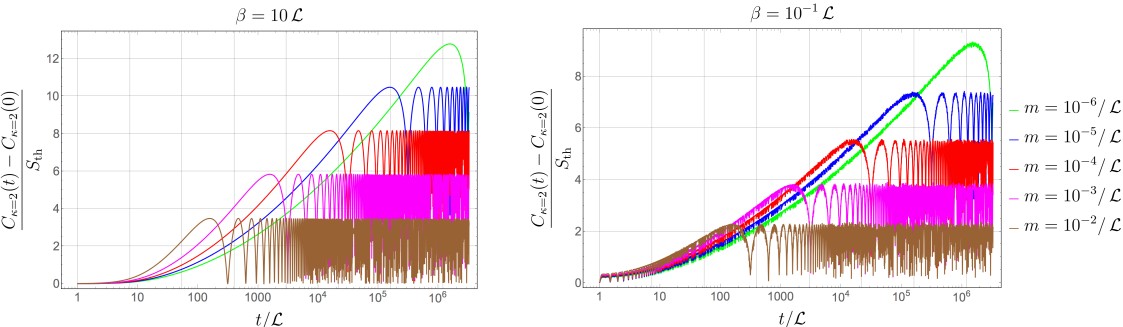

Figure 7: Time dependence of $\kappa = 2$ complexity with the initial value subtracted for a thermofield double state on a circle with circumference $\mathcal{L}$ with $\mu = 1/\mathcal{L}$, $\beta = 10\,\mathcal{L}$ (left) and $\beta = 0.1\,\mathcal{L}$ (right) and different decreasing masses, from $10^{-2}/\mathcal{L}$ to $10^{-6}/\mathcal{L}$. Similarly to fig. 6, we use 1601 lattice sites on each side. We see that complexity grows as $\log^2(t/\mathcal{L})$ up to times of the order of $1/m$ when it starts oscillating around the saturated value. We interpret this growth as originating from the presence of a zero mode in this setup. As should be apparent from fig. 6, the contribution from the zero mode becomes subdominant at large temperatures when effectively we make the circle size very large.

regime, eq. (192) for the zero mode reads

$$\mathcal{C}_{\kappa=2,\,k=0} \approx \frac{1}{4}\left(\log^2\left[\frac{\mu t^2}{\beta}\right] + \log^2\left[\frac{4\mu}{\beta m^2}\right]\right), \qquad (196)$$

which corresponds to the behaviour shown in figs. 6 and 7.

The last thing that we investigate in this subsection is the comparison between the $\kappa = 2$ complexity on a circle with fixed lattice spacing $\delta$, the analogous lattice calculation on the line, and a calculation using the cMERA inspired regularization, see subsection 5.4. As one sees in fig. 8, all of the approaches beautifully agree in their overlapping domains of applicability, *i.e.,* for times which are not so large that one becomes sensitive to finite size effects.

Let us briefly comment on why the dependence of complexity on the zero mode is not present in the decompactification limit, where the circumference of the circle becomes infinite.[30] The maximum complexity is reached at times $\pi/(2m)$, which from eq. (194) implies that the contributions from $f_0^{\pm}$ should be essentially the same. The complexity difference at times 0 and $\pi/(2m)$ then diverges logarithmically with the small masses $m$,

$$\mathcal{C}_{\kappa=2,\,k=0}(\pi/2m) - \mathcal{C}_{\kappa=2,\,k=0}(0) \approx \frac{1}{4}\left(2\log^2\left[\frac{2\mu}{\beta m^2}\right] - \log^2\left[\frac{4\mu}{\beta m^2}\right]\right) \approx \frac{1}{4}\log^2\left[\frac{\mu}{\beta m^2}\right] + \cdots. \quad (197)$$

Naively, this equation suggests that the zero mode should dominate the growth of complexity indefinitely. However, for a decompactified system, the contribution from the other modes in the sum given by eq. (192) will result in a scaling with the volume of the system, which in the decompactification limit is taken to infinity. Therefore, if we evaluate a ratio of quantities such as the difference in complexities divided by the thermal entropy of the system, we expect that the zero mode contribution will be subleading with respect to the other modes. Of course, the thermal entropy of the system increases with the temperature and so we expect the same suppression of the zero mode in the limit of high temperatures, as we see in fig. 6. We will explicitly confirm in section 5.4 that for decompactified theories in the continuum limit, there is not an unbounded growth.

---

[30]We can also think of this as a high temperature limit, because in the continuum with an arbitrarily small IR regulator mass, the physics is controlled by the combination $\mathcal{L}/\beta = \mathcal{L}T$.

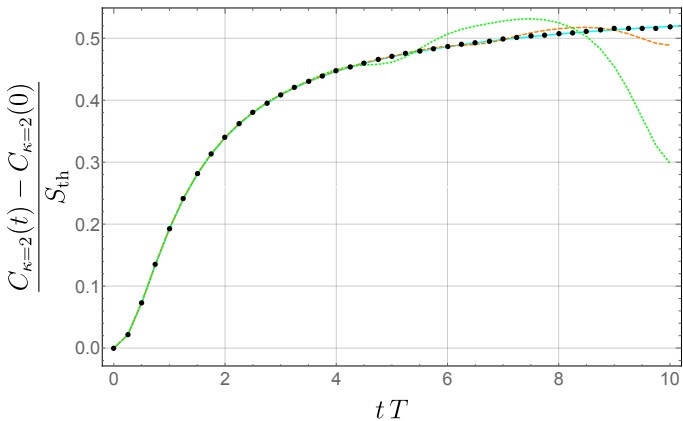

Figure 8: Time dependence of $\kappa = 2$ complexity with the initial value subtracted for a thermofield double state with fixed lattice spacing $\delta$, $m = 10^{-6}/\delta$, $\beta = 5/\delta$ and $\mu = 1/\delta$. The green curve represents the theory on a circle with the total number of sites $N = 101$ and the orange curve has $N = 201$. Cyan curve corresponds to $N = 401$ and is indistinguishable for this range of time from the results obtained using the cMERA inspired technique from subsection 5.4 with $\Lambda = 1/\delta$. Finally, black dots represent results obtained directly for a theory on an infinite line. The figure demonstrates that the decompactification limit of the circle quantitatively reproduces the results on an infinite line, and provides an example of the use of cMERA-inspired techniques in the context of complexity.

## 5.3   Complexity of formation with $\lambda_{\text{R}} = 1$

In this subsection, we evaluate the complexity of formation [27]. That is, we evaluate the extra complexity required to prepare the two copies of the scalar field theory in the TFD state compared to simply preparing each of the copies in the vacuum state. We consider the scalar in $d$ spacetime dimensions and with mass $m$, as well as the $m \to 0$ limit. For simplicity, we focus on the case $\lambda_{\text{R}} = 1$. We will start with the $L^1$-norm in the left-right (LR) basis in eq. (140), for which the complexity of formation in the continuum limit reads

$$\Delta \mathcal{C}_1^{(\text{LR})} = \text{vol} \int_{k \leq \Lambda} \frac{\mathrm{d}^{d-1}k}{(2\pi)^{d-1}} \, 2 \, |\alpha_k| = \text{vol} \int_{k \leq \Lambda} \frac{\mathrm{d}^{d-1}k}{(2\pi)^{d-1}} \, \log\left( \frac{1 + e^{-\beta \, \omega_k/2}}{1 - e^{-\beta \, \omega_k/2}} \right). \tag{198}$$

Here we have used the cMERA inspired regularization discussed at the end of subsection 5.1. One may worry that this integral will produce UV divergences. However, these potential divergences are eliminated because we have (positive) powers of $\Lambda$ competing against powers of $\exp[-\beta\Lambda]$ in these contributions, and so they actually vanish in the limit that $\Lambda \to \infty$. In fact, we can therefore remove the UV regulator altogether and integrating all the way to infinite momenta still leaves a finite result. As a result, one obtains

$$\Delta \mathcal{C}_1^{(\text{LR})} = \frac{\text{vol}}{\beta^{d-1}} f(\beta \, m), \tag{199}$$

with

$$f(x) = \frac{\Omega_{d-2}}{(2\pi)^{d-1}} \int_0^\infty \mathrm{d}u \, u^{d-2} \log\left( \frac{1 + e^{-\frac{1}{2}(u^2 + x^2)^{1/2}}}{1 - e^{-\frac{1}{2}(u^2 + x^2)^{1/2}}} \right), \tag{200}$$

where $\Omega_{d-2}$ is the volume of a $(d-2)$-sphere, i.e., $\Omega_{d-2} = 2\pi^{\frac{d-1}{2}}/\Gamma\left(\frac{d-1}{2}\right)$.

The complexity of formation for holographic CFTs in a flat decompactified space is directly proportional to the entropy, for $d \geq 3$ in both CA and CV proposals [27]. Therefore, it is natural to normalize the complexity of formation by the entropy. Let us then consider a gas of free bosons. Its partition function in $d$ dimensions reads

$$\log Z = -\text{vol} \int \frac{d^{d-1}k}{(2\pi)^{d-1}} \log\left(1 - e^{-\beta \omega_k}\right), \tag{201}$$

and hence the thermodynamic entropy is given by

$$S_{\text{th}} = \frac{\partial}{\partial T}(T \log Z) = \text{vol} \int \frac{d^{d-1}k}{(2\pi)^{d-1}} \left[\frac{\beta \omega_k}{e^{\beta \omega_k} - 1} - \log\left(1 - e^{-\beta \omega_k}\right)\right]. \tag{202}$$

We can rewrite this expression as

$$S_{\text{th}} = \frac{\text{vol}}{\beta^{d-1}} s_{\text{th}}(\beta m), \tag{203}$$

where $s_{\text{th}}(x)$ is defined as

$$s_{\text{th}}(x) = \frac{\Omega_{d-2}}{(2\pi)^{d-1}} \int_0^\infty du\, u^{d-2} \left[\frac{\sqrt{u^2 + x^2}}{e^{\sqrt{u^2+x^2}} - 1} - \log\left(1 - e^{-\sqrt{u^2+x^2}}\right)\right]. \tag{204}$$

The ratio of the complexity of formation and entropy is then simply the ratio of the two functions of $\beta m$ in eqs. (200) and (204),

$$\frac{\Delta C_1^{(\text{LR})}}{S_{\text{th}}} = \frac{f(\beta m)}{s_{\text{th}}(\beta m)}. \tag{205}$$

For the massless theory, the ratio has a simple analytic expression,

$$\left.\frac{\Delta C_1^{(\text{LR})}}{S_{\text{th}}}\right|_{\beta m=0} = \frac{2^d - 1}{d}. \tag{206}$$

As argued previously, we find a similar agreement with holography, where the complexity of formation scales like the entropy, with a dimensionless coefficient that increases with the dimension of spacetime. For the CA and CV proposals in holography, the coefficients of proportionality of complexity and entropy were [27]

$$\frac{\Delta C_A}{S_{\text{th}}} = \frac{(d-2)}{d\pi} \cot\left(\frac{\pi}{d}\right), \qquad \frac{\Delta C_V}{S_{\text{th}}} = 4\sqrt{\pi} \frac{(d-2)\Gamma\left(1 + \frac{1}{d}\right)}{(d-1)\Gamma\left(\frac{1}{2} + \frac{1}{d}\right)}. \tag{207}$$

The CA coefficient increases essentially linearly with the dimension, while the CV coefficient increases at first but very quickly saturates to a constant. The result of the massless free scalar in eq. (206) also increases with dimension, but it grows exponentially fast. Generally, comparing the QFT expression (206) with the holographic results (207), we see that in both frameworks, the complexity of formation is UV finite (*i.e.,* independent of the cutoff $\delta$), positive, and independent of the reference state scale $\mu$ (or of the normalization constant $\alpha$ or of the counterterm scale $\ell_{\text{ct}}$, in the case of holography). Of course, in both cases, we also have $\Delta C \propto S_{\text{th}}$ to leading order. However, one difference is that for $d = 2$, the holographic complexity of formation is a constant (*i.e.,* independent of the temperature), while eq. (206) is still proportional to the entropy for $d = 2$.

Next, we evaluate how the ratio of complexity of formation to entropy behaves for a massive scalar, given by eq. (205). We show in fig. 9 the numerical evaluation of how the coefficient

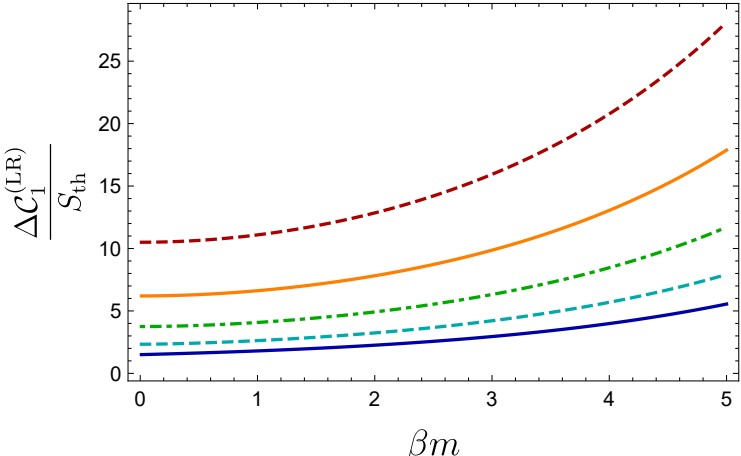

Figure 9: The complexity of formation in the left-right basis for the TFD state of a free scalar with a mass $m$. We show the dependence on the boundary dimension from $d = 2$ (bottom) to $d = 6$ (top). Despite both complexity of formation and entropy becoming exponentially small for large masses, their ratio increases exponentially as given by eq. (208).

of the complexity of formation in eq. (205) increases once the theory is massive, for various dimensions. Both the complexity of formation and the entropy go to zero as the parameter $\beta m$ increases, but the ratio increases exponentially as a function of $\beta m$. We can expand eq. (205) for large masses, resulting in

$$\frac{\Delta \mathcal{C}_1^{(\mathrm{LR})}}{S_{\mathrm{th}}}\bigg|_{\beta m \to \infty} = e^{\frac{\beta m}{2}}\left(\frac{2^{\frac{d+1}{2}}}{\beta m} + \frac{(d-5)(d+1)2^{\frac{d-5}{2}}}{\beta^2 m^2} + \mathcal{O}\left(\frac{1}{\beta^3 m^3}\right)\right). \tag{208}$$

**Comments on the diagonal basis**

We now turn our attention to evaluating the complexity of formation with the $L^1$-norm in the diagonal basis with $\lambda_{\mathrm{R}} = 1$, where each mode contributes according to eq. (140). The complexity of formation reads

$$\Delta \mathcal{C}_1^{(\pm)} = \mathrm{vol} \int_{|\vec{k}|<\Lambda} \frac{\mathrm{d}^{d-1}k}{(2\pi)^{d-1}}\left(\left|\frac{1}{2}\log\frac{\omega_k}{\mu} + \alpha_k\right| + \left|\frac{1}{2}\log\frac{\omega_k}{\mu} - \alpha_k\right| - \left|\log\frac{\omega_k}{\mu}\right|\right). \tag{209}$$

In order to evaluate the above expression, one has to study carefully how the sign changes inside each argument of the logarithms, and this behaviour depends strongly on the reference scale $\mu$. In appendix E, we break down carefully how one computes the complexity of formation in the diagonal basis for $\mu$ smaller than, equal to, and larger than the cutoff $\Lambda$ in the massless scalar theory (where $\omega_k = k$).

For now, let us focus on the case where the reference scale is higher than the cutoff ($\mu > \Lambda$) for a massless theory. In this regime, and for integration variable that ranges from 0 to $\Lambda$, $\frac{1}{2}\log\frac{k}{\mu} + \alpha_k$ changes sign at a value $k_f$ given by

$$k_f \coth\left(\frac{k_f}{4T}\right) = \mu. \tag{210}$$

There are two important limits to the above transcendental equation: when $k_f$ is very small and when $k_f$ is close to the cutoff scale $\Lambda$. Solving for the temperature in these two regimes,

we find

$$T_{c1} = \frac{\Lambda}{2} \frac{1}{\log\left(\frac{\mu+\Lambda}{\mu-\Lambda}\right)}, \qquad\qquad T_{c2} = \frac{\mu}{4}. \tag{211}$$

For $T < T_{c1}$, the arguments of all the absolute values in eq. (209) for $k$ in the range $[0, \Lambda]$ are negative. As a consequence, the complexity of formation is identically zero,

$$\Delta\mathcal{C}_1^{(\pm)}(T < T_{c1}) = 0. \tag{212}$$

Next, if the temperature is within the range $T_{c1} < T < T_{c2}$, there is a single solution $k_f$ to the transcendental equation (210) in the range $[0, \Lambda]$. We find that for $k < k_f$ the argument of the first absolute value is negative, and for $k > k_f$ the argument is positive. The complexity of formation in this situation is found by integrating only over modes larger than $k_f$,

$$\Delta\mathcal{C}_1^{(\pm)}(T_{c1} < T < T_{c2}) = \text{vol}\, \frac{\Omega_{d-2}}{(2\pi)^{d-1}} \int_{k_f}^{\Lambda} \mathrm{d}k\, k^{d-2}\left(2\alpha_k + \log\frac{k}{\mu}\right). \tag{213}$$

Finally, if the temperature is bigger than $T_{c2}$, we find that $\frac{1}{2}\log\frac{k}{\mu} + \alpha_k$ is always positive in the range of momenta $[0, \Lambda]$. Therefore, we have

$$\Delta\mathcal{C}_1^{(\pm)}(T > T_{c2}) = \text{vol}\, \frac{\Omega_{d-2}}{(2\pi)^{d-1}} \int_0^{\Lambda} \mathrm{d}k\, k^{d-2}\left(2\alpha_k + \log\frac{k}{\mu}\right). \tag{214}$$

We show in fig. 10 the integrated complexity of formation divided by the entropy for several dimensions. For small temperatures with respect to the cutoff scale, we find that the complexity of formation is exactly zero. For higher temperatures, there are some nontrivial cancellations between the circuits that introduce some dependence on $T$ and $\Lambda$ that contrasts with the LR basis and the holographic results of ref. [27]. Of course, the high temperature regime with $T \sim \Lambda$ should be regarded as unphysical and we are only presenting the results for illustrative purposes. Furthermore, we note that in this regime, the results are sensitive to the details of the UV regulator (which we simply chose as a hard cutoff on the momentum integral for the calculations here). In appendix E, we also analyze the other cases $\mu = \Lambda$ and $\mu < \Lambda$. Generally we still find that the complexity of formation in the diagonal basis vanishes for small temperatures and it is only a nontrivial function of the temperature with $T \sim \Lambda$.

**Comments on different cost functions**

Let us briefly comment on the integrated complexity of formation for different cost functions with $\lambda_R = 1$. For the $\kappa = 2$ cost function,[31] the one-mode complexity is simply given by eq. (139). The integrated complexity of formation then reads

$$\Delta\mathcal{C}_{\kappa=2} = \text{vol} \int_{k\leq\Lambda} \frac{\mathrm{d}^{d-1}k}{(2\pi)^{d-1}} 2\alpha_k^2 = \text{vol} \int_{k\leq\Lambda} \frac{\mathrm{d}^{d-1}k}{(2\pi)^{d-1}} \frac{1}{2}\left(\log\left(\frac{1+e^{-\beta\,\omega_k/2}}{1-e^{-\beta\,\omega_k/2}}\right)\right)^2. \tag{215}$$

We present the dependence on the dimension of the ratio of the complexity of formation to the entropy for different cost functions in fig. 11. We focus on the massless theory. The ratio decreases when the dimension increases for the $\kappa = 2$ cost function, in contrast to the exponential increase in the $L^1$ norm using the left-right basis, see eq. (206). In addition, the complexity of formation for the $F_2$ cost function in eq. (139) resembles the structure of the complexity of formation for the $L^1$ norm in the diagonal basis. We find again that the integrated complexity of formation in general depends on the cutoff scale $\Lambda$ and in the physically relevant regime where the temperature is much smaller than the cutoff scale, the ratio of complexity to entropy vanishes which does not match with holographic expectations.

---

[31]Recall that the complexity is basis independent with this cost function, *i.e.,* the same complexity results using either the left-right or diagonal basis in this case.

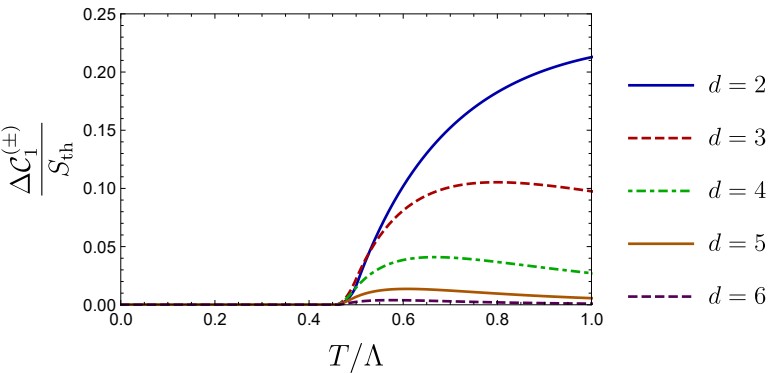

Figure 10: Complexity of formation normalized by the entropy in the diagonal basis for dimensions $d = 2, 3, 4, 5, 6$ (blue, dashed red, dot-dashed green, orange, dashed purple) and $\mu = 2\Lambda$. For $T < T_{c1}$, the complexity of formation is exactly zero (see eqs. (211) and (212)). This is the result obtained when holding the temperature fixed while sending the cutoff to infinity. For temperatures of the order of the UV cutoff, the complexity of formation in the diagonal basis develops a dependence on the temperature and the cutoff scale $\Lambda$. Of course, the results in this unphysical regime are sensitive to the UV regulator (and are only presented for illustrative purposes). The fact that the complexity of formation is either zero or not proportional to the entropy contrasts with the holographic results of ref. [27].

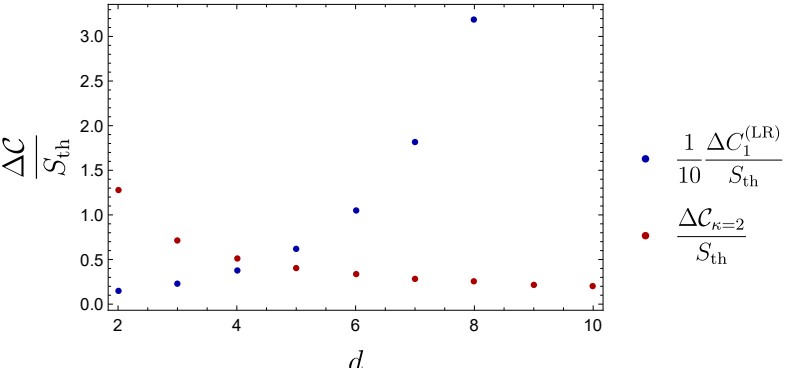

Figure 11: Comparison of the complexity of formation (normalized by the entropy) for the $\kappa = 2$ cost function and $L^1$ norm (using the left-right basis). Since the $L^1$ norm increases exponentially with $d$ in eq. (206), we divided by a factor of 10 to show both quantities in the same plot. With the $\kappa = 2$ measure, the ratio $\Delta C / S_{\text{th}}$ decreases as the spatial dimension increases, while in contrast the result for the $L^1$ norm increases.

## 5.4 Time dependence with $\lambda_{\text{R}} = 1$

Next, we investigate the time dependence of the TFD state in the continuum limit. The full time dependence in the $L^1$ norm in the LR basis with $\lambda_{\text{R}} = 1$ is given by integrating eqs. (151) and (152) over momenta.

### "Simple" limit

Let us start by studying the time dependence in the "simple" limit considered in eqs. (160) and (160). These simple results correspond taking either the limit $\lambda \to 0$ or $\lambda \to \infty$ (as well as setting $\lambda_{\text{R}} = 1$). For the first limit, it seems that in the corresponding field theory calculation, we would need to considering the case where the reference scale $\mu$ is much larger than the frequency $\omega_k$ of all possible modes. However, as we will see in a moment, very high frequency contributions are exponentially suppressed and so in fact, we need only consider $\mu \gg T = 1/\beta$. For the second limit, we would be considering $\mu \ll \omega_0 = m$, *i.e.,* the reference scale is much lower than all frequencies and so is much lower than the minimum frequency, which corresponds to the scalar field mass. In either case, we find

$$\Delta \mathcal{C}_1^{(\text{LR})} = \text{vol} \int_{k \leq \Lambda} \frac{d^{d-1}k}{(2\pi)^{d-1}} \log \left( \cosh 2\alpha_k + \sinh 2\alpha_k \, |\cos \omega_k t| \right) . \tag{216}$$

Recall that in this result, we are subtracting off the zero-temperature complexity (*i.e.,* subtracting off the contributions coming in the limit $\alpha_k \to 0$). Examining the above expression, we note that as in the complexity of formation calculation, the integrand in eq. (216) is exponentially suppressed for high energy modes. Effectively, this means we can consider the UV cutoff to be much higher than any other scale, and simply integrate up to infinity. Furthermore, this also means that the UV divergences in the complexity of the time-dependent TFD state still match those of (two copies of) the vacuum — see the discussion around eq. (198).

We present the time dependence of the complexity in this simple limit in figs. 12 and 13 for a massless theory in $d = 2$ and $d = 3$, respectively. In these plots, we present the difference between the complexity of the TFD at a given time $t$ and that at $t = 0$ by subtracting from eq. (216) its value at $t = 0$, *i.e.,* the complexity of formation, see eq. (209). We find that this difference of complexities actually decreases as a function of time, which can be understood from eq. (160). The maximum is at $t = 0$ since the contributions of all of the individual modes will take their maximum value at this time. However, the oscillating factors in eq. (160) will all become misaligned after this initial time. Hence the complexity begins by decreasing and it never recovers the maximum value. Instead, the complexity saturates at a new value that is still of the order of the entropy (for $m = 0$) on a time scale which is of the order of the inverse temperature. When summing over all modes, the exponential suppression (see discussion after eq. (152)) implies that the modes with frequencies of order $\beta$ or less are dominant in the summation. This means that the oscillations of different modes (with periodicity $2\pi/\omega_k$) will be maximally dephased after times of order $t \sim \beta$, which explains the saturation that we observe.

Of course, this is very different than what we see for holographic complexity, *i.e.,* where we see a linear growth at late times. However, this difference is not unexpected. The intuitive argument for the linear increase of the holographic complexity is that the effect of the chaotic/fast-scrambling Hamiltonian is like throwing random gates at the state, and in the large-$N$ limit, it is very unusual for a new gate to reduce the complexity. Since we have (almost) the simplest of possible Hamiltonians with the free theory in our QFT calculations, we should not expect to find analogous behaviour here — see further discussion in section 7.

Next, we investigate the influence of turning on the mass of the scalar field on the time dependence in the "simple" limit given by eq. (216). We show the time dependence of complexity

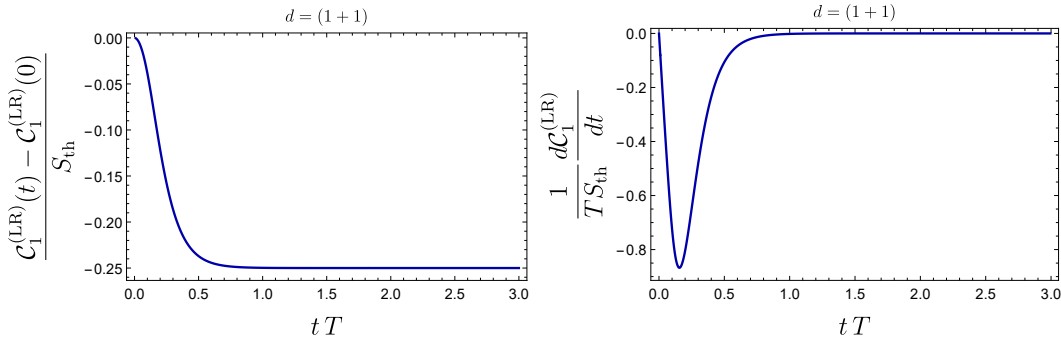

Figure 12: The time evolution of complexity of the TFD state for the $L^1$ norm in the left-right basis, for a massless scalar field in $d = 1 + 1$ dimensions. In contrast to holography, the rate of change is negative at first, then saturates to zero at times of the order of the inverse temperature.

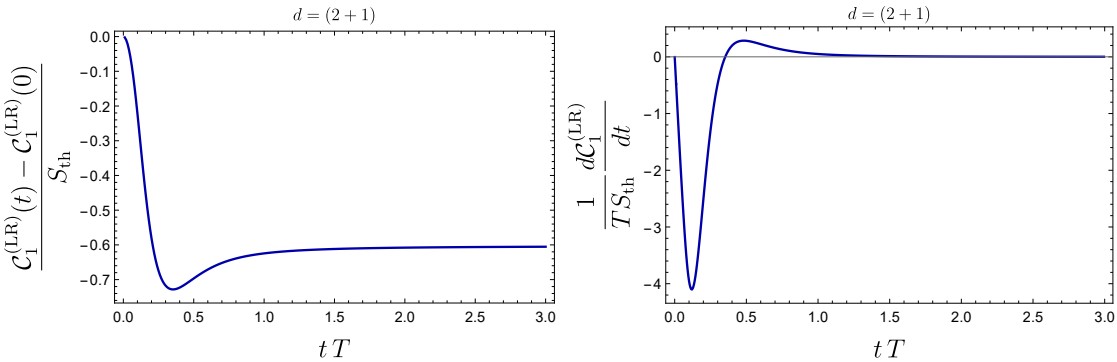

Figure 13: The time evolution of complexity of the TFD state for the $L^1$ norm in the left-right basis, for a massless scalar field in $d = 2 + 1$ dimensions. In contrast to holography, the rate of change is negative at first, then overshoots to positive values for a short amount of time, saturating to zero at times of the order of the inverse temperature.

for different masses in figs. 14 and 15 for $d = 2$ and $d = 3$, respectively. The complexity there shows damped oscillations, in particular if the mass scale is large with respect to the temperature. These oscillations arise because of the simple dependence on time $\cos \omega t$ in eq. (216), from which follows an oscillatory behaviour with period $\Delta t \approx \pi/m$.

**General reference scale**

We now turn our attention to the effect of the reference scale on the time evolution of complexity. That is, we will consider the time dependence for general values of $\lambda$. In order to take into account the reference scale, one has to integrate the complexity contributions in eqs. (151) and (152) over all momenta. For concreteness, let us focus on the massless case. We define the dimensionless variables

$$\tilde{k} := \beta k, \qquad \tilde{t} := \frac{t}{\beta}, \qquad \tilde{\gamma} := \frac{1}{\beta \mu}, \qquad (217)$$

where $\tilde{\gamma}\tilde{k}$ is simply the parameter $\lambda$ as in eq. (154). The two simple limits above are then $\tilde{\gamma} \to 0$ and $\tilde{\gamma} \to \infty$.

In figs. 16 and 17, we investigate the time evolution of complexity for different values of the dimensionless parameter related to the reference scale. For both small and large values

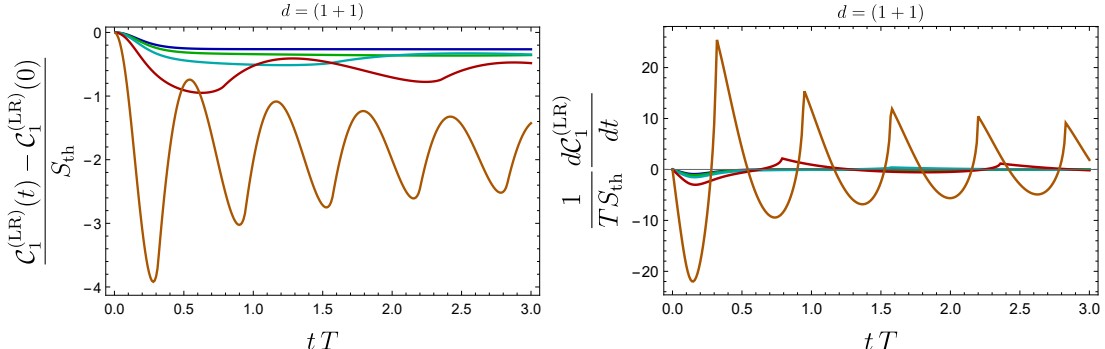

Figure 14: The time dependence of complexity for a massive theory in $d = 1 + 1$ dimensions. The complexity of the TFD state (left) and the time derivative (right) in units of the entropy of the theory, for $m = 0.1\,T$ (blue), $m = 0.5\,T$ (green), $m = 1\,T$ (cyan), $m = 2\,T$ (red) and $m = 5\,T$ (orange). For large masses with respect to the thermal scale, there is an oscillatory behaviour with period $\Delta t \approx \pi/m$. At late times, we observe a saturation to a constant value.

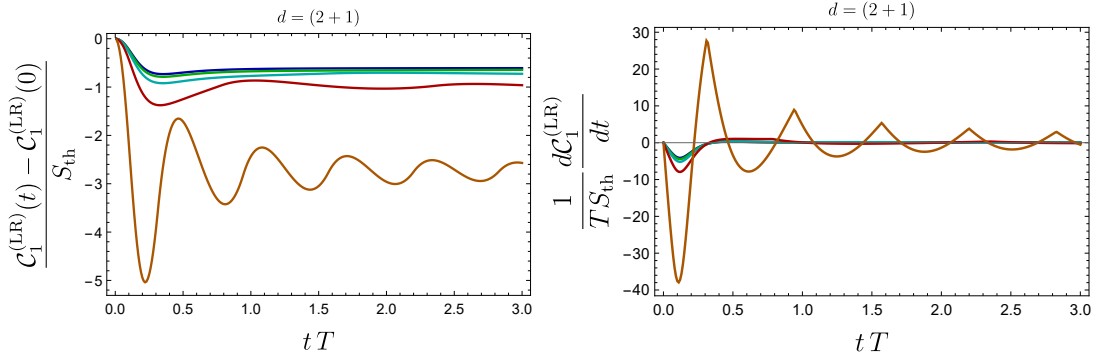

Figure 15: The time dependence of complexity for a massive theory in $d = 2 + 1$ dimensions. The complexity of the TFD state (left) and the time derivative (right) in units of the entropy of the theory, for $m = 0.1\,T$ (blue), $m = 0.5\,T$ (green), $m = 1\,T$ (cyan), $m = 2\,T$ (red) and $m = 5\,T$ (orange). For large masses with respect to the temperature scale, there is an oscillatory behaviour with period $\Delta t \approx \pi/m$. At late times, we observe a saturation to a constant value.

of $\tilde{\gamma}$, we recover a similar behaviour to the one discussed previously in figs. 12 and 13, with the complexity decreasing at first, then saturating to a constant. For intermediate values of $\tilde{\gamma}$, the complexity increases with respect to its value at $t = 0$, and then it again saturates to a constant value. Of course, despite increasing at first, the complexity does not continue increasing linearly with the energy for long times, as found for holographic complexity.

Let us comment further on another possible lesson from holography that is manifest in figs. 16 and 17. The time dependence of complexity for the TFD state dual to an eternal black hole exhibits non-universal behaviour at early times due to the normalization of the null normals to the boundary of the Wheeler-DeWitt patch in the complexity=action proposal. As discussed in ref. [28], the transient behaviour of the time derivative is controlled by a dimensionless parameter $\alpha$, as even under affine parametrization there is freedom in the overall scale of the null normals. Even if we fix reparametrization invariance by a boundary counterterm, as introduced in ref. [26] and argued to be necessary in order to reproduce desired properties of complexity in refs. [38,39], the transient behaviour is also controlled by an arbitrary dimen-

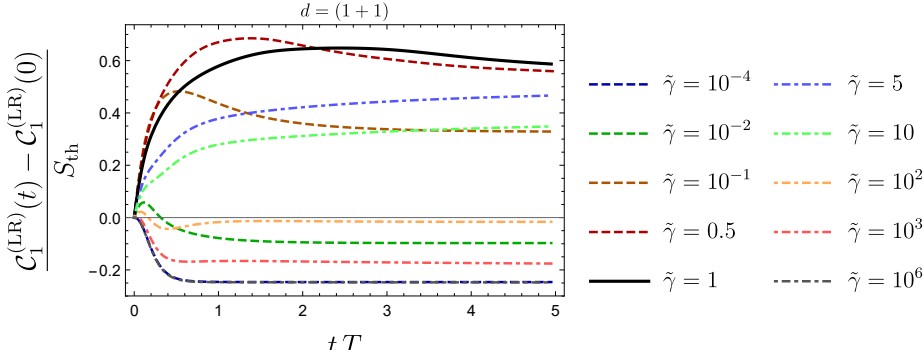

Figure 16: The time evolution of complexity with varying reference scale for the massless scalar in $d = 2$. The values of $\tilde{\gamma}$, see eq. (217), are $\tilde{\gamma} = 1$ (solid black), $\tilde{\gamma} < 1$ (dashed curves) and $\tilde{\gamma} > 1$ (dot-dashed curves). Both limits of large and small $\tilde{\gamma}$ recover the curves for the simple limit in figs. 12 and 13. By varying the reference scale, we obtain regimes where the complexity mostly increases with respect to the complexity of formation. For different values of $\tilde{\gamma}$, the complexity saturates to different constants at late times.

sionless parameter $\ell_{\rm ct}/L$.[32] In this sense, figs. 16 and 17 reproduce the intuition of holography, that at early times the evolution of complexity is dominated by non-universal effects, such as the scale of the reference state. The late time growth rate of complexity in holography was independent of these ambiguities in the null boundaries, which also seems to be a property in figs. 16 and 17. However, unlike holography, the growth rate of the complexity for the massless Gaussian TFD vanishes after times of the order of the inverse temperature. We will return to this point in the discussion in section 7.

**$\kappa = 2$ cost function**

We now investigate the time evolution for the $\kappa = 2$ cost function given by eq. (150). The integrated complexity has the simple expression

$$\Delta \mathcal{C}_{\kappa=2} = \text{vol} \int_{k \leq \Lambda} \frac{\mathrm{d}^{d-1}k}{(2\pi)^{d-1}} \left( \rho_{1+}^2 + \rho_{1-}^2 \right), \tag{218}$$

where $\rho_{1+}$ and $\rho_{1-}$ where defined in eq. (152). We show the time evolution for different reference scales $\tilde{\gamma}$ in fig. 18. Although the complexity still saturates to a constant at times of the order of a few inverse temperatures, the reference scale changes significantly the rate of change during the transient regimes at early times, and the time derivative grows as the reference scale $\tilde{\gamma}$ becomes very large or very small. This dependence on the reference scale in the time derivative is something that holographic complexity also exhibits when a counterterm is added to the null boundaries of the Wheeler-DeWitt patch (*e.g.,* see ref. [39]). Nonetheless, we should add that the UV structure for this cost function has an extra logarithmic factor compared to the UV divergences observed in holographic complexity, *i.e.,* $\mathcal{C}_{\kappa=2} \sim \frac{V}{\delta^{d-1}} \log^2(\ell/\delta)$ compared to $\mathcal{C}_{\rm holo} \sim \frac{V}{\delta^{d-1}} \log(\ell/\delta)$.

## 5.5 Comments on $\lambda_{\rm R} \neq 1$

Finally, we investigate the influence of varying the gate scale ($\lambda_{\rm R} \neq 1$) on the $D_{\kappa=2}$ cost function for a massless theory in $d = 2 + 1$ and $d = 1 + 1$ dimensions with reference scale $\mu\beta = 10$

---

[32]See, for instance, the discussion in appendix E of ref. [28] and the discussion section in ref. [39].

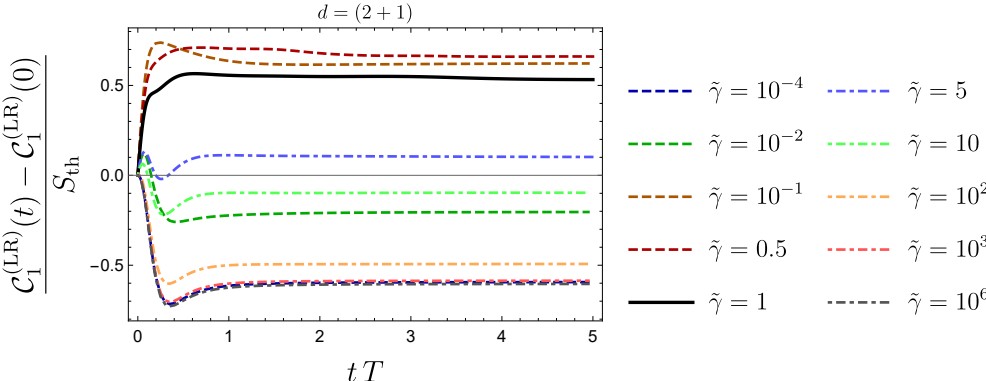

Figure 17: The time evolution of complexity with varying reference scale, for the massless scalar in $d = 3$. The values of $\tilde{\gamma}$, see eq. (217), are $\tilde{\gamma} = 1$ (solid black), $\tilde{\gamma} < 1$ (dashed curves) and $\tilde{\gamma} > 1$ (dot-dashed curves). Both limits of large and small $\tilde{\gamma}$ recover the curves for the simple limit in figs. 12 and 13. By varying the reference scale, we obtain regimes where the complexity mostly increases with respect to the complexity of formation. For different values of $\tilde{\gamma}$, the complexity saturates to different constants at late times.

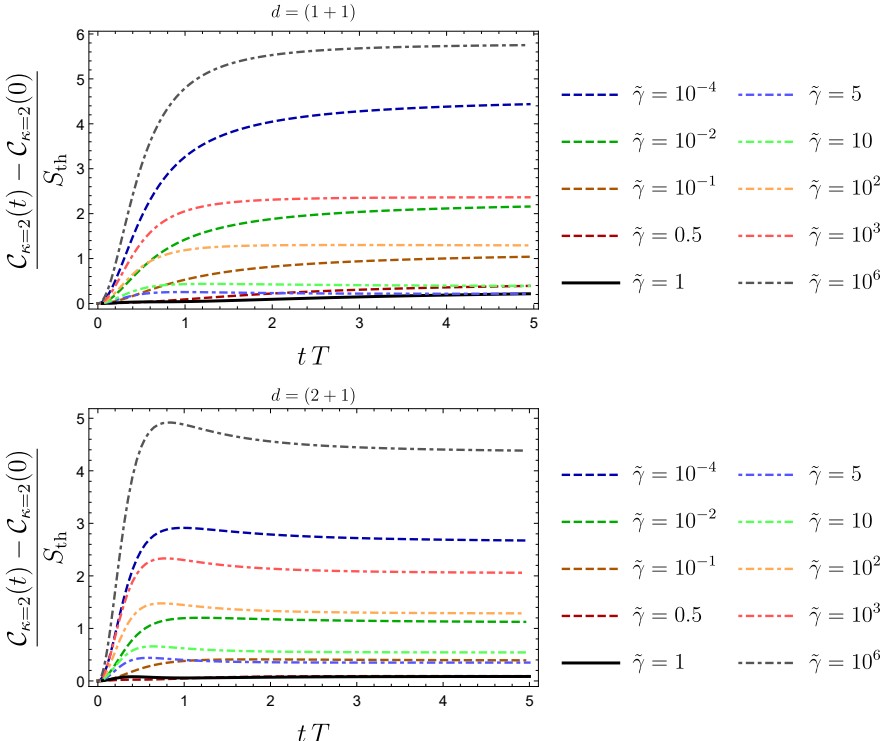

Figure 18: The time evolution of complexity with varying reference scale, for the massless scalar in $d = 2$ (top) and $d = 3$ (bottom), using the $\kappa = 2$ cost function. The values of $\tilde{\gamma}$, see eq. (217), are $\tilde{\gamma} = 1$ (solid black), $\tilde{\gamma} < 1$ (dashed curves) and $\tilde{\gamma} > 1$ (dot-dashed curves). Both limits of large and small $\tilde{\gamma}$ recover the curves for the simple limit in figs. 12 and 13. Here, we see that the complexity always increases with respect to the complexity of formation. For different values of $\tilde{\gamma}$, the complexity saturates to different constants at late times. We not that for large and small $\tilde{\gamma}$ we have a large time derivative during the transient period at early times.

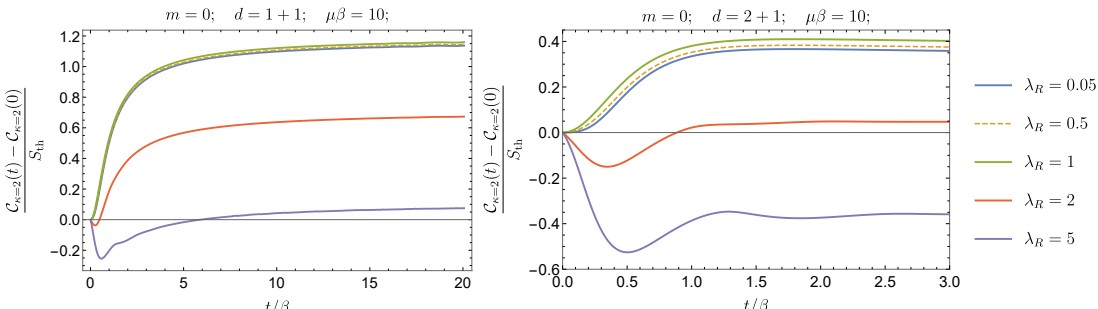

Figure 19: Integrated regularized $\kappa = 2$ complexity as a function of time for general $\lambda_R$ in 1+1 and 2+1 dimensions with $\mu\beta = 10$ and $m = 0$. Note that the $\lambda_R < 1$ curves are rather close to the $\lambda_R = 1$ curve and that the $\lambda_R = 5$ curve in $2 + 1$ dimensions is completely negative. The saturation time in $1 + 1$ dimensions is longer, similarly to what happened in fig. 18 for $\lambda_R = 1$ and $\tilde{\gamma} = 0.1$, however we stress that this is not a logarithmic growth.

in fig. 19. Varying the gate scale does not change the vanishing rate of change after a time of the order of the inverse temperature, however, it does change the growth rate at times $t \sim \beta$, as well as the final value of the complexity. The maximum value for the complexity is found for $\lambda_R = 1$, which suggests that the results we have obtained for $\lambda_R = 1$ can be understood as an upper limit on the complexity for other possible values of the gate scale. It would still be interesting to understand how the gate scale influences other properties of the complexity, such as the structure of the UV divergences, and whether there is a simple intuition as to why the circuit complexity has a maximum at $\lambda_R = 1$. We can also observe in the figure that for some values of $\lambda_R$ (in this case $\lambda_R = 5$ in $2 + 1$ dimensions) we obtain a complexity that is always lower than its value at $t = 0$ since the relevant curve is everywhere negative. When the reference scale is large, the curves are similar to those of $\lambda_R = 1$, for reasons that we do not fully understand at present. In $1 + 1$ dimensions, the transient regime is longer, similar to what happened for $\lambda_R = 1$ and $\tilde{\gamma} = 0.1$, see fig. 18. However, we stress that the plot does not exhibit a logarithmic growth and the complexity finally saturates, which we tested by plotting the exponent of the regularized complexity against time.

## 6 Entanglement production in TFD states

Among other motivations, circuit complexity was proposed as a quantity that can probe aspects of field theory states that cannot be captured by entanglement entropy.[33] In particular, when studying the thermofield double state of an interacting field theory with a holographic dual, it is expected that circuit complexity can probe the degrees of freedom which are encoded in the dual geometry deep in the interior of black holes. In holography two proposals have been made for the holographic dual of complexity – the CV and CA proposals and they both predict that the complexity keeps increasing for a very long time as long as the geometry is still well-approximated classically, *e.g.,* [13,14]. On the other hand, the entanglement entropy will not keep increasing for such a long time, and instead simply saturates at times of the order of the subregion size [3]. In this section, we will be primarily concerned with the time evolution of the entanglement entropy in the (1+1)-dimensional free field TFD state on a circle. We will consider the entanglement entropy of a subregion of the entire quantum system which

---

[33]Indeed, this was recently demonstrated in ref. [59] using a related model of complexity for quantum oscillators.

contains parts in both the left and right QFTs. However, our free field theory does not have a holographic dual and so we do not expect to see the same contrast in the behaviours of the circuit complexity and the entanglement entropy, as seen in holography. Still we believe that it is illuminating to analyze the entanglement dynamics where it can be done semi-analytically, and to compare our results with those for complexity.

## 6.1 Entanglement entropy for Gaussian states from covariance matrices

For bosonic Gaussian states, the knowledge of the covariance matrix $G_A$ for canonical coordinates supported in a subregion $A$ is equivalent to the knowledge of the reduced density matrix. Of course, $G_A$ itself admits a decomposition into normal modes,[34] and the von Neumann entropy and higher Rényi entropies of the reduced density matrix are sums of independent contributions from each of these modes. The entropies can be then computed by viewing each mode as an auxiliary thermal system.

The remaining question is then how to efficiently implement the above logic to evaluate the entanglement entropy. One can indeed easily find the expression for the entanglement entropy [87] $S(\rho_A(t))$ of the reduced density matrix $\rho_A(t)$ formed from the time-dependent global state $\rho(t) = |\psi(t)\rangle\langle\psi(t)|$, which is now a mixed quantum state. Before we perform the computation, let us start by re-emphasizing that the total subsystem consists of two spatially disconnected regions: an interval on the left $(I_L)$ and the corresponding (identical) interval on the right $I_R$, together constituting

$$A = I_L \cup I_R. \tag{219}$$

At the level of the covariance matrix of $A$, we can clearly see this decomposition in the associated block structure. The relevant covariance matrix $G_A$ can be viewed as consisting of four blocks with respect to the LR decomposition in real space. The time independent blocks $G_A^{L,L}$ and $G_A^{R,R}$ originate from the thermal density matrix (describing each side individually), reduced with respect to $I_L$ and $I_R$ (respectively), and are therefore time-independent and equal to each other because we assumed symmetric intervals. The time-dependence of the TFD state then manifests itself in the mixed L(eft)-R(ight) blocks $G_A^{R,L}(t)$ and $G_A^{L,R}(t)$. In this way, one has [78,79,87][35]

$$S(\rho_A(t)) = \frac{1}{2}\mathrm{tr}\left(s(|G_A^{1/2}(t)(i\Omega_A)G_A^{1/2}(t)|)\right). \tag{220}$$

Here, $\Omega_A$ is the symplectic matrix of the degrees of freedom belonging to $A$, $|\cdot|$ is the matrix absolute value, and the function $s : [1,\infty) \to [0,\infty)$ is defined as

$$s(\lambda) = \left(\frac{\lambda+1}{2}\right)\log\left(\frac{\lambda+1}{2}\right) - \left(\frac{\lambda-1}{2}\right)\log\left(\frac{\lambda-1}{2}\right). \tag{221}$$

$G_A$ is the principal submatrix of the covariance matrix $G$ that corresponds to $A$. In the same way, the Rényi entropies $S_q(\rho_A(t))$ for $q > 0$ can be computed by replacing the above function by $s_q : [1,\infty) \to [0,\infty)$ defined as

$$s_q(\lambda) = \frac{1}{q-1}\log\left[\frac{(\lambda+1)^q - (\lambda-1)^q}{2^q}\right]. \tag{222}$$

Of course, one recovers the familiar von Neumann entanglement entropy in the limit $S(\rho_A(t)) = \lim_{q\to 1} S_q(\rho_A(t))$. Furthermore, we find $S_2(\rho_A(t)) = \frac{1}{2}\log\det G_A(t)$, which follows from (222) with $s_2(\lambda) = \log\lambda$ and $\mathrm{tr}\log = \log\det$. Appendix G provides some useful closed form expressions for the relevant covariance matrices of TFD states.

---

[34]Note that in general, these are different from the normal modes of the full system.

[35]This means taking the absolute values of the eigenvalues $\lambda_i$ of $G_A^{1/2}(t)(i\Omega_A)G_A^{1/2}(t)$ and evaluating $S(\rho_A(t)) = \frac{1}{2}\sum_i s(|\lambda_i|)$ where the factor of 1/2 is due to the fact that the eigenvalues come in pairs.

## 6.2 Bounds on the entanglement entropy

Since our approach to computing entanglement entropy ultimately uses numerics to find the relevant symplectic eigenvalues, and we can therefore scan only a finite number of values of the underlying parameters including time, it is important to have additional guiding principles that constrain the problem at hand. A relevant question one can ask in this context is if it is possible to bound the maximal amount of entropy produced as the TFD is evolved in time.

We can use the subadditivity of the entanglement entropy applied to the decomposition (219) to upper bound $S(\rho_A(t))$ as

$$S(\rho_A(t)) \leq S(\rho_{I_L}(t)) + S(\rho_{I_R}(t)) . \tag{223}$$

The discussion in the previous subsection clarifies that $S(\rho_{I_L}(t)) = S(\rho_{I_R}(t))$, and that it is given by calculating the von Neumann entropy of a single interval in a finite temperature thermal state. Therefore, the upper bound is time-independent and depends only on the values of $m\mathcal{L}$, $\beta/\mathcal{L}$, the total number of sites and the size of the subsystem. It should be also clear that when the number of sites $N$ is sufficiently large (such as 1001 or 2001 sites, as we use in our numerics) and for large enough subsystems, subtracting from both sides of eq. (223) the initial ($t=0$) entanglement entropy gives rise to a bound that is to a good degree independent of the total number of sites and the size of the circle $\mathcal{L}$. The main advantage of the bound (223) in the context of our studies of the TFD state is that even if our numerics show a growing behaviour over a large range of times, we know that this growth will have to ultimately terminate since otherwise it would violate the inequality (223).

For completeness, we would also like to mention two other useful inequalities for entanglement entropy involving the Rényi entropy $S_2$ for $q = 2$. The first inequality, similarly to eq. (223), holds for any quantum state and is a lower bound

$$S_2 \leq S . \tag{224}$$

The second inequality is specific to Gaussian states and takes the form

$$S \leq S_2 + 2N_A(1 - \log 2) \approx S_2 + 0.614 N_A , \tag{225}$$

where $N_A$ is the number of bosonic degrees of freedom in the subregion $A$ on each side of the TFD [88]. It can be derived using $s(\lambda) \leq \log(\lambda) + (1 - \log 2)$ for all $\lambda \geq 1$. Note that, as opposed to inequality (223), the two bounds above are in general time-dependent. Note also that they can be viewed as providing a band in which the entanglement entropy necessarily resides, which is easier to calculate than the entanglement entropy itself, because we find a simple determinant in eq. (222) rather an expression involving eigenvalues. This will also be relevant for the asymptotic analysis in section 6.5, which is easier with a determinant than with the eigenvalues, for which we lack simple analytical formulas.

## 6.3 Quasiparticle picture of entanglement production

In our numerical studies in the next subsection we focus exclusively on the difference between the entanglement entropy at a given instant of time and the initial entanglement entropy normalized with respect to the thermodynamic entropy $S_{\text{th}}$ defined in eq. (202),

$$\frac{\Delta S(\rho_A(t))}{S_{\text{th}}} = \frac{1}{S_{\text{th}}} \left( S(\rho_A(t)) - S(\rho_A(0)) \right) . \tag{226}$$

Of course, the upper bound in eq. (223) now becomes

$$\Delta S(\rho_A(t)) \leq S(\rho_{I_L}(t)) + S(\rho_{I_R}(t)) - S(\rho_A(0)) . \tag{227}$$

Let us also note that $\Delta S(\rho_A(t))$ is a UV-finite quantity.

Our setup, following ref. [3], can be regarded as an unusual quantum quench involving two decoupled, albeit entangled via their initial conditions, subsystems. An important regime in quenches is the linear growth regime that occurs when the lattice spacing $\delta > 0$ is much smaller than the correlation length set in our case by $\beta > 0$, which itself is much smaller than the subsystem size $\ell$, *i.e.,*

$$\delta \ll \beta \ll \ell . \tag{228}$$

In the context of TFD states for holographic CFTs in $(1+1)$-dimensional Minkowski space,[36] ref. [3] observed in this regime a linear growth of $\Delta S(\rho_A(t))$ with the slope set by the associated thermodynamic entropy density. The growth terminates after a time equal to half the size of the interval, $t = \ell/2$, with $\Delta S(\rho_A(t))$ remaining constant afterwards. These features follow from the fact that the holographic entanglement entropy [89,90] (see, *e.g.,* ref. [91] for a review) is given by surfaces of minimal area (in the relevant case of AdS$_3$ these reduce to geodesics), and as $t$ approaches $\ell/2$ there is an exchange of dominance from geometric objects penetrating black hole interior to ones that remain outside. The latter case leads necessarily to a saturation, since the black hole exterior in the case of interest is static and equivalent geometric configurations contribute at every instance of time $t \geq \ell/2$. Of course, in this regime, the holographic entanglement entropy has precisely saturated the sub-additivity bound (223).

In standard quenches involving a rapid global change in a local Hamiltonian and a single interval of length $\ell$, the corresponding initial linear growth in $\Delta S(\rho_A(t))$ can be accurately captured using the quasi-particle picture introduced in refs. [92–94] and further developed in refs. [74,94–97]. In this picture, independent entangled pairs propagate in a ballistic fashion following an effective group velocity. This velocity is bounded from above by a Lieb-Robinson bound [94,95,98–100] (which also holds in the regime of harmonic infinite dimensional constituents discussed in ref. [95]), since any information propagation in a lattice model is upper bounded by a Lieb-Robinson bound [99,100]. The linear increase in the entanglement entropy arises from quasi-particles that move out of the interval while their partner is still inside (or alternatively, from quasi-particles created outside the interval entering while their partner remains outside).

Indeed, Alba and Calabrese in ref. [74] (see also ref. [75]) go as far as providing a formula for the change in the entanglement entropy as a function of time – which is largely independent of the underlying model as long as it is integrable – by counting the quasi-particles with a given weight contributing to the entanglement entropy for a given interval. Within the time scale of $\ell/2$, all the pairs created within the interval are distributed with one partner outside and the other inside. After this time, for some of these pairs, the second partner also leaves reducing the entanglement, in principle, but this effect is precisely counteracted by the increase produced by new quasi-particles coming into the interval. The net effect is a saturation of the entanglement entropy. At a quantitative level, this formula for standard quenches involving only one copy of a quantum many body system and homogeneous quenches is derived as follows: Suppose that we have a set of quasi-particles with labels $n$ (for the case of free QFTs, this is simply the momentum label) with a function $s_n$ characterizing their density times their contribution to the entanglement entropy. The non-trivial insight of refs. [74,75] was that the function $s_n$ can be simply related to the thermal entropy density $s_{\text{th}}$ characterizing the global equilibrium state with all conserved charges equal to the ones characterizing the post-quench set-up. Furthermore, it is assumed that pairs of quasi-particles are created, moving in opposite directions, each with velocity $v_n$ (as dictated by the conservation of momentum), without interacting with one another and that they are pairwise entangled. As a result, $s_n$ are treated as additive contributions to the entanglement entropy of a given interval with the rest

---

[36]Note again that we consider our theory to live on a Lorentzian cylinder, but when appropriate we will extrapolate our findings to Minkowski space.

of the system when one of the quasi-particles constituting a pair leaves the interval. Of course, the quasi-particle production is uniform through the interval because of homogeneity. With this set of assumptions, the problem becomes now a counting problem in which one needs to keep track of which quasi-particles are located within the interval at any given time.

Focusing on a single kind of quasi-particles of type $n$, the flux of quasi-particles across either end of the interval is proportional to $v_n t$, and because the interval has two ends, the relevant total number is proportional to $2 v_n t$.[37] Given the above logic, the contribution to the entanglement takes the form $t(2 v_n s_n)$. This process lasts until the time $t = \ell/(2 v_n)$, when the pairs created at the center the interval reach the ends and the entanglement saturates. Afterwards, with the entanglement constant, the contribution of this quasi-particle species to the total is just $s_n \ell$. Since by assumption quasi-particles are independent, the formula proposed is

$$\Delta S_{AC}(\rho_A(t)) = \sum_n \begin{cases} 2 s_n v_n t & \text{if } t < \frac{\ell}{2 v_n}, \\ s_n \ell & \text{if } t \geq \frac{\ell}{2 v_n}. \end{cases} \tag{229}$$

Of course, the contributions linear in time in the above equations are the ones driving the linear growth of entanglement. This simple formula was successfully tested in refs. [74, 75] in a variety of integrable models including free boson quenches. In the latter case, the authors considered theories where the boson mass is very large, *i.e.,* comparable to the inverse lattice spacing (which was set to one there). As a result, eq. (229) does not account for gapless or nearly gapless zero modes. In addition, the equation does not account for finite size effects when the system is confined to a circle, rather than to an infinite line.

It is possible to adapt the formula (229) to the case of the entanglement dynamics in the TFD state of free bosons living on a circle. In this case, the quasi-particle pairs are excitations of eigenmodes of a Hamiltonian with energies given by eq. (179) which move with group velocities given by[38]

$$v_n = \frac{\mathcal{L}}{2\pi} \partial_n \omega_n, \tag{230}$$

where $n$ is an integer running between 0 and $N$ (and of course, we assume $N \gg 1$). In the previous sections we have regarded the TFD state as consisting of two copies of the field theory living on two geometrically distinct spaces. Alternatively one can regard the TFD as a state living on a single space, but in which two copies of the fields reside. The relevant quasi-particle excitations would be combinations of excitations of the left and right degrees of freedom,[39] however, the simple count presented above still holds for these (more complicated) combined excitations. The only change needed is to recall that the time we used in the full Hamiltonian evolution $H_L + H_R$ for TFD states in the previous sections is not the parameter $t$ but rather $t/2$, see eq. (30). Evaluating eq. (229) with this change yields

$$\Delta S_{AC}^{\text{TFD}(t<\mathcal{L}-\ell)}(\rho_A(t)) = \sum_n \begin{cases} s_n^{\text{TFD}} v_n t & \text{if } t < \frac{\ell}{v_n}, \\ s_n^{\text{TFD}} \ell & \text{if } \frac{\ell}{v_n} \leq t < \frac{1}{v_n}(\mathcal{L}-\ell), \end{cases} \tag{231}$$

where the relevant function $s_n^{\text{TFD}}$ can be again deduced from the thermodynamic entropy density of this (double field) configuration

$$s_n^{\text{TFD}} = \frac{2}{\mathcal{L}} \left( \frac{\beta \omega_n}{e^{\beta \omega_n} - 1} - \log(1 - e^{-\beta \omega_n}) \right). \tag{232}$$

---

[37]The additional factor of proportionality needed in order to obtain the quasi-particle number is in fact the quasi-particle density which is encoded as a multiplicative factor in the function $s_n$.

[38]More precisely the group velocity is defined by differentiating $\omega_p$ in equation eq. (189) with respect to $p$ in eq. (190) and this introduces the extra factor of $\frac{\mathcal{L}}{2\pi}$ in eq. (230). Note that in the massless limit $v_n = 1$ for all species of quasi-particles.

[39]Naturally, those would be excitations of the $L + R$ and $L - R$ combinations of the fields, see eq. (22).

In the above equation, the portion within the large parentheses represents a contribution to the thermodynamic entropy of the free boson system at the inverse temperature $\beta$ from the mode $n$, cf. eq. (202), and the factor of $1/\mathcal{L}$ renders this an entropy density, and the overall factor of 2 stands for the two copies (Left and Right) of the fields.

Of course, this formula is valid only for times $t < \mathcal{L} - \ell$ before finite size effects appear. We can very easily include these by carefully tracking quasi-particles on a circle leaving and re-entering the interval, which leads to the final formula which is periodic in time with period $\mathcal{L}/v_n$ for any given species of quasi-particles[40]

$$
\Delta S_{\text{AC}}^{\text{TFD}}(\rho_A(t)) = \sum_n \begin{cases} s_n^{\text{TFD}} \, \mathcal{L} \, \text{frac}\left(\frac{v_n t}{\mathcal{L}}\right) & \text{if} \quad \mathcal{L} \, \text{frac}\left(\frac{v_n t}{\mathcal{L}}\right) < \ell, \\ s_n^{\text{TFD}} \, \ell & \text{if} \quad \ell \leq \mathcal{L} \, \text{frac}\left(\frac{v_n t}{\mathcal{L}}\right) < \mathcal{L} - \ell, \\ s_n^{\text{TFD}} \, \mathcal{L} \left(1 - \text{frac}\left(\frac{v_n t}{\mathcal{L}}\right)\right) & \text{if} \quad \mathcal{L} - \ell \leq \mathcal{L} \, \text{frac}\left(\frac{v_n t}{\mathcal{L}}\right), \end{cases} \tag{233}
$$

where frac denotes the fractional part, *e.g.,* frac$(3/2) = 1/2$. Of course in the almost-massless case $v_n = 1$ for all the different species and the entanglement production is approximately periodic, while for non vanishing value of the mass the different modes dephase and we obtain a picture that is not perfectly periodic (see fig. 21). Since eq. (233) is a phenomenological relation, it is interesting to see to what extent it quantitatively agrees with our set-up. Among other things, we will test it in the next section, where we study the dynamics of entanglement entropy numerically.

## 6.4 Numerical results

In the following, we evaluate $\Delta S(\rho_A(t))$, defined in eq. (226), numerically by using the techniques of subsections 6.1 for several values of the mass and temperature.[41] We work with each side being a circle of circumference $\mathcal{L}$ discretized into 1001 or 2001 lattice sites. We consider subsystems consisting of intervals on each side with lengths $\ell$ varying from $\ell = 0.1\mathcal{L}$ to $\ell = 0.5\mathcal{L}$ in increments of $0.05\mathcal{L}$. Since $\Delta S(\rho_A(t))$ is a UV-finite quantity, comparing the results for different numbers of lattice sites, but for the same relative subsystem size, provides us with a test of the numerical convergence of our calculations.

In fig. 20, we explore the linear regime and saturation of entanglement evolution on a circle for $N = 2001$ and $\beta = 0.01\mathcal{L}$, for masses ranging from $m\mathcal{L} = 0.001$ to $100$, where we increase the mass by a factor of 10 in each step. We plot times smaller than $\mathcal{L}/2$ where the system is not yet very sensitive to finite size effects (which will be explored in fig. 21). As expected in eq. (231), we observe a regime of linear growth that lasts until approximately $t \sim \ell$, followed by a saturation.[42] For $m\beta \ll 1$, the slope is equal with a good accuracy to the total thermal entropy density of two copies of the QFT (at inverse temperature $\beta$). Of course, this matches with the prediction of eq. (233), since for small $m$ the group velocity is $v_n = 1$ for all species of quasi-particles and the slope becomes the sum of entropy densities over all species, which is then simply the total entropy density. That is, with $v_n \simeq 1$, the prefactor in eq. (231) is given by eq. (202) divided by the spatial volume, *i.e.,* $\mathcal{L}$. For the larger masses, where $v_n$ varies significantly in the mode sum, the prefactor should not have this simple form and in fact, fig. 20 shows the slope is significantly less than $2S_{\text{th}}/\mathcal{L}$ for the largest mass $m = 100/\mathcal{L}$.

For times larger than $\mathcal{L} - \ell$, the entanglement entropy becomes very sensitive to finite size

---

[40]We would like to thank Pasquale Calabrese for pointing this out to us.

[41]In fact, we fix the values of the dimensionless parameters $m\mathcal{L}$ and $\beta/\mathcal{L}$.

[42]In fact, upon a closer examination of fig. 20, we have noticed that the linear regime does not start right away, and a different behaviour can be seen around $t = 0$. We believe this to be a manifestation of an expected quadratic growth at early times following a quantum quench, see e.g., refs. [101–105]. We thank the referee for pointing this out.

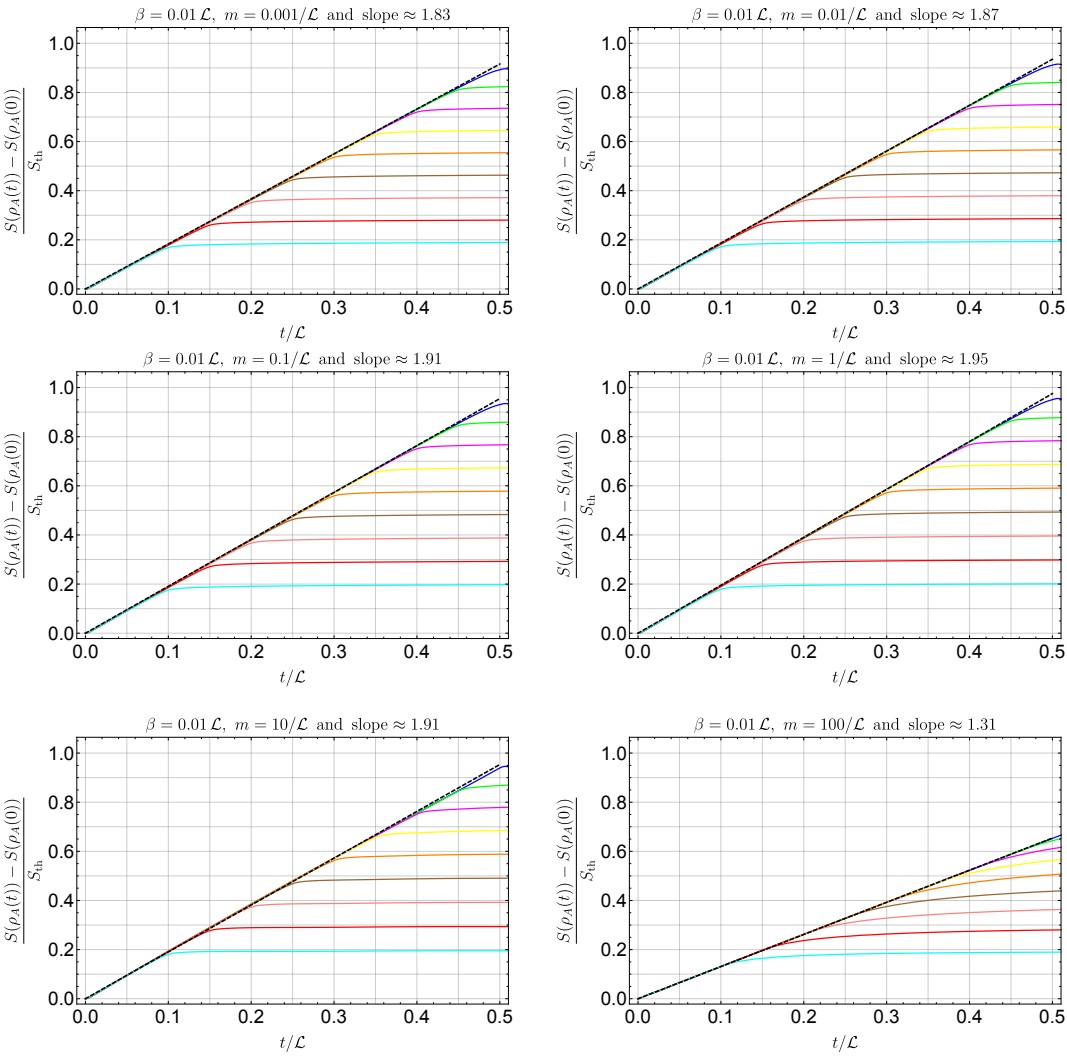

Figure 20: Linear regime of the entanglement entropy growth for $\beta = 0.01\,\mathcal{L}$ and masses from $m = 0.001/\mathcal{L}$ (upper left plot) to $m = 100/\mathcal{L}$ (bottom right plot), increasing by factors of 10, normalized by the thermal entropy of a single boson. Each plot presents the time dependence of the entanglement for intervals of varying lengths – from $\ell = 0.1\,\mathcal{L}$ (cyan, bottom curves) to $\ell = 0.5\,\mathcal{L}$ (blue, top curves) increasing in increments of $0.05\,\mathcal{L}$. Larger intervals are related to those already in the plot since the TFD is a pure state. The dashed black lines correspond to linear fits passing through the origin. We see that for small masses the slope is approximately 2. The linear regime terminates (albeit not in a sharp way for larger values of the mass) at times approximately equal to the size of an interval. The smooth transition for larger masses is due to dephasing of the different kinds of quasi-particels with different group velocities. See also fig. 21 where a longer period of time is plotted in order to study finite size effects.

effects, which manifest themselves as oscillations of periodicity $\mathcal{L}$ rather than a saturation.[43] This is depicted in fig. 21. For small masses, we observe a clear structure consisting of linear rise for a time $t \sim \ell$, followed by a plateau of width approximately $\mathcal{L} - 2\ell$ and then a linear decrease for another $t \sim \ell$. For large masses, the features of $\Delta S(\rho_A(t))$ change as time progresses, *i.e.,* the oscillations are rather irregular. We observe that for large masses and not too large intervals, the maxima of the oscillations lie close to the bound (227), see fig. 22.

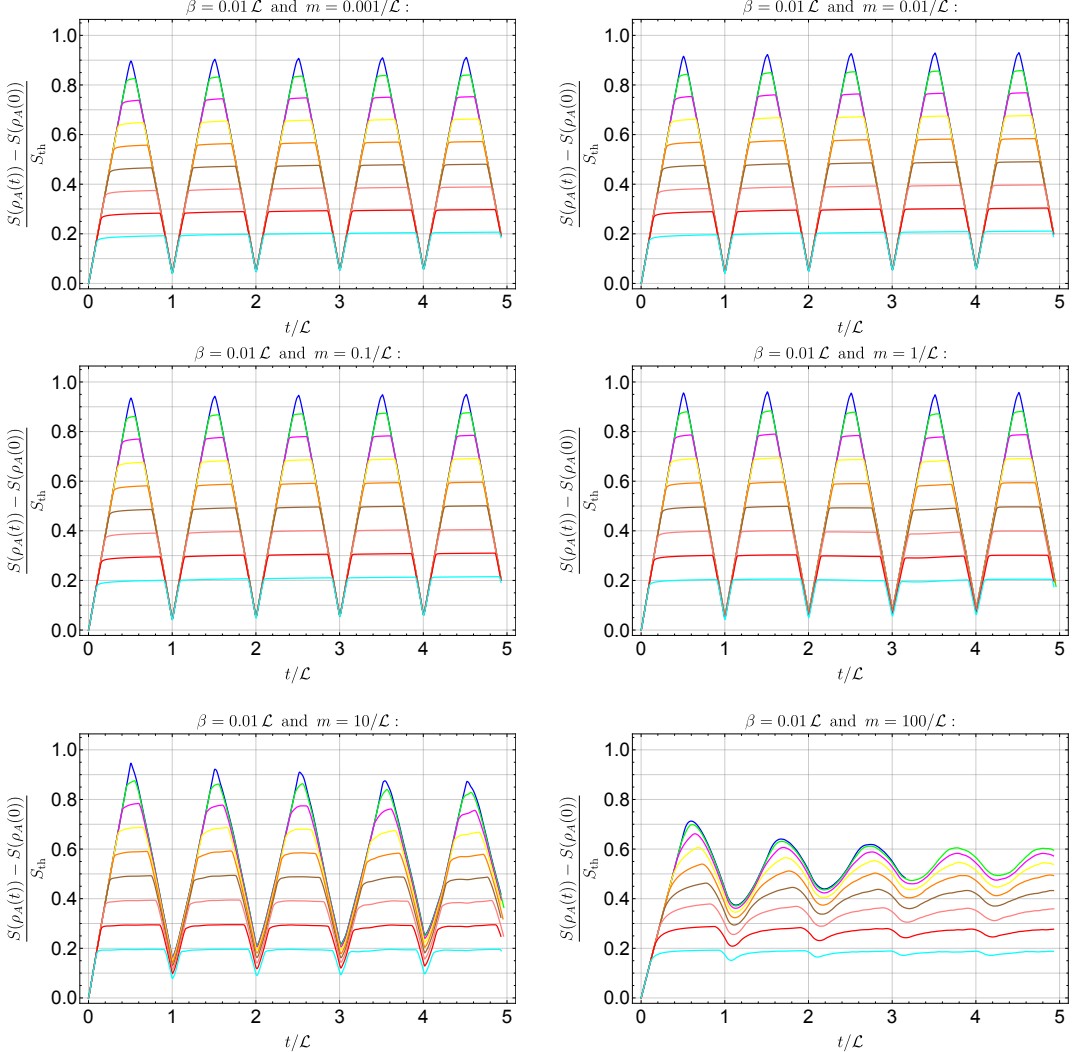

Figure 21: Data from fig. 20 plotted over a larger range of times. One sees finite size effects, which induce a periodic behaviour with periodicity $\mathcal{L}$. The plateaus for small values of the mass follow a logarithmic growth due to the presence of a zero mode, which we study further analytically in the next subsection.

Another interesting feature of the entanglement growth is hidden in the plateaus for small values of the masses. If we restrict our attention to these flat regimes, we observe a logarithmic growth with, to an excellent degree, unit coefficient, see fig. 22, where we show the relevant

---

[43]This is the time needed for a quasi-particle pair emitted in opposite directions to meet again at the opposite side of the circle, in which case they no longer contribute to the entanglement of the interval $\ell$, regardless of the position where they were created. Recall that our Hamiltonian is twice as slow and we in fact evolve the system with a time given by $t/2$.

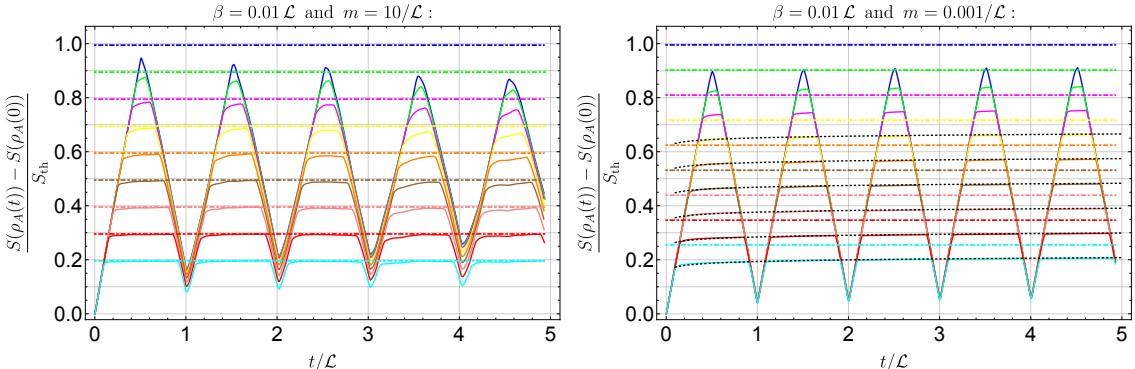

Figure 22: Time dependence of the entanglement entropy for $\beta = 0.01\,\mathcal{L}$ with $m = 10/\mathcal{L}$ (left), and $m = 0.001/\mathcal{L}$ (right) also displayed in figs. 20 and 21, plotted with the corresponding values of the bound (223) (dot-dashed curves in colours matching the relevant subsystems). For $m = 10/\mathcal{L}$ and with $\ell < 0.4\,\mathcal{L}$, we observe that the entanglement entropy reaches rather close to the upper bound in eq. (227) after approximately $t \sim \ell$. The entanglement does not, however, saturate at this value for the times we considered, but rather keeps oscillating. In the right plot, for $l < 0.4\,\mathcal{L}$, we also plot dotted black curves presenting very good fits to the quasi-plateau regimes for $t > \mathcal{L}$ that follow the behaviour (constant$+\log m\,t$)$/S_{\mathrm{th}}$ with unit coefficient in front of logarithm and with different constants for the different subsystem sizes. We interpret the latter behaviour as a manifestation of an approximately gapless zero mode on the circle as we explore further in the next subsection.

fit to the quasi-plateau regions after the first period, *i.e.,* for $t > \mathcal{L}$. Fits to the data shown in figs. 20 and 21 at $m = 0.001/\mathcal{L}$ for $\ell < t < \mathcal{L} - \ell$ point towards logarithmic growth with a smaller coefficient. In the next subsection we will develop an analytic understanding of this logarithmic growth and demonstrate that for $\ell < t < \mathcal{L} - \ell$ it has a prefactor of $1/2$. Let us also recall that a similar logarithmic growth was observed for the complexity on the circle in section 5.2.

A similar logarithmic growth of the entanglement entropy was reported earlier in ref. [106] for a global quench to a free massless bosonic quantum field theory on a circle, where it was attributed to the presence of an approximately gapless zero mode, namely the momentum mode with $k = 0$ in the massless limit.[44] The authors of ref. [106] claimed that an almost gapless zero mode does not lead to a ballistic propagation, as in the quasi-particle picture, but is instead of a diffusive nature. Since the times studied there were too short for the finite size effects to take effect, the logarithmic growth was observed with prefactor $1/2$. Indeed, in the next subsection, we confirm this statement and demonstrate how one is able to predict the logarithmic growth (with prefactors $1/2$ and $1$ for the different regimes) by studying analytically simple subsystems with a single degree of freedom on each side of the TFD. We also observed that this logarithmically growing contribution is absent if we break translational invariance by considering the system on an interval with, for example, Dirichlet instead of periodic boundary conditions.

In fig. 23, we compare our numerical predictions with the Alba-Calabrese formula [74, 75] adapted to our setup, *i.e.,* with eq. (233). We observe a remarkably good fit for the highest value of the mass we consider, namely $m = 100/\mathcal{L}$. For smaller values of the mass, even

---

[44]The authors have studied both a boundary state quench and a global mass quench focusing on cases where after the quench the mass of the scalar field is zero. See also refs. [107–109] for other relevant discussions of zero modes and entanglement entropy.

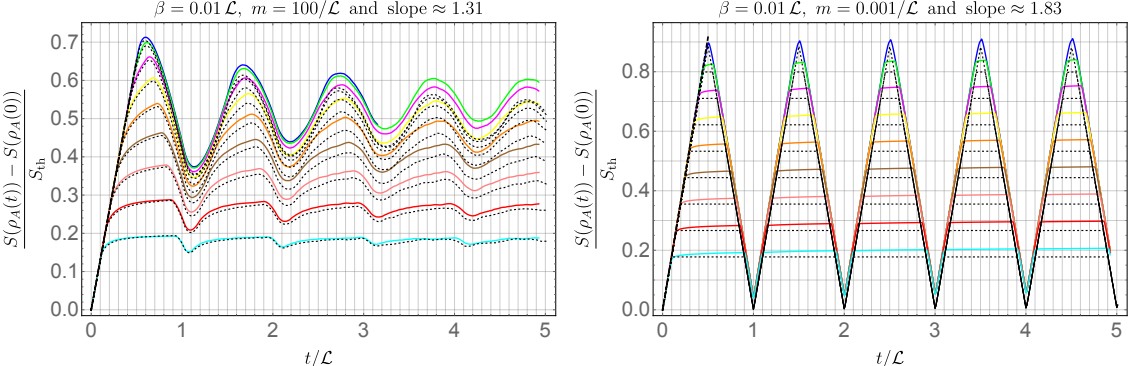

Figure 23: Top left and bottom right plots of fig. 21, superimposed with black dotted curves corresponding to the predictions of the Alba-Calabrese formula [74,75] adapted to our setup, see eq. (233). In the most massive case we considered (left), the formula works remarkably well, especially for the smaller subsystems. In particular, it predicts the slope of the linear regime to be approximately 1.30, a fraction of a percent from the observed value. For the larger subsystems we suspect that the mismatch is partially due to numerical errors. However, for lower values of masses all the way to $m = 0.001/\mathcal{L}$ depicted here, the formula predicts the slope only to about 5% accuracy (already for $m = 10/\mathcal{L}$), and does not lead to a logarithmic growth, due to not properly accounting for the zero mode which does not ballistically propagate. It would be interesting to further explore how to incorporate the zero mode effect into the Alba-Calabrese formula.

for $m = 10/\mathcal{L}$, we observe that the Alba-Calabrese formula, as it stands, misses the slope of the linear regime by about 5% and of course, due to the built-in saturation, it does not fit the logarithmic growth for $t \gg \ell$, since it does not account for a nearly gapless zero mode behaving non-ballistically. It would be very interesting to be able to account for the zero mode behaviour in a generalization of eq. (229). Note that just adding $\sim \log t$ would not work, because it behaves badly near $t = 0$. We leave these interesting issues for future work.

We can also use our derivation on the circle to obtain entanglement dynamics when the QFT lives on a line. This can be achieved by numerically studying a system on a circle with fixed lattice spacing $\delta$ and increasing the number of lattice sites $N$, while keeping the mass $m$, inverse temperature $\beta$ and subsystem size $\ell$ fixed ratios of the lattice spacing as was done for the complexity in subsection 5.2. Numerics with increasing $N$ reproduce more and more parts of a single curve which corresponds to the limit $\mathcal{L}/\delta \to \infty$. Note that this pushes the first oscillation period on the circle to larger and larger times. Fig. 24 shows these results for small masses. Note that here we observe a logarithmic growth of the entanglement entropy even when the theory lives on the line. In contrast for the complexity in section 5, we saw that the logarithmic growth initially found for the TFD on a circle did not survive in the decompactification limit.

## 6.5 Logarithmic contributions to the entanglement from the zero mode

Both refs. [74] and [106] argue that the logarithmic growth of the entanglement entropy for intermediate times could arise from logarithmic corrections due to the presence of a zero mode, *i.e.,* a mode of frequency $m \ll \mathcal{L}^{-1}$. In this subsection, we analyze the time dependence of the entanglement entropy for a single degree of freedom in the limit $m \to 0$ and find a logarithmic contribution. We then argue that the extracted asymptotic behaviour is due to

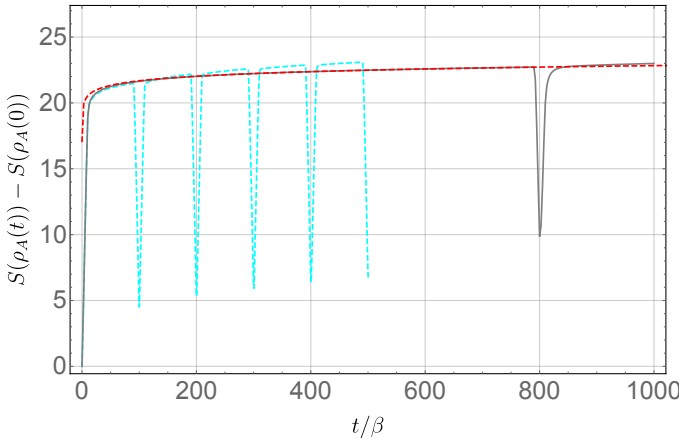

Figure 24: Entanglement dynamics on a circle extrapolated to a line with $m\beta = 10^{-5}$ and $m\ell = 10^{-4}$ (same as in the cyan curve in the top left panel of fig. 21), but now instead of $N = 2001$ as in fig. 21, we have used $N = 8001$. We see that the results of the $N = 2001$ from fig. 21 (dashed cyan curve) agree extremely well with the results of the $N = 8001$ (solid grey curve) up to the time when finite size effects kick in for the former. This obviously happens also when $N = 8001$, but at later times (this is the dip in the grey curve). Taking larger $N$ pushes finite size effects to later and later times, recovering more and more of the curve for a QFT on a line. As in fig. 22, we also observe a logarithmic growth, but now with coefficient equal to approximately 1/2 (dashed red curve), as reported earlier for standard quenches in ref. [106]. We interpret this as the effect of an almost gapless zero mode. This growth was harder to see before in fig. 21 since $\frac{1}{2}\log(t/\beta)$ becomes a good description only right before finite size effects kick in. Finally, we refer the reader to subsection 6.5 for further studies of the logarithmic growth with coefficients 1/2 and 1.

the zero mode and also applies to larger subsystem. The argument is based on overwhelming numerical evidence that the logarithmic growth with unit coefficient, as described in the previous subsection, persists for smaller subsystems, in particular for a subsystem $A$ consisting of a single local degree of freedom (single site) on each side. We comment on more general situations later. In this case of a single site $A$, we derive below the full asymptotic behaviour of the entanglement entropy in the limit $\beta, m^{-1} \ll \mathcal{L}$, which will turn out to be given by

$$S(\rho_A(t)) \sim \begin{cases} \frac{1}{2}\log(t/\mathcal{L}) + \text{const} & \delta \ll t < \mathcal{L}, \\ \log(t/\mathcal{L}) + \text{const} & \mathcal{L} < t \ll m^{-1}, \\ \log|\sin(mt)| + \text{const} & \mathcal{L} < t, \end{cases} \tag{234}$$

where the second case offers a simplification of the third case under the condition $t \ll m^{-1}$. The large time asymptotics given by oscillations with frequency $m$ is not surprising, because we already knew that the entanglement entropy cannot grow arbitrarily due to the upper bound provided by the thermal state, cf. eq. (223). Only for times $t \ll m^{-1}$ will we see the logarithmic asymptotics, while for longer times oscillations with frequency $m$ become visible. Another important regime is $t < \mathcal{L}$, which shows the characteristic behaviour for a field theory on a line ($\mathcal{L} \to \infty$) rather than circle ($\mathcal{L}$ fixed). We can see that these predictions match the different regimes in our numerical results in fig. 25.

We begin our argument by recalling from inequality (225) that for highly entangled states the von Neumann entropy $S(\rho_A)$ rapidly approaches $S_2(\rho_A) + 2N_A(1 - \log 2)$, where $S_2$ is the second Rényi entropy. In fact, we know that our error scales as $\exp(-2S_2(\rho_A))$, which means

it becomes exponentially small for highly entangled states, as explained in refs. [88,110]. The Rényi entropy of order 2 can be written as the determinant

$$S_2(\rho_A(t)) = \frac{1}{2} \log(\det(G_A(t)))\,, \tag{235}$$

which we already discussed in the context of equation (220). We will apply this formula to a single degree of freedom on each side of the TFD.

To do so, let us derive the covariance matrix for a subregion consisting of a single site at position $x$ on both sides of the TFD with respect to the dimensionless variables $\tilde{\xi}_k^a = (\tilde{q}_k^L, \tilde{q}_k^R, \tilde{p}_k^L, \tilde{p}_k^R)$, which are related to the discretized field variables in momentum space according to eqs. (37) and (176). The covariance matrix is derived by first decomposing the complex modes into real modes, as explained in appendix D. Then for each the real degree of freedom, we write the covariance matrix (104) with the substitutions $\omega \to \omega_k$ in eq. (179) and $\lambda \to \omega_k \mathcal{L}$ — see eq. (268),[45]

$$
\begin{aligned}
\tilde{G}_k^{a,b}(t) &= \langle \text{TFD}(t)|(\tilde{\xi}_k^a \tilde{\xi}_k^{\dagger b} + \tilde{\xi}_k^{\dagger b} \tilde{\xi}_k^a)|\text{TFD}(t)\rangle \\
&= \begin{pmatrix}
\frac{1}{\omega_k \mathcal{L}} \cosh(2\alpha_k) & \frac{1}{\omega_k \mathcal{L}} \cos(t\omega_k)\sinh(2\alpha_k) & 0 & -\sin(t\omega_k)\sinh(2\alpha_k) \\
\frac{1}{\omega_k \mathcal{L}} \cos(t\omega_k)\sinh(2\alpha_k) & \frac{1}{\omega_k \mathcal{L}} \cosh(2\alpha_k) & -\sin(t\omega_k)\sinh(2\alpha_k) & 0 \\
0 & -\sin(t\omega_k)\sinh(2\alpha_k) & \omega_k \mathcal{L} \cosh(2\alpha_k) & -\omega_k \mathcal{L} \cos(t\omega_k)\sinh(2\alpha_k) \\
-\sin(t\omega_k)\sinh(2\alpha_k) & 0 & -\omega_k \mathcal{L} \cos(t\omega_k)\sinh(2\alpha_k) & \omega_k \mathcal{L} \cosh(2\alpha)
\end{pmatrix}.
\end{aligned}
\tag{236}
$$

It may be surprising that this covariance matrix is real, even though the underlying operator $\tilde{\xi}_k^a \tilde{\xi}_k^{\dagger b} + \tilde{\xi}_k^{\dagger b} \tilde{\xi}_k^a$ is not Hermitian. However, if we remember $\tilde{\xi}_k^{\dagger a} = \tilde{\xi}_{-k}^a$, it follows from the translational invariance of the state that the expectation value should be invariant under the swap $k \leftrightarrow -k$, which implies that $\tilde{G}_k^{a,b}$ is real. Moreover, translational invariance also ensures that there cannot be any other cross correlations, so that the total covariance matrix is block diagonal. We can go to the position basis $\xi_x = (q_x^L, q_x^R, p_x^L, p_x^R)$ by applying the inverse Fourier transformation to $\tilde{\xi}_k^a$ given by

$$\xi_x^a = \sum_{k=1}^N \frac{1}{\sqrt{N}} e^{-i\frac{2\pi kx}{N}} \tilde{\xi}_k^a\,. \tag{237}$$

Conjugating this equation gives rise to an extra minus sign in the complex exponent. With this in hand, we can write the covariance matrix in position basis as

$$
\begin{aligned}
G_{x,y}^{a,b} &= \frac{1}{N} \sum_k e^{-i\frac{2\pi k(x-y)}{N}} \tilde{G}_k^{a,b} = \frac{1}{N} \sum_k e^{-i\frac{2\pi k(x-y)}{N}} \\
&\times \begin{pmatrix}
\frac{\cosh(2\alpha_k)}{\omega_k \mathcal{L}} & \frac{\cos(t\omega_k)\sinh(2\alpha_k)}{\omega_k \mathcal{L}} & 0 & -\sin(t\omega_k)\sinh(2\alpha_k) \\
\frac{\cos(t\omega_k)\sinh(2\alpha_k)}{\omega_k \mathcal{L}} & \frac{\cosh(2\alpha_k)}{\omega_k \mathcal{L}} & -\sin(t\omega_k)\sinh(2\alpha_k) & 0 \\
0 & -\sin(t\omega_k)\sinh(2\alpha_k) & \omega_k \mathcal{L} \cosh(2\alpha_k) & -\omega_k \mathcal{L} \cos(t\omega_k)\sinh(2\alpha_k) \\
-\sin(t\omega_k)\sinh(2\alpha_k) & 0 & -\omega_k \mathcal{L} \cos(t\omega_k)\sinh(2\alpha_k) & \omega_k \mathcal{L} \cosh(2\alpha)
\end{pmatrix},
\end{aligned}
\tag{238}
$$

where we have exploited the fact that the covariance matrix in momentum space is block diagonal. When setting $x = y$, this $4 \times 4$ matrix describes the quantum correlations and entanglement in a subsystem consisting of a single local degree of freedom on each side (left and right) of the theory. We will use this explicit representation to study the asymptotics of

---

[45]Here we use the indices $a, b \in \{L, R\}$. Further, let us note that, in contrast to the complexity, the entanglement entropy has no connection to the auxiliary gate scale. This means that we can choose any convenient scale to make the covariance matrix dimensionless. Effectively, our substitution $\lambda \to \omega_k \mathcal{L}$ replaces $\omega_g \to 1/\sqrt{\mathcal{L}\,\delta}$ or equivalently $\mu_g \to \mathcal{L}$ — see eqs. (184) and (187).

the entanglement entropy analytically, which complements our complexity studies of the TFD state. In particular, we will be able to identify the characteristic contribution of the zero mode ($m \to 0$ limit) to the production of the entanglement entropy.

Restricting to $x = y$ yields the following $4 \times 4$ covariance matrix

$$G_{x,x}^{a,b} = \frac{1}{N} \sum_{k=0}^{N-1} \begin{pmatrix} \frac{\cosh(2\alpha_k)}{\lambda_k} & \frac{\cos(t\omega_k)\sinh(2\alpha_k)}{\lambda_k} & 0 & -\sin(t\omega_k)\sinh(2\alpha_k) \\ \frac{\cos(t\omega_k)\sinh(2\alpha_k)}{\lambda_k} & \frac{\cosh(2\alpha_k)}{\lambda_k} & -\sin(t\omega_k)\sinh(2\alpha_k) & 0 \\ 0 & -\sin(t\omega_k)\sinh(2\alpha_k) & \lambda_k \cosh(2\alpha_k) & -\lambda_k \cos(t\omega_k)\sinh(2\alpha_k) \\ -\sin(t\omega_k)\sinh(2\alpha_k) & 0 & -\lambda_k \cos(t\omega_k)\sinh(2\alpha_k) & \lambda_k \cosh(2\alpha) \end{pmatrix}, \quad (239)$$

where we have defined $\lambda_k = \omega_k \mathcal{L}$. Schematically, the structure of this matrix is

$$G_{x,x}^{a,b}(t) = \begin{pmatrix} c_1 & F_1(t) & 0 & F_2(t) \\ F_1(t) & c_1 & F_2(t) & 0 \\ 0 & F_2(t) & c_2 & F_3(t) \\ F_2(t) & 0 & F_3(t) & c_2 \end{pmatrix}, \quad (240)$$

whose determinant can be explicitly computed as

$$\det\left(G_{x,x}^{a,b}(t)\right) = \underbrace{c_1^2 c_2^2 - c_2^2 F_1^2(t)}_{\text{Leading order}} + \underbrace{\left(F_2^2(t) - F_1(t)F_3(t)\right)^2 - 2c_1 c_2 F_2^2(t) - c_1^2 F_3^2(t)}_{\text{Subleading order}}. \quad (241)$$

One can observe that only the first two terms are leading order for $t \gg \delta = \mathcal{L}/N$ in the limit under consideration, namely $N \to \infty$, $m\mathcal{L} \ll 1$, and $\beta/\mathcal{L} \ll 1$. The three ingredients of the leading order piece are then given by

$$c_1 = \frac{1}{N} \sum_{k=0}^{N-1} \frac{\cosh 2\alpha_k}{\lambda_k}, \quad c_2 = \frac{1}{N} \sum_{k=0}^{N-1} \lambda_k \cosh 2\alpha_k,$$
$$F_1(t) = \frac{1}{N} \sum_{k=0}^{N-1} \frac{\sinh(2\alpha_k) \cos(\omega_k t)}{\lambda_k}. \quad (242)$$

We may use eq. (20) to show that $\sinh(2\alpha_k) = 1/\sinh(\beta\omega_k/2)$ and for $k \ll N$ and $\beta \ll \mathcal{L}$ we may effectively approximate $\sinh(2\alpha_k) \sim 2/(\beta\omega_k)$. Using these equalities we are able to approximate the function $F_1(t)$ for $t > 0$ as follows

$$F_1(t) = \frac{\sinh(2\alpha_0)\cos(mt)}{Nm\mathcal{L}} + \overbrace{\sum_{k=1}^{N-1} \frac{\sinh(2\alpha_k)}{N\lambda_k} \cos(\omega_k t)}^{\sim 2\sum_{k=1}^{N/2} \frac{\sinh(2\alpha_k)}{N\lambda_k} \cos(\omega_k t)} \quad (243)$$

$$\sim \frac{2}{N\beta m^2 \mathcal{L}} \left( \cos(mt) + \frac{m^2}{2} \underbrace{\sum_{k=1}^{\infty} \frac{\mathcal{L}^2}{k^2 \pi^2} \cos\left(\frac{2\pi k t}{\mathcal{L}}\right)}_{= \mathcal{P}_{\mathcal{L}}(t(t-\mathcal{L})) + \frac{\mathcal{L}^2}{6}} \right), \quad (244)$$

where we have used the fact that $\omega_k = \omega_{N-k} \sim 2\pi k/\mathcal{L}$ for $k \ll N$ and $m \ll \delta^{-1}$, see eq. (179), and ignored differences introduced for larger values of $k$ where both the terms in the above sum as well as $\sinh(2\alpha_k)$ are highly suppressed (the latter exponentially) in the limit $N \to \infty$. Furthermore, the sum over $\cos(2\pi k t/\mathcal{L})$ can be recognized as the Fourier series expansion of $t(t - \mathcal{L}) + \mathcal{L}^2/6$ over the interval $[0, \mathcal{L}]$. Due to the periodicity of the Fourier series, we define the function $\mathcal{P}_{\mathcal{L}}(x)$ as the periodic function that repeats its values from the interval $[0, \mathcal{L}]$ on all other intervals $[n\mathcal{L}, (n + 1)\mathcal{L}]$. Putting things together and using

$$c_1 \sim \frac{2}{N\beta m^2 \mathcal{L}} \left(1 + \frac{m^2 \mathcal{L}^2}{12}\right), \quad (245)$$

we find

$$
\begin{aligned}
\det\left(G_{xx}^{a,b}\right) &\sim c_2^2 \left(c_1^2 - F_1(t)^2\right) \\
&\sim \frac{4c_2^2}{N^2\beta^2 m^4 \mathcal{L}^2}\left[\left(1 + \tfrac{m^2\mathcal{L}^2}{12}\right)^2 - \left(\cos(mt) + \frac{m^2\mathcal{P}_{\mathcal{L}}(t(t-\mathcal{L}))}{2} + \frac{m^2\mathcal{L}^2}{12}\right)^2\right] \quad (246) \\
&\sim \frac{4c_2^2}{N^2\beta^2 m^4 \mathcal{L}^2}\left[\sin^2(mt) + m^2\mathcal{P}_{\mathcal{L}}(t(\mathcal{L}-t))\right],
\end{aligned}
$$

where we have neglected the square of $\mathcal{P}_{\mathcal{L}}(t(\mathcal{L}-t))$, and we approximated $\cos(mt) \sim 1$ in its product with $\mathcal{P}_{\mathcal{L}}(t(\mathcal{L}-t))$. These approximations are consistent as long as we are not too close to points where $mt$ is an integer multiple of $\pi$, since we are working in a regime where $m\mathcal{L} \ll 1$. Consequently, the entanglement entropy is given by

$$
S(\rho_A(t)) \sim 2(1-\log(2)) + \log\left(\frac{2c_2}{N\beta m}\right) + \frac{1}{2}\log\left[\frac{\sin^2(mt) + m^2\mathcal{P}_{\mathcal{L}}(t(\mathcal{L}-t))}{m^2\mathcal{L}^2}\right]. \quad (247)
$$

We can simplify this expression in three time regimes, namely $\delta \ll t < \mathcal{L}$, $\mathcal{L} < t \ll m^{-1}$, and $\mathcal{L} \ll t$. In these regimes, we find the asymptotic behaviour of the entanglement entropy to be given by

$$
S(\rho_A(t)) \sim 2(1-\log 2) + \log\left(\frac{2c_2}{N\beta m}\right) + \begin{cases} \frac{1}{2}\log(t/\mathcal{L}) & \delta \ll t < \mathcal{L} \\ \log(t/\mathcal{L}) & \mathcal{L} < t \ll m^{-1} \\ \log\left(\frac{|\sin mt|}{m\mathcal{L}}\right) & \mathcal{L} < t \end{cases}. \quad (248)
$$

This expression is well-matched by the numerical results in fig. 25. Note that the last case becomes inaccurate around its singularities, where $mt$ is a multiple of $\pi$. For large $N$, we can use the asymptotics $c_2 \sim 2\mathcal{L}/\beta$ to obtain further simplifications. We should emphasize that this expression will fail to describe the entanglement accurately in the regime $t < \delta$, in which the precise oscillations of the various modes will determine the time evolution.

This discussion gave a precise account of the logarithmic entanglement growth as a function of time for a single mode in $I_R$ and $I_L$, respectively, constituting the subsystem $A$. To argue about more general subsystems composed of many modes, let us first again state that due to the aforementioned upper and lower bounds to the von-Neumann entanglement entropy, and since we are in what follows not that much interested in pre-factors, we can basically express the entanglement entropy freely in terms of the first and second Rényi entropies, respectively. Let us denote with $\sigma(t)$ the reduced state of a single pair of sites in $R$ and $L$, so that eq. (248) applies to $S(\sigma(t))$. Hence, for a general region $A = I_L \cup I_R$, invoking the subadditivity of the von-Neumann entropy and making use of the above bound, we have as a rigorous upper bound

$$
S(\rho_A(t)) \leq \frac{N_A}{2}S(\sigma(t)) \leq \frac{N_A}{2}\left(S_2(\sigma(t)) + 2N_A(1-\log 2)\right). \quad (249)
$$

Hence, we find that $S(\rho_A(t))$ for the entire multi-mode region $A$ is again upper bounded by a constant plus a function that grows logarithmically in time, until saturation. At the same time, since the initial state contains short-ranged correlations, one expects to be able to decompose the system $A$ in largely uncorrelated regions, giving rise to each logarithmically growing contributions, with little error. That again has the consequence that one expects the behaviour $S(\rho_A(t)) \sim c(\text{constant} + \log m t)$ for some constant $c > 0$.

To close this discussion, let us emphasize that contributions similar to eq. (248) have already appeared earlier in section 5.2 in our discussion of the $\kappa = 2$ complexity on a circle, see eq. (196). There, the contribution of the zero mode had a fixed numerical prefactor and we argued that in the limit where we decompactify the circle, it is subdominant with respect to

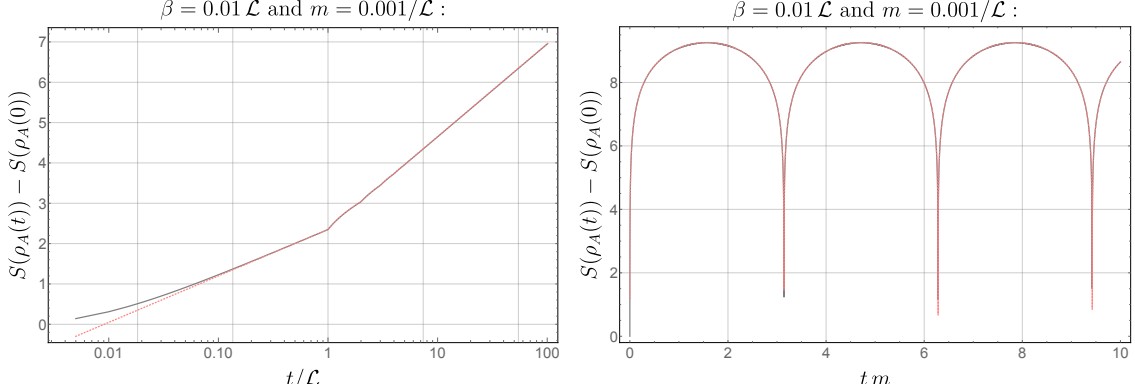

Figure 25: Comparison of the analytic formula (247) (dashed pink curves) with numerically computed entanglement entropy for subsystem consisting of a single site on each side of the TFD state (solid grey curves) for short (left) and long (right) times for $m = 10^{-3}/\mathcal{L}$ and $\beta = 0.01\,\mathcal{L}$. The total number of sites on each side is 1001. One observes a remarkable agreement. Note the different normalization of the time axis on each plot. Note also that for such small subsystems finite size effects manifest themselves differently than in the quasi-particle regime covered by fig. 21, *i.e.,* there are no pronounced oscillations. Note that we do not divide by the thermodynamic entropy $S_{\text{th}}$ here, since the entanglement entropy of two sites does not have a well-defined continuum limit.

the other modes contributing on the order of the thermodynamic entropy. As a result, unlike for the entanglement entropy, the zero mode played no role in the dynamics of complexity for quantum field theories in Minkowski space.

# 7 Discussion

In this work, we have studied the entanglement dynamics as well as the definition of circuit complexity for time-dependent TFD states (1) in free, non-interacting scalar QFTs. In our investigations, we have been directly motivated by the results of the proposed holographic duals of complexity and entanglement evaluated in eternal black hole backgrounds in asymptotically anti-de Sitter spacetimes, which are the dual manifestation of TFD states in a class of strongly coupled quantum field theories with large number of degrees of freedom.

In defining complexity, one needs to specify the elementary operations or gates whose number one counts according to some cost function which assigns a length to the circuit. The crucial insight in this respect is that for free systems, the TFD state introduced in section 2.2, the vacuum state, and simple, spatially disentangled reference states considered previously in refs. [23, 51] – are Gaussian. As a result, the natural choice of gates is the set of all Gaussian transformations (*i.e.,* the full set of Bogoliubov transformations), which for $N$ bosonic modes[46] is the symplectic group $Sp(2N, \mathbb{R})$. This is a larger set of elementary operations than considered previously in refs. [23, 51], but at the same time it is a small subset of all bosonic unitary transformations. A technical novelty of our work with respect to earlier works is phrasing the complexity calculation in the language of symplectic transformations acting on covariance matrices, which is the most efficient way of dealing with the full group of symplectic trans-

---

[46]For a quantum field theory, a finite number of modes arises from UV-regularizing the system by putting it on a lattice, see section 5.

formations on the Gaussian states, see section 3. An additional new feature with respect to ref. [23] is the need to make some of the new generators of the symplectic transformations dimensionless, see eqs. (37) and (91), which leads to the appearance of an additional scale in the problem – the so-called gate scale $\omega_g$.

Following the ideas of Nielsen and collaborators [70–72], we have represented the circuits which prepare the desired state as paths in a Riemannian geometry of the underlying Lie group, *i.e.,* $\text{Sp}(2N, \mathbb{R})$ with appropriate boundary conditions. The optimal circuits then become geodesics in this geometry. An important point to mention is that we are not just interested in representing or approximating a particular unitary operator as a circuit, but rather in considering a whole class of circuits, all of which will give rise to the desired target state. There is no unique choice of a circuit geometry.[47] In the bulk of the text, we focused on the distance measure induced by the Hilbert-Schmidt inner product between two infinitesimally separated group elements, see eq. (109) with $G = g = \mathbb{1}$. As we demonstrate in appendix F, when $\lambda_{\text{R}} = 1$ we can take advantage of a special property of this geometry to show that the optimal circuit does not mix different normal modes, which significantly simplifies the complexity calculation. Furthermore, in this situation where the scale controlling the on-site correlations in the reference state (*i.e.,* the reference state scale) is associated to the gate scale, the expression for complexity is a particularly simple function of the relative covariance matrix between the target and the reference state, see eqs. (86) and (88). In this case, in the basis in which both the reference and target state covariance matrices are diagonal, the optimal circuit amounts to squeezing each of the individual normal modes at a constant rate. This is reminiscent of earlier results obtained in refs. [23, 51]. However, we stress that while both of those references only optimized their respective circuits within some restricted gate set (*i.e.,* $\text{GL}(N, \mathbb{R})$ in ref. [23] and $\text{SU}(1, 1)^N$ in ref. [51]), we have shown that our shortest geodesics really optimize the corresponding circuits within the full family of $\text{Sp}(2N, \mathbb{R})$ generators, as proven in appendix F. Unfortunately, the simplicity of the optimal circuits is lost when the reference state scale and the gate scale are not simply related, *i.e.,* $\lambda_{\text{R}} \neq 1$. In this case, for each normal mode, one needs to find the optimal path numerically, see section 4.5. We must add here that to make the numerical problem tractable, we have still assumed that the optimal circuits did not mix the normal modes (although this was only rigorously proven for the case of $\lambda_{\text{R}} = 1$). Working in the normal mode basis implies that our circuits are constructed using spatially non-local gates. This is not unlike the action of MERA tensor networks [111], where entanglement at different scales is injected by gates in different layers of the circuit.[48] We might also add that a recent holographic work [36] suggests that if the holographic complexity proposals indeed capture some variant of a circuit complexity, then the optimal circuits must be utilizing spatially non-local gates.

Before we summarize the main physics lessons, let us remark that the complexity defined by the Riemannian geometry can be thought of as using the $\text{L}^2$ norm, *i.e.,* the $F_2$ cost function, when counting the number of gates, see eq. (8) (right). The results of refs. [23, 51] show UV behaviour closely aligned with the holographic results [29] when counting the gates using the $\text{L}^1$ norm, *i.e.,* the $F_1$ cost function, see eq. (8) (left) and the discussion below. Such counting leads to a much more complicated minimization problem, and in the present article we adopted the proviso from refs. [23, 51], *i.e.,* we used the $\text{L}^1$ norm to express the length of a circuit optimized with respect to the $\text{L}^2$ norm. In fact, for the preparation of the ground state, ref. [23] showed that this circuit also optimized the $\text{L}^1$ norm. But their proof does not extend to the present situation and so our results should be seen as an upper bound on the $F_1$

---

[47]One can make use of this freedom to impose further conditions on what kind of circuits are the optimal ones, such as locality; see the discussion in ref. [23].

[48]The similarity becomes even more apparent with the cMERA construction [112] for free scalars or fermions which also acts directly on the normal modes.

complexity. Setting aside the problem of optimality in the $L^1$ norm, there is also the issue of basis dependence, *i.e.,* the length of the path will depend on the basis chosen for the generators. While this may be seen as a shortcoming of this norm, we actually took advantage of this basis dependence in evaluating the complexity of formation in section 5.3. There we saw that using the physical left-right (LR) basis (but not the diagonal basis) produced a complexity of formation that compared well with the holographic results [27]—see further discussion below.

Having sketched the mathematical underpinnings of our work, let us now discuss the physics of complexity in the Gaussian TFD states that we have uncovered. We begin by re-iterating some of the key lessons stemming from studies of the holographic complexity proposals. First, the complexity of the vacuum state diverges as the spatial volume occupied by a boundary quantum field theory measured in the units of the UV cut-off with a non-universal prefactor [29]. Furthermore, for the CA proposal, there is a possibility that this leading divergence is also multiplied by a $|\log(\ell_{\rm ct}/L)|$ factor arising from the detailed structure of null boundary terms in the gravitational action.[49] Second, calculating holographic complexity for the TFD state (1) at $t_L = 0 = t_R$, and then subtracting twice the vacuum complexity, one finds that the remainder is proportional to the thermodynamic entropy, *i.e.,* the entanglement entropy between the left and right CFTs, see eq. (207) here and ref. [27] for details.[50] This remainder term is called the complexity of formation, *i.e.,* the extra complexity needed to prepare the left and right CFTs into the entangled TFD state, instead of a product state of two vacuum states (*i.e.,* the $\beta \to \infty$ limit of the TFD state). Finally, for the holographic TFD states both proposals predict a linear growth of the holographic complexity at late times, which is perhaps their most well-known feature.

Following refs. [23,51], we always take as our reference state a spatially disentangled state. Furthermore, we focus on the $L^1$ norm, *i.e.,* $F_1$ cost function, since evaluating the complexity of the vacuum state in the normal mode basis then yields a result with the same leading UV divergence as found in holographic complexity. However, it is also interesting to consider the complexity evaluated with the $\kappa = 2$ cost function since in this case the geodesics correctly describe optimal circuits.[51]

## 7.1 Complexity of formation and time evolution

Let us begin by discussing the complexity of formation in the case when the gate scale is equal to the reference state scale, *i.e.,* $\lambda_{\rm R} = 1$, for which the results have been presented in section 5.3. As we noted above, using the $L^1$ norm and physical LR basis, our result for the complexity of formation compared well with the holographic results [27]. Three general features shared with the holographic result are, $\Delta\mathcal{C}_1^{(\rm LR)}$ is UV finite, it is independent of the reference scale $\mu$, and it is positive.[52] Furthermore, if we set $m = 0$ to emulate a CFT, then $\Delta\mathcal{C}_1^{(\rm LR)}$ is proportional to the entanglement entropy between the left and right CFTs as found with holographic com-

---

[49]This expression includes a counter term introduced in ref. [26] in order to restore the reparametrization invariance of the gravitational action and which was found to be necessary in order to reproduce desired properties of complexity [38,39].

[50]Note that for AdS$_3$/CFT$_2$, the proportionality constant for this leading term vanishes and the remainder is simply a constant proportional to the central charge. Let us also add that there are curvature corrections for spherical and hyperbolic boundary geometries.

[51]Let us also add that the $\kappa$ measures 9 have been introduced in ref. [23] to find complexity models that would reproduce the leading $V\Lambda^{d-1}$ divergence found in holographic complexity. In general, apart from $\kappa = 1$ which coincides with the $L^1$ norm, these cost functions will not reproduce the logarithmic factor found with the CA proposal. However, this issue is evaded if we choose the reference frequency $\mu$ to be proportional to the cutoff scale, *e.g.,* see the discussion in refs. [23,29].

[52]A priori, $\Delta\mathcal{C} > 0$ is not an obvious result since while in preparing the TFD state, one introduces extra entanglement between the left and right copies of the QFT, one does not introduce as much long-range entanglement in either copy as one would in the vacuum state. For further discussion, see appendix E of ref. [27].

plexity when $d \geq 3$. However, there is a difference in the dependence of the proportionality constant on the number of spacetime dimensions compared to holography, *e.g.,* see eqs. (206) and (207). For two dimensions, our QFT complexity of formation is also proportional to the entropy and so grows with increasing temperature, *i.e.,* $\Delta\mathcal{C}_1^{(\mathrm{LR})} \propto VT$ for $d = 2$. Of course, this deviates from the holographic results where, as we explained above, $\Delta\mathcal{C}_{\mathrm{holo}} \propto c$ for $d = 2$, and so we might also classify this as part of the previous discrepancy in the constant of proportionality (between the complexity and the entropy), *i.e.,* this constant vanishes for $d = 2$ in holography while it remains finite in our QFT calculations. We might note that this coefficient had a very different dependence when the $\kappa = 2$ measure was applied, as shown in fig. 11, but that it still remained nonvanishing for $d = 2$ in this case as well. As a final comment, let us reiterate that when evaluating the complexity of formation to produce the results which compared well with holography, we have taken advantage of the basis dependence of the L$^1$ norm and in particular, evaluated it using the LR basis. In section 5.3 – with further details in appendix E – we have shown that using the diagonal basis (along with reasonable choices for the reference and gate scales) leads to the complexity of formation being suppressed with respect to the entropy at low temperatures (with respect to the cut-off $\Lambda$), peaking and decreasing as the temperature approaches the cut-off,[53] see figs. 10, 26, 27 and 28. Therefore, the good agreement with holography produced with the LR basis might actually be a clue as to the microscopic rules that implicitly define the holographic proposals for complexity.

Turning to the time-dependence of the TFD state, the complexity in our quantum field theory calculations is a combination of contributions from individual normal modes, each of which exhibits oscillatory behavior with a frequency set by the normal mode frequency, *e.g.,* see eqs. (149), (151) and (152). While the oscillations are all aligned at $t = 0$, they quickly dephase afterwards and as a result, summing over the modes yields a result in which contributions from different normal modes average out at times of the order of the inverse temperature $\beta$. This leads to a quick saturation in the TFD complexity of free QFTs, a feature which is very different from the late-time linear growth found for holographic complexity. Depending on the choice of parameters, the complexity can either grow or decrease after $t = 0$ before saturating with main features of the process independent of the dimensionality of the QFT. The dependence of the transient early time behaviour on $\mu$ is comparable to the sensitivity of the initial transients to the ambiguous scale arising in the null boundary terms in the CA proposal, similar to the aforementioned logarithmic divergence for the vacuum complexity [38, 39].

Since the late-time growth of holographic complexity was one of the hallmark features of the CA and CV proposals, it may seem somewhat disappointing that this feature is not recovered in our present complexity calculations. However, we must consider that the QFT for which we are calculating the complexity is a free theory, while the boundary CFTs which are described by holographic complexity are strongly coupled theories. In fact, the time evolution of the complexity of the TFD in these theories is predicted to initially show linear growth, *i.e.,* $\partial_t\mathcal{C} \sim M$, and to saturate only on a time scale of the order $e^{S_{\mathrm{BH}}}$ [13, 113, 114]. This expectation is based on the fact that these CFTs are fast scramblers and the evolution of the TFD state explores a larger and larger region within the Hilbert space of the boundary theory. Of course, this is clearly not the situation in the present case of a free scalar field theory. In fact, our complexity calculations have relied on the fact that the initial TFD state and time-evolved TFD are Gaussian states. Hence, the time evolution of the TFD state in our free scalar theory is confined to the submanifold in the full Hilbert space describing Gaussian states. From this perspective, it is perhaps not so surprising that we found that the complexity saturates on the thermal time scale. One could construct far more "complex" states by replacing the

---

[53]With an exception of $d = 2$, where the ratio is the largest for temperatures approaching the cut-off; one should note that this is an unphysical regime to consider.

regularly spaced phases, $(n + 1/2)\omega t$ in eq. (30), with random phases $\theta_n$. Furthermore upon integrating out one of the copies in these new entangled states, we would still recover the desired (time-independent) thermal density matrix. Unfortunately, while we expect the random phases increase the complexity of these states, we do not yet have the technology required to actually perform concrete calculations of their complexity.

One is tempted to draw here an analogy to the early studies of quark-gluon plasmas using holography, *e.g.,* see ref. [115] for an overview of these efforts. Such studies have revealed that the thermodynamics of $\mathcal{N} = 4$ super Yang-Mills theory at vanishing 't Hooft coupling constant differ only by 25% from the thermodynamics of the same model in the holographic regime of infinite coupling [116]. In our case, the complexity of formation for free QFTs behaved in a qualitatively similar way to the results of the holographic complexity proposals. However, the time-dependent properties of complexity in free QFTs turn out to be very different from the predictions of holographic complexity proposals. To close the analogy, it is well known that the time-dependence of holographic QFTs is very different from their weakly-coupled cousins, for example the value of the famous shear viscosity to the entropy density ratio is parametrically different between the two [117].[54]

Altering the ratio of the reference state scale to the gate scale away from unity, *i.e.,* choosing $\lambda_\text{R} \neq 1$, does not change our main conclusions here in a significant way (*e.g.,* as expected on general grounds, the complexity still saturates at times of the order of $\beta$), but it nevertheless leads to certain interesting effects. As shown in fig. 19, when $\lambda_\text{R}$ is either increased or decreased away from one, the complexity generally appears to decrease. This decrease seems to be a universal behaviour, which was already observed in the contributions of the individual modes in fig. 5. Hence, it appears that our results for the time dependence of the complexity with $\lambda_\text{R} = 1$ set an upper bound on the complexity with these more general parameters. Furthermore, we should keep in mind that fig. 19 shows the complexity at time $t$ relative to the complexity at the initial time. Given the results in fig. 4 for a single mode, we should also expect that varying $\lambda_\text{R}$ will reduce the initial complexity of the QFT by reducing the contributions for modes in a particular range of frequencies.

## 7.2 Comparison with entanglement dynamics

One of the most interesting aspects of the holographic proposals is the difference in the time evolution of the complexity and the entanglement entropy, as we have described in the introduction. This has motivated us to compare the time-dependent complexity with the behaviour of the entanglement entropy in time-dependent TFD states in free QFTs, where we have focused on 1+1 dimensional systems. We have studied the entanglement entropy associated to a subsystem consisting of two equal intervals on each side of the time-dependent TFD state. This setup corresponds to the free-field, non-interacting analogue of the holographic studies reported in ref. [3]. For convenience, we have worked with a theory on a spatial circle and considered two cases: first, keeping the size of the circle and physical parameters fixed, while making the lattice denser, and second, keeping the lattice spacing and physical parameters fixed, while increasing the number of lattice sites in order to approach the field theory on a line.

Our investigations have revealed an expected linear growth and saturation pattern, usually present in entanglement of quenched systems, as well as oscillations for systems on the circle

---

[54]One might wonder what features are expected in holographic complexity once more general features are added to the spacetime, such as higher curvature corrections. The investigation in this direction so far suggests that for holographic CFTs in Minkowski space as considered here the CA late time growth rate does not receive direct corrections for the Lovelock class of theories [47] (see refs. [25, 44] for discussion focused on Gauss-Bonnet gravity). It is an interesting question as to whether for more generic higher curvature theories this feature persists, or whether that is due to the special properties of the Lovelock action.

with periodicity given by the circle size. In addition, for small masses we have observed a logarithmic growth associated with the presence of a zero mode, both on the circle and on the line. This is in contrast with our findings for complexity on the line where a logarithmic growth was absent, see section 5. The complexity of the TFD on the circle does exhibit a logarithmic growth. This growth, however, does not survive the decompactification limit. We note however that the logarithmic growth of the entanglement cannot last forever given the upper bounds (223) (universal) and (225) (specific to Gaussian states) and the entanglement entropy has to eventually saturate.

In order to understand the linear phase of entanglement production, we related our results to previous studies of the entanglement entropy after global quenches in a wide class of bosonic systems. Building on earlier work [93, 94, 99], it has become clear that the linear growth of the entanglement entropy in time can be largely explained using a quasiparticle picture where pairs of entangled quasiparticle excitations are created and propagate ballistically at their group velocity in opposite directions [74, 95–97]. Initially, the flux of quasiparticles coming in or going out of the interval produces to an increase in the entanglement entropy and a careful accounting shows that the entanglement rises linearly until times $t \sim \ell$ where it saturates at a value which is proportional to the thermal entropy of the system. When studying the entanglement entropy on a circle, we find that it exhibits recurrences with periodically equal to the circle circumference. This can be related to quasiparticles going around the circle and reducing the correlations between the subsystem and its complement again once they meet on the opposite side of the circle. These features becomes particularly sharp in the massless limit where the quasiparticles effectively move with the speed of light [95], while for larger masses, the different quasiparticles have different group velocities and their contributions to the entanglement entropy quickly dephase. As a result, the variation of the entanglement entropy over time periods of $\Delta t \sim \mathcal{L}$ are greatly reduced.

Overall, we find that the effective model that has been proposed in ref. [74] provides a surprisingly accurate description of the linear slope and quasi-periodic behavior for large masses when adapted to our setting, see eq. (233). For smaller masses, we identified a logarithmic correction to the periodic behavior. Such a behavior was previously observed in refs. [74, 106] and attributed to the zero momentum mode in the massless limit. In fact, this contribution was purposefully removed through appropriate boundary conditions in ref. [74]. However, in the present work, we are able to present a largely analytical formula of this logarithmic growth, based on the determinant of submatrices of the covariance matrix, a contribution that we are convinced is interesting in its own right.

Specifically, to make an analytical analysis of the zero mode contribution feasible, we first focused on a subsystem consisting of a single lattice site on each side of the TFD state. We then argued that the observed asymptotics contribute additively to larger subsystems, due to the fact that the zero mode is completely non-local and therefore affects local subsystems in a similar way, regardless of their size. Our analytical study is based on a well-known relation between the von Neumann entropy and the Rényi entropy of order 2 that can be efficiently computed as the determinant of the covariance matrix for Gaussian states. In the case of a single site on each side of the TFD, this covariance matrix is a 4-by-4 matrix whose time dependence can be analyzed analytically. For a circle of circumference $\mathcal{L}$ and a small mass $m$, we find three distinct regimes, namely an initial logarithmic regime given by $S(\rho_A(t)) \sim \frac{1}{2} \log t/\mathcal{L} + \text{const}$ for $t < \mathcal{L}$, a second logarithmic regime given by $S(\rho_A(t)) \sim \log t/\mathcal{L} + \text{const}$ for $\mathcal{L} < t \ll m^{-1}$ and finally an oscillating regime given by $S(\rho_A(t)) \sim \log \sin mt + \text{const}$ when $t$ is of the same order as $m^{-1}$, see eq. (248). Note that we intentionally suppress the linear regime by choosing a subsystem size that vanishes in the continuum limit, so we can fully focus on understanding the logarithmic contribution analytically. In the non-compact limit, $\mathcal{L} \to \infty$, we expect to only see the initial contribution of $S(\rho_A(t)) \sim \frac{1}{2} \log t/\mathcal{L} + \text{const}$. Let us emphasize that to the

best of our knowledge, this asymptotic analysis provides one of the first analytical insights into the zero mode contributions. However, understanding its additive contribution for larger subsystems, *i.e.,* those that do not vanish in the continuum limit, will require further work if one wants to go beyond the rough analysis and heuristic arguments presented at the end of subsection 6.5.

## 7.3   Relation to other works

We note that the question of the complexity of the TFD state within a free scalar field theory has been studied previously in refs. [52, 53]. However, our methods differ in a variety of ways. For example, these works have not considered unitary circuits to prepare the target state; they used a different reference state, *i.e.,* their reference state was two unentangled copies of the vacuum state; they used a very restricted gate set, *i.e.,* their generators formed an $[\mathrm{Sp}(2, \mathbb{R})]^N$ algebra compared to $\mathrm{Sp}(2N, \mathbb{R})$ here;[55] the gate scale was implicitly set by the frequency of the individual modes, *i.e.,* $\omega_g^2 = M\omega_k$; and finally, they chose an unconventional cost function which was different than any of the cost functions considered in our paper. Given these extensive differences, it may seem remarkable if any of our results were to match with those in these earlier references. However, one finds an interesting parallel between our complexity of formation for a massless field (see eqs. (200) and (206)) and the complexity of the TFD relative to an unentangled product of two vacuum states evaluated in ref. [52], in that both results are proportional to $V T^{d-1}$. We might add that for the special case of $t_L = 0 = t_R$, *i.e.,* the TFD without any complex phases, the measure in ref. [52] coincides with the $F_1$ cost function, but their results do not match ours because of the other differences in our two approaches described above, *e.g.,* they would not produce the same complexities for a massive scalar field. Rather, the simple behaviour with $\Delta\mathcal{C} \propto V T^{d-1}$ seems to be a result of dimensional analysis. If we assume that the complexity is extensive, then certainly in a situation where the temperature is the only other dimensionful parameter, we must find $\Delta\mathcal{C} \propto V T^{d-1}$ (recall that we found similar results for both the $F_1$ and $\kappa = 2$ cost functions).

Our results for the complexity of the time-dependent TFD state differ quite dramatically from those found in ref. [53]. The latter reported a linear growth of complexity at late times (when applying the Nielsen approach), a behaviour which stands in stark contrast with our findings, *i.e.,* the saturation of complexity after roughly the thermal scale. The key difference in our approaches leading to this discrepancy is that ref. [53] adopts the unusual cost function introduced in ref. [52]: compare eq. (B.9) in ref. [53] with our eq. (8). With this choice for a unitary involving exponentiation of a phase factor, the cost function becomes arbitrarily large as the phase increases beyond $2\pi$, which directly leads to the aforementioned growth. We regard it as physically incorrect to assign complexity growth to a periodic behaviour, and indeed in the approach pursued in the present paper the complexity for each normal mode is simply periodic, as are the relevant unitaries.

We might add that it is straightforward to adapt our analysis to consider the complexity of the (time-dependent) TFD state relative to the reference state being two unentangled copies of the vacuum state, as in refs. [52, 53]. For each momentum mode, the reference state analogous to eq. (27) would be replaced by the vacuum, as in eq. (26). As a result, we would simply substitute $\omega_{\mathrm{R}}^2 \mapsto M\omega_k$ for each mode. The single-mode calculations would proceed as in section (4.5) with the substitutions $(\lambda, \lambda_{\mathrm{R}}) \mapsto (1, \lambda = M\omega_k/\omega_g^2)$. It would be interesting to investigate this relative complexity further, both for the individual modes and for the full field theory. However, as noted above, refs. [52, 53] go one step further and set the gate scale for each individual mode to be $\omega_g^2 = M\omega_k$.[56] If we added this choice in our approach, we would

---

[55]Hence refs. [52, 53] did not allow for any mode mixing in their approach, rather than proving that there was no mode mixing in the optimal circuit, as we did here.

[56]This choice is implicit because the generators for the gates are constructed in terms of annihilation operators,

also set $\lambda_{\text{R}} = 1$ and our analysis would reduce to that in section 4.4 with $\lambda = 1$. In this case, eq. (142) yields $\rho_1 = \alpha$ and $\theta_1 = \frac{\pi}{2} - \omega_k t$ (with $\tau_1 = 0$), and hence the complexity for the individual modes is simply a constant, *e.g.*, $\mathcal{C}_{\kappa=2} = \rho_1^2 = \alpha^2$. Furthermore, integrating over all the modes would yield a total complexity which precisely matches the expressions which we gave to the complexity of formation, *e.g.*, $\mathcal{C}_{\kappa=2}$ would be given by eq. (215). Hence the total complexity would be time-independent, and of course, our analysis still does not yield the linear growth found in ref. [53].

Ref. [53] also uses the Fubini-Study approach of ref. [51] to assign a complexity to the TFD state of a free scalar, again relative to the product of two vacua. With this approach, they report that the complexity initially grows but saturates in roughly the thermal scale. However, we still disagree with their calculations in this case. This result should correspond to considering eq. (144) together with eq. (142) evaluated for $\lambda = 1$ (and $\lambda_{\text{R}} = 1$), which again leads to a time-independent answer. While circuit complexity is *a priori* a different notion from the complexity of states, in the case of free QFTs (or more generally, for Gaussian states which are prepared with only squeezing gates[57]) they agree, see refs. [23, 51, 119], which motivated us to look into ref. [53] in more detail. And indeed, upon a closer examination one notices that eq. (4.32) in ref. [53], and following from it eq. (4.34), should have a phase factor pulled out from them. When this correction is included, the complexity remains constant throughout the time evolution, as suggested by the argument above.

The time-dependence of the TFD state discussed in the present manuscript is similar to a quantum quench in a free quantum field theory (in this context, evolution of a ground state under a Hamiltonian whose mass term becomes time-dependent for a certain period). Such set-ups have been discussed in the context of cMERA in refs. [120, 121] and it was reported there, in the language of ref. [51], that the Fubini-Study distance along the RG scale direction of quantum circuit modelling a state after a quench exhibits a linear growth, in contrast with our claims. However, there is no contradiction since the results of refs. [120, 121] can be regarded as measuring the circuit depth of a particular unitary rather than measuring the circuit depth of the optimal unitary and, as we demonstrated here, it does not exhibit any long-time growth.[58] When it comes to the complexity of quench setups in free QFTs, studies reported in recent ref. [58] define complexity using translationally-invariant Gaussian circuits and yield results that exhibit qualitative similarity to our results, *i.e.*, a short period of transient effects following a quench and a subsequent saturation or mild oscillations of complexity. Finally, ref. [59] demonstrated that complexity in free quantum field theory defined using some of the machinery introduced in the present manuscript can obey scaling relations as a function of the quench rate in quenches passing through a critical point, similar to an earlier analysis of one-point functions [122, 123] and entanglement entropy [124].

## 7.4 Open questions

Perhaps the most interesting open question that our work raises lies in defining circuit complexity in interacting QFTs in a calculable manner. This is, of course, related to a rather quick saturation of circuit complexity in free QFTs that we observed here, which is in stark contrast with the results of the holographic complexity proposals. It would be very interesting to understand if recent attempts of refs. [125, 126] to include interactions in the continuous version of MERA (cMERA) [84] can help in addressing this problem. See also the recent work [65].

---

which are tuned to annihilate the QFT vacuum state, and their conjugate creation operators.

[57]Note that recently examples have been found in the more general setting of coherent states where the two approaches produce different complexities and different optimal circuits [118].

[58]In particular, the time dependence of such unitaries is heavily dependent on the time dependence of a phase $\theta_k(t)$ that does not change the state, see, for instance, eq. (3.31) in ref. [121]

Another important problem is to understand if there are relations between circuit complexity as considered here and complexities defined using the Fubini-Study metric [51] and the path-integral optimization [60, 61, 127]. Free QFTs provide what we believe is a fruitful testing ground in which all three approaches become computable.

In the realm of Gaussian circuit complexity, it would be fascinating to explore more systematically the properties of optimal circuits in the presence of non-trivial penalty factors. An interesting question is whether insisting on (quasi-)locality of gates (even if it might not be the case in holography, see again ref. [36]) can significantly alter the results of existing complexity calculations, such as reported here or in refs. [23, 51, 55, 56, 58]. Such results can also have an interesting application in the field of tensor networks, namely if MERA [111] or cMERA [84, 125, 126], as tensor networks are in some sense an optimal way of preparing critical ground states from product states.

Taking a more quantum information perspective, one can quantitatively relate the notions of complexity with those of the entangling power of quantum gates. The complexity counts the number of gates from a specific gate set needed to prepare a quantum state. The entanglement over any cut can be upper bounded in terms of the number of gates that are supported nontrivially on both sides of the cut, discounted by the entangling power of these specific quantum gates. In a continuum limit, the latter can be captured in terms of entanglement rates [128]. The quantitative connection between upper bounds to entanglement over cuts quantified in terms of entanglement rates and circuit complexity will be explored in detail elsewhere.

Finally, our work raises many interesting open problems regarding nontrivial gate scales, *i.e.,* $\lambda_R \neq 1$. For instance, since every mode has a more intricate minimization procedure, it is possible that the vacuum UV divergence will get modified as well. Understanding whether the case $\lambda_R = 1$ still provides an upper bound, and exactly how the vacuum divergent terms get modified, could shed light on the nature of the gate scale, as well as whether holographic complexity has features which mimic this parameter. In addition, it would be interesting to directly investigate how the circuit gets modified once the geodesics deviate from the $\tau = 0$ plane. It is our hope that the present comprehensive analysis stimulates such further work.

# Acknowledgment

We would like to thank J. de Boer, P. Calabrese, P. Caputa, J. Hernandez, K. Papadodimas, S. M. Ruan, M. Smolkin, T. Takayanagi, E. Verlinde, M. Walter, and the members of the QGFI group at AEI for useful comments and discussions. We would also like to extend special thanks to F. Pastawski for collaboration during the initial stages of this project. Research at Perimeter Institute is supported by the Government of Canada through the Department of Innovation, Science and Economic Development and by the Province of Ontario through the Ministry of Research, Innovation and Science. JE, MPH, HM, and RCM thank the Kavli Institute for Theoretical Physics for its hospitality at various stages of this project. At the KITP, this research was supported in part by the National Science Foundation under Grant No. NSF PHY17-48958. SC and RCM also thank the Galileo Galilei Institute for Theoretical Physics for hospitality and the INFN for partial support where part of this work was carried out. MPH would also like to thank University of Amsterdam, Chulalongkorn University, Hanyang University, Hebrew University of Jerusalem, Mainz Institute for Theoretical Physics, National Taiwan University, University of Warsaw and Yukawa Institute for Theoretical Physics for extended hospitality during the completion of this project. LH thanks the Visiting Graduate Fellow program for hospitality at the Perimeter Institute during this project. LH was also supported by a Frymoyer fellowship, a Mebus fellowship, by the National Science Foundation under Grant No. PHY-1404204 awarded to Eugenio Bianchi and the Max Planck Harvard Research Center for Quantum Optics. RCM is

supported by funding from the Natural Sciences and Engineering Research Council of Canada and from the Simons Foundation through the "It from Qubit" collaboration. MPH and RJ acknowledge support from the Alexander von Humboldt Foundation and the Federal Ministry for Education and Research through the Sofja Kovalevskaja Award. JE is supported by the DFG (EI 519/14-1, EI 519/7-1, CRC 183, FOR 2724), the Templeton Foundation, and the ERC (TAQ). This work has also received funding from the European Union's Horizon 2020 research and innovation programme under grant agreement No 817482 (PASQUANS). SC acknowledges funding from the European Research Council (ERC) under starting grant No. 715656 (GenGeoHol) awarded to Diego M. Hofman.

# A   Table of notation and conventions

| Symbol | Meaning |
|---|---|
| $\beta$ | inverse temperature |
| $L, R$ | Left and Right sides of the thermofield double state |
| $Q, P$ | dimensionful conjugate variables |
| $q, p$ | dimensionless (rescaled) conjugate variables: $q = \omega_g Q$ and $p = P/\omega_g$ |
| $\omega_g$ | gate scale |
| $M$ | oscillator mass; in QFT, inverse lattice spacing |
| $\omega$ | oscillator frequency |
| $\mu$ | reference-state scale (frequency of reference state) |
| $\mu_g$ | field theory gate scale $\mu_g = \delta\,\omega_g^2$ |
| $\lambda$ | vacuum wavefunction parameter: $\lambda = \frac{M\omega}{\omega_g^2}$ |
| $\lambda_{\mathrm{R}}$ | scale ratio: $\lambda_{\mathrm{R}} = \frac{M\mu}{\omega_g^2}$ for single oscillator and $\lambda_{\mathrm{R}} = \frac{\mu}{\delta\omega_g^2}$ for field theory |
| $\xi^a$ | general rescaled canonical variables: $\xi^a \equiv (q, p)$ |
| $\alpha, \alpha_{\mathrm{R}}$ | $\alpha = \frac{1}{2}\log\left(\frac{1+e^{-\frac{\beta\omega}{2}}}{1-e^{-\frac{\beta\omega}{2}}}\right)$, $\alpha_{\mathrm{R}} = \frac{1}{2}\log\frac{\lambda}{\lambda_R}$ |
| $\widehat{K}$ | operator: Hermitian generator $\widehat{K} = \frac{1}{2}\xi^a k_{ab}\xi^b = \frac{1}{2}\xi k \xi$ |
| $K$ | matrix: Lie algebra generator $K^a{}_b = \Omega^{ac}k_{cb}$ |
| $\widehat{U}_K$ | operator: unitary group element $\widehat{U}_K = e^{-i\widehat{K}}$ |
| $U_K$ | matrix: Lie group element $U_K = e^K$ |
| $\widehat{V}, \widehat{Z}, \widehat{W}$ | operator: Hermitian generators $V = q^2$, $W = \frac{1}{2}(qp + pq)$, $Z = p^2$ |
| $V, Z, W$ | matrix: Lie algebra generators of $\mathrm{Sp}(2, \mathbb{R})$ |
| $\widehat{U}_V, \widehat{U}_Z, \widehat{U}_W$ | operator: infinitesimal unitary group elements $\widehat{U}_V = e^{-i\epsilon\widehat{V}}$, etc. |
| $U_V, U_Z, U_W$ | matrix: infinitesimal Lie group elements $U_V = e^{\epsilon V}$, etc. |
| $i, j, k$ | position and momentum indices, *i.e.*, $x_i$, $p_i$ |
| $a, b, c$ | phase space indices, *i.e.*, $\xi^a = (q_i, p_i)$. |
| $K_I$ | basis of generators |
| $G_{\mathrm{R}}, G_{\mathrm{T}}$ and $G_A$ | covariance matrix for reference, target state and subsystem $A$ |
| $\tilde{\gamma}$ | inverse mass of the reference state in the units of temperature: $\tilde{\gamma} = \frac{1}{\beta\mu}$ |
| $\mathcal{L}$ | circumference of a circle when we consider a theory on $\mathbb{R} \times S^1$ |
| $\ell$ | subsystem size when we consider a theory on $\mathbb{R} \times S^1$ ($\ell < \mathcal{L}$) |
| $S_{\mathrm{th}}$ | thermal entropy of full system at inverse temperature $\beta$ |
| $S(\rho)$ | von Neumann-entropy of mixed state $\rho$ |
| $S_q(\rho)$ | $q^{\mathrm{th}}$ Rényi entropy of mixed state $\rho$ |

| $N$ | number of lattice sites on each side of the TFD (left/right) |
| $N_A$ | number of sites in subsystem $A$ on each side of the TFD (left/right) |

## B  Unitary decomposition of the TFD state

In this appendix we introduce a useful decomposition formula which allows us to express the TFD state as a unitary operation on the product of vacuum states. We begin with the following operators[59]

$$k_- = a_L a_R, \quad k_+ = a_L^\dagger a_R^\dagger, \quad k_0 = \frac{1}{2}\left(a_L^\dagger a_L + a_R^\dagger a_R + 1\right), \tag{250}$$

which satisfy the algebra

$$[k_-, k_+] = 2k_0, \quad [k_0, k_\pm] = \pm k_\pm. \tag{251}$$

The decomposition formula taken from appendix 11.3.3 of ref. [129] reads

$$e^{\alpha_+ k_+ + \alpha_- k_- + \omega k_0} = e^{\gamma_+ k_+} e^{\log \gamma_0 k_0} e^{\gamma_- k_-},$$

$$\gamma_\pm = \frac{2\alpha_\pm \sinh \Xi}{2\Xi \cosh \Xi - \omega \sinh \Xi}, \quad \gamma_0 = \left(\cosh \Xi - \frac{\omega}{2\Xi}\sinh \Xi\right)^{-2}, \quad \Xi^2 \equiv \frac{\omega^2}{4} - \alpha_+ \alpha_-. \tag{252}$$

The requirement that the operator in eq. (252) be unitary implies $\alpha_+ = -\alpha_-^*$ and $\omega \in \mathbb{R}$. For the special case where $\omega = 0$ and $\alpha_+ = -\alpha_- = \alpha \in \mathbb{R}$ we obtain

$$e^{\alpha(k_+ - k_-)} = e^{\gamma_+ k_+} e^{\ln \gamma_0 k_0} e^{\gamma_- k_-} \quad \text{where} \quad \gamma_0 = \frac{1}{\cosh^2 \alpha}, \quad \gamma_\pm = \pm \tanh \alpha. \tag{253}$$

Now acting on the vacuum state $|0\rangle_L |0\rangle_R$, we have

$$k_- |0\rangle_L |0\rangle_R = 0, \quad k_0 |0\rangle_L |0\rangle_R = \frac{1}{2}|0\rangle_L |0\rangle_R. \tag{254}$$

Hence if we act with the expression in eq. (253), we obtain

$$\exp[\alpha(k_+ - k_-)]|0\rangle_L |0\rangle_R = \frac{1}{\cosh \alpha} \exp[\tanh \alpha \, k_+]|0\rangle_L |0\rangle_R. \tag{255}$$

Hence, if we identify

$$\tanh \alpha = \exp(-\beta \omega / 2), \tag{256}$$

which also implies $\cosh \alpha = (1 - e^{-\beta \omega})^{-1/2}$, then eq. (255) reproduces the desired thermofield double state (17)-(18). This formula also extends to the complex case with

$$\exp[z \, k_+ - z^* k_-]|0\rangle_L |0\rangle_R = \frac{1}{\cosh \alpha} \exp\left[e^{-i\omega t} \tanh \alpha \, k_+\right]|0\rangle_L |0\rangle_R, \tag{257}$$

where $z = \alpha e^{-i\omega t}$. As before, $\tanh \alpha$ is given by eq. (256) so that eq. (257) reproduces the time-dependent thermofield double state (30)-(31). Note however, that this expression does not capture the overall phase factor $\exp(-i\omega t/2)$ in eq. (30).

---

[59]Here $\pm$ does *not* stand for the diagonal basis.

## C    Matrix generators for $\mathrm{Sp}(4,\mathbb{R})$

To illustrate the ideas of subsection 3.5, let us consider $\mathrm{Sp}(4,\mathbb{R})$ as it will be useful for the discussion in section 4. We will be using the L(eft)-R(ight) basis associated with the canonical variables

$$\xi = (q_L, q_R, p_L, p_R). \tag{258}$$

The matrix generators for the $\mathrm{Sp}(4,\mathbb{R})$ are given by eq. (95). We can split them to the $\mathrm{Sp}(2,\mathbb{R})$ subalgebra acting on the left oscillator only

$$W_{L,L} = \begin{pmatrix} 1 & 0 & 0 & 0 \\ 0 & 0 & 0 & 0 \\ 0 & 0 & -1 & 0 \\ 0 & 0 & 0 & 0 \end{pmatrix}, \quad V_{L,L} = \begin{pmatrix} 0 & 0 & 0 & 0 \\ 0 & 0 & 0 & 0 \\ -\sqrt{2} & 0 & 0 & 0 \\ 0 & 0 & 0 & 0 \end{pmatrix}, \quad Z_{L,L} = \begin{pmatrix} 0 & 0 & \sqrt{2} & 0 \\ 0 & 0 & 0 & 0 \\ 0 & 0 & 0 & 0 \\ 0 & 0 & 0 & 0 \end{pmatrix}, \tag{259}$$

the $\mathrm{Sp}(2,\mathbb{R})$ subalgebra acting on the right oscillator only

$$W_{R,R} = \begin{pmatrix} 0 & 0 & 0 & 0 \\ 0 & 1 & 0 & 0 \\ 0 & 0 & 0 & 0 \\ 0 & 0 & 0 & -1 \end{pmatrix}, \quad V_{R,R} = \begin{pmatrix} 0 & 0 & 0 & 0 \\ 0 & 0 & 0 & 0 \\ 0 & 0 & 0 & 0 \\ 0 & -\sqrt{2} & 0 & 0 \end{pmatrix}, \quad Z_{R,R} = \begin{pmatrix} 0 & 0 & 0 & 0 \\ 0 & 0 & 0 & \sqrt{2} \\ 0 & 0 & 0 & 0 \\ 0 & 0 & 0 & 0 \end{pmatrix}, \tag{260}$$

and the remaining generators which entangle the two oscillators

$$W_{L,R} = \begin{pmatrix} 0 & 0 & 0 & 0 \\ 1 & 0 & 0 & 0 \\ 0 & 0 & 0 & -1 \\ 0 & 0 & 0 & 0 \end{pmatrix}, \qquad W_{R,L} = \begin{pmatrix} 0 & 1 & 0 & 0 \\ 0 & 0 & 0 & 0 \\ 0 & 0 & 0 & 0 \\ 0 & 0 & -1 & 0 \end{pmatrix},$$

$$V_{L,R} = \begin{pmatrix} 0 & 0 & 0 & 0 \\ 0 & 0 & 0 & 0 \\ 0 & -1 & 0 & 0 \\ -1 & 0 & 0 & 0 \end{pmatrix}, \qquad Z_{L,R} = \begin{pmatrix} 0 & 0 & 0 & 1 \\ 0 & 0 & 1 & 0 \\ 0 & 0 & 0 & 0 \\ 0 & 0 & 0 & 0 \end{pmatrix}. \tag{261}$$

## D    Comments on bases

In this appendix, we comment on a few details with respect to our description of complexity in the scalar field theory. First recall from eq. (175) that the Hamiltonian describing the lattice regulated field theory can be written as

$$H = \sum_{a=1}^{N} \left( \frac{\delta}{2} P_a^2 + \frac{m^2}{2\delta} Q_a^2 + \frac{1}{2\delta^3} (Q_a - Q_{a+1})^2 \right), \tag{262}$$

which takes the form of a chain of $N$ coupled harmonic oscillators. Implicitly, we are choosing periodic boundary conditions here with $Q_{N+1} = Q_1$ and $P_{N+1} = P_1$. We wish to note that because the initial theory involved a real scalar field, the $P_a$ and $Q_a$ describe real degrees of freedom, *i.e.,* $P_a^\dagger = P_a$ and $Q_a^\dagger = Q_a$. However, the next step in our analysis in section 5.1 was to pass from the position basis along the chain to the normal mode basis by Fourier transforming as in eq. (178). At this point, the variables $\tilde{P}_k$ and $\tilde{Q}_k$ are no longer real.[60] Rather, we have $\tilde{P}_k^\dagger = \tilde{P}_{-k}$ and $\tilde{Q}_k^\dagger = \tilde{Q}_{-k}$. This constraint indicates that the positive and negative momentum

---

[60]With the exception of $k = 0$, and for even $N$, also $k = N/2$.

modes are mixed, *e.g.,* with the canonical commutation relations $[\tilde{Q}_k, \tilde{P}_{-l}] = i\,\delta_{k,l}$. In other words, the two complex degrees of freedom labeled by $(\tilde{Q}_k, \tilde{P}_k)$ and $(\tilde{Q}_{-k}, \tilde{P}_{-k})$ only contain two real degrees of freedom. This mixing is also evident in the corresponding Hamiltonian (180), which takes the form

$$H = \sum_{k=0}^{N-1} \left( \frac{\delta}{2} \tilde{P}_k \tilde{P}_{-k} + \frac{\omega_k^2}{2\delta} \tilde{Q}_k \tilde{Q}_{-k} \right). \tag{263}$$

Of course, we can also define Hermitian combinations of these operators. If we assume that $N = 2n + 1$, we would have

$$
\begin{aligned}
\tilde{Q}_{\mathrm{R},k} &= \frac{1}{\sqrt{2}} \left( \tilde{Q}_k + \tilde{Q}_{-k} \right), & \tilde{P}_{\mathrm{R},k} &= \frac{1}{\sqrt{2}} \left( \tilde{P}_k + \tilde{P}_{-k} \right), \\
\tilde{Q}_{\mathrm{I},k} &= \frac{i}{\sqrt{2}} \left( \tilde{Q}_k - \tilde{Q}_{-k} \right), & \tilde{P}_{\mathrm{I},k} &= \frac{i}{\sqrt{2}} \left( \tilde{P}_k - \tilde{P}_{-k} \right),
\end{aligned}
\tag{264}
$$

where $1 \le k \le n$ in all of these expressions. Now $(\tilde{Q}_{\mathrm{R},k}, \tilde{P}_{\mathrm{R},k})$ and $(\tilde{Q}_{\mathrm{I},k}, \tilde{P}_{\mathrm{I},k})$ obey canonical commutation relations, but otherwise these operators commute with each other, as they correspond to different real modes.[61] Further, with these new variables, the Hamiltonian (263) becomes

$$H = \frac{\delta}{2} \tilde{P}_0^2 + \frac{m^2}{2\delta} \tilde{Q}_0^2 + \sum_{k=1}^{n} \left( \frac{\delta}{2} \tilde{P}_{\mathrm{R},k}^2 + \frac{\omega_k^2}{2\delta} \tilde{Q}_{\mathrm{R},k}^2 + \frac{\delta}{2} \tilde{P}_{\mathrm{I},k}^2 + \frac{\omega_k^2}{2\delta} \tilde{Q}_{\mathrm{I},k}^2 \right), \tag{265}$$

which is a sum of $N = 2n+1$ independent real harmonic oscillators. A similar line of arguments can be used in order to express the ultralocal Hamiltonian (182) in terms of real harmonic oscillators. It is really when the regulated scalar field theory is expressed in terms of these (Hermitian) operators that we can easily evaluate the complexity by drawing upon the results in section 4 for the TFD state comprised of two (real) harmonic oscillators. Recall that in general, the complexity of a single mode depends on the frequency of the oscillator. Now with the assumption (which we proved for the $L^2$ norm with $\lambda_{\mathrm{R}} = 1$) that the total complexity is given by the sum of the complexities of the individual modes, we have

$$\mathcal{C}_{\mathrm{tot}} = \mathcal{C}(\omega_0) + 2 \sum_{k=0}^{n} \mathcal{C}(\omega_k) = \sum_{k=0}^{N-1} \mathcal{C}(\omega_k), \tag{266}$$

where in the second equality we have used the fact that $\omega_k = \omega_{N-k}$. Using these arguments, eq. (192) follows immediately by substituting the different frequencies in eqs. (150) and (152), and a similar expression is obtained for the $\mathcal{C}_1^{(\mathrm{LR})}$ complexity using eq. (151).

The relation between the real and complex variables can also be used to express the covariance matrix directly in terms of the complex variables $\tilde{G}_k^{a,b}(t) = \langle \mathrm{TFD}(t) | (\tilde{\xi}_k^a \tilde{\xi}_k^{\dagger b} + \tilde{\xi}_k^{\dagger b} \tilde{\xi}_k^a) | \mathrm{TFD}(t) \rangle$, where $\tilde{\xi}_k = (\tilde{q}_k^L, \tilde{q}_k^R, \tilde{p}_k^L, \tilde{p}_k^R)$, see eq. (236). To obtain the explicit expression for this expectation value, we should consider the covariance matrix (104) for the real modes together in terms of an $8 \times 8$ matrix ordered according to $V := (\tilde{Q}_{\mathrm{R},k}^L, \tilde{Q}_{\mathrm{I},k}^L, \tilde{P}_{\mathrm{R},k}^L, \tilde{P}_{\mathrm{I},k}^L, \ldots^R)^T$, where the ellipsis indicates the same ordering of the right (R) degrees of freedom. We then rotate this matrix to the complex coordinates $\tilde{V} := (\tilde{Q}_k^L, \tilde{Q}_{-k}^L, \tilde{P}_k^L, \tilde{P}_{-k}^L, \ldots^R)^T$ using eq. (264), namely $\tilde{V} = AV$ where

$$A = \frac{1}{\sqrt{2}} \bigoplus_{\substack{L,R \\ \mathrm{I,R}}} \begin{pmatrix} 1 & -i \\ 1 & i \end{pmatrix}. \tag{267}$$

---

[61] These modes could be constructed by replacing the complex exponentials in the Fourier transform (178) with $\cos(2\pi ka/N)$ and $\sin(2\pi ka/N)$.

The new covariance matrix will be related to the real mode expression according to $\tilde{G} = AGA^T$ which results in a block diagonal form for the $k$ and $-k$ modes where for instance the $4 \times 4$ block for the $k$ modes reads

$$\tilde{G}_k^{a,b}(t) = \begin{pmatrix} \frac{\mu_g}{\omega_k}\cosh(2\alpha_k) & \frac{\mu_g}{\omega_k}\cos(t\omega_k)\sinh(2\alpha_k) & 0 & -\sin(t\omega_k)\sinh(2\alpha_k) \\ \frac{\mu_g}{\omega_k}\cos(t\omega_k)\sinh(2\alpha_k) & \frac{\mu_g}{\omega_k}\cosh(2\alpha_k) & -\sin(t\omega_k)\sinh(2\alpha_k) & 0 \\ 0 & -\sin(t\omega_k)\sinh(2\alpha_k) & \frac{\omega_k}{\mu_g}\cosh(2\alpha_k) & -\frac{\omega_k}{\mu_g}\cos(t\omega_k)\sinh(2\alpha_k) \\ -\sin(t\omega_k)\sinh(2\alpha_k) & 0 & -\frac{\omega_k}{\mu_g}\cos(t\omega_k)\sinh(2\alpha_k) & \frac{\omega_k}{\mu_g}\cosh(2\alpha) \end{pmatrix},$$

(268)

where as defined in eq. (187), we have substituted $\lambda = \omega_k/\mu_g$ into eq. (104). This also agrees with eq. (236) where implicitly, we made the choice $\mu_g = \mathcal{L}$, but this choice is immaterial to the evaluation of the entanglement entropy in section 6.

Next, we would like to consider expressing the generators of our quantum circuits in terms of annihilation and creation operators. Following [23], in eq. (91), we expressed these generators in terms of a basis of Hermitian operators $(Q_i, P_i)$ with the standard canonical commutation relations,

$$[Q_i, P_j] = i\,\delta_{i,j}. \tag{269}$$

Here, the indices $i, j = 1, \ldots, N$ may specify either the positions along the one-dimensional lattice, or the real and imaginary parts of the Fourier modes, described above. This choice will not be important for the following discussion. Now in working with QFTs, it is also natural to express the quantum circuits in terms of annihilation and creation operators, *e.g.,* see ref. [51–53]. Therefore let us introduce the operators

$$a_i = \frac{\overline{\mu}_i}{\sqrt{2}}\left(Q_i + i\,\frac{P_i}{\overline{\mu}_i^2}\right), \qquad a_i^\dagger = \frac{\overline{\mu}_i}{\sqrt{2}}\left(Q_i - i\,\frac{P_i}{\overline{\mu}_i^2}\right), \tag{270}$$

satisfying $[a_i, a_j^\dagger] = \delta_{i,j}$ as usual. Here, to produce dimensionless operators, we need to introduce scales $\overline{\mu}_i$ but we have left these scales unspecified for the moment. If operators were chosen in the momentum basis,[62] then it would be natural to choose $\overline{\mu}_i = \sqrt{\omega_i/\delta}$, as was done in refs. [51–53], in which case the $a_i$ annihilate the usual vacuum state of the (regulated) scalar field theory. Alternatively, in the present context of our study of complexity, it would also be natural to fix $\overline{\mu}_i = \sqrt{\mu/\delta}$ in which case the $a_i$ instead annihilate the unentangled reference state, *i.e.,* the ground state of the ultralocal Hamiltonian introduced in eq. (181). Of course, changing these scales can be accomplished with a subgroup of Bogoliubov transformations, $\mathbb{R}^N \subset \mathrm{Sp}(2N, \mathbb{R})$, where we parametrize the rescaling matrices $\mathrm{diag}(e^{s_1}, e^{-s_1}, \cdots, e^{s_N}, e^{-s_N})$ by $s \in \mathbb{R}^N$, *e.g.,* see ref. [56].

Turning now to the generators of our unitary circuits, we can write all of the quadratic combinations of these annihilation and creation operators as follows

$$\widehat{T}_{i,j} = \widehat{T}_{j,i} = \begin{cases} \sqrt{2}\,(a_i^\dagger)^2 = \frac{\overline{\mu}_i^2}{\sqrt{2}}Q_i^2 - \frac{P_i^2}{\sqrt{2}\overline{\mu}_i^2} - \frac{i}{\sqrt{2}}\left(P_iQ_i + Q_iP_i\right), & i = j \\ 2\,a_i^\dagger a_j^\dagger = \overline{\mu}_i\overline{\mu}_j Q_iQ_j - \frac{P_iP_j}{\overline{\mu}_i\overline{\mu}_j} - i\left(\frac{\overline{\mu}_j}{\overline{\mu}_i}P_iQ_j + \frac{\overline{\mu}_i}{\overline{\mu}_j}Q_iP_j\right), & i \neq j \end{cases}$$

$$\widehat{\overline{T}}_{i,j} = \widehat{\overline{T}}_{j,i} = \begin{cases} \sqrt{2}\,a_i^2 = \frac{\overline{\mu}_i^2}{\sqrt{2}}Q_i^2 - \frac{P_i^2}{\sqrt{2}\overline{\mu}_i^2} + \frac{i}{\sqrt{2}}\left(P_iQ_i + Q_iP_i\right), & i = j \\ 2\,a_i a_j = \overline{\mu}_i\overline{\mu}_j Q_iQ_j - \frac{P_iP_j}{\overline{\mu}_i\overline{\mu}_j} + i\left(\frac{\overline{\mu}_j}{\overline{\mu}_i}P_iQ_j + \frac{\overline{\mu}_i}{\overline{\mu}_j}Q_iP_j\right), & i \neq j \end{cases} \tag{271}$$

$$\widehat{A}_{i,j} = \begin{cases} \sqrt{2}\left(a_i^\dagger a_i + \frac{1}{2}\right) = \frac{\overline{\mu}_i^2}{\sqrt{2}}Q_i^2 + \frac{P_i^2}{\sqrt{2}\overline{\mu}_i^2}, & i = j \\ 2\,a_i^\dagger a_j = \overline{\mu}_i\overline{\mu}_j Q_iQ_j + \frac{P_iP_j}{\overline{\mu}_i\overline{\mu}_j} - i\left(\frac{\overline{\mu}_j}{\overline{\mu}_i}P_iQ_j - \frac{\overline{\mu}_i}{\overline{\mu}_j}Q_iP_j\right). & i \neq j \end{cases}$$

---

[62]Or rather constructed with the corresponding Hermitian basis in eq. (264) since we specified that $(Q_i, P_i)$ are Hermitian operators.

While we have not constructed Hermitian generators above, this is easily done by considering the combinations $\widehat{T} + \overline{\widehat{T}}$, $i(\widehat{T} - \overline{\widehat{T}})$, $\widehat{A} + \widehat{A}^\dagger$ and $i(\widehat{A} - \widehat{A}^\dagger)$. We note that the generators $\widehat{A}$ span the $\mathfrak{u}(N)$ subalgebra of the full $\mathfrak{sp}(2N,\mathbb{R})$ algebra. These are the compact generators while the remaining generators encoded in $\widehat{T}$ and $\overline{\widehat{T}}$ are all noncompact [77].

In comparing the expressions above with the $\mathfrak{sp}(2N,\mathbb{R})$ generators in eq. (91), we conclude that it is rather unnatural to choose different scales $\overline{\mu}_i$ for each of the $(a_i, a_i^\dagger)$ pairs. Following the more logical approach of setting all of these scales to be the same, we denote this common scale as the gate scale, *i.e.,* $\overline{\mu}_i = \omega_g$, and then we find

$$
\begin{aligned}
\widehat{T}_{i,j} &= \widehat{V}_{i,j} - \widehat{Z}_{i,j} - \begin{cases} i\sqrt{2}\,\widehat{W}_{i,i}, & i = j \\ i\left(\widehat{W}_{i,j} + \widehat{W}_{j,i}\right), & i \neq j \end{cases} \\
\overline{\widehat{T}}_{i,j} &= \widehat{V}_{i,j} - \widehat{Z}_{i,j} + \begin{cases} i\sqrt{2}\,\widehat{W}_{i,i}, & i = j \\ i\left(\widehat{W}_{i,j} + \widehat{W}_{j,i}\right), & i \neq j \end{cases} \\
\widehat{A}_{i,j} &= \widehat{V}_{i,j} + \widehat{Z}_{i,j} - i\left(\widehat{W}_{i,j} - \widehat{W}_{j,i}\right),
\end{aligned}
\tag{272}
$$

where $\widehat{W}$, $\widehat{V}$ and $\widehat{Z}$ are the generators in eq. (91). This gives us a new insight into the role of the gate scale which was simply introduced to ensure that the $\mathfrak{sp}(2N,\mathbb{R})$ generators were dimensionless. Here, we see that these generators are then naturally written in terms of annihilation and creation operators which annihilate a state where the degrees of freedom are all unentangled and the variance of all these Gaussians is set by $\omega_g$. Choosing this state to be the reference state may again seem like the natural choice, *i.e.,* it seems natural to choose $\overline{\mu}_i = \omega_g = \sqrt{\mu/\delta}$. These observations may give us some additional confidence in focusing on the case of $\lambda_\text{R} = 1$ in our analysis of the complexity in the main text.

## E  Complexity of formation in the diagonal basis

We have described in the main text the general behaviour of the complexity of formation in the diagonal basis given by eq. (209) if the reference scale is in the UV, with $\mu > \Lambda$. We will now analyze two more examples: when the reference scale is exactly equal to the cutoff $\mu = \Lambda$ and when the reference scale is in the IR, with $\mu < \Lambda$.

Let us start by the choice $\mu = \Lambda$. For instance, in cMERA circuits this is suggested as the natural reference scale for an ultralocal scalar theory. We will find that there are two important regimes to consider in this case. We focus on the only argument of the absolute value in eq. (209) that may change sign which is $\frac{1}{2}\log\frac{k}{\mu} + \alpha_k$. The sign flips when there is some $k_f$ in the range $[0,\Lambda]$ that satisfies the transcendental equation

$$
k_f \coth\left(\frac{k_f}{4T}\right) = \Lambda.
\tag{273}
$$

There are then two interesting ranges of temperatures namely, $0 < T < \Lambda/4$ and $T > \Lambda/4$. Focusing first on the case $0 < T < \Lambda/4$, we find that there is a $k_f$ in the range of integrated momenta that satisfies eq. (273). In this case for $k < k_f$, we have that $\frac{1}{2}\log\frac{k}{\mu} + \alpha_k$ is always negative, and the integrand in eq. (209) vanishes, while for $k > k_f$, we have that $\frac{1}{2}\log\frac{k}{\mu} + \alpha_k$ is always positive. The complexity of formation is then given by

$$
\Delta\mathcal{C}_1^{(\pm)}\left(T < \frac{\Lambda}{4}\right) = \text{vol}\,\frac{\Omega_{d-2}}{(2\pi)^{d-1}}\int_{k_f}^{\Lambda} dk\,k^{d-2}\left(2\alpha_k + \log\frac{k}{\mu}\right).
\tag{274}
$$

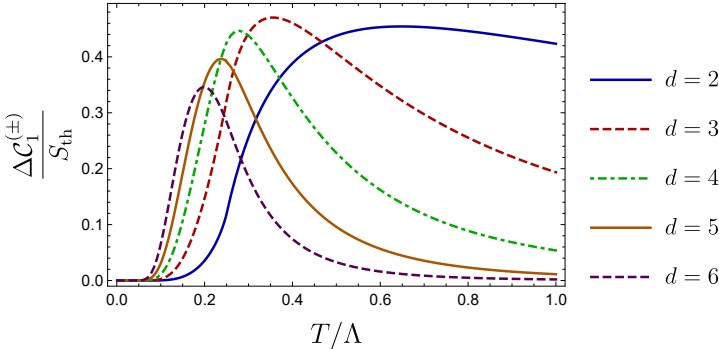

Figure 26: Complexity of formation normalized by the entropy in the diagonal basis for dimensions $d = 2, 3, 4, 5, 6$ (blue, dashed red, dot-dashed green, orange, dashed purple) and $\mu = \Lambda$. For small temperatures, the profile of the ratio of complexity of formation over temperature becomes small. For higher temperatures, it develops a dependence on the temperature and the cutoff scale $\Lambda$. The fact that the complexity of formation is either zero or not proportional to the entropy contrasts with the holographic results of ref. [27].

For $T > \Lambda/4$, we have that $\frac{1}{2} \log \frac{k}{\mu} + \alpha_k$ is always positive and does not change sign for $k$ in the range $[0, \Lambda]$. All the other arguments of the absolute values are negative, and the complexity of formation then reads

$$\Delta \mathcal{C}_1^{(\pm)}\left(T > \frac{\Lambda}{4}\right) = \text{vol} \frac{\Omega_{d-2}}{(2\pi)^{d-1}} \int_0^\Lambda dk\, k^{d-2}\left(2\alpha_k + \log \frac{k}{\mu}\right). \tag{275}$$

We show in fig. 26 the integrated complexity of formation in the diagonal basis divided by the entropy for several dimensions and $\mu = \Lambda$. For small temperatures with respect to the cutoff scale $\Lambda$, the ratio of the complexity of formation and entropy goes to zero, and for large temperatures it develops a nontrivial profile.

The sign flips in the arguments of the absolute values are slightly more complicated if the reference scale is in the IR, or $\mu < \Lambda$. There are many possible sign flips in the arguments of the absolute values in eq. (209). We denote by $k_{\text{sum}}$ values of the momentum related to the sign flip of the absolute value of $\frac{1}{2} \log \frac{k}{\mu} + \alpha_k$ and by $k_{\text{sub}}$ values of momentum related to the sign flip of absolute value of $\frac{1}{2} \log \frac{k}{\mu} - \alpha_k$. There are sign flips if $k_{\text{sum}}$ and $k_{\text{sub}}$ satisfy the transcendental equations

$$k_{\text{sum}} \coth\left(\frac{k_{\text{sum}}}{4T}\right) = \mu, \qquad k_{\text{sub}} \tanh\left(\frac{k_{\text{sub}}}{4T}\right) = \mu. \tag{276}$$

Therefore, there are two critical temperatures associated with sign flips given in eq. (276),

$$T_{c1}^{IR} = \frac{\mu}{4}, \qquad T_{c2}^{IR} = \frac{\Lambda}{2} \frac{1}{\log\left(\frac{\Lambda+\mu}{\Lambda-\mu}\right)}. \tag{277}$$

For the term in eq. (209) with $\frac{1}{2} \log \frac{k}{\mu} + \alpha_k$, if the temperature satisfies $T < T_{c1}^{IR}$, then one has to solve the transcendental equation (276) for $k_{\text{sum}}$ to find the momentum at which there is a sign flip. In this regime, the sum in question is negative for $k < k_{\text{sum}}$ and positive for $k > k_{\text{sum}}$. If $T > T_{c1}^{IR}$, then the sum $\frac{1}{2} \log \frac{k}{\mu} + \alpha_k$ is always positive. For the argument of the absolute value in eq. (276) given by $\frac{1}{2} \log \frac{k}{\mu} - \alpha_k$, there are the following cases to analyze. If $T < T_{c2}^{IR}$, one has to solve eq. (276) for $k_{\text{sub}}$, and the subtraction is negative for $k < k_{\text{sub}}$, and

positive for $k > k_{\text{sub}}$. If $T > T_{c2}^{IR}$, the subtraction is always negative in the range of momentum $[0, \Lambda]$. In order to study the overall behaviour of eq. (276) when $\mu < \Lambda$, we should break down the different cases depending on whether $T_{c1}^{IR}$ is smaller or bigger than $T_{c2}^{IR}$. Without loss of generality, we will consider $T_{c1}^{IR} < T_{c2}^{IR}$, which from eq. (277) is satisfied if $\mu < 0.833557\Lambda$. There are three separate regimes to consider in this case. If $T < T_{c1}^{IR}$, one has to solve for $k_{\text{sum}}$ and $k_{\text{sub}}$ in eq. (276) in order to find the momentum where the sign flips for all the individual terms. The complexity of formation in this regime reads

$$
\begin{aligned}
\Delta C_1^{(\pm)}(T < T_{c1}^{IR}) = \frac{\text{vol}\,\Omega_{d-2}}{(2\pi)^{d-1}} \bigg[ &- \int_0^{k_{\text{sum}}} dk\, k^{d-2} \left( \alpha_k + \frac{1}{2} \log\left(\frac{k}{\mu}\right) \right) \\
&+ \int_{k_{\text{sum}}}^{\Lambda} dk\, k^{d-2} \left( \alpha_k + \frac{1}{2} \log\left(\frac{k}{\mu}\right) \right) - \int_0^{k_{\text{sub}}} dk\, k^{d-2} \left( \frac{1}{2} \log\left(\frac{k}{\mu}\right) - \alpha_k \right) \\
&+ \int_{k_{\text{sub}}}^{\Lambda} dk\, k^{d-2} \left( \frac{1}{2} \log\left(\frac{k}{\mu}\right) - \alpha_k \right) + \int_0^{\mu} dk\, k^{d-2} \log\left(\frac{k}{\mu}\right) - \int_{\mu}^{\Lambda} dk\, k^{d-2} \log\left(\frac{k}{\mu}\right) \bigg].
\end{aligned}
\tag{278}
$$

Next, in the range $T_{c1} < T < T_{c2}$, the sum $\frac{1}{2} \log \frac{k}{\mu} + \alpha_k$ is always positive, and the subtraction $\frac{1}{2} \log \frac{k}{\mu} - \alpha_k$ changes sign at $k_{\text{sub}}$ given by eq. (276). The expression for the complexity of formation reads

$$
\begin{aligned}
\Delta C_1^{(\pm)}(T_{c1}^{IR} < T < T_{c2}^{IR}) = \frac{\text{vol}\,\Omega_{d-2}}{(2\pi)^{d-1}} \bigg[ &\int_0^{\Lambda} dk\, k^{d-2} \left( \alpha_k + \frac{1}{2} \log\left(\frac{k}{\mu}\right) \right) \\
&- \int_0^{k_{\text{sub}}} dk\, k^{d-2} \left( \frac{1}{2} \log\left(\frac{k}{\mu}\right) - \alpha_k \right) + \int_{k_{\text{sub}}}^{\Lambda} dk\, k^{d-2} \left( \frac{1}{2} \log\left(\frac{k}{\mu}\right) - \alpha_k \right) \\
&+ \int_0^{\mu} dk\, k^{d-2} \log\left(\frac{k}{\mu}\right) - \int_{\mu}^{\Lambda} dk\, k^{d-2} \log\left(\frac{k}{\mu}\right) \bigg].
\end{aligned}
\tag{279}
$$

Finally, if $T > T_{c2}$, then the sum $\frac{1}{2} \log \frac{k}{\mu} + \alpha_k$ is always positive and the subtraction $\frac{1}{2} \log \frac{k}{\mu} - \alpha_k$ is always negative in the range of momenta $[0, \Lambda]$. The complexity of formation then reads

$$
\begin{aligned}
\Delta C_1^{(\pm)}(T > T_{c2}^{IR}) = \frac{\text{vol}\,\Omega_{d-2}}{(2\pi)^{d-1}} \bigg[ &\int_0^{\Lambda} dk\, k^{d-2}\, 2\alpha_k + \int_0^{\mu} dk\, k^{d-2} \log\left(\frac{k}{\mu}\right) \\
&- \int_{\mu}^{\Lambda} dk\, k^{d-2} \log\left(\frac{k}{\mu}\right) \bigg].
\end{aligned}
\tag{280}
$$

We show the profile of the complexity of formation for $\mu = \Lambda/2$ in fig. 27. Analogously to the previous cases, the ratio of the complexity of formation and the entropy has a nontrivial profile that in principle depends on $T$ and the cutoff scale $\Lambda$. In fig. 28, we compare the ratio of the complexity of formation over the entropy for different reference state scales for $d = 4$. We notice that for smaller reference state scale, the peak of the profile is larger and happens at smaller temperatures.

## F Minimal geodesics for $N$ degrees of freedom with $\lambda_R = 1$

In section 5, we have proposed that the shortest geodesic (with respect to the unpenalized metric) connecting a Gaussian reference state $|G_R\rangle$ with a Gaussian target state $|G_T\rangle$ is just given by a combination of squeezing operations on independent normal modes. In this appendix, we prove this result for the full group $\text{Sp}(2N, \mathbb{R})$ but only for the $F_2$ or $D_{\kappa=2}$ norms with the choice $\lambda_R = 1$. While the latter choice does not modify the underlying geometry on the space

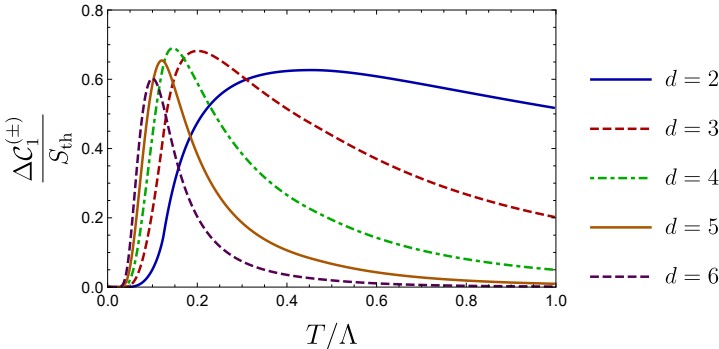

Figure 27: Complexity of formation normalized by the entropy in the diagonal basis for dimensions $d = 2, 3, 4, 5, 6$ (blue, dashed red, dot-dashed green, orange, dashed purple) and $\mu = \Lambda/2$. For small temperatures, the profile of the ratio of the complexity of formation over the entropy becomes small. For higher temperatures, it develops a dependence on the temperature and the cutoff scale $\Lambda$. Once again, the nontrivial profile of the ratio of the complexity of formation to the entropy contrasts with the holographic results of ref. [27].

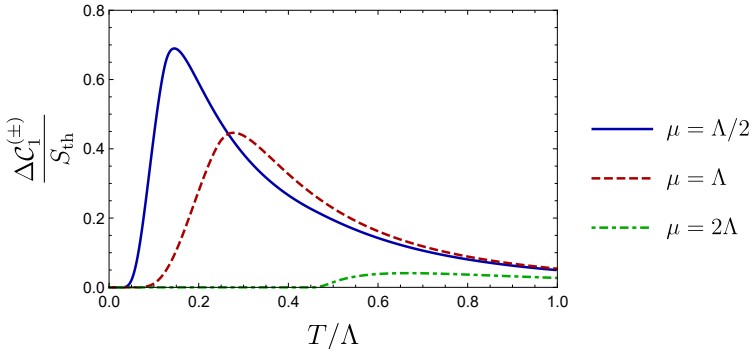

Figure 28: Complexity of formation normalized by the entropy in the diagonal basis for $d = 4$, varying the reference scale $\mu$. For the smaller $\mu$, the curve is shifted to the left and has a larger maximum value.

of unitaries, it does modify the boundary conditions describing a particular target state. Recall that the latter is not achieved by a single unitary transformation $U$ (acting on the reference state) but rather by a family of unitaries $U U_R$, where $U_R$ belongs to the stabilizer group (83) of the reference state. As we will see below, with $\lambda_R = 1$, the stabilizer group takes a particularly simple form and allows us to make use of a particular fiber bundle structure. Note that a very similar discussion of the fermionic case can be found in ref. [56].

Our proof requires some Lie group techniques together with some well-known decompositions of symplectic group elements. We also rely on the conventions established in section 3. We assemble the $N$ degrees of freedom together as $\xi := (q_1, \cdots, q_N, p_1, \cdots, p_N)$. Of course, with this choice of the standard basis of conjugate positions and momenta, the canonical commutation relations are given by $[\xi^a, \xi^b] = i\Omega^{a,b}$ where $\Omega^{a,b}$ is the canonical symplectic form in eq. (42). The Gaussian states $|G\rangle$ are then completely characterized by the symmetric part of their covariance matrix $G^{a,b}$ in eq. (43).[63] Finally recall that as described in section 3.2, we are naturally led to a representation of the symplectic group acting on the covariance matrix

---

[63]Recall that we only consider Gaussian states $|G\rangle$ satisfying $\langle G|\hat{\xi}^a|G\rangle = 0$. Further, we ignore the complex phase of a Gaussian state $|G\rangle$, because complexity is assigned to distinguishable quantum states, while $e^{i\phi}|G\rangle$ for different $\phi \in \mathbb{R}$ represents the same state.

where the generators $K \in \mathfrak{sp}(2N, \mathbb{R})$ are given by

$$K^a{}_b = \Omega^{a,c} k_{c,b} \, , \tag{281}$$

where $k_{a,b}$ define the quadratic operators appearing in the gates, as in eq. (48).

## Lie group geometry

In the Nielsen approach to complexity, we equip the Lie group $\mathrm{Sp}(2N, \mathbb{R})$ with a right invariant and positive metric. Such a metric is completely characterized by its value at the identity where we identify the tangent space $T_{\mathbb{1}} \mathrm{Sp}(2N, \mathbb{R})$ with its Lie algebra $\mathfrak{sp}(2N, \mathbb{R})$. We represent generators $A \in \mathfrak{sp}(2N, \mathbb{R})$ as matrices, namely linear maps $A^a{}_b$ on phase space.

In order to define our right invariant metric, we need to specify its value at the identity by giving a prescription how to compute the inner product $\langle A, B \rangle_{\mathbb{1}}$ between the two generators $A, B \in \mathfrak{sp}(2N, \mathbb{R})$. For a given Gaussian reference state $|G_R\rangle$, there is a natural metric that we call the *unpenalized metric* because it is the generalization of the unpenalized metric (109) on $\mathrm{Sp}(2, \mathbb{R})$ to systems with more degrees of freedom. This matrix is defined as

$$\langle A, B \rangle_{\mathbb{1}} = A^a{}_b \, G_R^{b,c} \, (B^\intercal)_c{}^d \, (g_R)_{d,a} = \mathrm{Tr}(A \, G_R \, B^\intercal \, g_R) \, . \tag{282}$$

Given two tangent vectors $X, Y \in T_U \mathrm{Sp}(2N, \mathbb{R})$ represented as matrices at a point $U \in \mathrm{Sp}(2N, \mathbb{R})$, we can find their inner product from the right-invariance by multiplying with $U^{-1}$ from the right, namely

$$\langle X, Y \rangle_U = \langle X U^{-1}, Y U^{-1} \rangle_{\mathbb{1}} \, , \tag{283}$$

where we can apply eq. (282).

## Fiber bundle structure

The choice of the reference state $|G_R\rangle$ equips the Lie group $\mathrm{Sp}(2N, \mathbb{R})$ with a fiber bundle structure. There exist symplectic group elements $U$ that leave the reference state invariant, namely the stabilizer subgroup $\mathrm{Sta}_{G_R} \subset \mathrm{Sp}(2N, \mathbb{R})$ in eq. (83). We explicitly introduce the choice $\lambda_R = 1$ because this simplifies the covariance matrix for the reference state to being a $2N \times 2N$ identity matrix, *i.e.*, $G_R = \mathbb{1}_{2N}$ for $\lambda_R = 1$ (compare to eq. (47) for $N = 1$). Then the elements of the stabilizer group are both symplectic (with respect to $\Omega$) and orthogonal (*i.e.*, the reference state is left invariant: $G_R = U G_R U^\intercal$ with $G_R = \mathbb{1}_{2N}$), and so they form the subgroup

$$\mathrm{Sta}_{G_R} = \mathrm{U}(N) = \mathrm{Sp}(2N, \mathbb{R}) \cap \mathrm{O}(2N) \, . \tag{284}$$

It is well known that $\mathrm{U}(N)$ is the largest subgroup of $\mathrm{Sp}(2N, \mathbb{R})$ and that the different choices of how to embed $\mathrm{U}(N)$ into $\mathrm{Sp}(2N, \mathbb{R})$ are in one-to-one correspondence to the different choices of metrics $G_R$. This is not surprising because every choice of a metric $G_R$ chooses a different subset $\mathrm{Sta}_{G_R}$ that are all isomorphic to $\mathrm{U}(N)$.

We define the equivalence relation $U \sim \tilde{U}$ if and only if $U G_R U^\intercal = \tilde{U} G_R \tilde{U}^\intercal$. This means acting with $U$ and $\tilde{U}$ on $G_R$ will give the same target state. In particular, the stabilizer subgroup, *i.e.*, $\mathrm{U}(N)$, is equal to the equivalence class $[\mathbb{1}]$ of the identity. Moreover, for every pair $U \sim \tilde{U}$, there exists a $u \in \mathrm{U}(N)$, such that $U u = \tilde{U}$.[64] Therefore, $\mathrm{Sp}(2N, \mathbb{R})$ becomes a fiber bundle where the fibers correspond to the different equivalence classes diffeomorphic to $\mathrm{U}(N)$ and the base manifold is given by the quotient

$$\mathcal{U} = \mathrm{Sp}(2N, \mathbb{R})/\sim = \mathrm{Sp}(2N, \mathbb{R})/\mathrm{U}(N) \, . \tag{285}$$

---

[64]Recall eq. (118) for the special case of $\mathrm{Sp}(2, \mathbb{R})$ in section 4.



This space is diffeomorphic to $\mathbb{R}^{N(N+1)}$ as a manifold and generally referred to as a symmetric space of type CI. We will refer to $\mathcal{U}$ as the space of pure Gaussian states and identify a point $[U] \in \mathcal{U}$ with the Gaussian state $|UG_{\mathrm{R}}U^{\mathsf{T}}\rangle$ up to an overall complex phase.

**Polar decomposition**

Identifying the Lie algebra $\mathfrak{sp}(2N, \mathbb{R})$ with the tangent space at the identity, we have a natural "vertical" subalgebra $\mathfrak{u}(N) \subset \mathfrak{sp}(2N, \mathbb{R})$ that is tangential to the fiber $[\mathbb{1}] = \mathrm{U}(N)$. A priori, there is no natural "horizontal" complement to write the Lie algebra as a direct sum of a vertical and a horizontal part. However, by equipping the Lie algebra with the inner product $\langle \cdot, \cdot \rangle_{\mathbb{1}}$ in eq. (282), we can choose the orthogonal complement

$$\mathrm{sym}(N) := \left\{ A \in \mathfrak{sp}(2N, \mathbb{R}) \big| \langle A, B \rangle_{\mathbb{1}} = 0 \, \forall B \in \mathfrak{u}(N) \right\}. \tag{286}$$

In contrast to $\mathfrak{u}(N)$, $\mathrm{sym}(N)$ is not a subalgebra. Its name stems from the fact that the decomposition

$$\mathfrak{sp}(2N, \mathbb{R}) = \mathrm{sym}(N) \oplus \mathfrak{u}(N) \tag{287}$$

is equivalent to splitting the set of generators into symmetric and antisymmetric matrices with respect to the metric $G_{\mathrm{R}}$ of the reference state:

- **Vertical subspace** $\mathfrak{u}(N)$
  A generator $B$ in the subspace $\mathfrak{u}(N)$ must preserve the reference state $G_{\mathrm{R}}$. It satisfies

$$B \, G_{\mathrm{R}} = -G_{\mathrm{R}} B^{\mathsf{T}}, \tag{288}$$

  which is equivalent to $B$ being an antisymmetric matrix in a basis where $G_{\mathrm{R}}$ is the identity.

- **Horizontal subspace** $\mathrm{sym}(N) = \mathfrak{u}_\perp(N)$
  A generator $A$ that is orthogonal to all elements $B \in \mathfrak{u}(N)$ satisfies

$$0 = \langle A, B \rangle_{\mathbb{1}} = \mathrm{Tr}(AG_{\mathrm{R}}B^{\mathsf{T}}g_{\mathrm{R}}). \tag{289}$$

  In a basis where $G_{\mathrm{R}}$ and $g_{\mathrm{R}}$ are just the identity, we are searching for a matrix $A$ that has zero trace when multiplied with any antisymmetric matrix $B$. This condition is equivalent to stating that $A$ is a symmetric matrix, namely satisfying

$$A G_{\mathrm{R}} = G_{\mathrm{R}} A^{\mathsf{T}}. \tag{290}$$

  We can refer to $\mathrm{sym}(N)$ as orthogonal complement of $\mathfrak{u}(N)$, *i.e.,* $\mathfrak{u}_\perp(N)$.

We can exponentiate the space $\mathrm{sym}(N)$ to define the $N(N+1)$-dimensional submanifold

$$\mathrm{Sym}_+(N) = \exp(\mathrm{sym}(N)) = \left\{ e^A \big| A \in \mathrm{sym}(N) \right\} \tag{291}$$

consisting of all symplectic group elements that are symmetric with respect to $G_{\mathrm{R}}$ and have positive eigenvalues. The fact that symmetric generators $A$ are diagonalizable with real eigenvalues implies that the exponential map provides a diffeomorphism and $\mathrm{Sym}_+(N)$ is thus diffeomorphic to $\mathbb{R}^{N(N+1)}$. Let us emphasize that $\mathrm{Sym}_+(N)$ is not flat with respect to our right-invariant metric, but rather some complicated embedded curved surface.

The polar decomposition of a symplectic group element $U$ is given by

$$U = Tu \quad \text{with} \quad T = \sqrt{UG_{\mathrm{R}}U^{\mathsf{T}}g_{\mathrm{R}}} \in \mathrm{Sym}_+(N) \quad \text{and} \quad u = T^{-1}U \in \mathrm{U}(N). \tag{292}$$

It is unique and provides a diffeomorphism between the symplectic group and the Cartesian product $Sym_+(N) \times U(N)$. In particular, it provides a trivialization of the fiber bundle on $Sp(2N, \mathbb{R})$ where the base manifold is identified with surface $Sym_+(N)$, from which we can move up and down along the fiber by multiplying with group elements $u \in U(N)$. Due to the fact that $Sym_+(N)$ is diffeomorphic to $sym(N)$, we can use the pair $(A, u)$ as generalized polar coordinates for any group element $U(A, u)$

$$U(A, u) = e^A u \quad \text{with} \quad A \in sym(N) \quad \text{and} \quad u \in U(N). \tag{293}$$

**Cylindrical foliation**

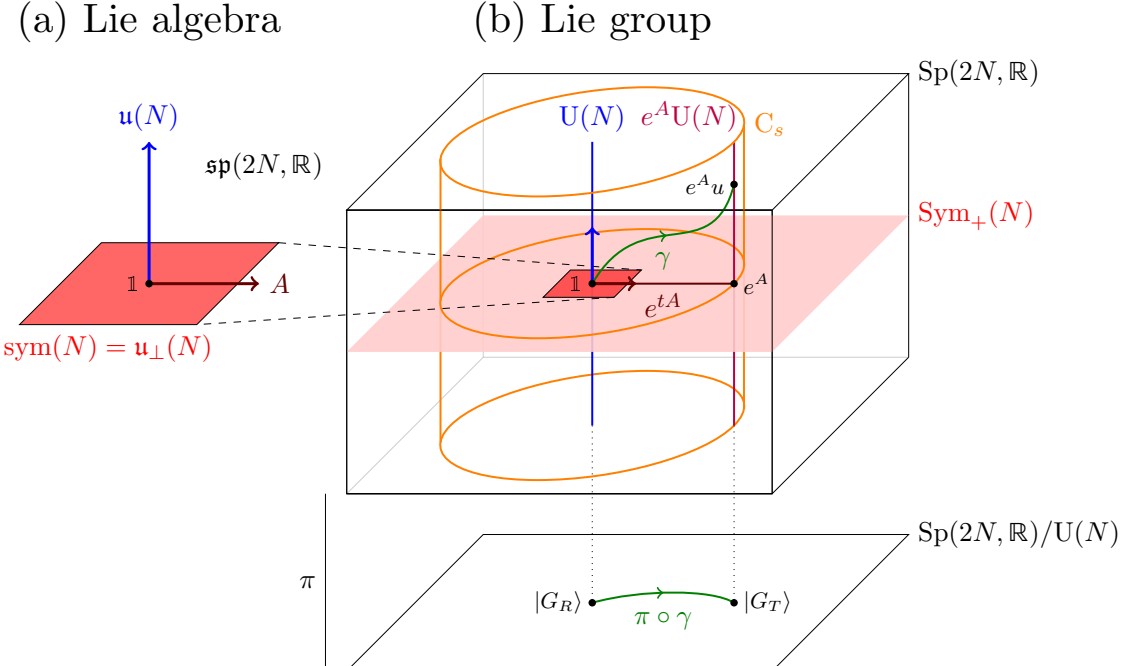

Figure 29: This sketch illustrates the geometry of the Lie algebra $\mathfrak{sp}(2N, \mathbb{R})$ and the Lie group $Sp(2N, \mathbb{R})$. (a) The Lie algebra can be decomposed as $\mathfrak{sp}(2N, \mathbb{R}) = \mathfrak{u}(N) \oplus sym(N)$, such that $sym(N)$ is the orthogonal complement $\mathfrak{u}_\perp(N)$ of $\mathfrak{u}(N)$. In particular, we can choose a vector $A \in sym(N)$. (b) The Lie group can be represented as fiber bundle over its quotient given by the symmetric space $Sp(2N, \mathbb{R})/U(N)$. This base manifold can be interpreted as the space of Gaussian quantum states. The fiber over the reference state $|G_R\rangle$ is given by the subgroup $U(N) \subset Sp(2N, \mathbb{R})$, while the fiber $e^A U(N)$ over any target state $|G_T\rangle$ is not a subgroup. We consider a path $\gamma$ in the group that connects $\mathbb{1}$ to some other group element $U = e^A u$. Such a point lies on the cylinder $C_s$ with $s = \|A\|$. Every curve $\gamma$ in the group can be projected down to the curve $\pi \circ \gamma$ in the base manifold. The vertical submanifold $Sym_+(N) = \exp(sym(N))$ is generated by exponentiating $sym(N)$ and it plays an important role because it contains the minimal geodesics. In particular, the straight line $e^{tA}$ connecting $\mathbb{1}$ with $e^A$ will turn out to be the minimal geodesic between $\mathbb{1}$ and the fiber $e^A U(N)$. We do *not* show the vector field $R$ consisting of radially outwards pointing unit vectors on the cylindrical surfaces $C_s$, such that the curves $e^{tA} u$ are its integral curves. Note that a similar sketch was used in ref. [56] to describe the setting of fermionic Gaussian states.

Using the polar coordinates (293), we can foliate the symplectic group by generalized

cylinders defined as

$$\mathsf{C}_s = \left\{ e^A u \,\middle|\, A \in \mathrm{sym}(N), \|A\| = s, u \in \mathrm{U}(N) \right\}, \tag{294}$$

with the topology $S^{N(N+1)-1} \times \mathrm{U}(N)$.[65] Moreover, we will define the radial vector field $R$ at point $U(A, u) \in \mathrm{Sp}(2N, \mathbb{R})$ given by

$$\mathcal{R}_{U(A,u)} = \frac{1}{\|A\|} A e^A u. \tag{295}$$

We will prove that this vector fields points radially outwards and is everywhere orthogonal to the cylindrical surfaces $\mathsf{C}_s$. We will show the orthogonality by considering different directions individually. Note that the normalization $1/\|A\|$ is irrelevant here.

- **Orthogonality to the $\mathrm{U}(N)$ fiber:**
  We show that $R$ is orthogonal to any vector pointing along the $\mathrm{U}(N)$ fiber. Let $X \in \mathfrak{u}(N)$, so that $e^A u X$ points in the direction of the $\mathrm{U}(N)$ fiber at point $U(A, u)$. We can compute the inner product

  $$\langle \mathcal{R}_{U(A,u)}, e^A u X \rangle = \frac{1}{\|A\|} \langle e^A A u, e^A u X \rangle_{e^A u}. \tag{296}$$

  We define $Y = uXu^{-1}$ which lies in $\mathfrak{u}(N)$ because $\mathfrak{u}(N)$ is a subgroup. This implies $uX = Yu$. We can therefore compute

  $$\langle e^A u X, e^A A u \rangle_{e^A u} = \langle e^A Y u, e^A A u \rangle_{e^A u} = \langle e^A Y, e^A A \rangle_{e^A} = \langle e^A T e^{-A}, A \rangle_{\mathbb{1}}. \tag{297}$$

  At this point, we can use the explicit form of the metric at the identity given by

  $$\langle e^A Y e^{-A}, A \rangle_{\mathbb{1}} = \mathrm{tr}\left( e^A Y e^{-A} G_{\mathrm{R}} A^{\mathsf{T}} g_{\mathrm{R}} \right) = \mathrm{tr}\left( e^A Y e^{-A} A \right) = \mathrm{tr}(YA) = 0, \tag{298}$$

  where we have used $G_{\mathrm{R}} A^{\mathsf{T}} g_{\mathrm{R}} = A$ for $A \in \mathrm{sym}(N)$.

- **Orthogonality to a generator $A \in \mathrm{sym}(N)$ preserving $\mathsf{C}_s$:**
  This second computation is slightly more involved. Let us look at a point $U = e^A u$ and ask what are the directions $B \in T_U \mathrm{Sp}(2N, \mathbb{R})$ that are tangential to the surface $\mathsf{C}_s$, but also to the surface $\exp\left(\mathrm{sym}(N)\right) u$. We can describe such elements by choosing a second generator $B \in \mathrm{sym}(N)$ that is orthogonal to $A$ with $\|A\| = \|B\|$. The circle

  $$\gamma(t) = e^{(\cos(t)A + \sin(t)B)} u \tag{299}$$

  lies in $\mathrm{Sym}_+(N)$ and on $\mathsf{C}_s$ with $s = \|A\| = \|B\|$. This gives rise to the tangent vector

  $$\dot{\gamma}(0) = \frac{d}{dt} e^{(A + tB)} \big|_{t=0}. \tag{300}$$

  We can compute the inner product with $\mathcal{R}_{U(A,u)}$ using $A = G_{\mathrm{R}} A^{\mathsf{T}} g_{\mathrm{R}}$ as

  $$\langle \mathcal{R}_{U(A,u)}, \dot{\gamma}(0) \rangle_{e^A u} = \frac{1}{\|A\|} \langle A, \dot{\gamma}(0) u^{-1} e^{-A} \rangle_{\mathbb{1}} = \frac{d}{dt} \mathrm{tr}(A e^{(A+tB)} e^{-A}). \tag{301}$$

  At this point, we write out the full exponential as

  $$\sum_{n,m=0}^{\infty} \frac{d}{dt} \frac{\mathrm{tr}[A(A+tB)^n(-A)^m]}{n!m!} \bigg|_{t=0} = \mathrm{tr}\left[ AB \sum_{n=1,m=0}^{\infty} \frac{(A)^{n-1}(-A)^m}{(n-1)!m!} \right] = \mathrm{tr}(AB) = 0, \tag{302}$$

  where we have used the fact that trace is cyclic and that $B$ was chosen orthogonal to $A$. Note that the sum just gives the identity.

---

[65] Note that since $A \in \mathrm{sym}(N)$, we may apply eq. (290) to find $\|A\|^2 = \langle A, A \rangle_{\mathbb{1}} = \mathrm{Tr}(A^2)$.

This proves that we have indeed a vector field $R$ that is everywhere orthogonal to the cylindrical surfaces $C_s$. Furthermore, we can quickly confirm that this vector field indeed has a constant length equal to 1, by computing

$$\langle \mathcal{R}_{U(A,u)}, \mathcal{R}_{U(A,u)} \rangle_{U(A,u)} = \frac{\langle e^A A u, e^A A u \rangle_{e^A u}}{\|A\|^2} = \frac{\langle A, A \rangle_{\mathbb{1}}}{\|A\|^2} = 1 \,. \tag{303}$$

Given a trajectory $\gamma : [0,1] \to \mathrm{Sp}(2,\mathbb{R}) : t \mapsto \gamma(t)$, we can compute how the coordinate $s(\gamma(t))$ changes. Due to the fact that the vector field $R$ is orthogonal to the surface $C_s$ of constant $s$ and correctly normalized, we have

$$ds = \langle \mathcal{R}_{\gamma(t)}, \dot{\gamma}(t) \rangle_{\gamma(t)} \,. \tag{304}$$

**Inequality for the geodesic length**

We will now use the cylindrical structure to bound the geodesic length from below. Given an arbitrary point $U(A,u) = e^A u$ on the cylinder $\mathcal{C}_s$, let us assume that we have already found the shortest path connecting the identity $\mathbb{1}$ with $U(A,u)$. This path may be given by $\gamma(t)$ with $\gamma(0) = \mathbb{1}$ and $\gamma(1) = U(A,u)$. We can compute the change $ds$ as the inner product

$$ds(t) = dt \, \langle \dot{\gamma}(t), \mathcal{R}_{\gamma(t)} \rangle_{\gamma(t)} \,. \tag{305}$$

Clearly, if we integrate this inner product we find how far we move in the $s$-direction. This follows directly from the fact that moving in the direction of $R$ increases $s$ with a constant rate, while moving along any orthogonal direction does not change $s$. Therefore, we have

$$s = \int_0^1 ds(t) = \int_0^1 dt \, \langle \dot{\gamma}(t), \mathcal{R}_{\gamma(t)} \rangle_{\gamma(t)} \,. \tag{306}$$

We can compare this with the actual length of the geodesic given by

$$\|\gamma\| := \int_0^1 dt \, \|\dot{\gamma}(t)\| \,. \tag{307}$$

At this point, we should note that $\langle \dot{\gamma}(t), \mathcal{R}_{\gamma(t)} \rangle_{\gamma(t)} \leq \|\dot{\gamma}(t)\|$ for all $t$. This follows from the fact that we are projecting onto the unit vector $R$, so this projection is at most the length of $\dot{\gamma}(t)$. We can combine these two equation to find the important inequality

$$s \leq \|\gamma\| \,, \tag{308}$$

stating compactly that any path connecting $\mathbb{1}$ with $U \in C_s$ must have a length of $s$ or more.

**Shortest path to a fiber $e^A \mathrm{U}(N)$**

At this point, we have not proven that for every $U \in \mathrm{Sp}(2N, \mathbb{R})$ there exists a path with length $s$ and there certainly are points $U$ where we cannot find such a shortest path. However, we are interested in the minimal geodesic that connects the identity $\mathbb{1}$ with an arbitrary point in the fiber $[U]$. This means that if we find a single path that does this with length $s$, we have proven that this is indeed *the* optimal path and there is no shorter one.

We can do this for the fiber of $e^A \mathrm{U}(N)$ by checking that the path

$$\gamma(t) = e^{tA} \tag{309}$$

satisfies exactly these conditions and reaches the representative $e^A$ at $t = 1$. This path has length $\|\gamma\| = \|A\| = s$. At this point, we have proven that for the "unpenalized" inner product discussed at the beginning, the shortest path is indeed always given by $e^{tA}$ with $A \in \text{sym}(N)$.

We can now ask how $A$ is related to the target state $G_\text{T}$. We must have

$$G_\text{T} = e^A G_\text{R} e^{A^\mathsf{T}}. \tag{310}$$

Now requiring that $A \in \text{sym}(N)$ implies that $U = e^A$ is symmetric with respect to the basis where $G_\text{R}$ is the identity. In an invariant language, we have

$$g_\text{R} U = U^\mathsf{T} g_\text{R}. \tag{311}$$

With this in hand, we can claim that the linear map $U = \sqrt{G_\text{T} g_\text{R}}$ will do the job. Importantly, $U$ satisfies $U G_\text{R} = G_\text{R} U^\mathsf{T}$. We can explicitly verify that

$$\sqrt{G_\text{T} g_\text{R}}\, G_\text{R} \left(\sqrt{G_\text{T} g_\text{R}}\right)^\mathsf{T} = \sqrt{G_\text{T} g_\text{R}}\, \sqrt{G_\text{T} g_\text{R}}\, G_\text{R} = G_\text{T} g_\text{R} G_\text{R} = G_\text{T}. \tag{312}$$

The algebra element that generates $U$ is given by $A = \log U = \frac{1}{2} \log G_\text{T} g_\text{R}$. We have $s = \|A\| = \frac{1}{2}\|G_\text{T} g_\text{R}\|$. Let us note at this point that all expressions, such as $\log G_\text{T} g_\text{R}$ and $\sqrt{G_\text{T} g_\text{R}}$ are well defined, because $G_\text{T} g_\text{R}$ is a positive symmetric, symplectic matrix in a basis where $g_\text{R}$ is the identity. This fact implies that $G_\text{T} g_\text{R}$ is (a) diagonalizable and (b) has positive non-zero eigenvalues.

Of course, the linear map $G_\text{T} g_\text{R}$ is precisely the relative covariance matrix (86) between our target state and reference state,

$$\Delta^a{}_b = (G_\text{T})^{a,c} (g_\text{R})_{c,b}. \tag{313}$$

This matrix encodes the invariant information about the relation between the reference state $|G_\text{R}\rangle$ and the target state $|G_\text{T}\rangle$. The eigenvalues of $\Delta$ come in conjugate pairs $(e_i, 1/e_i)$. We can compute the geodesic distance, which is equal to the norm $\|A\|$, directly[66] from $\Delta$:

$$s = \|A\| = \frac{1}{2}\sqrt{\text{Tr}[(\log \Delta)^2]}. \tag{314}$$

**Normal mode decomposition**

The result for the minimal geodesic is equivalent to stating that for any two Gaussian states $|G_\text{R}\rangle$ and $|G_\text{T}\rangle$, there exists a set of preferred normal modes, such that the optimal geodesic just corresponds to a linear combination of single-mode squeezing operations on these independent normal modes.

Let us consider a reference state $|G_\text{R}\rangle$ and a target state $|G_\text{T}\rangle$. We can choose a symplectic basis, such that the covariance matrix $G_\text{R}$ is simply the $2N \times 2N$ identity matrix:

$$G_R := \begin{pmatrix} \mathbb{1} & 0 \\ 0 & \mathbb{1} \end{pmatrix}. \tag{315}$$

This is always possible because $G_\text{R}$ is a positive, symmetric bilinear form. The covariance matrix $G_\text{T}$ will be another general symmetric matrix. However, we can still change basis by acting with a matrix $u$ in the stabilizer group of $G_\text{R}$, which leaves eq. (315) invariant. As in

---

[66]Another useful quantity that can be directly computed from $\Delta$ is the inner product

$$|\langle G_\text{R} | G_\text{T}\rangle|^2 = \det \frac{\sqrt{2}\Delta^{1/4}}{\sqrt{\mathbb{1} + \Delta}}.$$

eq. (284), $u$ is then an orthogonal matrix which acts by a similarity transformation on $G_{\mathrm{T}}$ and by choosing $u$ appropriately, we can put the latter in a diagonal form. Due to the fact that the covariance matrix of a pure Gaussian state is itself a symplectic matrix, the diagonal form of $G_{\mathrm{T}}$ will consist of conjugate pairs of eigenvalues, *i.e.,*

$$
G_{\mathrm{T}} := \begin{pmatrix} 1/e_1 & & & & & \\ & \ddots & & & & \\ & & 1/e_N & & & \\ & & & e_1 & & \\ & & & & \ddots & \\ & & & & & e_N \end{pmatrix}. \tag{316}
$$

We can refer to our final basis as $\xi := (q_1, \cdots, q_N, p_1, \cdots, p_N)$. In this basis, the matrix representation of $\Delta = G_{\mathrm{T}} g_{\mathrm{R}}$ will be same as the one of $G_{\mathrm{T}}$, because $G_{\mathrm{R}}$ and $g_{\mathrm{R}}$ are represented by the identity. However, the eigenvalues $e_i$ are only matrix invariants of $\Delta$, but not of $G_{\mathrm{R}}$ nor of $G_{\mathrm{T}}$.

The basis chosen above provides a normal mode decomposition, where each conjugate pair $(q_i, p_i)$ corresponds to a normal mode with a single degree of freedom. Both $|G_{\mathrm{R}}\rangle$ and $|G_{\mathrm{T}}\rangle$ can be written as tensor products over normal mode states,

$$
|G_{\mathrm{R}}\rangle = |0\rangle \otimes \cdots \otimes |0\rangle, \qquad |G_{\mathrm{T}}\rangle = |e_1\rangle \otimes \cdots \otimes |e_N\rangle, \tag{317}
$$

where the state $|0\rangle$ is the ground state of $H = \frac{1}{2}(p_i^2 + q_i^2)$, while the $|e_i\rangle$'s are the ground states of $H = \frac{1}{2}(p_i^2 + e_i^2 q_i^2)$. The generator $A$ is diagonal in the same basis and given by

$$
A := \frac{1}{2} \log \Delta = \frac{1}{2} \begin{pmatrix} -\log e_1 & & & & & \\ & \ddots & & & & \\ & & -\log e_N & & & \\ & & & \log e_1 & & \\ & & & & \ddots & \\ & & & & & \log e_N \end{pmatrix}. \tag{318}
$$

The squeezing operator producing $|G_{\mathrm{T}}\rangle = e^{-i\hat{A}}|G_{\mathrm{R}}\rangle$ is given by $\hat{A} = \frac{1}{2} \xi^a \omega_{a,c} A^c{}_d \xi^d$ (where $\omega_{a,c} \Omega^{c,b} = \delta_a^b$). This can be explicitly written as

$$
\hat{A} = \sum_{i=1}^{N} \underbrace{\frac{\log e_i}{4}(\hat{q}_i \hat{p}_i + \hat{p}_i \hat{q}_i)}_{\hat{A}_i} = \sum_{i=1}^{N} \frac{\log e_i}{2} \widehat{W}^{i,i}, \tag{319}
$$

where the last expression uses the notation in eq. (91). Hence the generator producing the optimal circuit simply consists of $N$ commuting single-mode squeezing operators $\hat{A}_i$ which squeeze each of the normal modes independently.

Furthermore, it is now straightforward to evaluate the complexity using the above results. However, let us first note that a similarity transformation was needed to put $G_{\mathrm{T}}$ in the diagonal form given in eq. (316). Hence we should focus on the $F_2$ or $\kappa = 2$ cost functions (given in eqs. (8) and (9), respectively) since they are invariant under rotations of the basis. From eq. (319) or by comparing the matrix generator $A$ in eq. (318) with the $\mathfrak{sp}(2N, \mathbb{R})$ generators in eq. (95), we see that the tangent vector to this circuit is simply given by

$$
Y^{W_{i,i}} = -\frac{1}{2} \log e_i, \tag{320}
$$

and hence the $F_2$ complexity is given by

$$\mathcal{C}_2(G_{\text{R}}, G_{\text{T}}) = \frac{1}{2}\sqrt{\sum_i (\log e_i)^2} = \frac{1}{2\sqrt{2}}\sqrt{\text{Tr}[(\log \Delta)^2]}, \tag{321}$$

where the extra factor of $1/\sqrt{2}$ appears in the second expression because the eigenvalues of the relative covariance matrix appear in conjugate pairs, *e.g.,* see eq. (316). This last expression gives a simple covariant formula for the $F_2$ complexity in terms of the relative covariance matrix, which can be applied for any reference and target states without any further calculation. In addition, since the tangent vector (320) is constant, it follows that the $\kappa = 2$ complexity is simply related to the above expression with

$$\mathcal{C}_{\kappa=2}(G_{\text{R}}, G_{\text{T}}) = [\mathcal{C}_2(G_{\text{R}}, G_{\text{T}})]^2 = \frac{1}{8}\text{Tr}[(\log \Delta)^2]. \tag{322}$$

**Application to** $\text{Sp}(2, \mathbb{R})$

In section 4.2, we examined the special case of $\text{Sp}(2, \mathbb{R})$ using the coordinates $(\rho, \theta, \tau)$ given in eq. (111). In eq. (119), it was found that the final state only depends on $\rho$ and the combination $\chi = \theta + \tau$. Further, it was found that the optimal circuit preparing a particular target state in the relevant equivalence class was given by the simple straight-line geodesic in eq. (121). That is,

$$\rho(\sigma) = \rho_1 \sigma, \quad \theta(\sigma) = \theta_1, \quad \tau(\sigma) = 0, \tag{323}$$

where $\rho_1$ and $\theta_1$ characterize the target state $G_{\text{T}}$. It is interesting to understand this result from the perspective presented in this appendix and so we consider here applying the previous analysis to the special case of $N = 1$.

First, we can consider $\text{Sp}(2, \mathbb{R})$ as a U(1) fiber bundle over the plane parameterized by $(\rho, \theta) = (\rho, \chi)$ with fixed $\tau$. The subgroup U(1) that preserves the reference state $G_{\text{R}} = \mathbb{1}$ is generated by

$$K_3 = \frac{V + Z}{\sqrt{2}} = \begin{pmatrix} 0 & 1 \\ -1 & 0 \end{pmatrix}, \tag{324}$$

which is an antisymmetric matrix in accord with eq. (288). The $U(1)$ fiber above the identity is then given by simple rotation matrices

$$e^{\tau K_3} = \begin{pmatrix} \cos \tau & -\sin \tau \\ \sin \tau & \cos \tau \end{pmatrix}, \tag{325}$$

which we recognize to agree with $U(0, 0, \tau)$ in eq. (111). This is simply the stabilizer group of the reference state in the case $\lambda_{\text{R}} = 1$, see eq. (85). This means that $K_3$ spans the subalgebra $\mathfrak{u}(N)$ for $N = 1$ whose orthogonal complement $\mathfrak{u}_\perp(N)$ with respect to the right-invariant inner product (283) is spanned by

$$K_1 = W = \begin{pmatrix} 1 & 0 \\ 0 & -1 \end{pmatrix} \quad \text{and} \quad K_2 = \frac{Z - V}{\sqrt{2}} = \begin{pmatrix} 0 & 1 \\ 1 & 0 \end{pmatrix}. \tag{326}$$

This subspace consists of symmetric matrices with respect to $G_{\text{R}}$, in accord with eq. (290). These two generators are referred to as $\text{sym}(N) = \mathfrak{u}_\perp(N)$ with $N = 1$. Applying the exponential map to an arbitrary generator $\rho(\cos \theta\, K_1 + \sin \theta\, K_2) \in \text{sym}(1)$ gives

$$e^{\rho(\cos \theta\, K_1 + \sin \theta\, K_2)} = \begin{pmatrix} \cosh \rho - \sin \theta \sinh \rho & \cos \theta \sinh \rho \\ \cos \theta \sinh \rho & \cosh \rho + \sin \theta \sinh \rho, \end{pmatrix} = U(\rho, \theta, 0), \tag{327}$$

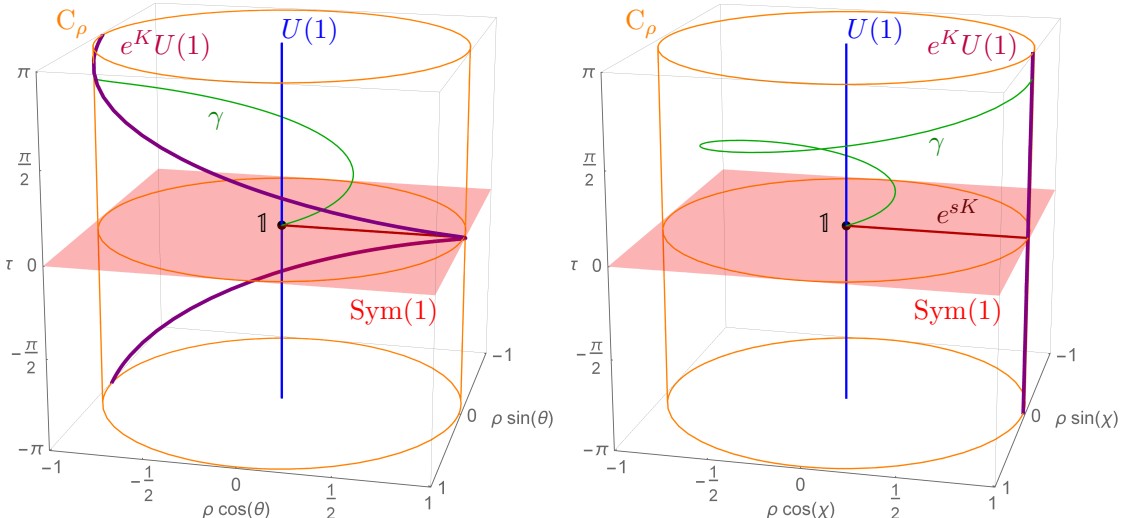

Figure 30: This figure illustrates the geometry of $\mathrm{Sp}(2,\mathbb{R})$ in the coordinates $(\rho,\theta,\tau)$ in the left picture and $(\rho,\chi,\tau)$ in the right picture with $\chi = \theta + \tau$. The identity element $\mathbb{1}$ is in the center on the surface $\mathrm{Sym}(1)$ corresponding to $\tau = 0$. The subgroup $U(1)$ corresponds to the vertical line with $\rho = 0$. The equivalence class $e^K U(1)$ of group elements that prepare the same target state are given by spiral (left picture) and vertical line (right picture). For a reference state with $|G(\rho,\chi)\rangle$, all group elements preparing this state necessarily lie on the cylindrical surface $\mathrm{C}_\rho$. A general geodesic $\gamma$ winds around, but the minimal geodesic corresponds to a straight line of the form $e^{sK}$.

which we recognize in the last equality to be given by elements in the plane with $\tau = 0$, again symmetric matrices. We therefore refer to this subset as $\mathrm{Sym}(N) = \exp(\mathrm{sym}(N))$ with $N = 1$. Let us emphasize that this is not a subgroup, because the product of two symmetric matrices will in general not be symmetric.

In this notation, the straight-line geodesic in eq. (323) preparing the state $G_{\mathrm{T}}(\rho = \rho_1, \chi = \theta_1)$ is described by

$$\gamma(\sigma) = U(\rho = \sigma\rho_1, \theta = \theta_1, \tau = 0) = e^{\sigma\rho_1(\cos\theta_1 K_1 + \sin\theta_1 K_2)}. \tag{328}$$

This path moves out radially in the $\tau = 0$ plane until it reaches the group element $\gamma(1) = U(\rho_1, \theta_1, 0)$. To understand the geometry better, it is useful to unwind the $(\rho, \theta, \tau)$ coordinate system by replacing $\theta$ with $\chi = \theta + \tau$ as second coordinate. Practically, we are rotating the $\tau = $ constant planes by $\tau$. The metric (112) with the coordinates $(\rho, \chi, \tau)$ becomes

$$\mathrm{d}s^2 = \mathrm{d}\rho^2 + \cosh(2\rho)\sinh^2\rho\,\mathrm{d}\chi^2 + \cosh(2\rho)\mathrm{d}\tau^2 + 2[2\cosh(2\rho)+1]\sinh^2\rho(\mathrm{d}\tau - \mathrm{d}\chi)\mathrm{d}\tau. \tag{329}$$

Now the cylindrical surfaces in eq. (294) are simply the surfaces of constant $\rho > 0$: $\mathrm{C}_\rho = \{U(\rho,\theta,\tau)|\theta,\tau \in [0,2\pi]\}$.[67] The radial vector field (295) becomes $\mathcal{R}(\rho,\theta,\tau) = \partial_\rho$, which is easily seen to be orthogonal to the cylinders $\mathrm{C}_\rho$ in the metric (329). We recognize that all group elements that prepare the target state $|G(\rho = \rho_1, \chi = \theta_1)\rangle$, namely $U(\rho_1, \theta_1 - \tau, \tau)$, lie on the cylinder $\mathrm{C}_{\rho_1}$. This feature is, of course, a consequence of our initial assumption that $\lambda_{\mathrm{R}} = 1$ underlying the proof in this appendix. While eq. (142) shows that $\rho_1$ is constant for $\lambda_{\mathrm{R}} = 1$, we can see from eq. (172) that with $\lambda_{\mathrm{R}} \neq 1$, the final radius $\rho_1$ changes as we change

---

[67]Topologically, $C_\rho$ is a torus because the upper and lower boundaries are identified due to the periodicity in $\tau$.

the boundary conditions by applying $U_\phi$ in eq. (167).[68] Further, in agreement with the preceding analysis in the appendix, the optimal geodesic (328) lies in the plane with $\tau = 0$, which corresponds to Sym(1). The length of this path is simply $\rho_1$, which corresponds to the norm of the generator in eq. (328), which agrees with the length given in eq. (314).

## G   Derivation of the TFD covariance matrix in terms of matrix functions

To be entirely self-contained, we rederive some of the previous results on the time-dependent covariance matrices in terms of *matrix functions*. These expressions render the numerical computation of entanglement entropies particularly simple and transparent. We start by making explicit the time dependence of the TFD state of two harmonic oscillators, where each oscillator is evolved according to eq. (10). We start from the Hamiltonian of a single degree of freedom

$$H = \frac{1}{2}\xi h \xi^\intercal, \tag{330}$$

with $\xi = (q, p)$ and

$$h = \mathrm{diag}(\eta, \delta), \tag{331}$$

where $\eta = M\omega^2$, $\delta = 1/M$, but for reasons that will become clear later, we keep the notation general at this point. The covariance matrix $G$ [78, 79] with entries $G^{i,j} = \langle 0|_L \langle 0|_R (\xi^i \xi^j + \xi^j \xi^i)|0\rangle_L |0\rangle_R$ of the vacuum state of two copies of this Hamiltonian is a $4 \times 4$ matrix

$$G = \mathrm{diag}\left(\frac{\delta^{1/2}}{\eta^{1/2}}, \frac{\eta^{1/2}}{\delta^{1/2}}, \frac{\delta^{1/2}}{\eta^{1/2}}, \frac{\eta^{1/2}}{\delta^{1/2}}\right), \tag{332}$$

in the following convention for the coordinates $\xi = (q_\mathrm{L}, p_\mathrm{L}, q_\mathrm{R}, p_\mathrm{R})$. Note these are the original coordinates of the left and right copies which one uses to the define the TFD. The covariance matrix of the TFD state (18) is found to be

$$G(\alpha) = W(\alpha) G W(\alpha)^\intercal, \tag{333}$$

where

$$W(\alpha) = \begin{pmatrix} \cosh(\alpha) & 0 & -\sinh(\alpha) & 0 \\ 0 & \cosh(\alpha) & 0 & \sinh(\alpha) \\ -\sinh(\alpha) & 0 & \cosh(\alpha) & 0 \\ 0 & \sinh(\alpha) & 0 & \cosh(\alpha) \end{pmatrix}, \tag{334}$$

following the framework of ref. [95]. This is obtained by acknowledging that the TFD in eq. (18) has the form of a pure vacuum state, time evolved under a quadratic Hamiltonian with the real number $\alpha$ encoding the inverse temperature formally taking the role of a time. This is a convenient form of the covariance matrix of the TFD of a single decoupled mode. One can easily verify that $W(\alpha) \Omega W(\alpha)^\intercal = \Omega$, where $\Omega$ is the symplectic form

$$\Omega = \begin{pmatrix} 0 & 1 \\ -1 & 0 \end{pmatrix} \oplus \begin{pmatrix} 0 & 1 \\ -1 & 0 \end{pmatrix} \tag{335}$$

in this convention, so that indeed $W(\alpha) \in \mathrm{Sp}(4, \mathbb{R})$ for all values of $\alpha$. Again following ref. [95], the time evolved TFD state (31) can be expressed as a matrix exponential

$$G(\alpha, t) = e^{tK} G(\alpha) e^{tK^\intercal}, \tag{336}$$

---

[68]Recall that the $\rho_1$ being a constant was an essential property allowing us to demonstrate that the straight-line geodesics were in fact the optimal circuits in section 4.2, *i.e.,* see discussion around eq. (125).

with $K$ having components $K^a{}_b = \Omega^{a,c}(h \oplus h)_{c,b}$. This is a concise form of the covariance matrix of the time-dependent TFD state of a single mode. Having paved the ground in the case of a single mode and its double, we now turn to the TFD state of the full quantum field theory with the Hamiltonian (175). This Hamiltonian can be written with respect to the following choice of coordinates

$$\xi = (q_{L,1}, q_{L,2}, \ldots, q_{L,N}, p_{L,1}, p_{L,2}, \ldots, p_{L,N}, q_{R,1}, q_{R,2}, \ldots, q_{R,N}, p_{R,1}, p_{R,2}, \ldots, p_{R,N}), \quad (337)$$

as

$$H = \frac{1}{2} \xi k \xi^{\mathsf{T}}, \quad (338)$$

where

$$k = \delta \mathbb{1}_N \oplus x \oplus \delta \mathbb{1}_N \oplus x \quad (339)$$

and

$$x = \delta^{-1} m^2 \mathbb{1}_N + \delta^{-3} \operatorname{Toeplitz}(2, -1, \ldots, -1). \quad (340)$$

The latter is a banded matrix with the entry 2 on the main diagonal and $-1$ on the first off diagonal. This is a concise way of expressing the Hamiltonian of the UV regularized quantum field theory. The matrix $x$ can be diagonalized as

$$O x O^{\mathsf{T}} = D, \quad (341)$$

with a suitable $O \in \mathrm{O}(N)$, so that $k$ can be diagonalized according to

$$V k V^{\mathsf{T}} = \operatorname{diag}(D, \mathbb{1}_N \delta, D, \mathbb{1}_N \delta), \quad (342)$$

with $V := O \oplus O \oplus O \oplus O \in \mathrm{Sp}(4N, \mathbb{R})$. This is easy to see: The coupling is in the positions only, so the momenta will be left unchanged and captured by the diagonal matrix $\mathbb{1}_N \delta$ by this real orthogonal transformation, while the positions are diagonalized to $D$. This way presents an alternative to the direct complex Fourier transform. We now turn to a closed form of the covariance matrix of the entire time dependent thermofield double state of the quantum field theory. We will keep the expressions entirely in the form of matrix exponentials. Using eqs. (332) and (342), the covariance matrix of the initial vacuum state is found to be

$$G = \operatorname{diag}\left(\delta^{1/2} x^{-1/2}, \delta^{-1/2} x^{1/2}, \delta^{1/2} x^{-1/2}, \delta^{-1/2} x^{1/2}\right) \quad (343)$$

in terms of the matrix square root of $x$ and its inverse [87]. The covariance matrix of the full $N$-mode thermofield double is $G(\alpha) = W(\alpha) G W(\alpha)^{\mathsf{T}}$, where now

$$W(\alpha) = \begin{pmatrix} \cosh(\alpha)\mathbb{1}_N & 0 & -\sinh(\alpha)\mathbb{1}_N & 0 \\ 0 & \cosh(\alpha)\mathbb{1}_N & 0 & \sinh(\alpha)\mathbb{1}_N \\ -\sinh(\alpha)\mathbb{1}_N & 0 & \cosh(\alpha)\mathbb{1}_N & 0 \\ 0 & \sinh(\alpha)\mathbb{1}_N & 0 & \cosh(\alpha)\mathbb{1}_N \end{pmatrix}. \quad (344)$$

This is again a convenient form that reflects the fact that the transformation that diagonalizes the position part of the Hamiltonian commutes with the transformation that maps the vacuum onto its TFD state for single modes. Following eq. (30) we obtain the final expression for the covariance matrix of the time-dependent TFD state which reads

$$G(\alpha, t) = \exp(t k / 2) G(\alpha) \exp(t k^{\mathsf{T}} / 2), \quad (345)$$

where the factor of $1/2$ is a result of the convention taken in eq. (30). Here, the expression is concise, as the Fourier transform does not have to be explicitly performed, the covariance matrix being expressed directly as a matrix function of the coupling matrix.

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
