# Peer review of "Complexity and entanglement for thermofield double states"

_SciPost Physics, doi:SciPost Phys. 6, 034 (2019)_

## Round 2 · Referee Report · Anonymous (Referee 1) · 2018-12-18

Strengths

1: The paper is detailed and well-written. 2: The complexity of the time-dependent thermofield double states for the free scaler field theory is discussed comprehensively.

Weaknesses

1: The results are not sufficient to either confirm or rule out either the CV or CA proposals. 2: It is not clear how to generalize the results to interacting field theories.

Report

The authors of the paper "Complexity and entanglement for thermofield double states"
study the circuit complexity for the thermofield double (TFD) states in the free scalar quantum field theories using the Nielsen's proposal. The main idea is to provide some insights into the CA and CV proposals. Although the results are not sufficient
to either confirm or rule out either the CV or CA proposals, the authors believe that the results may suggest some degree of universality.
The time-independent part of the article is an extension of the reference [23] using the language of covariance matrices which is easier and more elegant. Here, the authors conclude that the complexity of the formation of the TFD is proportional to the thermodynamic entropy in accordance with the conclusions of the references [52,53]. I am not sure the time-independent section offers many new physical insights into the conclusions of [23,52].

The complexity of the time-dependent thermofield double states has been also studied before in [53]. However, the authors not only extend the results of [52,53] (and also correct some mistakes) they make many detailed calculations for variety of cases and conclude that the complexity of formation of the TFD state does not exhibit the late-time growth characteristic of holographic theories/fast scramblers due to the Gaussian nature of the TFD state for the free scalar theories.

The paper is well-written and although some presented results are not original, it has enough merit to be published in Scipost. In particular, since the paper has a pedagogical style of writing it can be a useful reference for the researchers that just want to start working on this field.

Requested changes

Major comments: 1: In page 34 why for small $t$ the growth of $\mathcal{C}_1^{LR}$ is initially linear, while that for $\mathcal{C}_2$ is quadratic? 2: I have similar question for the Entanglement evolution: In the context of harmonic oscillators(HO) it was argued in ( Phys. Rev. A 90,062330 (2014)) and later confirmed numerically in ( Phys. Rev. B 90, 205438 (2014)) that there is a quadratic initial growth in the EE. For similar conclusions in the context of the holography and CFT see ( Phys. Rev. Lett.112, 011601 (2014)) and (J. Cardy, KITP Conference: Closing the entanglement gap: Quantum information, quantum matter, and quantum fields (2015)). A comment is needed addressing the reason for the absence of this regime in the paper. 3: In the section (6.5) the authors attempt to connect the logarithmic contributions to the evolution of the entanglement to the zero mode. To the best of my reading, they do not isolate the contribution of the zero mode. Of course, the zero mode is there and they find a log behaviour but it is absolutely not clear from the calculation that they are related. I think the authors either need to isolate the zero mode contribution and successfully prove their point or change the claim. 4: At page 63 the authors say that the quasiparticle picture was introduced in 92-97. The LR bound gives an upper bound to the entanglement growth and more than that I think the LR velocity has nothing to do with the quasi-particle velocity. I recommend the authors to be more careful here in the citation.

There are also few typos and minor issues in the paper: 1: In the page 3 define $t_R$ and $t_L$. 2: References [74] and [75] are repeated. 3: In Figure 4, the vertical label seems to me strange. 4: In the equations (196) and (197) I am not sure in the indices the $n=0$ is defined!

Attachment

  • validity: high
  • significance: good
  • originality: ok
  • clarity: top
  • formatting: perfect
  • grammar: perfect

Author:  Lucas Hackl  on 2019-02-11  [id 436]

(in reply to Report 2 on 2018-12-18)

To begin, we would like to thank the referee for his/her careful reading of the paper and for providing useful insights. As a general comment relevant for all three reports, we agree with all of the referees that our analysis of circuit complexity for Gaussian states has limitations, but we view our investigation as one of the initial steps in a much longer research program aimed at defining complexities for general quantum field theory states, and in particular, for situations which could be of relevance to holography. However, the circuit complexity of quantum field theory states is very new concept for researchers both in high energy physics and quantum information science. Therefore performing explicit calculations in simple situations, such as with Gaussian states, provides a good forum where we can begin to develop some familiarity and physical intuition for these new ideas. In this regard, we would like to point out that there have been a number of (surprising) qualitative results where agreement was found between complexities in free field theories and the predictions of the holographic conjectures. In particular, the free field analysis of [1707.08570, 1707.08582] helped in understanding the structure of UV divergences found in the holographic calculations of [arXiv:1612.00433]. Further, Gaussian states have been successfully applied as a starting point of a perturbative analysis in the weak coupling limit, e.g., see [1808.03105]. Of course, we fully agree that the problem of defining circuit complexity in the strong coupling regime is an important open problem, in particular, in order to make contact with the CA and CV conjectures. Some interesting steps in this direction have already been taken in refs. [1706.07056, 1807.04422]. Further, we would like to emphasize that the setup of the current paper is not merely a rephrasing the results of [23] (Jefferson & Myers) in the language of covariance matrices, but rather we lay down a systematic framework which allows one to study, in a manageable way, general Gaussian states which are reached from the reference state by gates spanning the full group $\mathrm{Sp}(2N,\mathbb{R})$, rather than just $\mathrm{GL}(N,\mathbb{R})$ as in [23]. In particular, we explicitly compute the minimal geodesic distance for arbitrary $N$. We would also like to emphasize that we presented a variety of new results about the entanglement entropy in the TFD state for the free scalar theory, including a comparison with the circuit complexity behaviour and how it can be understood from a quasi-particle picture from refs. [74,75], which have not been explored in the past.

Let us now turn to the referee's list of major comments under Requested changes": (1) Motivated by the referee's comment, we took a closer look at the early linear growth of $\Delta\mathcal{C}_1^{(\mathrm{LR})}$ (compared to the quadratic growth of $\Delta\mathcal{C}_2$) and noticed that in thesimple limit" presented on pages 32-34, the growth of $\Delta\mathcal{C}_1^{(\mathrm{LR})}$ also becomes quadratic. This change arises because the linear growth of $\Delta\mathcal{C}_1^{(\mathrm{LR})}$ in eq. (151) is, in the limit $\hat \alpha \rightarrow -\infty$, given by $-4 e^{2 \hat{\alpha}} \hat \alpha \sinh (2 \alpha ) (\sinh (2 \alpha )+\sqrt{2} \cosh (2 \alpha )) \omega t$, which vanishes in this limit. This is actually within an absolute value, and so if we continue to negative times, we obtain $|t|$ behaviour at the origin. We have added a comment on this issue below eq. (160). Unfortunately, we do not have further physical insights for the cause of this effect beyond the mathematical derivation but hope to develop one in the future. (2) The referee asks about the quadratic growth of the entanglement entropy at early times. Indeed upon closer examination of figure 20, we noticed that in a very narrow region near $t=0$, the behaviour deviates from the linear growth. We believe this is a manifestation of the quadratic growth mentioned by the referee but unfortunately we do not have enough resolution in this region to examine the behaviour more carefully at the moment. Again, we hope to return to this question in the future. We have added a clarification on this point in footnote 42, including the references pointed out by the referee. (3) In keeping with the referee's comments, we weakened our initial claim on what we can say analytically about the zero mode: In this subsection, we analyze the time dependence of the entanglement entropy for a single degree of freedom in the limit $m\to 0$ and find a logarithmic contribution. We then argue that the extracted asymptotic behavior is due to the zero mode and also applies to larger subsystem.'' We showed analytically for a subsystem of a single degree of freedom how a logarithmic contribution (in excellent agreement with the numerics) arises in the limit of small $m$ (zero mode limit) and then argue from there that such a contribution of a de-localized mode should contribute additively to larger subsystems. We do not present a rigorous proof for the latter, but our derivation explicitly shows how the zero mode is responsible for the appearance of a logarithmic term in $m\to 0$ limit. (4) The referee writesThe LR bound gives an upper bound to the entanglement growth and more than that I think the LR velocity has nothing to do with the quasi-particle velocity." We have now extended this section and elaborated on this point in more detail. We give citation credit only to the works which discuss the quasi-particle picture in the context of equilibration in quantum many-body systems. We also state more clearly what the Lieb-Robinson bound precisely limits: It provides an upper bound to the fastest group velocity in the system, and in fact to the speed of any information propagation, up to exponentially suppressed terms. As such, Lieb-Robinson bounds provide information about entanglement growth, but also about the speed of quasi-particles: Any excitation can only propagate at most with the Lieb-Robinson velocity. This is now stated more elaborately in the draft.

Regarding the typos and minor issues: - We have added a comment under eq. (1). - When it comes to repeated reference in refs. [74]-[75], what we meant is ref. [74] being the first paper in the series, i.e. 1608.00614, and [75] stay as it was. We added 1608.00614 to our references and replaced the citation. - We have added a clarification in the caption below the figure. - Using the letter $n$ was a typo from previous notation. We replaced $n=0$ with $k=0$ here, which is the same notation for the zero mode as used in eq. (192).

Again, we thank the referee for his/her careful reading of the manuscript.

---

## Round 2 · Referee Report · Anonymous (Referee 2) · 2018-12-22

Strengths

1- The paper is well-written and the explanations are clear. 2- It is self-contained, all necessary definitions and derivations are provided. 3- The question studied, namely how to define and compute the complexity of a highly entangled quantum mechanical state, is important.

Weaknesses

The prospects of the method beyond the Gaussian states look rather limited.

Report

The authors tackle an important problem: How to define and compute the complexity of a quantum state in an extended system. The state of interest is thermofield double in a bosonic system described by a free field theory. The authors adopt the circuit complexity as a definition of state complexity and use a geometric approach (originally due to Nielsen) which essentially maps the problem of finding complexity to an optimization problem. The authors took steps towards a more general framework beyond earlier papers (written by some of the current authors) by framing complexity calculations in terms of covariance matrices and the associated symplectic transformations, which provide a natural way for dealing with Gaussian states.

The study of thermofield double states is timely, especially in lights of connections to holographic theories and the proposals for quantifying the size of the Einstein-Rosen bridge dual to holographic complexity. Compared to the holographic results, the complexity of formation shares the similarity that it is proportional to the thermodynamic entropy, while it differs from holographic scenarios as it saturates after a time of the order of the inverse temperature.

This work is valuable as it possibly provides a toolbox to explore the complexity of the quantum field theory side of holographic theories. However, it is not clear how the definition of complexity in terms of Gaussian unitary circuits will be eventually helpful for interacting theories. Moreover, the unitary circuits here are implemented in momentum space which effectively gives rise to a circuit comprising of highly non-local gates. I should note that the latter two remarks are rather open-ended at this stage.

Requested changes

Some minor changes/suggestions:

1-Typo: second line after (130), the analysis *for* the $x_-$ mode.
2-Fig. 7, it would be better to include a legend as opposed to using confusing terms such as top/bottom curves, as curves cross and their order reverses over in each plot.

  • validity: high
  • significance: high
  • originality: good
  • clarity: high
  • formatting: excellent
  • grammar: perfect

Author:  Lucas Hackl  on 2019-02-11  [id 435]

(in reply to Report 1 on 2018-12-22)

To begin, we would like to thank the referee for his/her careful reading of the paper and for providing useful insights.
As a general comment relevant for all three reports, we agree with all of the referees that our analysis of circuit complexity for Gaussian states has limitations, but we view our investigation as one of the initial steps in a much longer research program aimed at defining complexities for general quantum field theory states, and in particular, for situations which could be of relevance to holography. However, the circuit complexity of quantum field theory states is very new concept for researchers both in high energy physics and quantum information science. Therefore performing explicit calculations in simple situations, such as with Gaussian states, provides a good forum where we can begin to develop some familiarity and physical intuition for these new ideas. In this regard, we would like to point out that there have been a number of (surprising) qualitative results where agreement was found between complexities in free field theories and the predictions of the holographic conjectures. In particular, the free field analysis of [1707.08570, 1707.08582] helped in understanding the structure of UV divergences found in the holographic calculations of [arXiv:1612.00433]. Further, Gaussian states have been successfully applied as a starting point of a perturbative analysis in the weak coupling limit, e.g., see [1808.03105]. Of course, we fully agree that the problem of defining circuit complexity in the strong coupling regime is an important open problem, in particular, in order to make contact with the CA and CV conjectures. Some interesting steps in this direction have already been taken in refs. [1706.07056, 1807.04422].
The referee also observed that our circuits involve ``highly non-local gates". This is of course true, but we would like to add that it has been argued in [1801.01137] that holographic complexity must also involve a non-local gates, for both holographic conjectures. Further, this is also the case for MERA tensor networks (e.g., [cond-mat/0512165]) where the gates introduce entanglement at different scales at different steps of the circuit and which have been conjectured to be related to the structure of holographic spacetimes [0905.1317].
We also took care of the smaller comments and typos that the referee pointed out. We thank the referee again for his/her careful reading of the manuscript.

---

## Round 2 · Referee Report · Anonymous (Referee 3) · 2019-1-11

Strengths

  1. Clear presentation

Weaknesses

  1. Discussion limited to harmonic oscillator and free fields
  2. Presentation is too long for the limited applicability of the results.

Report

The study of complexity in field theories is argued to be interesting based on observations made in the holographic context about black hole interiors. The present paper aims to understand a particular notion of complexity for thermofield double states of free fields. They exploit various simple features of Gaussian states, and analyze certain proposals based on geodesic distance in the space of unitaries acting on such states and find some reasonable answers.

It is at present unclear how robust the results derived in this limited context are and whether they hold a deep lesson for the relevance of complexity in quantum field theory. The paper does contain also some results about entanglement evolution which is of interest.

Requested changes

The paper seems fine in its current form.

  • validity: good
  • significance: good
  • originality: good
  • clarity: high
  • formatting: excellent
  • grammar: excellent

Author:  Lucas Hackl  on 2019-02-11  [id 437]

(in reply to Report 3 on 2019-01-11)

To begin, we would like to thank the referee for his/her careful reading of the paper and for providing useful insights.
As a general comment relevant for all three reports, we agree with all of the referees that our analysis of circuit complexity for Gaussian states has limitations, but we view our investigation as one of the initial steps in a much longer research program aimed at defining complexities for general quantum field theory states, and in particular, for situations which could be of relevance to holography. However, the circuit complexity of quantum field theory states is very new concept for researchers both in high energy physics and quantum information science. Therefore performing explicit calculations in simple situations, such as with Gaussian states, provides a good forum where we can begin to develop some familiarity and physical intuition for these new ideas. In this regard, we would like to point out that there have been a number of (surprising) qualitative results where agreement was found between complexities in free field theories and the predictions of the holographic conjectures. In particular, the free field analysis of [1707.08570, 1707.08582] helped in understanding the structure of UV divergences found in the holographic calculations of [arXiv:1612.00433]. Further, Gaussian states have been successfully applied as a starting point of a perturbative analysis in the weak coupling limit, e.g., see [1808.03105]. Of course, we fully agree that the problem of defining circuit complexity in the strong coupling regime is an important open problem, in particular, in order to make contact with the CA and CV conjectures. Some interesting steps in this direction have already been taken in refs. [1706.07056, 1807.04422].

---

## Round 3 · Author Response

Dear Editor,

We would like to thank the referees for their careful reading and the detailed reports. We replied to all points raised by the referees in our direct replies. We believe that the updated version has improved clarify and we hope that it now accepted for publication in SciPost.

Best regards,
Shira Chapman, Jens Eisert, Lucas Hackl, Michal P. Heller, Ro Jefferson, Hugo Marrochio, Robert C. Myers

---

## Round 3 · List of Changes

Based on the three referee reports, we implemented the following changes (for more details see our reply to the referees):
- We corrected the typo (or -> for) pointed out.
- We adjusted figure 7, where we now clearly refer to individual curves.
- We included a definition for $t_R$ and $t_L$ after eq. (1).
- We clarified refs. [74-75] and also added 1608.00614.
- We added a clarification regarding the vertical axis of figure 4.
- We corrected a typo (n -> k) in eq. (196) and (197).
- We added a comment regarding the early linear growth of $\Delta\mathcal{C}_1^{\mathrm{LR}}$ below eq. (160).
- We added a clarification (new footnote 42) regarding the early growth shown in figure 20.
- Following the referee's suggestion, we weakened our initial claim on what we can say analytically about the zero more, which now states: “In this subsection, we analyze the time dependence of the entanglement entropy for a single degree of freedom in the limit m → 0 and find a logarithmic contribution. We then argue that the extracted asymptotic behavior is due to the zero mode and also applies to larger subsystem."
- We elaborated on the relationship between the Lieb-Robinson bound and the quasi-particle picture, which can be found on page 63.

---

## Round 4 · Author Response

We apologize for missing the reference 'P. Calabrese and J. Cardy, J. Stat. Mech. P04010 (2005)'. This work should indeed be cited before Ref. [92] and [93] when the introduction of the quasi-particle picture is discussed. Again, we apologize for the omission.

---

## Round 4 · List of Changes

We added 'P. Calabrese and J. Cardy, J. Stat. Mech. P04010 (2005)' as new reference [92].

---

## Editorial Decision

published